# Optimal Top-Two Method for Best Arm Identification and Fluid Analysis

**Agniv Bandyopadhyay**
TIFR Mumbai, India
agniv.bandyopadhyay@tifr.res.in

**Sandeep Juneja**
Ashoka University, India
sandeep.juneja2010@gmail.com

**Shubhada Agrawal**
Georgia Institute of Technology, USA
sagrawal362@gatech.edu

## Abstract

Top-2 methods have become popular in solving the best arm identification (BAI) problem. The best arm, or the arm with the largest mean amongst finitely many, is identified through an algorithm that at any sequential step independently pulls the empirical best arm, with a fixed probability $\beta$, and pulls the best challenger arm otherwise. The probability of incorrect selection is guaranteed to lie below a specified $\delta > 0$. Information theoretic lower bounds on sample complexity are well known for BAI problem and are matched asymptotically as $\delta \to 0$ by computationally demanding plug-in methods. The above top 2 algorithm for any $\beta \in (0, 1)$ has sample complexity within a constant of the lower bound. However, determining the optimal $\beta$ that matches the lower bound has proven difficult. In this paper, we address this and propose an optimal top-2 type algorithm. We consider a function of allocations anchored at a threshold. If it exceeds the threshold then the algorithm samples the empirical best arm. Otherwise, it samples the challenger arm. We show that the proposed algorithm is optimal as $\delta \to 0$. Our analysis relies on identifying a limiting fluid dynamics of allocations that satisfy a series of ordinary differential equations pasted together and that describe the asymptotic path followed by our algorithm. We rely on the implicit function theorem to show existence and uniqueness of these fluid ode's and to show that the proposed algorithm remains close to the ode solution.

## 1 Introduction

Stochastic best arm identification (BAI) problem has attracted a great deal of attention in the multi armed bandit community (see [21], [10], [3] for some early references in BAI). The basic problem involves a finite number of unknown probability distributions or arms that can be sampled from independently and the aim is to identify the arm with the largest mean. We consider a popular fixed confidence version of the problem where the sampling is sequential and the aim is to minimise sample complexity while guaranteeing that the probability of selecting the wrong arm is restricted to a pre-specified $\delta > 0$. Applications are many including in healthcare and recommendation systems.

[11] developed asymptotically (as $\delta \to 0$) tight lower bound on sample complexity of $\delta-$correct algorithms for these BAI problems under the assumption that arms belong to a single parameter exponential family (SPEF). This assumption reduces a probability distribution to a single parameter and allows the analysis to better focus on certain aspects of the problem structure. We retain it for similar reasons. The sample complexity lower bounds involve solving an optimization problem that also identifies optimal proportion of allocations across arms. They also propose a track-and-stop

38th Conference on Neural Information Processing Systems (NeurIPS 2024).

algorithm that plugs in the empirical estimates of the distribution parameters in the lower bound and tracks the resulting approximations to optimal proportions of arms to sample. Although, this plug-in algorithm was shown to asymptotically match the lower bound, it involves repeatedly solving an optimization problem and is computationally demanding. [14] consider linear Gaussian bandits, [1] consider bandits with general distributions. Both the references propose track and stop algorithms where computation is sped up through batch processing.

Substantial literature has come up on 'top-2' based, alternative faster and intuitively appealing algorithms to identify the best arm (see [25], [26] for Bayesian approaches; [24], [22], [16] for frequentist approaches). The algorithms essentially proceed by identifying at each stage an empirical winner arm, that is, an arm with the largest mean, and its closest challenger. The empirical arm is pulled with probability $\beta$, and the challenger arm with the complimentary probability. In the frequentist setting, in [16], the challenger arm is the one with the smallest 'index function'. Heuristically, this index function measures the likelihood of the challenger arm actually being the best one. The smaller the index function, more the likelihood. Further, with high probability, the index function increases with increased allocations to the corresponding arm. As is standard (see, e.g., [11], [19]), the algorithm is terminated when the generalized log-likelihood ratio (GLLR, given in Section 3) statistic exceeds a specified threshold. These algorithms are shown to be $\beta$ optimal in the sense that they match the lower bound on sample complexity satisfied by algorithms that pull the best arm $\beta$ fraction of times (see [15] for non-asymptotic analysis when $\beta = 1/2$). However, determining optimal $\beta$ has been an open problem that has generated considerable activity and that we address in this paper.

**Contributions - Algorithm:** The key insight from index based top-2 algorithm is that once a sample is given to a challenger arm with the smallest index, its index function increases. The net effect is that as the algorithm progresses, the challenger arm indexes tend to come close to each other and move together. We build upon the above insight. Through the first order conditions associated with the lower bound problem, we identify a function $g$ that equals zero under optimal allocation when the underlying arm distributions are known. We propose an *anchored* top-2 type algorithm where when $g > 0$, the empirical winner arm is pulled and that tends to decrease $g$. When $g < 0$, our algorithm pulls a challenger arm (arm with the smallest index function), and that typically increases $g$. We observe that the indexes of challenger arms that have been pulled, tend to rise up together until they catch up with arms with higher indexes. Once challenger arms associated with all the indexes have been pulled, call this the time to stability, then, since $g$ is close to zero and indexes of all challenger arms are close together, it can be seen that the proportionate samples to the empirical winner and the remaining arms are close to the optimal proportions as per the lower bound. This continues until the GLLR statistic exceeds a threshold, roughly of order $\log(1/\delta)$. The time to stability can be bounded from above by a random time with finite expectation independent of $\delta$, while the time from stability till the GLLR statistic hits a threshold scales with $\log(1/\delta)$ with a constant that matches the lower bound.

**Fluid model:** Our other key contribution is to capture the above intuitive description through constructing an idealized *fluid* dynamics where $g$ stays equal to zero once it touches zero and where the indexes that have been pulled, remain equal and rise together as the algorithm progresses. We further show that the resulting equations have an invertible Jacobian. Implicit function theorem (IFT) (see [20, Appendix A.6] for an introduction to IFT) then becomes an important tool in analyzing this idealized fluid system as it allows the arm allocations $(N_a : a \in K)$ to be unique functions of the overall allocation $N$. IFT further allows us to identify the ordinary differential equations satisfied by the derivatives $N'_a = \frac{dN_a}{dN}$ as the allocations $N$ increase. The overall path till stability is constructed by pasting together the ode paths followed by arm allocations as the set of indexes that have already been pulled and are increasing together with $N$, meet another higher index. Once all the indexes have been pulled, our ode stabilizes so that the proportions $N_a/N$ thereafter remain constant and equal the optimal proportions as $N$ increases. IFT further helps show that the proposed algorithm remains close to the fluid dynamics, and matches the lower bound for small $\delta$. For completeness, in Appendix E.2, we also identify the ode paths under fluid dynamics for $\beta$ top-2 algorithms. A great deal of technical analysis goes into showing that the algorithm, observed after sufficiently large amount of samples so that the sample means are close to the true means, is close to the fluid process and they both converge to the same limit.

**Other related literature:** [22] also develop a top-2 type algorithm for a single parameter family of distributions. There algorithm decides between the empirical best and the challenger arm based on directional change in a certain index (related to the LB) when the underlying allocation proportions

are perturbed. It is less directly connected to the first order conditions in the LB problem compared to our algorithm. Empirically, we observe that the our proposed algorithm has lower sample complexity, and is computationally substantially faster (Their algorithm can be sub-optimal. We discuss this in Appendix D.2). [7] consider an algorithm structurally similar to ours. They focus on the BAI fixed budget (FB) setting where the total number of samples are fixed and the aim is to allocate samples to minimise the probability of incorrect selection. Unlike the fixed confidence (FC) setting (the one that we consider), the FB setting requires optimizing the first argument of relative entropy functions that appear in the lower bound. In FC setting, the second argument is optimized ([7] vary the first argument). Fundamentally, this is because FB is concerned with sample allocations that control the probability of the data conducting a large deviations to arrive at an incorrect conclusion, while FC is concerned with controlling sample allocations on high probability paths and gathering enough evidence to rule out the likelihood that the observed data is a result of large deviations. Furthermore, [7] prove weaker a.s. convergence results for associated indexes although not for allocations, and since they focus on FB settings, they do not provide sample complexity bounds or probabilistic false selection guarantees. Our analysis is more nuanced and structurally detailed, and we prove that the sample complexity of the proposed algorithm is asymptotically optimal. [28] study the best-$k$-arm identification problem in the BAI setting with fixed confidence and bring out the structural complexities that arise in lower bound analysis when $k > 1$. For $k = 1$, they develop an asymptotically optimal top-2 algorithm when arm distributions are restricted to be Gaussian. [27] consider related pure exploration problems using Frank-Wolfe algorithm. Their implementation involves solving a linear program at each iteration. [17], [13], [6] provide algorithms that provide finite $\delta$ sample complexity guarantees, however they are order optimal and do not match the constant in the lower bound.

Finally, while fluid analysis is common in many settings including mean field analysis and games (e.g., [4]), stochastic approximation (e.g., [5]) and queuing theory (e.g., [8]), to the best of our knowledge little or no work exists that arrives at it through IFT.

**Roadmap:** In Section 2, we describe the problem and develop lower bound related analysis. The proposed algorithm and our main result, Theorem 3.1, demonstrating algorithm's efficacy are stated in Section 3 where we also develop the relevant IFT framework. Section 4 spells out the fluid dynamics associated with the algorithm. Key steps involved in proving Theorem 3.1 are outlined in Section 5. We describe the numerical experiments in Section 6. Detailed proof of all results are in the appendix.

**Key limitations:** The proposed algorithm extends from SPEF to bounded random variables in a straightforward manner. While we do not provide supporting analysis (this limitation is due to space constraints), our numerical results in Appendix J suggest that our algorithm improves upon existing ones even in this setting. As is standard in the bandit literature, we also assume that samples from arm distributions are independent. Further, another limitation is the assumption of stationarity of the underlying distributions. This may be true when relatively short sampling horizons are involved.

## 2  Problem description and lower bound

**Distributional assumption:** As mentioned earlier, we focus on arm distributions that belong to a known SPEF. Let $\mathcal{S} \subset \mathbb{R}$ denote the open set of possible means of the SPEF under consideration. The details related to SPEF are reviewed in Appendix B.

**Fixed confidence BAI set-up:** Consider an instance with $K$ unknown probability distributions or *arms*, denoted by the mean vector $\boldsymbol{\mu} = (\mu_1, \ldots, \mu_K)$, where each $\mu_i \in \mathcal{S}$ (we refer to each $\mu_i$ interchangeably as a distribution as well as its mean in the SPEF context). As is standard in the BAI framework, we assume that there is a unique arm with the largest mean. Thus, without loss of generality $\mu_1 > \max_{i \geq 2} \mu_i$. One way to handle the case where 2 or more arms are tied for the largest mean is to look for an $\epsilon$-best arm (an arm whose mean is within $\epsilon$ of the best arm). However, that is technically a significantly more demanding problem (see [12]). Assuming uniqueness of the best arm and focusing on the best arm identification allows us to highlight the simple fluid dynamics underlying the proposed algorithm.

**Algorithm:** Given an unknown bandit instance $\mu$, we consider algorithms that sequentially generate samples - if $A_N$ denotes the arm pulled at sample $N$, and $X_N$ denotes the associated reward generated independently from distribution $\mu_{A_N}$, then $A_N$ is chosen sequentially and adaptively as a function of generated $(A_n, X_n : n = 1, 2, \ldots, N-1)$. Further, an algorithm stops at some finite random stopping time $\tau$ and announces the best arm. $\delta-$**correct algorithm** is an algorithm that, given a $\delta > 0$,

stops at time $\tau_\delta > 0$ and outputs a best arm estimate $k_{\tau_\delta}$ such that $\mathbb{P}(\tau_\delta < \infty,\ k_{\tau_\delta} \neq 1) \leq \delta$. That is, it identifies the arm with highest mean with probability at least $1 - \delta$. Our interest is in identifying a $\delta$-correct algorithm that minimizes $\mathbb{E}[\tau_\delta]$. To this end lower bounds on sample complexity of $\delta$-correct algorithms are established using, e.g., the data processing inequality (see, e.g., [18]). We see that

$$\inf_x\ \{\mathbb{E}[N_1]d(\mu_1, x) + \mathbb{E}[N_a]d(\mu_a, x)\} \geq \log(1/(2.4\delta))$$

with $d(\nu, x)$ denoting the Kullback-Leibler divergence between two distributions in $\mathcal{S}$ with means $\nu$ and $x$, and the expectation is under measure $\mathbb{P}_{\boldsymbol{\mu}}$ associated with $\boldsymbol{\mu}$. The infimum above is solved at $x^\star = \frac{\mu_1\mathbb{E}[N_1]+\mu_a\mathbb{E}[N_a]}{\mathbb{E}[N_1]+\mathbb{E}[N_a]}$. With this, we obtain the lower bound $\mathbb{E}[\tau_\delta] \geq T^\star(\boldsymbol{\mu}) \log \frac{1}{2.4\delta}$, where $T^\star(\boldsymbol{\mu})$ is the reciprocal of the optimal value of a max-min problem,

$$(T^\star(\boldsymbol{\mu}))^{-1}\ =\ \max_{\boldsymbol{\omega}=(\omega_a: a\in[K])\in\Sigma_K}\ \min_{a\neq 1}\ (\omega_1 d(\mu_1, x_{1,a}) + \omega_a d(\mu_a, x_{1,a}))\,, \tag{1}$$

where $x_{1,a} = (\omega_1\mu_1 + \omega_a\mu_a)/(\omega_1 + \omega_a)$ and $\Sigma_K$ denotes a simplex in $K$ dimension.

The popular **plug-in track and stop** algorithm involves solving the max-min problem (1) repeatedly for optimal weights with empirical distribution plugged in for $\boldsymbol{\mu}$ above. The algorithm at each stage $t$, generates the next sample from an arm so that the proportion of arms sampled closely match the resulting optimal weights while ensuring an adequate, sub-linear exploration (e.g., each arm gets at least $\sqrt{t}$ samples at each stage $t$).

Propositions 2.1 and 2.2 below are crucial for our analysis. Proposition 2.1 helps in constructing the fluid dynamics in Section 4. Proposition 2.2 provides a characterization of the unique optimal allocation $\boldsymbol{\omega}^\star = (\omega_a^\star : a \in [K])$ which motivates our algorithm's sampling strategy. Before stating the two propositions, we need some notation. Let $B \subseteq [K]/\{1\}$, and $\overline{B} = B \cup \{1\}$. Whenever $\overline{B}^c \neq \emptyset$, let $\boldsymbol{N}_{\overline{B}^c} = (N_a \geq 0 : a \in \overline{B}^c)$ denote an allocation of samples to the arms in $\overline{B}^c$ and we treat this as a constant in the following discussion and also in the statement of the two propositions. We define the quantity $N_{1,1}$ depending on $\boldsymbol{N}_{\overline{B}^c}$ in the following way: **1)** If $\overline{B}^c = \emptyset$ or $\sum_{a\in\overline{B}^c} N_a = 0$, $N_{1,1}$ is zero. **2)** Otherwise, $N_{1,1}$ is the value of $N_1$ at which $\sum_{a\in\overline{B}^c} \frac{d(\mu_1, x_{1,a})}{d(\mu_a, x_{1,a})} = 1$ for the given allocation $\boldsymbol{N}_{\overline{B}^c}$. To see existence of such $N_{1,1}$, observe that whenever $\overline{B}^c \neq \emptyset$ and $\sum_{a\in\overline{B}^c} N_a > 0$, the function $N_1 \to \sum_{a\in\overline{B}^c} \frac{d(\mu_1, x_{1,a})}{d(\mu_a, x_{1,a})}$ is continuous and it monotonically decreases from $\infty$ to $0$ as $N_1$ increases from $0$ to $\infty$. Hence, a unique $N_1$ exists where this function equals 1. We define the quantity $N_{\min} = N_{1,1} + \sum_{a\in\overline{B}^c} N_a$.

**Proposition 2.1.** *For every positive $N$ satisfying $N \geq N_{\min}$, there is a unique set of variables $\boldsymbol{N}_{\overline{B}}(N) = (N_a(N) : a \in \overline{B})$ and $I_B(N)$ satisfying the following conditions*

$$\left.\begin{array}{c} \sum_{a\neq 1} \frac{d(\mu_1, x_{1,a})}{d(\mu_a, x_{1,a})}\ =\ 1, \quad where \quad x_{1,a} = \frac{N_1(N)\cdot\mu_1 + N_a(N)\cdot\mu_a}{N_1(N)+N_a(N)}, \quad \sum_{a\in[K]} N_a(N) = N, \\ and, \quad for\ every\ a\in B, \quad N_1(N)\cdot d(\mu_1, x_{1,a}) + N_a(N)\cdot d(\mu_a, x_{1,a})\ =\ I_B(N). \end{array}\right\} \tag{2}$$

*Furthermore, $\boldsymbol{N}_{\overline{B}}(\cdot)$ and $I_B(\cdot)$ are continuously differentiable w.r.t. $N$ for $N > N_{\min}$.*

**Proposition 2.2.** *Upon taking $B = [K]/\{1\}$ and $N = 1$, $\boldsymbol{N}_{\overline{B}}(1)$, as defined in Proposition 2.1 is same as the unique allocation $\boldsymbol{\omega}^\star$ solving the max-min problem in (1). Further, $I_B(1) = T^\star(\boldsymbol{\mu})^{-1}$. Moreover, for every $N > 0$, if $B = [K]/\{1\}$, the unique solution $\boldsymbol{N}_{\overline{B}}(N) = (N_a(N) : a \in [K])$ satisfies $N_a(N) = N\omega_a^\star$.*

Proposition 2.1 is proved by applying the Implicit function theorem (IFT). Proposition 2.2 is subsumed by [11, Theorem 5], but we prove it using a different set of tools by applying the IFT. See Appendix D for the detailed arguments.

For two vectors $\boldsymbol{\nu} = (\nu_a \in \mathcal{S} : a \in [K])$ and $\boldsymbol{N} = (N_a \in \mathbb{R}_{\geq 0} : a \in [K])$ define the *anchor function*, $g(\boldsymbol{\nu}, \boldsymbol{N}) = \sum_{a\in[K]/\{\hat{j}\}} \frac{d(\nu_{\max}, z_a)}{d(\nu_a, z_a)} - 1$, where $\hat{j} = \arg\max_a \nu_a$, $\nu_{\max} = \max_a \nu_a$, and $z_a = (N_{\hat{j}}\nu_{\max} + N_a\nu_a)/(N_{\hat{j}} + N_a)$ for all $a \neq \hat{j}$.

**Remark 2.1.** It follows from Proposition 2.2 that the anchor function $g(\boldsymbol{\mu}, \boldsymbol{\omega}) = 0$ and all the indexes $\omega_1 d(\mu_1, x_{1,a}) + \omega_a d(\mu_a, x_{1,a})$ equal to each other, uniquely identify the optimal proportion $\boldsymbol{\omega}^\star$ solving the max-min problem (1) (see Appendix D.1 for an easier and more insightful derivation

of these conditions). The algorithm proposed in Section 3 ensures that the empirical version of the anchor function $g(\cdot)$ quickly becomes close to zero and thereafter remains close to zero. Further, the indexes sequentially come close to each other and once they are close, they stay close through the remaining steps of the algorithm.

## 3 Anchored Top-2 (AT2) Algorithm

**Notation:** Some notation is needed to help state the proposed algorithm. For every arm $a \in [K]$ and iteration $N$, $\widetilde{N}_a(N)$ denotes the number of times arm $a$ has been drawn till iteration $N$, and $\widetilde{\boldsymbol{N}}(N) = (\widetilde{N}_a(N) : a \in [K])$. Thus, $N = \sum_a \widetilde{N}_a(N)$. Let $\widetilde{\boldsymbol{\mu}}(N) = (\widetilde{\mu}_a(N) : a \in [K])$ where $\widetilde{\mu}_a(N)$ denotes the sample mean of arm $a$ at time $N$, i.e., $\widetilde{\mu}_a(N) = \sum_{t=1}^{N} \mathbb{I}(A_t = a) \cdot X_t / \widetilde{N}_a(N)$, and $\hat{i}_N = \arg\max_{a \in [K]} \widetilde{\mu}_a(N)$, with an arbitrary tie breaking rule.

For every pair of arms $a, b$, define

$$x_{a,b}(N) = \frac{\widetilde{N}_a(N) \cdot \mu_a + \widetilde{N}_b(N) \cdot \mu_b}{\widetilde{N}_a(N) + \widetilde{N}_b(N)}, \quad \text{and} \quad \widetilde{x}_{a,b}(N) = \frac{\widetilde{N}_a(N) \cdot \widetilde{\mu}_a(N) + \widetilde{N}_b(N) \cdot \widetilde{\mu}_b(N)}{\widetilde{N}_a(N) + \widetilde{N}_b(N)}.$$

Let, $I_{a,b}(N) = \widetilde{N}_a(N) \cdot d(\mu_a, x_{a,b}(N)) + \widetilde{N}_b(N) \cdot d(\mu_b, x_{a,b}(N))$, and $\mathcal{I}_{a,b}(N) = \widetilde{N}_a(N) \cdot d(\widetilde{\mu}_a(N), \widetilde{x}_{a,b}(N)) + \widetilde{N}_b(N) \cdot d(\widetilde{\mu}_b(N), \widetilde{x}_{a,b}(N))$. For $a \neq \hat{i}_N$, we call $I_{\hat{i}_N,a}(N)$, and $\mathcal{I}_{\hat{i}_N,a}(N)$, respectively, *actual index* (or, simply *index*) and *empirical index* of arm $a$ at iteration $N$, and denote them using $I_a(N)$, and $\mathcal{I}_a(N)$. For notational simplicity, we hide the dependency on $N$ whenever it doesn't cause confusion. Note that $\mathcal{I}_a(N)$ is a function of $\widetilde{N}_{\hat{i}_N}(N)$, $\widetilde{N}_a(N)$, $\widetilde{\mu}_{\hat{i}_N}(N)$ and $\widetilde{\mu}_a(N)$.

**Stopping Rule:** As is typical in this literature, in our algorithm below, we follow a generalized log likelihood ratio (GLLR) to decide when to stop the algorithm. It is easy to check that $\min_{a \in [K]/\{\hat{i}\}} \mathcal{I}_a(N)$ denotes the GLLR, that is log of likelihood function (LF) evaluated at maximum likelihood estimator (MLE) divided by the LF evaluated at MLE of parameters restricted to alternate set with a different best arm compared to MLE (see [11, Section 3.2] for a detailed derivation). Define stopping time $\tau_\delta = \inf\{N | \text{for all } a \in [K]/\{\hat{i}\}, \mathcal{I}_a(N) > \beta(N, \delta)\}$, for an appropriate choice of threshold $\beta(N, \delta)$. After stopping at $\tau_\delta$, the algorithm outputs $\hat{i}_{\tau_\delta}$ as the best arm. [19, Eq. 25, Section 5.1] argued that for instances in SPEF, upon choosing $\beta(N, \delta) \approx \log((K-1)/\delta) + 6 \log(\log(N/2) + 1) + 8 \log(1 + 2\log((K-1)/\delta))$, the GLLR based stopping rule is $\delta$-correct for any sampling strategy including the one we propose. In our numerical experiments, we follow [11] and choose a smaller threshold, $\beta(N, \delta) = \log((1 + \log N)/\delta)$.

**Description of the AT2 and Improved AT2 (IAT2) Algorithm:** The AT2 algorithm takes in confidence parameter $\delta > 0$ and exploration parameter $\alpha \in (0, 1)$ as inputs, and executes the following steps at iteration $N$:

1. Let $\mathcal{V}_N \overset{\text{def.}}{=} \{a \in [K] \mid \widetilde{N}_a(N - 1) < N^\alpha\}$ be the set of under-explored arms.

2. If $\mathcal{V}_N \neq \emptyset$, choose $A_N = \arg\min_{a \in [K]} \widetilde{N}_a(N - 1)$, and go to step 5.

3. Else, if $g(\widetilde{\boldsymbol{\mu}}(N - 1), \widetilde{\boldsymbol{N}}(N - 1)) > 0$, choose the empirically best arm *i.e.* $A_N = \hat{i}_{N-1}$, and go to step 5.

4. Else, if $g(\widetilde{\boldsymbol{\mu}}(N - 1), \widetilde{\boldsymbol{N}}(N - 1)) \leq 0$, choose the challenger arm *i.e.* $A_N = \arg\min_{a \in [K]/\{\hat{i}_{N-1}\}} \mathcal{I}_a(N - 1)$ using some arbitrary tie breaking rule, and go to step 5.

5. Sample $X_N$ from $A_N$ and update $\widetilde{\boldsymbol{\mu}}(N)$ and $\widetilde{\boldsymbol{N}}(N)$ using $X_N, A_N$.

6. If $\min_{a \in [K]/\{\hat{i}_N\}} \mathcal{I}_a(N) > \beta(N, \delta)$, terminate and return $\hat{i}_N$.

Inspired from the Improved Transportation Cost Balancing (ITCB) policy for selecting the challenger arm in [16], Improved AT2 (IAT2) algorithm has the same input and follows the same strategy for exploration (step 1 and 2) and choosing the best arm (step 3) as AT2. IAT2 differs from AT2 only by its choice of the challenger arm in step 4, where IAT2 samples from the arm $A_N = \arg\min_{a \in [K]/\{\hat{i}_{N-1}\}} (\mathcal{I}_a(N - 1) + \log \widetilde{N}_a(N - 1))$.

Empirically, we see that typically IAT2 performs better than AT2 with respect to sample complexity. In the appendix, we provide pseudo-codes of AT2 and IAT2 in Algorithms 1 and 2, respectively.

## 3.1 Theoretical guarantees

Proposition 3.1 below shows that the allocations made by AT2 and IAT2 algorithms converge to the optimal allocations $\boldsymbol{\omega}^\star$ w.p. 1 in $\mathbb{P}_{\boldsymbol{\mu}}$. For every $a \in [K]$ we define $\widetilde{\omega}_a(N) = \widetilde{N}_a(N)/N$, and use $\widetilde{\boldsymbol{\omega}}(N) = (\widetilde{\omega}_a(N) : a \in [K])$ to denote the algorithm's proportion at iteration $N$.

**Proposition 3.1** (**Convergence to optimal proportions**). *There exists a random time $T_{stable}$ and a constant $C_1 > 0$ depending on $\boldsymbol{\mu}, \alpha$, and $K$, and independent of $\delta$, such that, $\mathbb{E}_{\boldsymbol{\mu}}[T_{stable}] < \infty$, and for every $N \geq T_{stable}$ and arm $a \in [K]$,*

$$|\widetilde{\omega}_a(N) - \omega_a^\star| \leq C_1 N^{\frac{-3\alpha}{8}}, \quad and \quad |\widetilde{\mu}_a(N) - \mu_a| \leq \epsilon(\boldsymbol{\mu}) N^{\frac{-3\alpha}{8}},$$

*where $\epsilon(\boldsymbol{\mu})$ is a positive constant depending only on $\boldsymbol{\mu}$ and defined in Appendix B.*

**Theorem 3.1** (*Asymptotic optimality of AT2 and IAT2*). *Both AT2 and IAT2 are $\delta$-correct over instances in $\mathcal{S}$, and are asymptotically optimal, i.e., for both the algorithms, the corresponding stopping times satisfy, $\limsup_{\delta \to 0} \frac{E_{\boldsymbol{\mu}}[\tau_\delta]}{\log(1/\delta)} \leq T^\star(\boldsymbol{\mu})$, and $\limsup_{\delta \to 0} \frac{\tau_\delta}{\log(1/\delta)} \leq T^\star(\boldsymbol{\mu})$ a.s. in $\mathbb{P}_{\boldsymbol{\mu}}$. Moreover, we can find a constant $C > 0$ depending on the instance $\boldsymbol{\mu}$ and $\alpha$, such that,*

$$\tau_\delta \leq \max\{ T_{stable}, \ T^\star(\boldsymbol{\mu}) \cdot \log(1/\delta) + C (\log(1/\delta))^{1-3\alpha/8} \} \quad a.s. \ in \quad \mathbb{P}_{\boldsymbol{\mu}}.$$

**Proof idea of Theorem 3.1:** We assume Proposition 3.1 and sketch the argument by which asymptotic optimality follows from it. For $N \geq T_{stable}$, and $a \in [K]$, from Proposition 3.1, $\widetilde{N}_a(N) \approx \omega_a^\star N$ and $\widetilde{\mu}_a(N) \approx \mu_a$. As a result, from Proposition 2.2, after $T_{stable}$, $\mathcal{I}_a(N) \approx N I_{[K]/\{1\}}(1) = N T^\star(\boldsymbol{\mu})^{-1}$ for every $a \neq 1$. Therefore, if $\min_{a \neq 1} \mathcal{I}_a(N)$ crosses $\beta(N, \delta)$ at $N = \tau_\delta$, since $\beta(N, \delta) = \log(1/\delta) + O(\log \log(1/\delta) + \log \log(N))$, we have $\tau_\delta T^\star(\boldsymbol{\mu})^{-1} \approx_{\delta \to 0} \log(1/\delta) + O(\log \log(1/\delta) + \log \log(\tau_\delta))$, which gives $\limsup_{\delta \to 0} \frac{\tau_\delta}{\log(1/\delta)} \leq T^\star(\boldsymbol{\mu})$ a.s. in $\mathbb{P}_{\boldsymbol{\mu}}$. Detailed proof is in Appendix H. $\square$

We outline the key steps of the proof of Proposition 3.1 for AT2 in Section 5, and the detailed proof is in Appendix G.2. Similar arguments hold for IAT2. Considerable technical effort goes in proving this proposition due to the noise in the empirical estimate $\widetilde{\boldsymbol{\mu}}(N)$, resulting in noise in the anchor function and the empirical indexes. However, before presenting the proof sketch, in the next section, we first observe the algorithm's dynamics in the limiting fluid regime where this noise is zero. Several of the important proof steps for the algorithmic allocations rely on insights from the simpler fluid model.

## 4 Fluid dynamics

**Motivation:** The fluid dynamics idealizes our algorithm's evolution through making assumptions at each iteration $N$ that hold for the algorithm in the limit as the number of samples increase to infinity. Unlike the real setting with discrete samples, here we treat samples as a continuous object getting distributed between different arms as the sampling budget (also referred to as 'time') evolves. We denote the no. of samples allocated to an arm $a \in [K]$ at some time $N > 0$ using $N_a(N)$, and define the tuple $\boldsymbol{N}(N) = (N_a(N) : a \in [K])$. Note that $\sum_{a \in [K]} N_a(N) = N$. The rate $N_a'(N)$ at which samples get allocated to arm $a$ at time $N$ depends on a continuous version of the AT2 algorithm, which we refer to as the algorithm's fluid dynamics. We define the index of arm $a \neq 1$ at time $N$ as, $I_a(N) = N_1(N) \cdot d(\mu_1, x_{1,a}(N)) + N_a(N) \cdot d(\mu_a, x_{1,a}(N))$, where $x_{1,a}(N) = \frac{N_1(N)\mu_1 + N_a(N)\mu_a}{N_1(N) + N_a(N)}$. Notice that $I_a(N)$ defined in Section 3 is the index of arm $a$ with respect to the algorithm's allocation $\widetilde{\boldsymbol{N}}(N)$, whereas in our current context, $I_a(N)$ represents the index with respect to the fluid allocations $\boldsymbol{N}(N)$.

**Description of the fluid dynamics:** First we explain the fluid dynamics in words. We formally characterize the fluid allocation $\boldsymbol{N}(N)$ via a system of ODEs in Theorem 4.1. Later in this section, we exploit the obtained ODEs to argue that, after starting the fluid dynamics from some time $N^0 > 0$, the allocations $\boldsymbol{N}(N)$ reach the optimal proportions $\boldsymbol{\omega}^\star = (\omega_a^\star : a \in [K])$ by a time

atmost $\left(\min_{a\in[K]}\omega_a^\star\right)^{-1}\cdot N^0$. In other words, for $N\geq\left(\min_{a\in[K]}\omega_a^\star\right)^{-1}\cdot N^0$, we have $N_a(N)=\omega_a^\star\cdot N$ for every arm $a\in[K]$ irrespective of the initial allocation we had at time $N^0$.

For notational simplicity, we hide the dependency on $N$, whenever it doesn't cause any confusion. Recall the anchor function $g(\cdot)$ introduced in Section 2. We use $g$ to denote $g(\boldsymbol{\mu},\boldsymbol{N}(N))$. For every subset $A\subseteq[K]/\{1\}$, we use $\overline{A}$ to denote $A\cup\{1\}$.

We start the fluid dynamics at time $N^0>0$ with some initial allocation $\boldsymbol{N}^0=(N_a^0\geq 0:a\in[K])$. We assume that the vector of true means $\boldsymbol{\mu}$ is known. The fluid dynamics evolves according to the following steps at a given total allocation $N\geq N^0$: **1)** If $g>0$, then $N_1$ increases with $N$ while other $N_a$'s for $a\neq 1$ are held constant till $g=0$ ($g$ can be seen to be a monotonically decreasing function of $N_1$). **2)** If $g=0$, let $B$ denote the set of minimum indexes. Thus, $I_a(N)$ are equal for all $a\in B$ (the equal value is denoted by $I_B(N)$) and $I_a(N)>I_B(N)$ for all $a\in\overline{B}^c$. Then, as $N$ increases, allocations $N_1$ and $(N_a:a\in B)$ increase such that $g$ remains equal to zero, while the indexes in $B$ remain equal. In Proposition 2.1, we characterize and prove existence of such allocations, which the fluid dynamics will track. Later we observe that, $I_B$ increases atleast at a linear rate and indexes of arms in $\overline{B}^c$ stay bounded from above by a constant. **3)** If $g<0$, let $B$ be the set of minimum index arms and $I_B$ be the index of arms in $B$. In this situation, $(N_a:a\in B)$ increase with $N$ keeping index of the arms in $B$ equal, while $N_1$ and $(N_a:a\in\overline{B}^c)$ are unchanged. With this $g$ also increases, since $g$ is a strictly increasing function of $N_a$ for every $a\in B$. The dynamics in this case are simple and described in Proposition E.1 of Appendix E. **4)** Once, $g=0$, and $B=\{2,\ldots,K\}$, we show that each allocation increases linearly with $N$ such that $N_a'=\omega_a^\star$.

**The fluid ODEs:** In Appendix E, we argue that if the fluid dynamics has $g\neq 0$ at time $N^0$, then $g$ becomes zero within a finite time by following step 1 or step 3 of the description. This is easy to observe when $g>0$ at $N^0$, because $g$ is strictly decreasing in $N_1$, and $g\to-1$ as $N_1\to\infty$. Therefore, following step 1, $g$ becomes 0 at some finite $N$. We now consider the situation where $g=0$ at some $N>N^0$. Setting $B$ to the set of minimum index arms, the algorithm evolves by tracking the allocation $\boldsymbol{N}_{\overline{B}}(N)=(N_a(N):a\in\overline{B})$ defined through the system (2) in Proposition 2.1. By Proposition 2.1, $\boldsymbol{N}_{\overline{B}}(N)$ is continuously differentiable w.r.t. $N$. Applying IFT to (2), we obtain the ODEs via which the allocations and the indexes evolve and present them in Theorem 4.1.

**Some definitions:** Let $f(\boldsymbol{\mu},a,N)=-\frac{\partial}{\partial x}\left(\frac{d(\mu_1,x)}{d(\mu_a,x)}\right)\Big|_{x=x_{1,a}}$. $f(\boldsymbol{\mu},a,\boldsymbol{N})$ is strictly positive because $\frac{d(\mu_1,x)}{d(\mu_a,x)}$ is strictly decreasing with $x$ for $x\in(\mu_a,\mu_1)$.

Let $\Delta_a=\mu_1-\mu_a$, and $h_a(\boldsymbol{\mu},N_1,N_a)=f(\boldsymbol{\mu},a,\boldsymbol{N})\frac{N_1^2\Delta_a}{(N_1+N_a)^2}$. For notational simplicity, we denote $h(\boldsymbol{\mu},N_1,N_a)$ by $h_a$. Further, for each $a$, we denote $d(\mu_1,x_{1,a})$ by $d_{1,a}$ and $d(\mu_a,x_{1,a})$ by $d_{a,a}$. Recall that for given allocations $(N_a:a\in[K])$, $B$ denotes a set such that $N_1d_{1,a}+N_ad_{a,a}=I_B(N)$ for all $a\in B$, and $N_1d_{1,a}+N_ad_{a,a}>I_B(N)$ for all $a\in\overline{B}^c$. Let $h(B)=\sum_{a\in B}h_ad_{a,a}^{-1}$, $h(N)=\sum_{a\in\overline{B}^c}h_aN_a$, and $d_B=\left(\sum_{a\in B}d_{a,a}^{-1}\right)^{-1}$.

**Theorem 4.1** (**Fluid ODEs**). *If at total allocation $N\geq N^0$, we have $g=0$, and $B$ is the set of minimum index arms, i.e., $B=arg\min_{a\in[K]/\{1\}}I_a(N)$, then the following holds true:*

*1. As $N$ increases, and till $I_B(N)$ increases to hit an index in $\overline{B}^c$,*

$$N_1'=\frac{N_1h(B)}{(N_1+\sum_{a\in B}N_a)h(B)+d_B^{-1}h(N)},\quad\text{and}\quad N_b'=\frac{N_bh(B)+d_{b,b}^{-1}h(N)}{(N_1+\sum_{a\in B}N_a)h(B)+d_B^{-1}h(N)},$$
(3)

*for all $b\in B$. It follows that, $I_B'(N)=\frac{I_B(N)h(B)+h(N)}{(N_1+\sum_{a\in B}N_a)h(B)+d_B^{-1}h(N)}$.*

*2. Furthermore, for $a\in\overline{B}^c$, $I_a'(N)=N_1'd_{1a}=\frac{N_1h(B)d_{1a}}{(N_1+\sum_{a\in B}N_a)h(B)+d_B^{-1}h(N)}$.*

*3. There exists a $\beta>0$, independent of $N$ such that $I_B'(N)>\beta$. In addition, for $a\in\overline{B}^c$, $N_a'=0$, $I_a(N)\leq N_a^0d(\mu_a,\mu_1)$, thus the index is bounded from above. Thus, if $\overline{B}^c\neq\emptyset$, $I_B(N)$ eventually catches up with another index in $\overline{B}^c$. In this way, the set $B$ grows into $\{2,\ldots,K\}$.*

**Indexes once they meet must stay together:** In Appendix F.1 we argue via contradiction that in our fluid dynamics, once a set of smallest indexes that are equal, increase and catch up with another index, their union then remains equal and increases together with $N$. This argument is important as it motivates the proof in our algorithm that after sufficient amount of samples, once a sub-optimal arm is pulled, its index stays close to indexes of the other arms that have been pulled.

**Bounding the time to reach optimal proportion:** We define $N^\star$ to be smallest time after $N^0$ at which the fluid dynamics has both $B = \{2, \ldots, K\}$ and $g = 0$. Let $(N_a^\star : a \in [K])$ be the allocation at $N^\star$. We first argue that: *there exists $i \in [K]$ such that $N_i^\star = N_i^0$*. We have argued before that if $g \neq 0$ at $N^0$, then $g$ becomes zero by some finite time, which we call $M$. By definition $M \leq N^\star$. Now if $B \neq \{2, \ldots, K\}$ at $M$ or $M = N^0$, then after time $M$ the fluid dynamics evolve by the ODEs in (3) and $N^\star$ is the time at which $B$ becomes $\{2, \ldots, K\}$, which is finite by statement 3 of Theorem 4.1. In this case $i$ is the last element to be added to $B$. Otherwise if $B = \{2, \ldots, K\}$ at $M$ and $M > N^0$, the only way this can happen is $g < 0$ in $[N^0, M)$. In this case, $i = 1$ and $M = N^\star$. Since $g = 0$ and $B = \{2, \ldots, K\}$ at time $N^\star$, Proposition 2.2 implies $N_a^\star = \omega_a^\star N^\star$ for all $a$. Combining our observations, we have $\omega_i^\star N^\star = N_i^\star = N_i^0 \leq N^0$. Hence $N^\star \leq \frac{N^0}{\omega_i^\star} \leq (\min_{a \in [K]} \omega_a^\star)^{-1} N^0$. Thus $N^\star$ is within a constant times of $N^0$. We bound $T_{stable}$ of Proposition 5.1 using a similar argument.

**Remark 4.1** (**Incorporating the stopping rule into the fluid dynamics**). At stopping time the idealized GLLR (which is the GLLR defined in Section 3 by replacing the estimated means with the true means) just exceeds $\log(1/\delta)$. Since the idealized GLLR grows linearly with the allocated samples, the stopping time increases linearly with $\log(1/\delta)$. Since the time for fluid dynamics to reach stability is independent of $\delta$, for small $\delta$, stability will be reached before the algorithm stops.

**Remark 4.2** ($\beta$-**fluid dynamics**). In Appendix E.2, we construct the fluid dynamics for the $\beta$-EB-TCB algorithm [16] using IFT. We prove that, for every $\beta \in (0, 1)$, the $\beta$-fluid dynamics started at some time $N^0 > 0$ reach the $\beta$-optimal proportion (which is the solution to the max-min problem 1 with the added constraint $\omega_1 = \beta$) by a time which is a constant times $N^0$.

# 5 Convergence of algorithmic allocations to the optimal proportions

We now outline the proof steps for Proposition 3.1. To simplify our analysis, we analyze the AT2 algorithm after the random time $T_0$ defined as,

$$T_0 = \inf \left\{ N' \geq 1 \,\middle|\, \forall a \in [K] \text{ and } N \geq N', \, |\widetilde{\mu}_a(N) - \mu_a| \leq \epsilon(\boldsymbol{\mu}) \cdot N^{-3\alpha/8} \right\},$$

after which the estimates $\widetilde{\boldsymbol{\mu}}(N)$ are converging to $\boldsymbol{\mu}$. Recall that $\alpha \in (0, 1)$ is the exploration parameter, and $\epsilon(\boldsymbol{\mu}) > 0$ is a constant depending only on $\boldsymbol{\mu}$. By the definition of $\epsilon(\boldsymbol{\mu})$ in Appendix B, we have $\widetilde{\mu}_a(N) < \widetilde{\mu}_1(N)$ for all $a \neq 1$ and $N \geq T_0$. As a result, arm 1 becomes the empirically best arm after $T_0$. In Appendix G.3, we use Chernoff's bound to prove that $\mathbb{P}_{\boldsymbol{\mu}}(T_0 = n + 1) = \exp(-\Omega(n^{\alpha/4}))$, which implies $\mathbb{E}_{\boldsymbol{\mu}}[T_0] < \infty$. In the following discussion, all the results mentioned are true for both AT2 and IAT2 algorithms.

Proposition 5.1 shows that the allocations made by the proposed algorithm converges to the first order condition satisfied by the optimal proportion $\boldsymbol{\omega}^\star = (\omega_a^\star : a \in [K])$ at a rate $O(N^{-3\alpha/8})$, where $\alpha$ is the exploration parameter.

**Proposition 5.1.** *There exists a random time $T_{stable} \geq T_0$ satisfying $\mathbb{E}_{\boldsymbol{\mu}}[T_{stable}] < \infty$ and a constant $C_2 > 0$ depending on $\boldsymbol{\mu}, \alpha$ and $K$, and independent of the sample paths, such that, for $N \geq T_{stable}$*

$$|g(\boldsymbol{\mu}, \widetilde{\boldsymbol{\omega}}(N))| = \left| \sum_{a \neq 1} \frac{d(\mu_1, x_{1,a}(N))}{d(\mu_a, x_{1,a}(N))} - 1 \right| \leq C_2 N^{-3\alpha/8}, \tag{4}$$

$$\text{and} \quad \max_{a,b \in [K]/\{1\}} |I_a(N) - I_b(N)| \leq C_2 N^{1-3\alpha/8}. \tag{5}$$

Before outlining the proof of Proposition 5.1, we explain how Proposition 3.1 follows from Proposition 5.1 just using the IFT.

**Proof idea of Proposition 3.1:** If our algorithm follows optimal proportions at time $N$, *i.e.*, $\widetilde{N}_a(N) = \omega_a^\star N$ for all $a \in [K]$, RHS of (4) and (5) becomes zero by Proposition 2.2. Moreover, by Proposition 2.2 $\boldsymbol{\omega}^\star$ uniquely satisfies the conditions: anchor function is zero and all alternative arms have equal index. (4) and (5) imply that, $\widetilde{\omega}(N)$ satisfies these conditions upto a perturbation of $C_2 N^{-3\alpha/8}$ after $T_{stable}$. Using the IFT, we prove that the algorithm's allocation is a Lipschitz continuous function of the perturbation when it is sufficiently small. Hence, by choosing $T_{stable}$ large enough and using Lipschitzness, we get $\max_{a\in[K]} |\widetilde{\omega}_a(N) - \omega_a^\star| = O(N^{-3\alpha/8})$. Closeness of $\widetilde{\boldsymbol{\mu}}(\cdot)$ to $\boldsymbol{\mu}$ follows from the fact that $T_{stable} \geq T_0$. $\qquad\square$

**Proof idea of Proposition 5.1:** We separately outline the proofs of (4) and (5) in Proposition 5.1. In the following discussion, constants hidden in $O(\cdot)$, $\Omega(\cdot)$ and $\Theta(\cdot)$ notations are independent of the sample path after time $T_{stable}$. To simplify our analysis, we choose $T_{stable}$ such that exploration stops after $T_{stable}$, *i.e.*, $\mathcal{V}_N = \emptyset$ for $N \geq T_{stable}$ (see the discussion before Definition G.1 in Appendix G.1.1 for justification).

***Key ideas in the proof of (4):*** We prove (4) via induction. We prove the existence of a constant $D > 0$ such that at every $N \geq T_{stable}$, whenever the actual anchor value $g(\boldsymbol{\mu}, \widehat{\boldsymbol{N}}(N))$ (we denote using $g(N)$) satisfies $|g(N)| > DN^{-3\alpha/8}$, our algorithm pushes $g(\cdot)$ towards zero by $\Theta(1/N)$ in the next iteration through steps 3 and 4. Whereas the interval $[-C_2 N^{-3\alpha/8}, C_2 N^{-3\alpha/8}]$ shrinks by $O(N^{-(1+3\alpha/8)})$ from both ends. Since $N^{-(1+3\alpha/8)} << N^{-1}$, we choose the constant $C_2$ large enough such that $g(\cdot)$ stays in the said interval in iteration $N+1$.

***Key ideas in the proof of (5):*** The following lemma forms a crucial part of the argument for proving closeness of the indexes in the non-fluid setting.

**Lemma 5.1.** *There exists a random time $T_{good} \in [T_0, T_{stable}]$ such that the algorithm picks all the alternative arms in $[K]/\{1\}$ atleast once between the iterations $T_{good}$ and $T_{stable}$. Moreover, for $N \geq T_{good}$, if the algorithm picks some arm $a \in [K]/\{1\}$ at iteration $N$, then it picks arm $a$ again within a next $O(N^{1-3\alpha/8})$ iterations.*

Proof of Lemma 5.1 (in Appendix G.1.2) is technically involved and requires proving several supplementary lemmas. Several of the key steps of this proof borrow insights from the fluid dynamics, and we outline them in Appendix F. Here we assume Lemma 5.1 and sketch the argument by which (5) follows from it for the AT2 algorithm. For any $a, b \neq 1$ and after any $N \geq T_{stable}$, $\mathcal{I}_a(\cdot)$ and $\mathcal{I}_b(\cdot)$ crosses each other before the algorithm picks both $a, b$ atleast once. We can show that, for $j = a, b$ and $N \geq T_{stable}$, $\mathcal{I}_j(N)$ and $I_j(N)$ differ by $O(N^{1-3\alpha/8})$. As a result, when $\mathcal{I}_a(\cdot)$ crosses $\mathcal{I}_b(\cdot)$ at $N+R$, we have $|I_a(N+R) - I_b(N+R)| = O((N+R)^{1-3\alpha/8}) = O(N^{1-3\alpha/8})$ since $R = O(N^{1-3\alpha/8})$. For $j = a, b$, the partial derivatives of $I_j(\cdot)$ w.r.t. $\widetilde{N}_1$ and $\widetilde{N}_j$ are non-negative and bounded from above by $\max\{d(\mu_1, \mu_j), d(\mu_j, \mu_1)\} = O(1)$. As a result, $|I_j(N+R) - I_j(N)| = O(R) = O(N^{1-3\alpha/8})$ for $j = a, b$. Hence, $|I_a(N) - I_b(N)| \leq |I_a(N+R) - I_b(N+R)| + \sum_{j=a,b} |I_j(N+R) - I_j(N)| = O(N^{1-3\alpha/8})$. $\qquad\square$

**Bounding $T_{stable}$:** In Appendix G.2, we choose $T_{good}$ and $T_{stable}$ such that $T_{stable}$ is the time after $T_{good}$ by which the algorithm picks all the sub-optimal arms atleast once. By Proposition 3.1, the algorithm approximately matches $\boldsymbol{\omega}^\star$ after $T_{stable}$. Using an argument similar to the one for bounding time to reach the optimal proportion in the fluid dynamics of Section 4, we can prove that $T_{stable} \lesssim (\omega_{\min}^\star)^{-1} T_{good}$ a.s. in $\mathbb{P}_{\boldsymbol{\mu}}$, where $\omega_{\min}^\star = \min_{a\in[K]} \omega_a^\star$ (Lemma G.4, Appendix G.2.1).

**Role of forced exploration in analysis:** As we observe in the numerical results in Appendix J.4, forced exploration (step 1 of our algorithm) does not increase the observed sample complexity. We emphasize that without the forced exploration, Propositions 5.1 and 3.1 continue to hold if we can show that the proposed algorithm perform sufficient exploration over the instance. That is, after a random time $T$ depending on the instance and satisfying $\mathbb{E}[T] < \infty$, every arm has $\widetilde{N}_a(N) = \Omega(\sqrt{N})$. As a result, upon proving sufficient exploration, asymptotic optimality will follow without the forced exploration step.

In Appendix J.4, we see in the numerical experiments, when there is no forced exploration, AT2's sample complexity blows up over instances where multiple sub-optimal arms have equal mean. On

the other hand, IAT2 performs optimally over the same instances and its sample complexity remains unaffected even when there is no forced exploration. To understand AT2's sample complexity blow up when multiple sub-optimal arms have the same mean, consider the sample paths where the best arm observes unusually small values in the first few samples. As a result, with positive probability, AT2 confuses one of the multiple sub-optimal arms with equal mean as the best arm and stay stuck sampling between those sub-optimal arms forever. However, IAT2 avoids such situation because of the exploration of every sub-optimal arm induced by the extra logarithmic term in the index. Based on these observations, we make the following conjectures: **1)** AT2 performs sufficient exploration over instances where the means of all the sub-optimal arms are distinct, and **2)** IAT2 performs sufficient exploration over all instances including when some of the sub-optimal arms may have equal means.

## 6    Numerical results

In this section, we numerically demonstrate the dynamics followed by the algorithm AT2, and also compare its performance against the $\beta$-EB-TCB algorithm of [16] for different values of $\beta$, and TCB algorithm of [22]. We consider 4 armed Gaussian bandit with unit variance and mean vector $\mu = [10, 8, 7, 6.5]$. We simulate one sample path of the AT2 without stopping rule, and plot the value of normalized indexes of the sub-optimal arms. Figure 1 demonstrates that the normalised indexes once close remain close, and hence, AT2 closely mimics the fluid path. In Figure 2, we plot the sample complexities of the (I)AT2, (I)TCB, and $\beta$-EB-(I)TCB, for different choices of $\beta$, and observe that (I)AT2 outperforms all the other algorithms. Note that we use the same forced exploration rule and stopping rule for all algorithms. In Appendix J, we demonstrate by several examples that both

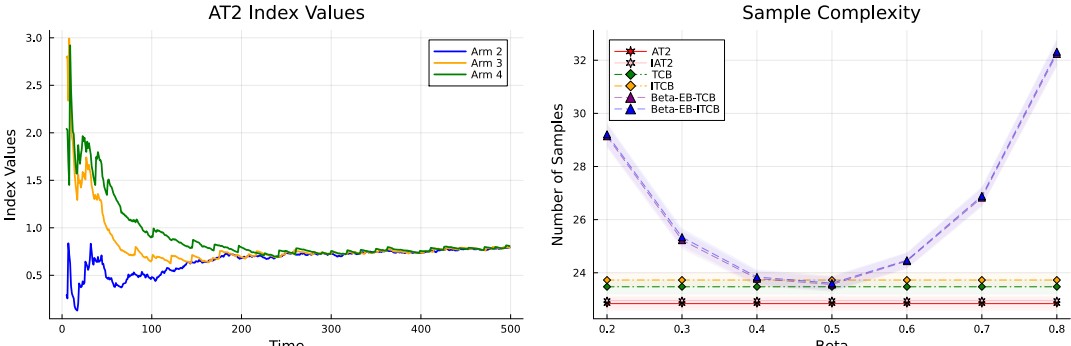

Figure 1: Normalised index on 1 sample path.        Figure 2: Sample complexity comparison.

the AT2 and IAT2 algorithms significantly outperform the $\beta$-EB-TCB and $\beta$-EB-ITCB of [16] when $\beta$ is chosen different from the optimal $\beta$. We also illustrate that the AT2 and IAT2 algorithms have average sample complexity significantly lesser than the TCB and ITCB policies of [22]. In fact, we observe numerically, that (I)TCB doesn't quite satisfy the asymptotic optimality conditions (Figure 4, Appendix J). Next, in Appendix J.4, we study the effect of choice of the forced exploration parameter $\alpha$ on the sample complexities of AT2 and IAT2. Additionally, we conduct simulations for natural extensions of these algorithms to bandits with distributions supported in $[0, 1]$ (Appendix J.5).

## 7    Conclusion

We considered the best-arm identification problem under the popular top-2 framework. In the literature, top-2 framework involves sequentially identifying the empirical best arm and the most-likely challenger arm, and sampling the empirical best with probability $\beta$ and the other with the complimentary probability. However, optimal $\beta$ was not known. [22] recently proposed a deterministic rule for deciding between the empirical best and the challenger arm. In this paper, we have provided a most natural first order optimality condition based rule to help decide between the two. We showed that our associated algorithm is asymptotically optimal, and empirically performs better than [22] both in sample and computational complexity. Our another key contribution was to identify the underlying limiting ordinary differential equation based fluid dynamics that our algorithm tracks. This structure also provides important insights which help prove convergence of the proposed algorithm.

**Acknowledgments:** *We thank Arun Suggala and Karthikeyan Shanmugam from Google Research Bangalore for initial discussions on this project. The second and the third author initiated this work while visiting Google Research in Bangalore.*

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

# Appendix

## A  Outline

Below we provide a brief outline of the topics presented in the appendices.

1. Algorithm (1) and (2) are, respectively, the pseudocodes of the AT2 and IAT2 algorithms introduced in Section 3.

2. Appendix B:  We define the single parameter exponential family (SPEF) of distributions, and prove several inequalities bounding the index function, anchor function, and the derivatives of the anchor function, which are crucial for our analysis.

3. Appendix C:  We introduce a framework using which we apply the implicit function theorem for proving several properties related to the fluid dynamics and the algorithm's allocations.

4. Appendix D:  We prove Propositions 2.1 and 2.2 from Section 2.

5. Appendix E:  We provide the proofs of the results mentioned in Section 4, and also construct the fluid dynamics for the $\beta$-EB-TCB algorithm ([16]) in Appendix E.2.

6. Appendix F:  We provide a heuristic argument to show that if the minimum index meets with the index of some sub-optimal arm $a \neq 1$ following the ODEs in Theorem 4.1, then arm $a$ must be incorporated into the set of minimum index arms. We argue via contradiction to show that, if this is not the case then index of arm $a$ becomes strictly less than the minimum index of the arms, which implies a contradiction. Several of the key steps in the proof of Lemma 5.1 are extensions of the said argument to the non-fluid setting of the algorithm with additional terms because of the noise in the estimates.

7. Appendix G:  We prove Proposition 3.1, 5.1, and provide detailed proofs of all the results mentioned in Section 5.

8. Appendix H:  We provide the detailed proof of Theorem 3.1.

9. Appendix I:  We describe a natural extension of the proposed AT2 and IAT2 algorithms to the class of distributions with support contained in $[0, 1]$. We do not theoretically analyze this algorithm owing to space constraints. However we compare the proposed algorithm with existing algorithms experimentally in Appendix J.5.

10. Appendix J:  We compare the performance of the proposed algorithms against existing algorithms through numerical experiments. We also illustrate how our algorithm follows the fluid dynamics as the no. of samples increase.

---

**Algorithm 1:** Anchored Top-Two (AT2) Algorithm

**Input :** Confidence parameter $\delta > 0$, exploration parameter $\alpha \in (0, 1)$

1 **for** $N \geq 1$ **do**

2 $\quad \mathcal{V}_N \leftarrow \left\{ a \in [K] \mid \widetilde{N}_a(N-1) < N^\alpha \right\}$

3 $\quad$ **if** $\mathcal{V}_N \neq \emptyset$ **then**

4 $\quad\quad A_N \leftarrow \arg\min_{a \in [K]} \widetilde{N}_a(N-1)$        `// Forced exploration`

5 $\quad$ **else if** $g(\widetilde{\boldsymbol{\mu}}(N-1), \widetilde{\boldsymbol{N}}(N-1)) > 0$ **then**

6 $\quad\quad A_N \leftarrow \hat{i}_{N-1}$        `// Choosing leader`

7 $\quad$ **else**

8 $\quad\quad A_N \leftarrow \arg\min_{a \in [K]/\{\hat{i}_{N-1}\}} \mathcal{I}_a(N-1)$        `// Choosing challenger`

9 $\quad$ Sample $X_N$ from $A_N$ and compute $\widetilde{\boldsymbol{\mu}}(N)$ and $\widetilde{\boldsymbol{N}}(N)$

     `/* Generalized Log-Likelihood Ratio (GLLR) Test                    */`

10 $\quad$ **if** $\min_{a \in [K]/\{\hat{i}_N\}} \mathcal{I}_a(N) > \beta(N, \delta)$ **then**

11 $\quad\quad$ **return** $\hat{i}_N$

---

**Algorithm 2:** Improved Anchored Top-Two (IAT2) Algorithm

**Input :** Confidence parameter $\delta > 0$, exploration parameter $\alpha \in (0, 1)$

1   **for** $N \geq 1$ **do**

2     $\mathcal{V}_N \leftarrow \left\{ a \in [K] \mid \widetilde{N}_a(N-1) < N^\alpha \right\}$

3     **if** $\mathcal{V}_N \neq \emptyset$ **then**

4       $A_N \leftarrow \arg\min_{a \in [K]} \widetilde{N}_a(N-1)$             // Forced exploration

5     **else if** $g(\widetilde{\boldsymbol{\mu}}(N-1), \widetilde{\boldsymbol{N}}(N-1)) > 0$ **then**

6       $A_N \leftarrow \hat{i}_{N-1}$                            // Choosing leader

7     **else**

8       $A_N \leftarrow \arg\min_{a \in [K]/\{\hat{i}_{N-1}\}} \mathcal{I}_a(N-1) + \log \widetilde{N}_a(N-1)$    // Choosing challenger

9     Sample $X_N$ from $A_N$ and compute $\widetilde{\boldsymbol{\mu}}(N)$ and $\widetilde{\boldsymbol{N}}(N)$

      /* Generalized Log-Likelihood Ratio (GLLR) Test                 */

10    **if** $\min_{a \in [K]/\{\hat{i}_N\}} \mathcal{I}_a(N) > \beta(N, \delta)$ **then**

11       **return** $\hat{i}_N$

## B   Single parameter exponential family of distributions

We consider single parameter exponential family (SPEF) of distributions of the form

$$d\nu_\theta(x) = \exp(\theta x - b(\theta)) d\rho(x)$$

where $\rho$ is a dominating measure which we assume to be degenerate, $\theta$ lies in the interior of set $\Theta$ defined below (denoted by $\Theta^o$):

$$\Theta = \left\{ \theta \mid \int_{\mathbb{R}} \exp(\theta x) d\rho(x) < \infty \right\},$$

and

$$b(\theta) = \log \left( \int_{\mathbb{R}} \exp(\theta x) d\rho(x) \right)$$

is the log-moment generating function of the measure $\rho(\cdot)$.

For $\theta, \widetilde{\theta} \in \Theta^o$, the KL-divergence between the measures $\nu_\theta$ and $\nu_{\widetilde{\theta}}$ is,

$$KL(\nu_\theta, \nu_{\widetilde{\theta}}) = (\theta - \widetilde{\theta}) b'(\theta) - b(\theta) + b(\widetilde{\theta}).$$

The mean under $\nu_\theta$ is given by $b'(\theta)$. Let $\mathcal{S}$ be the image of the set $\Theta^o$ under the mapping $b'(\cdot)$. Note that $\mathcal{S}$ is an open interval. Also, since $b''(\cdot) > 0$ in $\Theta^o$, $b'(\cdot)$ is strictly increasing in $\Theta^o$, and is a bijection between $\Theta^o$ and $\mathcal{S}$. This implies we can parameterize the distributions in the SPEF using their means as well.

Let $\theta_\mu$ be the unique $\theta$ satisfying $b'(\theta) = \mu$ for some $\mu \in \mathcal{S}$. Clearly, $\theta_\mu$ is a strictly increasing function of $\mu$. This follows since $b'(\cdot)$ is strictly increasing in $\Theta^o$. Additionally, all the higher derivatives of $b(\cdot)$ exist in the set $\Theta^o$ (see Exercise 2.2.24 in [9]).

For $\mu, \widetilde{\mu} \in \mathcal{S}$ we define $d(\mu, \widetilde{\mu})$ as,

$$d(\mu, \widetilde{\mu}) = KL(\nu_{\theta_\mu}, \nu_{\theta_{\widetilde{\mu}}}) = (\theta_\mu - \theta_{\widetilde{\mu}})\mu - b(\theta_\mu) + b(\theta_{\widetilde{\mu}}).$$

We define $\mu_{\inf} \in \mathbb{R} \cup \{-\infty\}$ and $\mu_{\sup} \in \mathbb{R} \cup \{+\infty\}$, respectively, to be the infimum and supremum of the interval $\mathcal{S}$. Then, $\mathcal{S} = (\mu_{\inf}, \mu_{\sup})$.

**Definition B.1.** For $\boldsymbol{\mu} = (\mu_1, \mu_2, \ldots, \mu_K) \in \mathcal{S}^K$, define $r_{\min}(\boldsymbol{\mu}) = \min_{i \in [K]} \{\min\{\mu_i - \mu_{\inf}, \mu_{\sup} - \mu_i\}\}$.

Since $\mathcal{S}$ is an open interval, $r_{\min}(\boldsymbol{\mu})$ is positive for every $\boldsymbol{\mu} \in \mathcal{S}^K$, and can be $\infty$ if both $\mu_{\inf}$ and $\mu_{\sup}$ are $\infty$.

**Partial derivatives of $d(\mu, \widetilde{\mu})$:** For every pair $\mu, \widetilde{\mu} \in \mathcal{S}$, the partial derivatives of $d(\mu, \widetilde{\mu})$ with respect to the first argument is

$$d_1(\mu, \widetilde{\mu}) \stackrel{\text{def.}}{=} \frac{\partial}{\partial \mu} d(\mu, \widetilde{\mu}) = \theta_\mu - \theta_{\widetilde{\mu}},$$

and that with respect to the second argument is,

$$d_2(\mu, \widetilde{\mu}) \stackrel{\text{def.}}{=} \frac{\partial}{\partial \widetilde{\mu}} d(\mu, \widetilde{\mu}) = \frac{\widetilde{\mu} - \mu}{b''(\theta_{\widetilde{\mu}})}.$$

**Enveloping the KL-divergence:** In the following discussion, we try to bound the KL-divergence $d(\mu, \widetilde{\mu})$ from both sides using the squared distance $|\mu - \widetilde{\mu}|^2$ after imposing some restrictions on the choice of $\mu, \widetilde{\mu} \in \mathcal{S}$. For an instance $\boldsymbol{\mu} \in \mathcal{S}^K$, we define the constants $\Delta_{\min}(\boldsymbol{\mu})$ and $\epsilon(\boldsymbol{\mu})$ as

$$\Delta_{\min}(\boldsymbol{\mu}) = \min_{i \in [K]/\{1\}} (\mu_1 - \mu_i) \quad \text{and} \quad \epsilon(\boldsymbol{\mu}) = \min\left\{\frac{\Delta_{\min}(\boldsymbol{\mu})}{4}, \frac{r_{\min}(\boldsymbol{\mu})}{2}\right\}.$$

We further define

$$\mathcal{H}(\boldsymbol{\mu}) = \bigcup_{i \in [K]/\{1\}} [\mu_i - \epsilon(\boldsymbol{\mu}), \mu_1 + \epsilon(\boldsymbol{\mu})],$$

$$\sigma_{\max}(\boldsymbol{\mu}) = \max_{\mu \in \mathcal{H}(\boldsymbol{\mu})} b''(\theta_\mu) \quad \text{and} \quad \sigma_{\min}(\boldsymbol{\mu}) = \min_{\mu \in \mathcal{H}(\boldsymbol{\mu})} b''(\theta_\mu).$$

Since $\mathcal{H}(\boldsymbol{\mu}) \subset \mathcal{S}$, all distributions with mean in $\mathcal{H}(\boldsymbol{\mu})$ have positive variance. Note that $b''(\theta_\mu)$ represents the variance of the distribution with mean $\mu$. As a result, since $\mathcal{H}(\boldsymbol{\mu})$ is a compact set, both $\sigma_{\max}(\boldsymbol{\mu})$ and $\sigma_{\min}(\boldsymbol{\mu})$ are positive constants.

Hence, $b(\cdot)$ is $\sigma_{\min}(\boldsymbol{\mu})$-strongly convex and $b'(\cdot)$ is $\sigma_{\max}(\boldsymbol{\mu})$-Lipschitz on the set $(b')^{-1}(\mathcal{H}(\boldsymbol{\mu}))$. Thus, using [23, Theorems 2.1.5 and 2.1.10], for every $\theta_1, \theta_2 \in (b')^{-1}(\mathcal{H}(\boldsymbol{\mu}))$, we have

$$\frac{|b'(\theta_1) - b'(\theta_2)|^2}{2\sigma_{\max}(\boldsymbol{\mu})} \leq b(\theta_2) - b(\theta_1) - b'(\theta_1) \cdot (\theta_2 - \theta_1) = d(\nu_{\theta_1}, \nu_{\theta_2}) \leq \frac{|b'(\theta_1) - b'(\theta_2)|^2}{2\sigma_{\min}(\boldsymbol{\mu})},$$

and hence, for $\mu, \widetilde{\mu} \in \mathcal{H}(\boldsymbol{\mu})$,

$$\frac{|\mu - \widetilde{\mu}|^2}{2\sigma_{\max}(\boldsymbol{\mu})} \leq d(\mu, \widetilde{\mu}) \leq \frac{|\mu - \widetilde{\mu}|^2}{2\sigma_{\min}(\boldsymbol{\mu})}. \tag{6}$$

**Bounding the partial derivatives:** We now introduce bounds on the partial derivatives $d_1$ and $d_2$ introduced earlier. Consider $\mu, x, \widetilde{\mu} \in \mathcal{H}(\boldsymbol{\mu})$ such that $\mu > \widetilde{\mu}$, and $x \in [\widetilde{\mu}, \mu]$. Recall

$$d_1(\mu, x) = \theta_\mu - \theta_x \quad \text{and} \quad d_1(\widetilde{\mu}, x) = -(\theta_x - \theta_{\widetilde{\mu}}).$$

Since $\theta_{(\cdot)} = (b')^{-1}(\cdot)$, we have,

$$\frac{\mu - x}{\sigma_{\max}(\boldsymbol{\mu})} \leq d_1(\mu, x) \leq \frac{\mu - x}{\sigma_{\min}(\boldsymbol{\mu})}, \quad \text{and} \quad -\frac{x - \widetilde{\mu}}{\sigma_{\max}(\boldsymbol{\mu})} \geq d_1(\widetilde{\mu}, x) \geq -\frac{x - \widetilde{\mu}}{\sigma_{\min}(\boldsymbol{\mu})}. \tag{7}$$

Similarly, since,

$$d_2(\mu, x) = -\frac{\mu - x}{b''(\theta_x)} \quad \text{and} \quad d_2(\widetilde{\mu}, x) = \frac{x - \widetilde{\mu}}{b''(\theta_x)},$$

we have,

$$-\frac{\mu - x}{\sigma_{\max}(\boldsymbol{\mu})} \geq d_2(\mu, x) \geq -\frac{\mu - x}{\sigma_{\min}(\boldsymbol{\mu})}, \quad \text{and} \quad \frac{x - \widetilde{\mu}}{\sigma_{\min}(\boldsymbol{\mu})} \leq d_2(\widetilde{\mu}, x) \leq \frac{x - \widetilde{\mu}}{\sigma_{\min}(\boldsymbol{\mu})}. \tag{8}$$

## B.1 Enveloping the anchor and index functions under noisy estimates of the rewards

For every arm $a \in [K]$, let $\widetilde{\mu}_a$ be an estimate of $\mu_a$ satisfying $|\widetilde{\mu}_a - \mu_a| \leq \epsilon(\boldsymbol{\mu})$ and $\widetilde{\boldsymbol{\mu}} = (\widetilde{\mu}_a)_{a \in [K]}$. Since $\epsilon(\boldsymbol{\mu}) \leq \frac{\Delta_{\min}(\boldsymbol{\mu})}{4}$, the empirically best arm with respect to the estimates $\widetilde{\boldsymbol{\mu}}$ is the first arm. Also let $N_a$ be the no. of times arm $a$ has been pulled and $\boldsymbol{N} = (N_a)_{a \in [K]}$.

**Enveloping the anchor function:** As we introduced in Section 2, the anchor function is,

$$g(\widetilde{\boldsymbol{\mu}}, \boldsymbol{N}) = \sum_{a \neq 1} \frac{d(\widetilde{\mu}_1, \widetilde{x}_{1,a})}{d(\widetilde{\mu}_a, \widetilde{x}_{1,a})} - 1,$$

where $\widetilde{x}_{1,a} = \frac{N_1 \widetilde{\mu}_1 + N_a \widetilde{\mu}_a}{N_1 + N_a}$. Note that $\widetilde{\mu}_a, \widetilde{x}_{1,a} \in \mathcal{H}(\boldsymbol{\mu})$ for every $a \in [K]/\{1\}$. Therefore, using (6), we have,

$$\frac{\sigma_{\min}(\boldsymbol{\mu})}{\sigma_{\max}(\boldsymbol{\mu})} \sum_{a \neq 1} \frac{(\widetilde{\mu}_1 - \widetilde{x}_{1,a})^2}{(\widetilde{x}_{1,a} - \widetilde{\mu}_a)^2} - 1 \ \leq \ g(\widetilde{\boldsymbol{\mu}}, \boldsymbol{N}) \ \leq \ \frac{\sigma_{\max}(\boldsymbol{\mu})}{\sigma_{\min}(\boldsymbol{\mu})} \sum_{a \neq 1} \frac{(\widetilde{\mu}_1 - \widetilde{x}_{1,a})^2}{(\widetilde{x}_{1,a} - \widetilde{\mu}_a)^2} - 1,$$

Putting $\widetilde{x}_{1,a} = \frac{N_1 \widetilde{\mu}_1 + N_a \widetilde{\mu}_a}{N_1 + N_a}$, we obtain,

$$\frac{\sigma_{\min}(\boldsymbol{\mu})}{\sigma_{\max}(\boldsymbol{\mu})} \sum_{a \neq 1} \frac{N_a^2}{N_1^2} - 1 \ \leq \ g(\widetilde{\boldsymbol{\mu}}, \boldsymbol{N}) \ \leq \ \frac{\sigma_{\max}(\boldsymbol{\mu})}{\sigma_{\min}(\boldsymbol{\mu})} \sum_{a \neq 1} \frac{N_a^2}{N_1^2} - 1, \tag{9}$$

Now $\widetilde{\mu}_1 - \widetilde{x}_{1,a} = \frac{N_a}{N_1 + N_a}(\widetilde{\mu}_1 - \widetilde{\mu}_a)$ and $\widetilde{x}_{1,a} - \widetilde{\mu}_a = \frac{N_1}{N_1 + N_a}(\widetilde{\mu}_1 - \widetilde{\mu}_a)$. Using these, we can say,

$$g(\widetilde{\boldsymbol{\mu}}, \boldsymbol{N}) \ = \ \Theta\left(\sum_{a \neq 1} \frac{N_a^2}{N_1^2}\right) - 1, \tag{10}$$

whenever $|\widetilde{\mu}_a - \mu_a| \leq \epsilon(\boldsymbol{\mu})$ for all $a \in [K]$. The constants hidden in the $\Theta(\cdot)$ depends only on $\boldsymbol{\mu}$ and obviously the choice of the SPEF family.

**Enveloping the index:** Following the definition of empirical index $\mathcal{I}_a(\cdot)$ in Section 3, we define the index of any alternative arm $a \in [K]/\{1\}$ with respect to the estimates $\widetilde{\boldsymbol{\mu}} = (\widetilde{\mu}_a : a \in [K])$ and allocation $\boldsymbol{N} = (N_a : a \in [K])$ is,

$$\mathcal{W}_a(\widetilde{\boldsymbol{\mu}}, \boldsymbol{N}) \ = \ N_1 d(\widetilde{\mu}_1, \widetilde{x}_{1,a}) + N_a d(\widetilde{\mu}_a, \widetilde{x}_{1,a}).$$

Observe that, the empirical index $\mathcal{I}_a(N)$ and index $I_a(N)$ introduced in Section 3 are, respectively, equivalent to the quantities $\mathcal{W}_a(\widetilde{\boldsymbol{\mu}}(N), \widetilde{\boldsymbol{N}}(N))$, and $\mathcal{W}_a(\boldsymbol{\mu}, \widetilde{\boldsymbol{N}}(N))$, where $\widetilde{\boldsymbol{\mu}}(N)$ is the empirical estimate and $\widetilde{\boldsymbol{N}}(N)$ is the algorithm's allocation at iteration $N$.

Using (6) we have,

$$\frac{1}{2\sigma_{\min}(\boldsymbol{\mu})}(N_1(\widetilde{\mu}_1 - \widetilde{x}_{1,a})^2 + N_a(\widetilde{x}_{1,a} - \widetilde{\mu}_a)^2) \ \leq \ \mathcal{W}_a(\widetilde{\boldsymbol{\mu}}, \boldsymbol{N}) \ \leq \ \frac{1}{2\sigma_{\min}(\boldsymbol{\mu})}(N_1(\widetilde{\mu}_1 - \widetilde{x}_{1,a})^2$$
$$+ N_a(\widetilde{x}_{1,a} - \widetilde{\mu}_a)^2).$$

Putting $\widetilde{x}_{1,a} = \frac{N_1 \widetilde{\mu}_1 + N_a \widetilde{\mu}_a}{N_1 + N_a}$, we get,

$$\frac{(\widetilde{\mu}_1 - \widetilde{\mu}_a)^2}{2\sigma_{\max}(\boldsymbol{\mu})} \frac{N_1 N_a}{N_1 + N_a} \ \leq \ \mathcal{W}_a(\widetilde{\boldsymbol{\mu}}, \boldsymbol{N}) \ \leq \ \frac{(\widetilde{\mu}_1 - \widetilde{\mu}_a)^2}{2\sigma_{\min}(\boldsymbol{\mu})} \frac{N_1 N_a}{N_1 + N_a}, \text{ which implies,}$$

$$\frac{(\mu_1 - \mu_a - 2\epsilon(\boldsymbol{\mu}))^2}{2\sigma_{\max}(\boldsymbol{\mu})} \frac{N_1 N_a}{N_1 + N_a} \ \leq \ \mathcal{W}_a(\widetilde{\boldsymbol{\mu}}, \boldsymbol{N}) \ \leq \ \frac{(\mu_1 - \mu_a + 2\epsilon(\boldsymbol{\mu}))^2}{2\sigma_{\min}(\boldsymbol{\mu})} \frac{N_1 N_a}{N_1 + N_a}.$$

We define $\Delta_{\max}(\boldsymbol{\mu}) = \max_{a \in [K]/\{1\}} \Delta_a$. Since $\epsilon(\boldsymbol{\mu}) \leq \frac{1}{4}\Delta_{\min}(\boldsymbol{\mu})$, we have,

$$\frac{\Delta_{\min}(\boldsymbol{\mu})^2}{8\sigma_{\max}(\boldsymbol{\mu})} \frac{N_1 N_a}{N_1 + N_a} \ \leq \ \mathcal{W}_a(\widetilde{\boldsymbol{\mu}}, \boldsymbol{N}) \ \leq \ \frac{(\Delta_{\max}(\boldsymbol{\mu}) + 2\epsilon(\boldsymbol{\mu}))^2}{2\sigma_{\min}(\boldsymbol{\mu})} \frac{N_1 N_a}{N_1 + N_a}. \tag{11}$$

Using the same notation, as we used in (10), we have,

$$\mathcal{W}_a(\widetilde{\boldsymbol{\mu}}, \boldsymbol{N}) = \Theta\left(\frac{N_1 N_a}{N_1 + N_a}\right) \tag{12}$$

for every arm $a \in [K]/\{1\}$.

**Enveloping the partial derivatives of anchor function with respect to $\boldsymbol{N}$:** While analyzing the AT2 and IAT2 algorithms, we need to show that the anchor function converges to zero at a uniform rate as no. of iteration goes to infinity. During this step, we need to bound the partial derivatives of the anchor function $g(\boldsymbol{\mu}, \boldsymbol{N})$ with respect to $\boldsymbol{N}$. Below we evaluate the partial derivatives of $g(\boldsymbol{\mu}, \boldsymbol{N})$ with respect to $N_a$ for different arms $a \in [K]$.

$$\frac{\partial g}{\partial N_1}(\boldsymbol{\mu}, \boldsymbol{N}) = -\sum_{a \neq 1} f(\boldsymbol{\mu}, a, \boldsymbol{N}) \frac{N_a \Delta_a}{(N_1 + N_a)^2}, \quad \text{and}$$

$$\forall a \neq 1, \quad \frac{\partial g}{\partial N_a}(\boldsymbol{\mu}, \boldsymbol{N}) = f(\boldsymbol{\mu}, a, \boldsymbol{N}) \frac{N_1 \Delta_a}{(N_1 + N_a)^2}, \tag{13}$$

where $\Delta_a = \mu_1 - \mu_a$ and

$$f(\boldsymbol{\mu}, a, \boldsymbol{N}) = -\frac{\partial}{\partial x}\left(\frac{d(\mu_1, x)}{d(\mu_a, x)}\right)\Big|_{x = x_{1,a}} = -\frac{d_2(\mu_1, x_{1,a})}{d(\mu_a, x_{1,a})} + \frac{d(\mu_1, x_{1,a}) d_2(\mu_a, x_{1,a})}{(d(\mu_a, x_{1,a}))^2}.$$

Using (8), and (6), we have,

$$-d_2(\mu_1, x_{1,a}) = \Theta(\mu_1 - x_{1,a}), \quad d_2(\mu_a, x_{1,a}) = \Theta(x_{1,a} - \mu_a),$$

$$d(\mu_1, x_{1,a}) = \Theta((\mu_1 - x_{1,a})^2), \quad \text{and} \quad d(\mu_a, x_{1,a}) = \Theta((x_{1,a} - \mu_a)^2),$$

where the constants hidden in $\Theta(\cdot)$ depend only on $\boldsymbol{\mu}$ and are independent of the sample path. Therefore,

$$f(\boldsymbol{\mu}, a, \boldsymbol{N}) = \Theta\left(\frac{\mu_1 - x_{1,a}}{(x_{1,a} - \mu_a)^2} + \frac{(\mu_1 - x_{1,a})^2}{(x_{1,a} - \mu_a)^3}\right)$$

$$= \Theta\left(\frac{N_a}{N_1}\left(1 + \frac{N_a}{N_1}\right)^2\right). \tag{14}$$

As a consequence, we have,

$$\frac{\partial g}{\partial N_1}(\boldsymbol{\mu}, \boldsymbol{N}) = -\Theta\left(\sum_{a \in [K]/\{1\}} \frac{N_a^2}{N_1^3}\right), \quad \text{and}$$

$$\forall a \neq 1, \quad \frac{\partial g}{\partial N_a}(\boldsymbol{\mu}, \boldsymbol{N}) = \Theta\left(\frac{N_a}{N_1^2}\right). \tag{15}$$

## C   Framework for applying the Implicit function theorem (IFT)

In this appendix we explain a general framework using which we later apply the Implicit function theorem for the following purposes:

1. Constructing the fluid dynamics for the AT2 and $\beta$-EB-TCB algorithms in Appendix E.

2. Proving convergence of the algorithm's allocations to the optimal proportions in Appendix G.2.2.

We introduce the variables: $\boldsymbol{N} = (N_a \in \mathbb{R}_{\geq 0} : a \in [K])$, $I \in \mathbb{R}$, $\boldsymbol{\eta} = (\eta_a \in \mathbb{R} : a \in [K])$, and $N \geq 0$. After fixing some instance $\boldsymbol{\mu} \in \mathcal{S}^K$ ($\mathcal{S}$ is defined in Appendix B), we define the

following functions:

$$\Phi_1(\boldsymbol{N}, \boldsymbol{\eta}) = \sum_{a \in [K]/\{1\}} \frac{d(\mu_1, x_{1,a}(N_1, N_a))}{d(\mu_a, x_{1,a}(N_1, N_a))} - 1 - \eta_1,$$

$$\text{for } a \in [K]/\{1\}, \ \Phi_a(\boldsymbol{N}, I, \boldsymbol{\eta}) = N_1 d(\mu_1, x_{1,a}(N_1, N_a)) + N_a d(\mu_a, x_{1,a}(N_1, N_a)) - I - \eta_a,$$

$$\text{and} \quad \Phi_{K+1}(\boldsymbol{N}, N) = \sum_{a \in [K]} N_a - N,$$

where

$$x_{1,a}(N_1, N_a) = \frac{N_1 \mu_1 + N_a \mu_a}{N_1 + N_a}.$$

For every non-empty subset $B \subseteq [K]/\{1\}$, we define the vector valued functions,

$$\boldsymbol{\Phi}_B(\boldsymbol{N}, I, \boldsymbol{\eta}, N) = [\ \Phi_1(\boldsymbol{N}, \boldsymbol{\eta}), \quad (\Phi_a(\boldsymbol{N}, I, \boldsymbol{\eta}))_{a \in B}, \quad \Phi_{K+1}(\boldsymbol{N}, N)\ ], \quad \text{and}$$

$$\widetilde{\boldsymbol{\Phi}}_B(\boldsymbol{N}, I, \boldsymbol{\eta}) = [\ \Phi_1(\boldsymbol{N}, \boldsymbol{\eta}), \quad (\Phi_a(\boldsymbol{N}, I, \boldsymbol{\eta}))_{a \in B}\ ].$$

We denote $\boldsymbol{\Phi}_{[K]/\{1\}}(\cdot)$ just using $\boldsymbol{\Phi}(\cdot)$.

In the definitions of $\boldsymbol{\Phi}_B(\cdot)$ and $\widetilde{\boldsymbol{\Phi}}_B(\cdot)$, without loss of generality, we assume that, the functions $\Phi_a(\cdot)$ in the tuple $(\Phi_a(\cdot))_{a \in B}$ are enumerated in the increasing order of $a \in B$, *i.e.*, if we have $B = \{a_1, a_2, \ldots, a_{|B|}\}$ with $1 < a_1 < a_2 < \ldots < a_{|B|}$, then,

$$\boldsymbol{\Phi}_B = \begin{bmatrix} \Phi_1, & \Phi_{a_1}, & , \Phi_{a_2}, & \Phi_{a_3}, & \ldots & \Phi_{a_{|B|}}, & \Phi_{K+1} \end{bmatrix}, \quad \text{and}$$

$$\widetilde{\boldsymbol{\Phi}}_B = \begin{bmatrix} \Phi_1, & \Phi_{a_1}, & , \Phi_{a_2}, & \Phi_{a_3}, & \ldots & \Phi_{a_{|B|}} \end{bmatrix}.$$

Before stating the main result of this section in Lemma C.1, we define some notation that are essential for the lemma statement. For any $A \subseteq [K]$, we use the notation $\boldsymbol{N}_A$ to denote the tuple of variables $(N_a : a \in A)$. For some vector valued function $\boldsymbol{G}$ depending on $\boldsymbol{N}$, denote the Jacobian of $G(\cdot)$ with respect to the tuple of variables $\boldsymbol{N}_A$ using $\frac{\partial \boldsymbol{G}}{\partial \boldsymbol{N}_A}$.

For any non-empty $B \subseteq [K]/\{1\}$, we define $\overline{B} = B \cup \{1\}$. For every $k \geq 1$, $\boldsymbol{0}_k$ and $\boldsymbol{1}_k$, respectively, refers to a $k$-dimensional vector with entries 0 and 1. We define $\mathcal{Z}_B$ to be the set of tuples $(\boldsymbol{N}, I, \boldsymbol{0}_K, N)$ with $\boldsymbol{N} \in \mathbb{R}^K_{\geq 0}$, $I \in \mathbb{R}$, and $N \in \mathbb{R}_{>0}$, which satisfy

$$\boldsymbol{\Phi}_B(\boldsymbol{N}, I, \boldsymbol{0}_K, N) = \boldsymbol{0}_{|B|+2}.$$

**Lemma C.1 (Invertibility of the Jacobian).** *For all non-empty subset $B \subseteq [K]/\{1\}$, the following statements hold true at every tuple $(\boldsymbol{N}, I, \boldsymbol{0}_K, N)$ in the set $\mathcal{Z}_B$,*

1. *The Jacobian $\frac{\partial \widetilde{\boldsymbol{\Phi}}_B}{\partial \boldsymbol{N}_{\overline{B}}}$ is invertible at $(\boldsymbol{N}, I, \boldsymbol{0}_K)$.*

2. *We have*

$$\boldsymbol{1}^T_{|B|+1} \left( \frac{\partial \widetilde{\boldsymbol{\Phi}}_B}{\partial \boldsymbol{N}_{\overline{B}}} \right)^{-1} \frac{\partial \widetilde{\boldsymbol{\Phi}}_B}{\partial I} \leq -\sum_{a \in B} \frac{1}{d(\mu_a, \mu_1)},$$

*at $(\boldsymbol{N}, I, \boldsymbol{0}_K)$.*

3. *The Jacobian $\frac{\partial \boldsymbol{\Phi}_B}{\partial (\boldsymbol{N}_{\overline{B}}, I)}$ defined as,*

$$\frac{\partial \boldsymbol{\Phi}_B}{\partial (\boldsymbol{N}_{\overline{B}}, I)} = \begin{bmatrix} \frac{\partial \widetilde{\boldsymbol{\Phi}}_B}{\partial \boldsymbol{N}_{\overline{B}}} & \frac{\partial \widetilde{\boldsymbol{\Phi}}_B}{\partial I} \\ \hline \frac{\partial \Phi_{K+1}}{\partial \boldsymbol{N}_{\overline{B}}} & \frac{\partial \Phi_{K+1}}{\partial I} \end{bmatrix}$$

*is invertible at $(\boldsymbol{N}, I, \boldsymbol{0}_K, N)$.*

*Proof.* **Statement 1:** The Jacobian $\frac{\partial \widetilde{\boldsymbol{\Phi}}_B}{\partial \boldsymbol{N}_{\overline{B}}}$ is equivalent to,

$$\frac{\partial \widetilde{\boldsymbol{\Phi}}_B}{\partial \boldsymbol{N}_{\overline{B}}} = \begin{bmatrix} \frac{\partial \Phi_1}{\partial N_1} & \frac{\partial \Phi_1}{\partial N_{a_1}} & \frac{\partial \Phi_1}{\partial N_{a_2}} & \cdots & \frac{\partial \Phi_1}{\partial N_{a_{|B|}}} \\ \frac{\partial \Phi_{a_1}}{\partial N_1} & \frac{\partial \Phi_{a_1}}{\partial N_{a_1}} & \frac{\partial \Phi_{a_1}}{\partial N_{a_2}} & \cdots & \frac{\partial \Phi_{a_1}}{\partial N_{a_{|B|}}} \\ \vdots & \vdots & \vdots & \cdots & \vdots \\ \frac{\partial \Phi_{a_{|B|}}}{\partial N_1} & \frac{\partial \Phi_{a_{|B|}}}{\partial N_{a_1}} & \frac{\partial \Phi_{a_{|B|}}}{\partial N_{a_2}} & \cdots & \frac{\partial \Phi_{a_{|B|}}}{\partial N_{a_{|B|}}} \end{bmatrix}. \tag{16}$$

Now we observe the following properties about the sign of the entries of the above Jacobian matrix,

- We have $N > 0$ and $\Phi_1(\boldsymbol{N}, \boldsymbol{0}_K) = 0$. This implies $N_1 > 0$ and $\max_{a \in [K]/\{1\}} N_a > 0$. As a result, using (15), we have $\frac{\partial \Phi_1}{\partial N_1} < 0$ and $\frac{\partial \Phi_1}{\partial N_{a_i}} \geq 0$ for every $i \in \{1, 2, \ldots, |B|\}$.

- For $i \in \{2, \ldots, |B| + 1\}$, in the $i$-th row, the only non-zero entries can be the first and the $i$-th entry. The first entry is $\frac{\partial \Phi_{a_i}}{\partial N_1} = d(\mu_1, x_{1,a_i}(N_1, N_{a_i})) \geq 0$. The $i$-th entry is $\frac{\partial \Phi_{a_i}}{\partial N_{a_i}} = d(\mu_{a_i}, x_{1,a_i}(N_1, N_{a_i}))$. Since we have $N_1 > 0$, using (6), we have $d(\mu_{a_i}, x_{1,a_i}(N_1, N_{a_i})) > 0$, making the $i$-th entry positive.

Therefore, considering only the sign of the elements, the matrix in (16) is of the form,

$$\begin{bmatrix} -- & + & + & + & \cdots & + \\ + & ++ & 0 & 0 & \cdots & 0 \\ + & 0 & ++ & 0 & \cdots & 0 \\ + & 0 & 0 & ++ & \cdots & 0 \\ \vdots & \vdots & \vdots & \vdots & \cdots & \vdots \\ + & 0 & 0 & 0 & \cdots & ++ \end{bmatrix}, \tag{17}$$

where the symbols $++$, $--$ and $+$ implies that the corresponding entries are positive, negative and non-negative.

We now argue that a matrix of the above structure has a rank $|B| + 1$. To see that, by subtracting some appropriate constant times the $i$-th column from the first column, we can make the entries in position $i \in \{2, 3, \ldots, |B| + 1\}$ in the first column zero. As a result of these transformations, since we are subtracting non-negative quantities from the first entry of the first column, that entry remains negative. The matrix we obtain after this sequence of transformations has a structure,

$$\begin{bmatrix} -- & + & + & + & \cdots & + \\ 0 & ++ & 0 & 0 & \cdots & 0 \\ 0 & 0 & ++ & 0 & \cdots & 0 \\ 0 & 0 & 0 & ++ & \cdots & 0 \\ \vdots & \vdots & \vdots & \vdots & \cdots & \vdots \\ 0 & 0 & 0 & 0 & \cdots & ++ \end{bmatrix}. \tag{18}$$

Clearly a matrix of the above structure has a rank $|B| + 1$ and therefore invertible.

**Statement 2:** Using statement 1 of Lemma C.1, if we take $\boldsymbol{v} = \left( \frac{\partial \widetilde{\boldsymbol{\Phi}}_B}{\partial \boldsymbol{N}_{\overline{B}}} \right)^{-1} \frac{\partial \widetilde{\boldsymbol{\Phi}}_B}{\partial I}$, then we have,

$$\frac{\partial \widetilde{\boldsymbol{\Phi}}_B}{\partial \boldsymbol{N}_{\overline{B}}} \boldsymbol{v} = \frac{\partial \widetilde{\boldsymbol{\Phi}}_B}{\partial I}.$$

Note that, the RHS of the above linear system *i.e.* $\frac{\partial \widetilde{\boldsymbol{\Phi}}_B}{\partial I}$ is a $|B| + 1$ dimensional vector with zero in its first entry and $-1$ in every other entry. Using (17), $\boldsymbol{v} = [v_1, v_2, v_3, \ldots, v_{|B|+1}]^T$ satisfies a linear system with coefficients having the following signs,

$$(--)v_1 + (+)v_2 + (+)v_3 + (+)v_4 + \ldots + (+)v_{|B|+1} = 0, \quad \text{and}$$

$$\text{for } i \in \{2, \ldots, |B| + 1\}, \quad (+)v_1 + (++)v_i = (--),$$

where $++$, $--$ and $+$ represents quantities which are positive, negative and non-negative.

For every $i \in \{2, \ldots, |B| + 1\}$, we can eliminate $v_i$ from the first equation by subtracting some positive constant times the $i$-th equation from it. After eliminating $v_2, v_3, \ldots, v_{|B|+1}$ from the first equation following the mentioned procedure, we will be left with an equation of the form $(--)v_1 = (+)$, implying $v_1 \leq 0$.

Now the $i$-th equation of the system, for $i \in \{2, \ldots, |B| + 1\}$ is,

$$\frac{\partial \Phi_{a_{i-1}}}{\partial N_1} v_1 + \frac{\partial \Phi_{a_{i-1}}}{\partial N_{a_{i-1}}} v_i = -1. \tag{19}$$

We know that, $\frac{\partial \Phi_{a_{i-1}}}{\partial N_1} = d(\mu_1, x_{1,a_{i-1}})$ and $\frac{\partial \Phi_{a_{i-1}}}{\partial N_{a_{i-1}}} = d(\mu_{a_{i-1}}, x_{1,a_{i-1}})$, where $x_{1,a} = \frac{N_1 \mu_1 + N_a \mu_a}{N_1 + N_a}$ for every $a \in [K]/\{1\}$.

Now putting the derivatives in (19), we have

$$d(\mu_1, x_{1,a_{i-1}})v_1 + d(\mu_{a_{i-1}}, x_{1,a_{i-1}})v_i = -1,$$

which implies,

$$v_i = -\frac{1}{d(\mu_{a_{i-1}}, x_{1,a_{i-1}})} - \frac{d(\mu_1, x_{1,a_{i-1}})}{d(\mu_{a_{i-1}}, x_{1,a_{i-1}})}v_1,$$

for every $i \in \{2, \ldots, |B| + 1\}$.

Now adding both sides for $i \in \{2, 3, \ldots, |B| + 1\}$, we get,

$$
\begin{aligned}
\sum_{i=2}^{|B|+1} v_i &= -\sum_{i=2}^{|B|+1} \frac{1}{d(\mu_{a_{i-1}}, x_{1,a_{i-1}})} - v_1 \sum_{i=2}^{|B|+1} \frac{d(\mu_1, x_{1,a_{i-1}})}{d(\mu_{a_{i-1}}, x_{1,a_{i-1}})} \\
&= -\sum_{a \in B} \frac{1}{d(\mu_a, x_{1,a})} - v_1 \sum_{a \in B} \frac{d(\mu_1, x_{1,a})}{d(\mu_a, x_{1,a})} \\
&\leq -\sum_{a \in B} \frac{1}{d(\mu_a, x_{1,a})} - v_1 \sum_{a \in [K]/\{1\}} \frac{d(\mu_1, x_{1,a})}{d(\mu_a, x_{1,a})} \quad (\text{since } v_1 \leq 0) \\
&= -\sum_{a \in B} \frac{1}{d(\mu_a, x_{1,a})} - v_1,
\end{aligned}
$$

where the last step follows from the fact that $\sum_{a \in [K]/\{1\}} \frac{d(\mu_1, x_{1,a})}{d(\mu_a, x_{1,a})} = \Phi_1(\mathbf{N}, \mathbf{0}_K) + 1 = 1$. Taking $v_1$ on the LHS, we have,

$$\sum_{i=1}^{|B|+1} v_i \leq -\sum_{a \in B} \frac{1}{d(\mu_a, x_{1,a})}.$$

Note that the LHS of the above inequality is same as $\mathbf{1}^T \mathbf{v} = \mathbf{1}^T \left(\frac{\partial \widetilde{\boldsymbol{\Phi}}_B}{\partial \mathbf{N}_{\overline{B}}}\right)^{-1} \frac{\partial \Phi_{K+1}}{\partial I}$. In the RHS, since $d(\mu_a, x_{1,a}) \leq d(\mu_a, \mu_1)$, we conclude the desired result.

**Statement 3:** We have,

$$\frac{\partial \boldsymbol{\Phi}_B}{\partial (\mathbf{N}_{\overline{B}}, I)} = \left[ \begin{array}{c|c} \dfrac{\partial \widetilde{\boldsymbol{\Phi}}_B}{\partial \mathbf{N}_{\overline{B}}} & \dfrac{\partial \widetilde{\boldsymbol{\Phi}}_B}{\partial I} \\ \hline \dfrac{\partial \Phi_{K+1}}{\partial \mathbf{N}_{\overline{B}}} & \dfrac{\partial \Phi_{K+1}}{\partial I} \end{array} \right] = \left[ \begin{array}{c|c} \dfrac{\partial \widetilde{\boldsymbol{\Phi}}_B}{\partial \mathbf{N}_{\overline{B}}} & \dfrac{\partial \widetilde{\boldsymbol{\Phi}}_B}{\partial I} \\ \hline \mathbf{1}^T_{|B|+1} & 0 \end{array} \right]$$

We do the following determinant preserving column operation on $\frac{\partial \boldsymbol{\Phi}_B}{\partial(\boldsymbol{N}_{\overline{B}}, I)}$,

$$\left[\frac{\partial \boldsymbol{\Phi}_B}{\partial(\boldsymbol{N}_{\overline{B}}, I)}\right]_{:,|B|+2} \Longleftarrow \left[\frac{\partial \boldsymbol{\Phi}_B}{\partial(\boldsymbol{N}_{\overline{B}}, I)}\right]_{:,|B|+2} - \left[\frac{\partial \boldsymbol{\Phi}_B}{\partial(\boldsymbol{N}_{\overline{B}}, I)}\right]_{:,1:|B|+1} \left(\frac{\partial \widetilde{\boldsymbol{\Phi}}_B}{\partial \boldsymbol{N}_{\overline{B}}}\right)^{-1} \frac{\partial \widetilde{\boldsymbol{\Phi}}_B}{\partial I},$$

where $\left[\frac{\partial \boldsymbol{\Phi}_B}{\partial(\boldsymbol{N}_{\overline{B}}, I)}\right]_{:,|B|+2}$ and $\left[\frac{\partial \boldsymbol{\Phi}_B}{\partial(\boldsymbol{N}_{\overline{B}}, I)}\right]_{:,1:|B|+1}$, respectively, denotes the $|B|+2$-th column and left $(|B|+2) \times (|B|+1)$ submatrix of $\frac{\partial \boldsymbol{\Phi}_B}{\partial(\boldsymbol{N}_{\overline{B}}, I)}$.

The above column operation gives us the matrix,

$$\left[\begin{array}{c|c} \frac{\partial \widetilde{\boldsymbol{\Phi}}_B}{\partial \boldsymbol{N}_{\overline{B}}} & \boldsymbol{0}_{|B|+1} \\ \hline \boldsymbol{1}_{|B|+1}^T & -\boldsymbol{1}_{|B|+1}^T \left(\frac{\partial \widetilde{\boldsymbol{\Phi}}_B}{\partial \boldsymbol{N}_{\overline{B}}}\right)^{-1} \frac{\partial \widetilde{\boldsymbol{\Phi}}_B}{\partial I} \end{array}\right],$$

which has the same determinant as $\frac{\partial \boldsymbol{\Phi}_B}{\partial(\boldsymbol{N}_{\overline{B}}, I)}$. Therefore,

$$\det\left(\frac{\partial \boldsymbol{\Phi}_B}{\partial(\boldsymbol{N}_{\overline{B}}, I)}\right) = \left(-\boldsymbol{1}_{|B|+1}^T \left(\frac{\partial \widetilde{\boldsymbol{\Phi}}_B}{\partial \boldsymbol{N}_{\overline{B}}}\right)^{-1} \frac{\partial \widetilde{\boldsymbol{\Phi}}_B}{\partial I}\right) \times \det\left(\frac{\partial \widetilde{\boldsymbol{\Phi}}_B}{\partial \boldsymbol{N}_{\overline{B}}}\right).$$

Using statement 1 and 2 of Lemma C.1, both the quantities in the above product are non-zero, making the Jacobian $\frac{\partial \boldsymbol{\Phi}_B}{\partial(\boldsymbol{N}_{\overline{B}}, I)}$ invertible for every tuple in $\mathcal{Z}_B$. □

## D   Proofs from Section 2

Theorem D.1 is essential for proving Propositions 2.1 and 2.2 in Section 2. Before stating Theorem D.1 we state an alternative formulation of the max-min problem 1 which we call **O**.

$$\mathbf{O} : \quad \min \sum_{a=1}^{K} N_a$$

$$\text{s.t. } \forall a \neq 1, \ W_a(N_1, N_a) := N_1 d(\mu_1, x_{1,a}) + N_a d(\mu_a, x_{1,a}) \geq \log \frac{1}{2.4\delta}, \quad (20)$$

where $N_a \geq 0$ for all $a$, and each $x_{1,a} = \frac{\mu_1 N_1 + \mu_a N_a}{N_1 + N_a}$.

The optimal value of the problem **O** is of the form $T^\star(\boldsymbol{\mu}) \log(1/(2.4\delta))$, where $T^\star(\boldsymbol{\mu})$ is the reciprocal of the optimal value of (1). If $\boldsymbol{N}^\star = (N_a^\star : a \in [K])$ is an optimal allocation solving **O**, then $\omega_a^\star = \frac{N_a^\star}{\sum_{b \in [K]} N_b^\star}$ is an optimal proportion solving (1). Similarly if $\boldsymbol{\omega}^\star$ is an optimal proportion solving (1) then $\boldsymbol{N}^\star = (N_a^\star : a \in [K])$ with $N_a^\star = \omega_a^\star T^\star(\boldsymbol{\mu})$ is an optimal allocation solving **O**. Theorem D.1 implies uniqueness to the solution of **O** which also implies uniqueness of the solution of (1).

Some notations are needed before stating Theorem D.1. Let $B \subset [K]/1$ and $\overline{B} = B \cup \{1\}$. Let $\boldsymbol{\nu} = (\nu_a : a \in [K]) \in \mathcal{S}^K$ be some instance with $\nu_1 > \max_{a \neq 1} \nu_a$. Let $(I_a : a \in B)$ each be strictly positive. If $\overline{B}^c \neq \emptyset$ then let $(N_a \in \mathbb{R}_{\geq 0} : a \in \overline{B}^c)$ be the no. of samples allocated to arms in $\overline{B}^c$. We define $N_{1,1}$ as zero when $\sum_{a \in \overline{B}^c} N_a = 0$ or $\overline{B}^c = \emptyset$. Otherwise, we take $N_{1,1}$ to be the unique $N_1 > 0$ that solves,

$$\sum_{a \in \overline{B}^c} \frac{d(\nu_1, x_{1,a})}{d(\nu_a, x_{1,a})} = 1, \quad (21)$$

where $x_{1,a} = \frac{\nu_1 N_1 + \nu_a N_a}{N_1 + N_a}$. Note that if $\overline{B}^c \neq \emptyset$ and $\sum_{a \in \overline{B}^c} N_a > 0$, there exists an $a \in \overline{B}^c$ with $N_a > 0$. As a result, the LHS of (21) decreases from $\infty$ to 0 as $N_1$ increases from 0 to $\infty$. This implies the existence of a unique $N_1 > 0$ solving (21).

Set $N_{1,2} := \max_{a \in B} I_a d(\nu_1, \nu_a)^{-1}$.

**Theorem D.1.** *There exists a unique solution $N_1^\star \geq 0$ and $(N_a^\star \geq 0 : a \in B)$ satisfying*

$$\sum_{a \neq 1} \frac{d(\nu_1, x_{1,a}^\star)}{d(\nu_a, x_{1,a}^\star)} - 1 = 0 \quad \text{where} \quad x_{1,a}^\star = \frac{\nu_1 N_1^\star + \nu_a N_a^\star}{N_1^\star + N_a^\star}, \tag{22}$$

$$\text{and} \quad N_1^\star d(\nu_1, x_{1,a}^\star) + N_a^\star d(\nu_a, x_{1,a}^\star) = I_a \quad \forall a \in B. \tag{23}$$

*Further, $N_1^\star \geq \max(N_{1,1}, N_{1,2})$ and each $N_a^\star \geq \frac{I_a}{d(\nu_a, \nu_1)}$.*

*Furthermore, The optimal solution to $\mathbf{O}$ is uniquely characterized by the solution above with $B = \{2, \ldots, K\}$ and each $I_a = \log(1/(2.4\delta))$ and constraints (20) tight, that is, indexes of all the sub-optimal arms being equal to $\log(1/(2.4\delta))$. Further, $N_1^\star \geq \max_{a \in [K] \backslash 1} \frac{\log(1/(2.4\delta))}{d(\nu_1, \nu_a)}$ and each $N_a^\star \geq \frac{\log(1/(2.4\delta))}{d(\nu_a, \nu_1)}$.*

**Proof:** First observe that every solution $N_1^\star$ and $(N_a^\star : a \in B)$ to the system (22) and (23) must satisfy $N_1 \geq \max\{N_{1,1}, N_{1,2}\}$ and $N_a^\star \geq \frac{I_a}{d(\nu_a, \nu_1)}$, for every $a \in B$.

If $\overline{B}^c \neq \emptyset$, (22) implies,

$$\sum_{a \in \overline{B}^c} \frac{d(\nu_1, x_{1,a})}{d(\nu_a, x_{1,a})} \quad \leq \quad \sum_{a \in [K]/\{1\}} \frac{d(\nu_1, x_{1,a})}{d(\nu_a, x_{1,a})} \quad = \quad 1.$$

If $\sum_{a \in \overline{B}^c} N_a = 0$, then $N_1 \geq 0 = N_{1,1}$. Otherwise, we can find an $a \in \overline{B}^c$ with $N_a > 0$, making $\sum_{a \in \overline{B}^c} \frac{d(\nu_1, x_{1,a})}{d(\nu_a, x_{1,a})}$ strictly decreasing in $N_1$. As a result, by the definition of $N_{1,1}$ we have $N_1^\star \geq N_{1,1}$.

Now, for every $a \in B$, we have

$$I_a = N_1^\star d(\nu_1, x_{1,a}) + N_a^\star d(\nu_a, x_{1,a}) = \min_{x \in [\nu_a, \nu_1]} \{N_1^\star d(\nu_1, x) + N_a^\star d(\nu_a, x)\}.$$

Note that RHS of the above inequality is upper bounded by $\min\{N_1^\star d(\nu_1, \nu_a), N_a^\star d(\nu_a, \nu_1)\}$. As a result, for every $a \in B$, $N_a^\star \geq \frac{I_a}{d(\nu_a, \nu_1)}$, and $N_1^\star \geq \frac{I_a}{d(\mu_1, \mu_a)}$. This further implies, $N_1^\star \geq \max_{a \in B} \frac{I_a}{d(\nu_1, \nu_a)} = N_{1,2}$.

Now we prove existence of a unique $N_1^\star$ and $(N_a^\star : a \in B)$. For every $N_1 \geq N_{1,2}$ and $a \in B$, as $N_a$ increases from 0 to $\infty$, $N_1 d(\nu_1, x_{1,a}) + N_a d(\nu_a, x_{1,a})$ increases monotonically from 0 to $N_1 d(\nu_1, \nu_a)$. Note that $N_1 d(\nu_1, \nu_a) \geq N_{1,2} d(\nu_1, \nu_a) \geq I_a$ (by the definition of $N_{1,2}$). As a result, we can find a unique $N_a$ for which $N_1 d(\nu_1, x_{1,a}) + N_a d(\nu_a, x_{1,a}) = I_a$. For every $a \in B$, we call that unique $N_a$ as $N_a(N_1)$. Observe that, since $I_a > 0$, $N_a(N_1)$ is strictly decreasing in $N_1$, and if $I_a = 0$, then $N_a(N_1) = 0$. Also, if $a = \arg\max_{b \in B} \frac{I_b}{d(\nu_1, \nu_b)}$, then $N_a(N_{1,2}) = \infty$ and $\lim_{N_1 \to \infty} N_a(N_1) = \frac{I_a}{d(\nu_a, \nu_1)}$.

We now consider the allocation where every arm $a \in B$ has $N_a(N_1)$ samples, and consider the function,

$$h(N_1) = \sum_{a \in [K]/\{1\}} \frac{d(\nu_1, x_{1,a})}{d(\nu_a, x_{1,a})}.$$

Observe that, for all $a \in B$, $\frac{d(\nu_1, x_{1,a})}{d(\nu_a, x_{1,a})}$ is strictly decreasing for $N_1 \geq N_{1,2}$. Also if $\overline{B}^c$ is non-empty, then every term $\frac{d(\nu_1, x_{1,a})}{d(\nu_a, x_{1,a})}$ with $N_a > 0$ is strictly decreasing in $N_1$. As a result, the overall function $N_1 \to h(N_1)$ is strictly decreasing.

Moreover, as $N_1$ increases to $\infty$, $N_a(N_1)$ converges to $\frac{I_a}{d(\nu_a, \nu_1)}$ for every $a \in B$. Hence, $h(N_1)$ decreases to 0 as $N_1 \to \infty$. Therefore, if we can show that $h(\max\{N_{1,1}, N_{2,1}\}) \geq 1$, then we can find a unique $N_1^\star \geq \max\{N_{1,1}, N_{1,2}\}$ at which $h(N_1^\star) = 1$, and can take $N_a^\star = N_a(N_1^\star)$ for every $a \in B$. Following this, to prove uniqueness, it is sufficient to show $h(\max\{N_{1,1}, N_{1,2}\}) \geq 1$.

If $N_{1,2} \geq N_{1,1}$, then some $a \in B$ has $N_a(N_{1,2}) = \infty$. As a result, we have $h(N_{1,2}) = \infty \geq 1$. Otherwise, if $N_{1,1} \geq N_{1,2} > 0$, then by definition of $N_{1,1}$, we have

$$h(N_{1,1}) \geq \sum_{a \in \overline{B}^c} \frac{d(\nu_1, x_{1,a})}{d(\nu_a, x_{1,a})} = 1.$$

Hence we finish proving the first part of Theorem D.1.

To see the necessity of the stated optimality conditions for **O** observe that we cannot have $N_1^\star = 0$ or $N_a^\star = 0$ as that implies that the index $W_a(N_1^\star, N_a^\star)$ is zero. Further, if $W_a(N_1^\star, N_a^\star) > \log(1/(2.4\delta))$, the objective improves by reducing $N_a^\star$. Thus the constraints (20) must be tight. To see the tightness of (20) again note that the derivative of $W_a(N_1, N_a)$ with respect to $N_1$ and $N_a$, respectively, equals $d(\mu_1, x_{1,a})$ and $d(\mu_a, x_{1,a})$.

Now, perturbing $N_1$ by a tiny $\epsilon$ and adjusting each $N_a$ by $\frac{d(\mu_1, x_{1,a})}{d(\mu_a, x_{1,a})}\epsilon$ maintains the value of $W_a(N_1^\star, N_a^\star)$. Thus, at optimal $\boldsymbol{N}^\star$, necessity of tightness of inequalities in (20) follows. This condition can also be seen through the Lagrangian (see [2]).

The fact that these three criteria uniquely specify the optimal solution follows from our analysis above. Since the convex problem **O** has a solution, the uniqueness of the solution above satisfying the necessary conditions implies that this uniquely solves **O**. $\square$

To prove Propositions 2.1 and 2.2, we need to use the Implicit function theorem. For that, we define the following functions,

$$J_1(\boldsymbol{N}) = g(\boldsymbol{\mu}, \boldsymbol{N}) = \sum_{a \neq 1} \frac{d(\mu_1, x_{1,a}(N_1, N_a))}{d(\mu_a, x_{1,a}(N_1, N_a))} - 1,$$

$$\forall a \in [K]/\{1\} \quad J_a(\boldsymbol{N}, I) = N_1 d(\mu_1, x_{1,a}(N_1, N_a)) + N_a d(\mu_a, x_{1,a}(N_1, N_a)) - I,$$

$$J_{K+1}(\boldsymbol{N}, N) = \sum_{a \in [K]} N_a - N,$$

where $\boldsymbol{N} = (N_a \in \mathbb{R}_{\geq 0} : a \in [K])$, $I \in \mathbb{R}$, $N \in \mathbb{R}_+$, and, for every $a \in [K]/\{1\}$, $x_{1,a}(N_1, N_a) = \frac{N_1\mu_1 + N_a\mu_a}{N_1 + N_a}$.

Using these functions, for every non-empty $B \subseteq [K]/\{1\}$, we define the vector valued function

$$\boldsymbol{J}_B(\boldsymbol{N}, I, N) = [J_1(\boldsymbol{N}), \ (J_a(\boldsymbol{N}, I))_{a \in B}, \ J_{K+1}(\boldsymbol{N}, N)].$$

We call $\boldsymbol{J}_{[K]/\{1\}}(\cdot)$ as $\boldsymbol{J}(\cdot)$. Recall that $\overline{B}$ denotes $B \cup \{1\}$.

For every $m \geq 1$, we use the notation $\boldsymbol{0}_m$ to denote a $m$-dimensional vector with all entries set to zero. Observe that, for every $B \subseteq [K]/\{1\}$, $\boldsymbol{J}_B(\boldsymbol{N}, I, N) = \boldsymbol{\Phi}_B(\boldsymbol{N}, I, \boldsymbol{0}_K, N)$, where the function $\boldsymbol{\Phi}_B(\cdot)$ is defined in Appendix C.

Lemma D.1 is essential for proving Proposition 2.1.

**Lemma D.1.** *For every $N > 0$ and non-empty $B \subseteq [K]/\{1\}$, if $\hat{\boldsymbol{N}} = (\hat{N}_a \in \mathbb{R}_{\geq 0} : a \in [K])$ satisfies $\sum_{a \in \overline{B}^c} \hat{N}_a < N$, and $\boldsymbol{J}_B(\hat{\boldsymbol{N}}, \hat{I}, N) = \boldsymbol{0}_{|\overline{B}|+1}$, then, the Jacobian of $\boldsymbol{J}_B(\cdot)$ with respect to the arguments $(\boldsymbol{N}_{\overline{B}}, I)$ is invertible at $(\hat{\boldsymbol{N}}, \hat{I}, N)$.*

*Proof.* We have $\boldsymbol{J}_B(\boldsymbol{N}, I, N) = \boldsymbol{\Phi}_B(\boldsymbol{N}, I, \boldsymbol{0}_K, N)$, for every tuple $\boldsymbol{N}, I, N$, and non-empty $B \subseteq [K]/\{1\}$. As a result, Lemma D.1 follows from statement 3 of Lemma C.1. $\square$

For every non-empty subset $B \subseteq [K]/\{1\}$, we define the function $\widetilde{\boldsymbol{J}}_B(\cdot)$ to be the first $|B| + 1$ components of the vector valued function $\boldsymbol{J}_B(\cdot)$, or in other words,

$$\widetilde{\boldsymbol{J}}_B(\cdot) = [\ J_1(\cdot), \ (J_a(\cdot))_{a \in B}\ ].$$

Observe that $\widetilde{\boldsymbol{J}}_B(\cdot)$ depends only on the tuple $\boldsymbol{N}$ and $I$, and doesn't depend on $N$. Also for every tuple $(\boldsymbol{N}, I) \in \mathbb{R}_{\geq 0}^K$, $\widetilde{\boldsymbol{J}}_B(\boldsymbol{N}, I) = \widetilde{\boldsymbol{\Phi}}_B(\boldsymbol{N}, I, \boldsymbol{0}_K)$, where $\widetilde{\boldsymbol{\Phi}}_B$ is defined in Appendix C.

We now proceed on proving Propositions 2.1 and 2.2.

**Proof of Proposition 2.1:** Observe that, for every non-empty $B \subseteq [K]/\{1\}$, solving the system (2) is equivalent to solving for the pair $\mathbf{N}_{\overline{B}}, I_B$ in $\mathbf{J}_B((\mathbf{N}_{\overline{B}}, \mathbf{N}_{\overline{B}^c}), I_B, N) = 0$.

For every $I \geq 0$, by Theorem D.1, there is a unique $\mathbf{N}_{\overline{B}} = (N_a \in \mathbb{R}_{\geq 0} : a \in \overline{B})$ for which, $\widetilde{\mathbf{J}}_B((\mathbf{N}_{\overline{B}}, \mathbf{N}_{\overline{B}^c}), I) = \mathbf{0}_{|B|+1}$. We denote that solution using $\mathbf{N}_{\overline{B}}(I)$ (we supress the dependence on $\mathbf{N}_{\overline{B}^c}$ for cleaner presentation, and also because we will be treating $\mathbf{N}_{\overline{B}^c}$ like a constant in the rest of the proof). Since, $\widetilde{\mathbf{J}}_B(\mathbf{N}, I) = \widetilde{\mathbf{\Phi}}_B(\mathbf{N}, I, \mathbf{0}_K)$, by the Implicit function theorem and using statement 1 of Lemma C.1, the function $I \to \mathbf{N}_{\overline{B}}(I)$ is continuously differentiable. Also, we have

$$\mathbf{N}'_{\overline{B}}(I) = -\left(\frac{\partial \widetilde{\mathbf{J}}_B}{\partial \mathbf{N}_{\overline{B}}}\right)^{-1} \frac{\partial \widetilde{\mathbf{J}}_B}{\partial I} = -\left(\frac{\partial \widetilde{\mathbf{\Phi}}_B}{\partial \mathbf{N}_{\overline{B}}}\right)^{-1} \frac{\partial \widetilde{\mathbf{\Phi}}_B}{\partial I},$$

where the right most quantity is evaluated at the tuple $((\mathbf{N}_{\overline{B}}(I), \mathbf{N}_{\overline{B}^c}), I, \mathbf{0}_K)$. Moreover, using statement 2 of Lemma C.1, we have,

$$\sum_{a \in \overline{B}} N'_a(I) = \mathbf{1}^T_{|B|+1} \mathbf{N}'_{\overline{B}}(I) = -\mathbf{1}^T_{|B|+1} \left(\frac{\partial \widetilde{\mathbf{\Phi}}_B}{\partial \mathbf{N}_{\overline{B}}}\right)^{-1} \frac{\partial \widetilde{\mathbf{\Phi}}_B}{\partial I} \geq \sum_{a \in B} \frac{1}{d(\mu_a, \mu_1)} > 0$$

As a result, the function $\sum_{a \in \overline{B}} N_a(I)$ is strictly increasing in $I$ with a derivative atleast $\sum_{a \in B} \frac{1}{d(\mu_a, \mu_1)}$. Also, for $I = 0$, the unique solution is $N_1(0) = N_{1,1}$ and $N_a(0) = 0$ for every $a \in B$. As a result, as $I$ increases from $0$ to $\infty$, $\sum_{a \in \overline{B}} N_a(I)$ increases from $N_{11}$ to $\infty$ monotonically. Hence, for every $N \geq N_{11} + \sum_{a \in B^c} N_a$, we can find a unique $I_B$ for which $\sum_{a \in \overline{B}} N_a(I_B) + \sum_{a \in \overline{B}^c} N_a = N$. Therefore $\mathbf{N}_{\overline{B}}(I_B), I_B$ becomes the unique tuple to satisfy, $\mathbf{J}_B((\mathbf{N}_{\overline{B}}, \mathbf{N}_{\overline{B}^c}), I_B, N) = \mathbf{0}_{|B|+2}$. $\qquad \square$

**Proof of Proposition 2.2:** Recall that solving $\mathbf{O}$ in (20) is equivalent to solving (1). With this observation Proposition 2.2 follows directly from the definition of $\mathbf{N}_{[K]/\{1\}}(1), I_{[K]/\{1\}}(1)$ and the second statement of Theorem D.1. $\qquad \square$

## D.1 Single variable formulation of the lower bound problem and intuition behind the anchor function

Now we show that the $K$-variable convex optimization problem $\mathbf{O}$ defined in (20) can be reduced to a single variable convex optimization problem involving only $N_1$. To see this, observe that $\mathbf{O}$ is equivalent to the problem

$\mathbf{O}_1 \quad \min \sum_{a \in [K]} N_a$

$\qquad$ s.t. $\quad \forall a \neq 1, \ W_a(N_1, N_a) = N_1 d(\mu_1, x_{1,a}) + N_a d(\mu_a, x_{1,a}) = \log(1/(2.4\delta))$

$\qquad$ where $x_{1,a} = \dfrac{N_1 \mu_1 + N_a \mu_a}{N_1 + N_a}$, and $\forall a \in [K], \ N_a \geq 0$.

This follows from the proof of Theorem D.1.

We first make some observations about $\mathbf{O}_1$.

1. Upon fixing some $N_1 > 0$, $N_a \to W_a(N_1, N_a)$ is strictly increasing for every $a \neq 1$. Moreover, we have

$$W_a(N_1, 0) = 0 \quad \text{and} \quad \lim_{N_a \to \infty} W_a(N_1, N_a) = N_1 d(\mu_1, \mu_a).$$

As a result, every feasible solution of $\mathbf{O}_1$ must satisfy $N_1 d(\mu_1, \mu_a) > \log(1/(2.4\delta))$ for every $a \neq 1$, which implies $N_1 > \frac{\log(1/(2.4\delta))}{\min_{a \neq 1} d(\mu_1, \mu_a)}$.

2. Using the preceding observation, for every $a \neq 1$ and $N_1 > \frac{\log(1/(2.4\delta))}{\min_{a \neq 1} d(\mu_1, \mu_a)}$, we can find a unique $N_a$ such that $W_a(N_1, N_a) = \log(1/(2.4\delta))$. We use $\bar{N}_a(N_1)$ to denote that unique $N_a$ as a function of $N_1$. Using the Implicit function theorem, it is easy to prove that the function $N_1 \to \bar{N}_a(N_1)$ is differentiable for every $N_1 > \frac{\log(1/(2.4\delta))}{\min_{a \neq 1} d(\mu_1, \mu_a)}$ with the derivative being

$$\bar{N}_a'(N_1) = -\frac{d(\mu_1, z_a(N_1))}{d(\mu_a, z_a(N_1))} \quad \text{where} \quad z_a(N_1) = \frac{N_1 \mu_1 + \bar{N}_a(N_1) \mu_a}{N_1 + \bar{N}_a(N_1)}.$$

It follows that $\mathbf{O}_1$ is equivalent to the single variable optimization problem:

$$\mathbf{O}_2 \quad \min f(N_1) \overset{\text{def.}}{=} N_1 + \sum_{a \neq 1} \bar{N}_a(N_1)$$

$$\text{s.t. } N_1 > \frac{\log(1/(2.4\delta))}{\min_{a \neq 1} d(\mu_1, \mu_a)}.$$

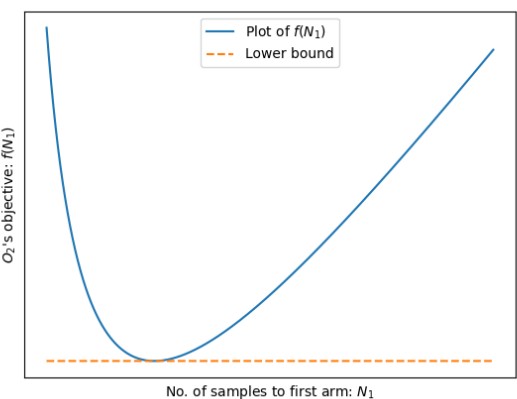

Figure 3: Illustrative plot of $\mathbf{O}_2$'s objective $f(N_1)$

Further,

$$f'(N_1) = 1 + \sum_{a \neq 1} \bar{N}_a'(N_1)$$

$$= 1 - \sum_{a \neq 1} \frac{d(\mu_1, z_a(N_1))}{d(\mu_a, z_a(N_1))},$$

which is exactly negative of the anchor function defined in Section 2 w.r.t. the allocation where arm 1 gets $N_1$ samples and every sub-optimal arm $a \neq 1$ gets $\bar{N}_a(N_1)$ samples. Furthermore, $f'(N_1)$ is strictly increasing in $N_1$ and increases from $-\infty$ to 1 as $N_1$ increases from $\frac{\log(1/(2.4\delta))}{\min_{a \neq 1} d(\mu_1, \mu_a)}$ to $\infty$. As a result, $\mathbf{O}_2$ is a convex problem w.r.t. $N_1$ and the optimal $N_1$ solving $\mathbf{O}_2$ is uniquely identified by the condition:

$$\sum_{a \neq 1} \frac{d(\mu_1, z_a(N_1))}{d(\mu_a, z_a(N_1))} - 1 = 0,$$

which is equivalent to saying that the anchor function $g$ must be zero when evaluated at the allocation where the first arm gets $N_1$ samples and every arm $a \neq 1$ gets $\bar{N}_a(N_1)$ samples. Figure 3 shows an illustrative plot of $\mathbf{O}_2$'s objective $f(N_1)$.

This observation also implies that the unique $\beta$ which makes the $\beta$-EB-TCB(I) policy in [16] asymptotically optimal is uniquely identified by the first order conditions: 1) the anchor function $g$ must be zero and 2) indexes of the sub-optimal arms must be equal. Hence allocations made by every

asymptotically optimal sampling policy must converge to these first order conditions. Both the fluid dynamics in Section 4 and Proposition 5.1 in Section 5, respectively, show that the sampling policies of (I)AT2 algorithm converges to these first order conditions in a fluid model and in the proposed algorithms.

## D.2 Sub-optimality of TCB(I)

[22] implicitly assumes that the optimal proportion is uniquely identified by the condition that indexes of every sub-optimal arm under the proportion are equal. Note that the optimal proportion is a $K$ dimensional vector and the conditions mentioned in [22, Lemma 2, statement 2] has $K - 1$ equations ($K - 2$ equations to maintain equality of $K - 1$ indexes and one more equation to make sure that all the entries in the $K$-dimensional vector add up to 1). Moreover, in Appendix D.1, changing the variables of the problem $\mathbf{O}_2$ to capture proportion of samples allocated to every arm, it is not hard to prove that for every value of $w_1 \in (0, 1)$, we can get a unique proportion $(w_1, w_2, \ldots, w_K)$ such that $\sum_{a \in [K]} w_a = 1$ and index of all sub-optimal arms under the proportion are equal. By the argument in Appendix D.1, the unique optimal proportion out of these infinite no. of proportions satisfying equality of the indexes, is uniquely identified by the necessary and sufficient condition that the anchor function $g(\cdot)$ evaluated at that proportion must be zero. Without this condition, the allocation cannot be optimal. In the numerical experiments of Appendix J.2, we see that the anchor function doesn't always converge to zero for the TCB(I) algorithm. As a result, allocations made by TCB(I) can be sub-optimal.

# E   Proofs from Section 4

**Proof of Theorem 4.1:** We first prove all the steps of Theorem 4.1 except for showing the existence of $\beta > 0$ and independent of $N$ such that $I_b'(N) > \beta$. That requires intermediate lemmas and is done separately.

First suppose that $B$ contains a singleton index $b$. Define $N_1(N)$ and $N_b(N)$ using IFT through the equations

$$\sum_{a \neq 1} \frac{d(\mu_1, x_{1,a})}{d(\mu_a, x_{1,a})} - 1 = 0 \tag{24}$$

and $\sum_a N_a = N$. For each $a$, letting $x_{1,a}'$ denote the derivative of $x_{1,a}$ with respect to $N_1$, $\widetilde{x}_{1,b}'$ denote the derivative of $x_{1,b}$ with respect to $N_b$. It is easy to check that

$$\widetilde{x}_{1,b}' = -\frac{N_1}{N_a} x_{1,b}', \tag{25}$$

and each $x_{1,a}' = \frac{N_a \Delta_a}{(N_1 + N_a)^2}$. Differentiating (24) with respect to $N$, observing that $N_b' = 1 - N_1'$, we get

$$N_1' \sum_{a \neq 1} h_a N_a = N_1 h_b (1 - N_1').$$

It follows that

$$N_1' = \frac{N_1 h_b}{\sum_{a \neq 1} h_a N_a + N_1 h_b}$$

as stipulated. Also $N_b' = \frac{\sum_{a \neq 1} h_a N_a}{\sum_{a \neq 1} h_a N_a + N_1 h_b}$.

Now consider the case where $g = 0$, and we have set $B \subset [K]/1$ of indices where the indexes are equal, they are higher for the remaining set. Cardinality of $B$ is at least 2. We want to argue that as $N$ increases, and the equality of indexes in $B$ is maintained along with $g = 0$, then the tied indexes will increase with $N$.

We have for $b, a \in B$

$$N_1 d(\mu_1, x_{1,b}) + N_b d(\mu_b, x_{1,b}) = N_1 d(\mu_1, x_{1,a}) + N_a d(\mu_a, x_{1,a}). \tag{26}$$

Furthermore, $g = 0$, i.e.,

$$\sum_{a \neq 1} \frac{d(\mu_1, x_{1,a})}{d(\mu_a, x_{1,a})} = 1. \tag{27}$$

Keeping a particular $b \in B$ fixed, differentiating with respect to $N$, (since for each $a$, by definition of $x_{1,a}$, $N_1 d'(\mu_1, x_{1,a}) + N_a d'(\mu_a, x_{1,a}) = 0$) we see from (26) that

$$N_1' d(\mu_1, x_{1,b}) + N_b{}' d(\mu_b, x_{1,b}) = N_1' d(\mu_1, x_{1,a}) + N_a{}' d(\mu_a, x_{1,a}).$$

Using (26) again in the above equality,

$$N_a' = \frac{1}{N_1 d(\mu_a, x_{1,a})} \left( N_a d(\mu_a, x_{1,a}) - N_b d(\mu_b, x_{1,b}) \right) N_1' + \frac{d(\mu_b, x_{1,b})}{d(\mu_a, x_{1,a})} N_b'. \tag{28}$$

Then from (27), we have that

$$N_1' \sum_{a \neq 1} f(\boldsymbol{\mu}, a, \boldsymbol{N}) x_{1,a}' + \sum_{a \in B} f(\boldsymbol{\mu}, a, \boldsymbol{N}) \widetilde{x}_{1,a}' N_a' = 0. \tag{29}$$

(Recall that for each $a$, $x_{1,a}'$ denotes the derivative of $x_{1,a}$ with respect to $N_1$ and $\widetilde{x}_{1,a}'$ denotes the derivative of $x_{1,a}$ with respect to $N_a$.)

Plugging (28) and (25) in (29), multiplying each term by $N_1^2$, we see that $N_1'$ is a ratio of

$$N_1 N_b' d_{b,b} h_B$$

with

$$h(N) + N_b d_{b,b} h_B.$$

Then,

$$N_b' = N_1' \frac{h(N) d_{b,b}^{-1} + N_b h_B}{N_1 h_B}.$$

In particular, since, $\sum_a N_a' = 1$, (3) follow. Statement 3 of Theorem 4.1 follows from (3) and expression for $I_a(N)$. Since $I_a'(N) > 0$ index is non-decreasing in $N$. Furthermore, $\lim_{N \to \infty} I_a(N) = N_a d(\mu_a, \mu_1)$. $\qquad \square$

To prove the existence of $\beta > 0$ and independent of $N$ such that $I_b'(N) > \beta$, we need Lemmas (E.1), (E.2) and (E.3). In Lemma E.3, we argue that the indexes in set $B$ grow linearly with the number of samples. Since index for arm $a \in \overline{B}^c$ are bounded, eventually indexes in set $B$ catch-up with other indexes.

Some notation first. Observe that $d(\mu_1, x) - d(\mu_a, x)$ is a continuous and strictly decreasing function of $x \in [\mu_a, \mu_1]$. It equals $d(\mu_1, \mu_a)$ for $x = \mu_a$ and $-d(\mu_a, \mu_1)$ for $x = \mu_1$. Let $x_a \in (\mu_a, \mu_1)$ be such that

$$d(\mu_1, x_a) = d(\mu_a, x_a).$$

Furthermore, let

$$\widetilde{a} = \arg\max_a \frac{d(\mu_1, x_{1,a})}{d(\mu_a, x_{1,a})}.$$

Let $x(\widetilde{a})$ be such that

$$d(\mu_1, x(\widetilde{a})) = (K - 1)^{-1} d(\mu_a, x(\widetilde{a})).$$

It is guaranteed to exist since the ratio $d(\mu_a, x)/d(\mu_1, x) \in (0, \infty)$ is monotonic in $x$.

Next, let

$$d_a = \frac{\mu_1 - x_a}{x_a - \mu_a} \quad \text{and} \quad d(\widetilde{a}) = \frac{\mu_1 - x(\widetilde{a})}{x(\widetilde{a}) - \mu_a}.$$

Since $K \geq 2$, we have $x(\widetilde{a}) \geq x_{\widetilde{a}}$, and $d(\widetilde{a}) \leq d_{\widetilde{a}}$.

**Lemma E.1.** *Suppose that $g = 0$. Then,*

$$\left(1 + \sum_{a \neq 1} d_a\right)^{-1} N \leq N_1 \leq (1 + d(\widetilde{a}))^{-1} N. \tag{30}$$

**Lemma E.2.** *Suppose that $g = 0$. Then,*

    1. *there exist constants $H$ and $D$ such that $h_a \leq H$ for all $a$, and $d_{a,a}^{-1} \leq D$ for all $a$.*

    2. *Further, there exists an $\widetilde{a}$ such that $N_{\widetilde{a}} > \alpha N$ for some $\alpha > 0$ and the corresponding $h_{\widetilde{a}}$ is bounded from below by a positive constant.*

**Lemma E.3.** *Suppose that $g = 0$, and for $B \subset [K]/1$ the indexes are all equal and are strictly higher for the remaining set. Then there exists a constant $\beta > 0$ such that*

$$N_1' d(\mu_1, x_{1,a}) + N_a' d(\mu_a, x_{1,a}) = I_B'(N) > \beta.$$

**Proof of Lemma (E.1):** Since $g = 0$, it follows that for each $a \in [K] \setminus 1$,

$$\frac{d_{1,a}}{d_{a,a}} \leq 1.$$

Thus, $x_{1,a} \geq x_a$. This in turn implies that for each $a$,

$$N_1 \geq N_a d_a^{-1}.$$

The above follows from substituting for $x_{1,a}$ in the inequality $x_{1,a} \geq x_a$, and from the definition of $d_a$. Moreover, since $g = 0$, it also follows that for each $a \neq 1$, $x((\widetilde{a}) \geq x_{1,a}$, implying

$$N_1 d((\widetilde{a})) \leq N_a.$$

Then,

$$N_1 \left(1 + \sum_{a \neq 1} d_a\right) \geq N$$

and

$$N_1(1 + d(\widetilde{a})) \leq N,$$

and the result follows. $\qquad\qquad\square$

**Proof of Lemma (E.2):** Recall the definitions of $h_a$ and $f(\boldsymbol{\mu}, a, \boldsymbol{N})$ from Section 4.

Since, $g = 0$ implies that $x_{1,a} \geq x_a$ for all $a$, it follows that $d_{a,a} = d(\mu_a, x_{1,a}) \geq d(\mu_a, x_a)$. In particular, for all $a$

$$d_{a,a}^{-1} \leq D$$

for $D = \max_a d(\mu_a, x_a)^{-1}$.

Further, $d'(\mu_a, x_{1,a})$ is continuous in $x_{1,a}$ and is bounded from above by $\sup_{x_a \leq x_{1,a} \leq \mu_1} d'(\mu_a, x_{1,a})$. Similarly, $-d'(\mu_1, x_{1,a})$ is bounded from above by $\sup_{x_a \leq x_{1,a} \leq \mu_1} -d'(\mu_1, x_{1,a})$. This implies that $f(\boldsymbol{\mu}, a, \boldsymbol{N})$ is bounded from above by a positive constant and hence so is $h_a$.

To see part 2, observe from definition of $x(\widetilde{a})$ that $x_{1,\widetilde{a}} \leq x(\widetilde{a})$. It follows that

$N_{\widetilde{a}} \geq N_1 d_{\widetilde{a}}^{-1}$. Therefore,

$$N_{\widetilde{a}} \geq N d_{\widetilde{a}}^{-1}(1 + \sum_{a \neq 1} d_a)^{-1}. \tag{31}$$

Again, $x_{1,\widetilde{a}} \leq x(\widetilde{a})$. Therefore, $d_{\widetilde{a},\widetilde{a}} = d(\mu_{\widetilde{a}}, x_{1,\widetilde{a}}) \leq d(\mu_{\widetilde{a}}, x(\widetilde{a}))$ and $d_{1,\widetilde{a}} = d(\mu_1, x_{1,\widetilde{a}}) \geq d(\mu_1, x(\widetilde{a}))$.

Further, $d'(\mu_{\widetilde{a}}, x_{1,\widetilde{a}})$ is continuous in $x_{1,\widetilde{a}}$ and is bounded from below by

$$\inf_{x_{\widetilde{a}} \leq x_{1,\widetilde{a}} \leq x(\widetilde{a})} d'(\mu_{\widetilde{a}}, x_{1,\widetilde{a}}).$$

Similarly, $-d'(\mu_1, x_{1,\widetilde{a}})$ is bounded from below by $\inf_{x_{\widetilde{a}} \leq x_{1,\widetilde{a}} \leq x(\widetilde{a})} -d'(\mu_1, x_{1,\widetilde{a}})$.

Thus, $f(\boldsymbol{\mu}, \widetilde{a})$ is bounded from below. Further, since each $N_a \leq d_a N_1$, $N_1^2/(N_1 + N_{\widetilde{a}})^2 \geq (1 + d_{\widetilde{a}})^{-2}$, hence, $h_{\widetilde{a}}$ is bounded from below by a positive constant. $\qquad\square$

**Proof of Lemma (E.3):** Recall from (31) that

$$N_{\widetilde{a}} \geq N d_{\widetilde{a}}^{-1} (1 + \sum_{a \neq 1} d_a)^{-1}. \tag{32}$$

Also, recall the definition of $f(\boldsymbol{\mu}, a, \boldsymbol{N})$ and $h_a$ from Section 4.

Because of (30) and (32), and since $N_{\widetilde{a}} \leq N$, we see that $f(\boldsymbol{\mu}, \widetilde{a})$ is bounded from below by a positive constant, and the same is true for $h_{\widetilde{a}}$.

If $\widetilde{a} \in B$, recall that $h_B = \sum_{a \in B} h_a d_{a,a}^{-1}$. Thus, $h_B$ is greater than a constant times $N$. This ensures that $N'_{\widetilde{a}}$ is bounded from below by a positive constant. Since $d_{\widetilde{a},\widetilde{a}}$ is also bounded from below by a positive constant, we conclude that there exists a $\beta > 0$ such that $I'_B(N) > \beta$.

If $\widetilde{a} \notin B$, then recalling that $h(N) = \sum_{a \in B^c/1} h_a N_a$, we conclude that $N'_{\widetilde{a}}$ is bounded from below by a positive constant. This implies that as $N$ increases by a positive fraction, so does each $N_a$ for $a \in B$. This in turn ensures that then $h_B$ is thereafter bounded from below by a positive constant. In particular, after some delay we have $I'_B(N) > \beta$ for some $\beta > 0$. $\qquad\square$

## E.1 Fluid dynamics starting at $g < 0$

Proposition E.1 provides us the ODEs by which the fluid allocations evolve in step 3 of the description of fluid dynamics when $g < 0$.

**Proposition E.1.** *Now consider the case where $g < 0$ at total allocations $N$, and $B$ again denotes the set of arms whose indexes have the minimum value. Then, till $I_B(N)$ increases with $N$ to either hit an index in $\overline{B}^c$, or for $g$ to equal zero, whichever is earlier, $I'_B(N) = \left(\sum_{a \in B} d_{a,a}^{-1}\right)^{-1}$, and for $a \in B$, $N'_a = d_{a,a}^{-1} \left(\sum_{a \in B} d_{a,a}^{-1}\right)^{-1}$. In particular, $I_B(N)$ and each $(N_a, a \in B)$, are increasing functions of $N$.*

*Proof.* Let $i_1$ denote the arm corresponding to a minimum index. Recall that $\boldsymbol{\omega}^\star = (\omega_a^\star : a \in [K])$ denote the optimal proportions to the lower bound problem. Consider $\hat{N}_a = \frac{\omega_a^\star}{\omega_1^\star} N_1$. Recall that at these samples, $g = 0$ and all the indexes are equal. Let $\hat{I}$ denote the corresponding value of the indexes at this allocation.

First we argue that $N_{i_1} < \hat{N}_{i_1}$.

Suppose this is not true, then $g < 0$ implies that for $N_1$ fixed, there exists an arm $a$ so that $N_a < \hat{N}_a$, else if each $N_a \geq \hat{N}_a$ then since $g$ increases with $N_a$, we would have $g \geq 0$. This contradiction implies that index for arm $a$ is $< \hat{I}_B(N)$. It follows that the index corresponding to $i_1$ is $< \hat{I}_B(N)$. Since the index increases with $N_{i_1}$, it follows that $N_{i_1} < \hat{N}_{i_1}$.

Thus, initially $N$ increases due to increase in $N_{i_1}$. Let $B = \{i_1\}$. Suppose, iteratively that $B = \{i_1, \ldots, i_{j-1}\}$, denotes the smallest indexes that are equal and increase with $N$ and $g < 0$. Proof follows by observing that the derivative of each index $a \in B$ satisfies the relation $N'_a d_{a,a} = I'_B(N)$. Further, $\sum_{a \in B} N'_a = 1$.

Thus, as $N$ increases, each $N_a(N), a \in B$ increases, so that $g$ increases. Since all indexes corresponding to $\overline{B}^c$ are constant, as $N$ increases, either $g = 0$ first or another index $I_j$ becomes equal to $I_B(N)$. $\qquad\square$

## E.2 Fluid dynamics of the $\beta$-EB-TCB algorithm ([16])

For every $\beta \in (0, 1)$ and allocation $\boldsymbol{N} = (N_a \in \mathbb{R}_{\geq 0} : a \in [K])$, we define the $\beta$-anchor function as,

$$g(\boldsymbol{N}; \beta) = \beta - \frac{N_1}{\sum_{a \in [K]} N_a}.$$

Note that, if $g(\boldsymbol{N}; \beta) = 0$, then $\beta$-fraction of the total no. of samples in the allocation $\boldsymbol{N}$ is allocated to the first arm. The fluid dynamics for the $\beta$-EB-TCB algorithm (see [16]) can be constructed similarly to that of the Anchored Top Two algorithm, by replacing the anchor function $g(\boldsymbol{\mu}, \boldsymbol{N})$ with the $\beta$-anchor function $g(\boldsymbol{N}; \beta)$ in Section 4.

**Existence of fluid dynamics:** Recall that, for every $B \subseteq [K]/\{1\}$, $\overline{B}$ denotes the set $B \cup \{1\}$. Lemma E.4 and Proposition E.2 are essential for constructing the fluid behavior for $\beta$-EB-TCB algorithm.

**Lemma E.4.** *Given a non-empty $B \subseteq [K]/\{1\}$, some tuple $\boldsymbol{N}_{\overline{B}^c} = (N_a \in \mathbb{R}_{\geq 0} : a \in \overline{B}^c)$ and $I \geq 0$, there is a unique tuple $\boldsymbol{N}_{\overline{B}} = (N_a \in \mathbb{R}_{\geq 0} : a \in \overline{B})$ which satisfies,*

$$N_1 = \beta \sum_{a \in [K]} N_a, \quad and$$

$$for\ every\ a \in B, \quad N_1 d(\mu_1, x_{1,a}) + N_a d(\mu_a, x_{1,a}) = I,$$

*where $x_{1,a} = \frac{N_1 \mu_1 + N_a \mu_a}{N_1 + N_a}$.*

*Moreover, if we define $N_{1,1} = \beta \sum_{a \in \overline{B}^c} N_a$ and $N_{1,2} = \frac{I}{\min_{a \in B} d(\mu_1, \mu_a)}$, then $N_1 \geq \max\{N_{1,1}, N_{1,2}\}$.*

*Proof.* Proof of this lemma follows an argument similar to the proof of Theorem D.1.

First we fix some $I \geq 0$ and $N_1 \geq N_{1,2}$. Note that for every $a \in B$, as $N_a$ increases from $0$ to $\infty$, $N_1 d(\mu_1, x_{1,a}) + N_a d(\mu_a, x_{1,a})$ increases monotonically from $0$ to $N_1 d(\mu_1, \mu_a)$. Since $N_1 \geq N_{1,2} \geq \frac{I}{d(\mu_1, \mu_a)}$, we have $N_1 d(\mu_1, \mu_a) \geq I$. This implies, for a given $N_1$, there is a unique value of $N_a$ for which $N_1 d(\mu_1, x_{1,a}) + N_a d(\mu_a, x_{1,a}) = I$. We call this value $N_a(N_1)$ for every $a \in B$.

Observe that $N_1 \to N_a(N_1)$ is a strictly decreasing function of $N_1$, and if $N_1 = N_{1,2}$, then there exists an $a \in B$ for which $N_a(N_{1,2}) = \infty$. On the other hand, if $N_1 \to \infty$, $N_a(N_1) \to \frac{I}{d(\mu_a, \mu_1)}$ for every $a \in B$.

For every $N_1$, we consider the function

$$h(N_1; \beta) = \beta - \frac{N_1}{N_1 + \sum_{a \in B} N_a(N_1) + \sum_{a \in \overline{B}^c} N_a}.$$

Note that $h(N_1; \beta)$ is the value of $g(\boldsymbol{N}; \beta)$, when the tuple $\boldsymbol{N}$ has $N_a = N_a(N_1)$ for every $a \in B$. Note that $N_1 \to h(N_1; \beta)$ is strictly decreasing for $N_1 \geq N_{1,2}$. Moreover, as $N_1 \to \infty$, $h(N_1; \beta) \to \beta - 1 < 0$. In the rest of the argument, we show that $h(\max\{N_{1,1}, N_{1,2}\}; \beta) \geq 0$. After we prove this, we can find a unique $N_1$ for which $h(N_1; \beta) = 0$. Using this, we take $N_a = N_a(N_1)$ for every $a \in B$ to obtain our unique tuple $\boldsymbol{N}_{\overline{B}}$.

Now we consider two cases.

**Case 1:** If $N_{1,1} \geq N_{1,2}$, then at $N_1 = N_{1,1}$,

$$N_{1,1} = \beta \sum_{a \in \overline{B}^c} N_a \leq \beta\left(N_{1,1} + \sum_{a \in [K]/\{1\}} N_a\right).$$

As a result,

$$\frac{N_{1,1}}{N_{1,1} + \sum_{a \in \overline{B}^c} N_a + \sum_{a \in B} N_a(N_{1,1})} \leq \beta,$$

which implies $h(N_{1,1}; \beta) \geq 0$.

**Case 2:** If $N_{1,2} \geq N_{1,1}$, then, as we argued before, there exists an $a \in B$ for which $N_a(N_{1,2}) = \infty$. As a result,

$$\frac{N_{1,2}}{N_{1,2} + \sum_{a \in \overline{B}^c} N_a + \sum_{a \in B} N_a(N_{1,2})} = 0$$

implying $h(N_{1,2}; \beta) = \beta > 0$. $\qquad\qquad\qquad\square$

Proposition E.2 stated below is crucial for constructing the fluid dynamics of the $\beta$-EB-TCB policy and is analogous to Proposition 2.1 used for constructing the fluid dynamics of the anchored top-two algorithm.

**Proposition E.2.** *For every non-empty $B \subseteq [K]/\{1\}$, tuple $\boldsymbol{N}_{\overline{B}^c} = (N_a \in \mathbb{R}_{\geq 0} : a \in \overline{B}^c)$, and $N \geq \frac{1}{1-\beta} \sum_{a \in \overline{B}^c} N_a$, there exists a unique tuple $\boldsymbol{N}_{\overline{B}} = (N_a \in \mathbb{R}_{\geq 0} : a \in \overline{B})$ and $I_B \geq 0$ for which,*

$$N_1 = \beta N, \quad \sum_{a \in [K]} N_a = N,$$

$$\text{for every } a \in B, \quad N_1 d(\mu_a, x_{1,a}) + N_a d(\mu_a, x_{1,a}) = I_B, \quad \text{and}$$

$$\text{where} \quad x_{1,a} = \frac{N_1 \mu_1 + N_a \mu_a}{N_1 + N_a} \quad \text{for every } a \in [K]/\{1\}.$$

*If we denote that tuple by $\boldsymbol{N}_{\overline{B}}(N)$ and $I_B(N)$, then the functions $N \to \boldsymbol{N}_{\overline{B}}(N), I_B(N)$ are continuously differentiable with respect to $N$.*

*Proof.* Proof of Proposition E.2 follows by an argument similar to the one used in the proof of Proposition 2.1, by using Lemma E.4 instead of Theorem D.1. Observe that the $\beta$-anchor function $g(\boldsymbol{N}; \beta)$ is strictly decreasing in $N_1$ and strictly increasing in $N_a$, when $N_1 > 0$. As a result, statement 1 of Lemma C.1 in Appendix C holds true upon having,

$$\Phi_1(\boldsymbol{N}, \boldsymbol{\eta}) = g(\boldsymbol{N}; \beta) - \eta_0,$$

and defining the set $\mathcal{Z}_B$ using the modified function $\boldsymbol{\Phi}_B$.

With the above modification, if we can find a constant $\gamma > 0$ such that, $\mathbf{1}_{|B|+1}^T \left( \frac{\partial \widetilde{\boldsymbol{\Phi}}_B}{\partial \boldsymbol{N}_B} \right)^{-1} \frac{\partial \widetilde{\boldsymbol{\Phi}}_B}{\partial I} \leq -\gamma < 0$ for every tuple in $\mathcal{Z}_B$, then statements 2 and 3 of Lemma C.1 also hold true for this modified $\boldsymbol{\Phi}_B$. As a result, Proposition E.2 follows using Lemma E.4 by the same argument used for proving Proposition 2.1 using Theorem D.1.

In the rest of the proof we argue the existence of such a constant $\gamma > 0$.

Let $\boldsymbol{v} = (v_a : a \in \overline{B}) \in \mathbb{R}^{|B|+1}$ be the solution to the system

$$\frac{\partial \widetilde{\boldsymbol{\Phi}}_B}{\partial \boldsymbol{N}_B} \boldsymbol{v} = \frac{\partial \widetilde{\boldsymbol{\Phi}}_B}{\partial I}.$$

We have $\frac{\partial \Phi_1}{\partial \boldsymbol{N}_{\overline{B}}} \boldsymbol{v} = \frac{\partial \Phi_1}{\partial I}$, and $N_1 = \beta \sum_{a \in [K]} N_a$, which after some algebraic manipulation implies,

$$-\left( \frac{1}{\beta} - 1 \right) v_1 + \sum_{a \in B} v_a = 0. \tag{33}$$

(33) further implies,

$$\mathbf{1}_{|B|+1}^T \boldsymbol{v} = \sum_{i=1}^{|B|+1} v_i = \frac{v_1}{\beta}.$$

Therefore proving that $v_1$ is upper bounded by a negative constant is sufficient for proving the desired result.

For every $a \in B$, we have

$$\frac{\partial \Phi_a}{\partial \boldsymbol{N}_{\overline{B}}} \boldsymbol{v} = \frac{\partial \Phi_a}{\partial I} = -1,$$

which implies

$$v_1 d(\mu_1, x_{1,a}) + v_a d(\mu_a, x_{1,a}) = -1, \tag{34}$$

where $x_{1,a} = \frac{N_1 \mu_1 + N_a \mu_a}{N_1 + N_a}$. We use $d_{1,a}$ and $d_{a,a}$, respectively, to denote $d(\mu_1, x_{1,a})$ and $d(\mu_a, x_{1,a})$.

For every $a \in B$, we can eliminate $v_a$ for (33) using (34). After this procedure, we get,

$$
\begin{aligned}
-v_1 &= \frac{\sum_{a \in B} \frac{1}{d_{a,a}}}{\frac{1}{\beta} - 1 + \sum_{a \in B} \frac{d_{1,a}}{d_{a,a}}} \\
&\geq \frac{\sum_{a \in B} \frac{1}{d_{a,a}}}{\frac{1}{\beta} - 1 + \sum_{a \neq 1} \frac{d_{1,a}}{d_{a,a}}} \\
&\geq \frac{\sum_{a \in B} \frac{1}{d_{a,a}}}{\frac{1}{\beta} - 1 + \sum_{a \neq 1} \frac{d(\mu_1, \mu_a)}{d_{a,a}}}. \quad \text{(since } d_{1,a} \leq d(\mu_1, \mu_a)\text{)} \tag{35}
\end{aligned}
$$

Since $N_1 = \beta \sum_{a \in [K]} N_a$, using (6) we have $d_{a,a} = \Theta(1)$ for all $a \in [K]/\{1\}$, where the constant hidden in $\Theta(\cdot)$ is independent of $\boldsymbol{N}$. As a result, by (35), $-v_1 = \Omega(1)$. Hence we conclude the proof. □

**Constructing the fluid ODEs:** Without loss of generality, we assume that the fluid dynamics starts from a state $\boldsymbol{N}$ where $g(\boldsymbol{N}; \beta) = 0$. Otherwise,

1. If $g(\boldsymbol{N}; \beta) > 0$, the algorithm gives samples to arm 1 till $g(\boldsymbol{N}; \beta) = 0$.
2. If $g(\boldsymbol{N}; \beta) < 0$, the $\beta$-EB-TCB algorithm follows the dynamics in Proposition E.1, and reaches $g(\boldsymbol{N}; \beta) = 0$ in a finite amount of time.

Following Proposition E.2, the algorithm tracks the allocation $\boldsymbol{N}_{\overline{B}}(N)$ at a given time $N$, where $B$ denotes the set of minimum index arms. We now determine the ODEs by which the state of the algorithm evolves.

To simplify the notations, we use $g_\beta$ as a shorthand for $g(\boldsymbol{N}(N); \beta)$ at a given time $N$. For every $a \in [K]$, we use $N_a, N_a', I_B$, and $I_B'$, respectively, to denote $N_a(N), N_a'(N), I_B(N)$ and $I_B'(N)$. For every $a \in \overline{B}^c$, we use $I_a$ to denote $I_a(N)$. Also for every $a \in [K]/\{1\}$, we adopt the notation $d_{1,a}$ and $d_{a,a}$, respectively, for the quantities $d(\mu_1, x_{1,a})$ and $d(\mu_a, x_{1,a})$, where $x_{1,a} = \frac{N_1 \mu_1 + N_a \mu_a}{N_1 + N_a}$.

For every non-empty $B \subseteq [K]/\{1\}$ we define the quantity,

$$d_B = \left( \sum_{a \in B} \frac{1}{d_{a,a}} \right)^{-1}.$$

We now show the fluid ODEs in the following proposition.

**Proposition E.3 (Fluid ODEs for $\beta$-EB-TCB).** *Let us assume the algorithm starts from a state* $\boldsymbol{N}(N^0) = (N_a^0 : a \in [K])$ *with* $\sum_{a \in [K]} N_a^0 = N^0$, $N_1^0 = \beta N^0$ *and* $N^0 > 0$. *Let* $B \subseteq [K]/\{1\}$ *be the set of arms having minimum index at a given time* $N \geq N^0$. *The following statements hold true about the allocation* $\boldsymbol{N}_{\overline{B}}(N) = (N_a(N) : a \in \overline{B})$ *made by* $\beta$-*EB-TCB algorithm,*

1. *The allocation* $\boldsymbol{N}(N) = (N_a(N) : a \in [K])$ *evolves by the following system of ODEs,*

$$N_1' = \beta, \quad \text{and}$$

$$\text{for every} \quad b \in B, \quad N_b' = \frac{((1 - \beta)N - \sum_{a \in B} N_a) d_B + N_b d_{b,b}}{N d_{b,b}}. \tag{36}$$

2. *The index $I_B(N)$ of the arms in $B$ evolves by the following ODE,*

$$I'_B = \left(1 - \beta - \frac{\sum_{a \in B} N_a}{N}\right) d_B + \frac{I_B}{N}. \qquad (37)$$

3. *There exists a constant $c > 0$ such that, $I'_B \geq c$. On the other hand, indexes of the arms in $\overline{B}^c$ remains upper bounded by $N_a^0 d(\mu_a, \mu_1)$.*

*Proof.* **Statement 1:** $N'_1 = \beta$ follows directly from the fact that $g_\beta = 0$.

By definition of $B$, we have

$$N_1 d_{1,a} + N_a d_{a,a} = N_1 d_{1,b} + N_b d_{b,b} \qquad (38)$$

for every $a, b \in [K]$. Taking derivative on both sides, we get,

$$N'_1 d_{1,a} + N'_a d_{a,a} = N'_1 d_{1,b} + N'_b d_{b,b},$$

which implies,

$$N'_a = \frac{d_{1,b} - d_{1,a}}{d_{a,a}} N'_1 + \frac{d_{b,b}}{d_{a,a}} N'_b. \qquad (39)$$

Using (38), we have

$$d_{1,b} - d_{1,a} = \frac{N_a d_{a,a} - N_b d_{b,b}}{N_1}.$$

Using the above expression in (39), we get,

$$N'_a = \frac{N_a d_{a,a} - N_b d_{b,b}}{N_1 d_{a,a}} N'_1 + \frac{d_{b,b}}{d_{a,a}} N'_b.$$

Since $N'_1 = \beta$ and $N_1 = \beta N$ (which follows from $g_\beta = 0$), the above equation implies,

$$N'_a = \frac{N_a d_{a,a} - N_b d_{b,b}}{N d_{a,a}} + \frac{d_{b,b}}{d_{a,a}} N'_b.$$

Adding both sides for $a \in B$, we get,

$$1 - \beta = \sum_{a \in B} N'_a = \sum_{a \in B} \frac{N_a d_{a,a} - N_b d_{b,b}}{N d_{a,a}} + d_{b,b} N'_b \sum_{a \in B} d_{a,a}^{-1}$$

$$= \frac{\sum_{a \in B} N_a}{N} - \frac{N_b}{N} d_{b,b} d_B^{-1} + N'_b d_{b,b} d_B^{-1},$$

which implies

$$N'_b = \frac{((1 - \beta)N - \sum_{a \in B} N_a) d_B + N_b d_{b,b}}{N d_{b,b}}.$$

**Statement 2:** We know

$$I'_B = N'_1 d_{1,b} + N'_b d_{b,b} \quad \text{for every } b \in B.$$

Putting in the derivatives from (36), we obtain,

$$I'_B = \beta d_{1,b} + \left(1 - \beta - \frac{\sum_{a \in B} N_a}{N}\right) d_B + \frac{N_b d_{b,b}}{N}$$

$$= \left(1 - \beta - \frac{\sum_{a \in B} N_a}{N}\right) d_B + \frac{\beta N d_{1,b} + N_b d_{b,b}}{N}$$

$$= \left(1 - \beta - \frac{\sum_{a \in B} N_a}{N}\right) d_B + \frac{I_B}{N}. \quad \text{(since } N_1 = \beta N \text{ and } I_b = I_B \text{ for every } b \in B\text{)}$$

**Statement 3:** For the following argument, the constants hidden in $O(\cdot)$, $\Omega(\cdot)$ and $\Omega(\cdot)$ are independent of $N$.

Note that $N_1 = \beta N$. As a result, using (6), $d_{a,a} = \Theta(1)$ for every $a \in [K]/\{1\}$. This implies, we can find a constant $c_1 > 0$ such that $d_B \geq c_1$. On the other hand, using (12), we have,

$$I_B = \Theta\left(\frac{N_1 N_a}{N_1 + N_a}\right)$$

for every $a \in B$. Since $N_1 = \beta N$ and $N_a \leq N$, we have

$$I_B = \Theta(N_a).$$

Adding for all $a \in B$, we get $I_B = \Theta(\sum_{a \in B} N_a)$. Therefore, $I_B \geq c_2 \sum_{a \in B} N_a$, for some constant $c_2 > 0$.

Now using (37),

$$
\begin{aligned}
I'_B &= \left(1 - \beta - \frac{\sum_{a \in B} N_a}{N}\right) d_B + \frac{I_B}{N} \\
&\geq c_1 \times \left(1 - \beta - \frac{\sum_{a \in B} N_a}{N}\right) + c_2 \times \frac{\sum_{a \in B} N_a}{N} \\
&\geq \min\{c_1, c_2\} \times (1 - \beta).
\end{aligned}
$$

Taking $c = \min\{c_1, c_2\} > 0$, we have the desired result.

Now for arms $a \in \overline{B}^c$, note that $x_{1,a} = \arg\min_{x \in [\mu_a, \mu_1]} \left(N_1 d(\mu_1, x) + N_a^0 d(\mu_a, x)\right)$ and $I_a = N_1 d(\mu_1, x_{1,a}) + N_a^0 d(\mu_a, x_{1,a})$.

As a result, putting $x = \mu_1$, we have $I_a \leq N_a^0 d(\mu_a, \mu_1)$. $\qquad\square$

**Reaching $\beta$-optimal proportions:** By statement 3 of Proposition E.3, the indexes of the arms in $B$ increase at a linear rate, whereas the indexes of the arms in $\overline{B}^c$ stay bounded above by a constant. As a result, by some finite time, $I_B$ crosses the index of some arm in $a \in \overline{B}^c$, after which $B$ gets updated to $B \cup \{a\}$. The same process then continues with the updated $B$. In this way, eventually $B = [K]/\{1\}$ and the fluid dynamics reaches the $\beta$-optimal proportion $\boldsymbol{\omega}^\star(\beta) = (\omega_a^\star(\beta) : a \in [K])$ ($\beta$-optimal proportion is the solution to the max-min problem (1) with the added constraint $\omega_1 = \beta$) where,

$$\frac{N_1}{N} = \omega_1^\star(\beta) = \beta \quad \text{and} \quad \frac{N_a}{N} = \omega_a^\star(\beta) \quad \text{for every } a \in [K]/\{1\}.$$

Applying the same argument as used to bound the time to reach optimal proportions in Section 4, if the fluid dynamics start at some time $N^0$ with state $\boldsymbol{N}(N^0) = (N_a(N^0) : a \in [K])$, then it reaches stability by a time atmost $\frac{N^0}{\min_{a \in [K]} \omega_a^\star(\beta)}$.

# F Intuitions based on fluid dynamics applied to algorithmic behavior

## F.1 Indexes once they meet do not separate

In the fluid dynamics described in Theorem 4.1, once the indexes meet thereafter they move up together by construction. It turns out that $I'_B(N)$ is positive. Below we give a heuristic argument that in our fluid dynamics, once a set of indexes that are equal, increase and catch up with another index, their union then remains equal and increases together with $N$. This argument provides important insights which help us later to prove that after after a random time of finite expectation, if our algorithm picks a suboptimal arm, then it picks that arm again in a periodic manner, which helps us prove closeness of indexes w.r.t. the algorithmic allocations (see Lemma 5.1). Differentiating $g = 0$ with respect to $N$, we see that,

$$N_1' \sum_{a \neq 1} f(\boldsymbol{\mu}, a, \boldsymbol{N}) \frac{N_a \Delta_a}{(N_1 + N_a)^2} = \sum_{a \neq 1} N_a' f(\boldsymbol{\mu}, a, \boldsymbol{N}) \frac{N_1 \Delta_a}{(N_1 + N_a)^2}. \tag{40}$$

Inductively, suppose that a set $B$ of indexes are moving up together and they run into another index $b$ at time $N$. Upon assuming contradiction, we can have a neighbourhood $[N, N + \Delta N]$ where the algorithm only allocates to a subset $C \subset B \cup \{b\}$ and doesn't allocate to arms in $D = B \cup \{b\} - C \neq \emptyset$. Then the allocations follow the ODEs in (3) of Theorem 4.1 with $B = C$, in the interval $[N, N + \Delta N]$.

Consider the probability vector $(p_a : a \in [K]/\{1\})$ where,

$$p_a = \frac{f(\boldsymbol{\mu}, a, \boldsymbol{N}) \frac{N_a \Delta_a}{(N_1 + N_a)^2}}{\sum_{b \neq 1} f(\boldsymbol{\mu}, b, \boldsymbol{N}) \frac{N_b \Delta_b}{(N_1 + N_b)^2}}.$$

Note that $p_a > 0$ for every $a \in [K]/\{1\}$. We have from (40) that

$$\frac{N_1'}{N_1} = \sum_{a \in C} \frac{N_a'}{N_a} p_a \tag{41}$$

Letting $b = \arg\max_{a \in C} \frac{N_a'(N)}{N_a(N)}$ (where $N_a'(N)$ is the derivative in (3) of Theorem 4.1, upon putting $B = C$), we have

$$\frac{N_1'}{N_1} \leq \left( \sum_{a \in C} p_a \right) \frac{N_b'}{N_b} \overset{(1)}{<} \frac{N_b'}{N_b}, \tag{42}$$

where the strict inequality in (1) follows from the fact that $D = B \cup \{b\} - C \neq \emptyset$, causing $\sum_{a \in C} p_a < 1$.

We now argue that $D$ must be empty. Suppose instead that $D \neq \emptyset$ and $a \in D$. Because all indexes in $B$ are equal at time $N$, we have, $N_1 d(\mu_1, x_{1,a}) + N_a d(\mu_a, x_{1,a}) = N_1 d(\mu_1, x_{1,b}) + N_b d(\mu_b, x_{1,b})$ at N. Observe that for any arm $d \in [K]/\{1\}$, derivative of its index with respect to $N$ equals $N_1' d(\mu_1, x_{1,d}) + N_d' d(\mu_d, x_{1,d})$ (since, by definition of $x_{1,d}$, $N_1 d_2(\mu_1, x_{1,d}) + N_d d_2(\mu_d, x_{1,d}) = 0$). Since arm $a$ gets no sample in $[N, N + \Delta N]$, we have $N_a' = 0$, which implies

$$I_a' = N_1' d(\mu_1, x_{1,a}) \text{ in } [N, N + \Delta N].$$

By our previous discussion

$$I_b' = N_1' d(\mu_1, x_{1,b}) + N_b' d(\mu_b, x_{1,b}). \tag{43}$$

We now argue that $N_1' d(\mu_1, x_{1,a})$ is strictly less than (43) at $N$. As a result, if $\Delta N > 0$ is picked sufficiently small, index of $b$, which is the minimum index, outruns index of $a$ in $[N, N + \Delta N]$.

Consider the difference

$$N_1' d(\mu_1, x_{1,a}) - N_1' d(\mu_1, x_{1,b}) - N_b' d(\mu_b, x_{1,b}) = N_1'((\mu_1, x_{1,a}) - d(\mu_1, x_{1,b})) - N_b' d(\mu_b, x_{1,b}). \tag{44}$$

We want show that this is strictly negative. We consider two cases,

**Case I:** If $d(\mu_1, x_{1,a}) - d(\mu_1, x_{1,b}) \leq 0$, it follows trivially.

**Case II:** If $d(\mu_1, x_{1,a}) - d(\mu_1, x_{1,b}) > 0$, since $N_b' > 0$, using (42) we can upper bound (44) by

$$N_b' \cdot \left( \frac{N_1}{N_b} (d(\mu_1, x_{1,a}) - d(\mu_1, x_{1,b})) - d(\mu_b, x_{1,b}) \right). \tag{45}$$

Since the two indexes are equal at this point, we have

$$N_1(d(\mu_1, x_{1,a}) - d(\mu_1, x_{1,b})) = N_b d(\mu_b, x_{1,b}) - N_a d(\mu_a, x_{1,a}).$$

Substituting this in (45), the latter equals

$$N_b' \cdot \left( \frac{1}{N_b} (N_b d(\mu_b, x_{1,b}) - N_a d(\mu_a, x_{1,a})) - d(\mu_b, x_{1,b}) \right) \leq -N_b' \cdot \frac{N_a}{N_b} d(\mu_a, x_{1,a}) < 0.$$

We thus have our contradiction. Therefore, indexes of the arms in $B \cup \{b\}$ move together.

## F.2 Proof sketch of Lemma 5.1

We consider the situation where the algorithm picks some arm $a \in [K]/\{1\}$ at iteration $N$ and doesn't pick $a$ for the next $R(N) \geq 1$ iterations. For better readability, we denote $R(N)$ using $R$. In the following argument, we try to bound $R$ from above using $N$. We can prove that $\widetilde{N}_j(N) = \Theta(N)$ for every $j \in [K]$ and $N \geq T_{good}$ (see Remark G.1 in Appendix G.2). As a result, $R$ is almost $O(N)$ for $N \geq T_{good}$. Below, we improve the upper bound to $O(N^{1-3\alpha/8})$ by a refined analysis.

Let us define $\Delta \widetilde{N}_j(N, t) = \widetilde{N}_j(N + t) - \widetilde{N}_j(N)$ for every $j \in [K]$ and $N, t \geq 1$. By our choice of $T_{good}$, we have $|g(\boldsymbol{\mu}, \widetilde{\boldsymbol{N}}(N))| = O(N^{-3\alpha/8})$ for $N \geq T_{good}$ (see Remark G.1 in Appendix G.2). By applying mean value theorem on $g(\cdot)$ for $N \geq T_{good}$, we have,

$$\left| \frac{\Delta \widetilde{N}_1(N, t)}{\widetilde{N}_1(N)} - \sum_{j \neq 1, a} \hat{p}_j(N, t) \frac{\Delta \widetilde{N}_j(N, t)}{\widetilde{N}_j(N)} \right| = O(N^{-3\alpha/8}), \quad \text{for every } t \leq R, \qquad (46)$$

where $(\hat{p}_j(N, t) : j \in [K]/\{1\})$ is a probability distribution over the set $[K]/\{1\}$, depending on $N$ and $t$ (this distribution is not important to the discussion and is spelt out at Appendix G.6). Taking $b_t = \arg\max_{a \neq 1} \frac{\Delta \widetilde{N}_a(N,t)}{\widetilde{N}_a(N)}$, and using (46), we obtain,

$$\frac{\Delta \widetilde{N}_1(N, t)}{\Delta \widetilde{N}_{b_t}(N, t)} \leq \frac{\widetilde{N}_1(N)}{\widetilde{N}_{b_t}(N)} + O(N^{1-3\alpha/8}) \quad \text{for every } t \leq R. \qquad (47)$$

Observe that (46) and (47), respectively, resembles (41) and (42) from Appendix F.1, except for a $O(N^{1-3\alpha/8})$ term due to the noise in $\widetilde{\boldsymbol{\mu}}$.

Applying the mean value theorem, we can bound the difference between the empirical indexes of arm $a$ and $b_t$ at iteration $N + t$ by,

$$\mathcal{I}_a(N + t) - \mathcal{I}_{b_t}(N + t) \leq \Delta \widetilde{N}_1(N, t) \cdot (d(\widetilde{\mu}_1, \widetilde{x}_{1,a}) - d(\widetilde{\mu}_1, \widetilde{x}_{1,b_t}))$$
$$- \Delta \widetilde{N}_{b_t}(N, t) \cdot d(\widetilde{\mu}_{b_t}, \widetilde{x}_{1,b_t}) + O(N^{1 - \frac{3\alpha}{8}}). \qquad (48)$$

Now if $d(\widetilde{\mu}_1, \widetilde{x}_{1,a}) - d(\widetilde{\mu}_1, \widetilde{x}_{1,b_t}) < 0$, (48) implies,

$$\mathcal{I}_a(N + t) - \mathcal{I}_{b_t}(N + t) \leq -\Delta \widetilde{N}_{b_t}(N, t) \cdot d(\widetilde{\mu}_{b_t}, \widetilde{x}_{1,b_t}) + O(N^{1-3\alpha/8}).$$

Otherwise, if $d(\widetilde{\mu}_1, \widetilde{x}_{1,a}) - d(\widetilde{\mu}_1, \widetilde{x}_{1,b_t}) \geq 0$, using (47) we have

$$\mathcal{I}_a(N + t) - \mathcal{I}_{b_t}(N + t)$$
$$\leq \Delta \widetilde{N}_{b_t}(N, t) \cdot \left( \frac{\widetilde{N}_1(N)}{\widetilde{N}_{b_t}(N)} (d(\widetilde{\mu}_1, \widetilde{x}_{1,a}) - d(\widetilde{\mu}_1, \widetilde{x}_{1,b_t})) - d(\widetilde{\mu}_{b_t}, \widetilde{x}_{1,b_t}) \right) + O(N^{1-3\alpha/8}), \quad (49)$$

Note that (48) resembles (45) in Appendix F.1. Since $\mathcal{I}_a(N - 1) \leq \mathcal{I}_{b_t}(N - 1)$, we can prove using the mean value theorem that $\mathcal{I}_a(N) \leq \mathcal{I}_{b_t}(N) + O(N^{1-3\alpha/8})$. Now expanding the empirical indexes, we get

$$\widetilde{N}_1(N) \cdot d(\widetilde{\mu}_1, \widetilde{x}_{1,a}) + \widetilde{N}_a(N) \cdot d(\widetilde{\mu}_a, \widetilde{x}_{1,a}) \leq \widetilde{N}_1(N) \cdot d(\widetilde{\mu}_1, \widetilde{x}_{1,a}) + \widetilde{N}_{b_t}(N) \cdot d(\widetilde{\mu}_{b_t}, \widetilde{x}_{1,b_t})$$
$$+ O(N^{1-3\alpha/8}).$$

Now dividing both sides by $\widetilde{N}_{b_t}(N)$ and using the fact that $\widetilde{N}_{b_t}(N) = \Theta(N)$, we have

$$\frac{\widetilde{N}_1(N)}{\widetilde{N}_{b_t}(N)} (d(\widetilde{\mu}_1, \widetilde{x}_{1,a}) - d(\widetilde{\mu}_1, \widetilde{x}_{1,b_t})) - d(\widetilde{\mu}_{b_t}, \widetilde{x}_{1,b_t}) \leq -\frac{\widetilde{N}_a(N)}{\widetilde{N}_{b_t}(N)} d(\widetilde{\mu}_a, \widetilde{x}_{1,a}) + O(N^{-3\alpha/8}).$$

Using the above inequality in (49), we have,

$$\mathcal{I}_a(N+t) - \mathcal{I}_{b_t}(N+t) \leq -\varDelta \widetilde{N}_{b_t}(N,t) \cdot \frac{\widetilde{N}_a(N)}{\widetilde{N}_{b_t}(N)} d(\widetilde{\mu}_a, \widetilde{x}_{1,a}) + O(N^{1-\frac{3\alpha}{8}})$$
$$+ O(\varDelta \widetilde{N}_{b_t}(N,t) \cdot N^{-\frac{3\alpha}{8}}),$$
$$\leq -\varDelta \widetilde{N}_{b_t}(N,t) \cdot \frac{\widetilde{N}_a(N)}{\widetilde{N}_{b_t}(N)} d(\widetilde{\mu}_a, \widetilde{x}_{1,a})$$
$$+ O(N^{1-\frac{3\alpha}{8}}) \quad (\text{since } \varDelta \widetilde{N}_{b_t}(N,t) \leq R = O(N)), \qquad (50)$$

whenever $d(\widetilde{\mu}_1, \widetilde{x}_{1,a}) - d(\widetilde{\mu}_1, \widetilde{x}_{1,b_t}) \geq 0$.

Since $\widetilde{N}_j(N) = \Theta(N)$ and $\widetilde{\mu}_j(N) \approx \mu_j$ for all $j \in [K]$ and $N \geq T_{good}$, the coefficient of $\varDelta \widetilde{N}_{b_t}(N,t)$ in (49) and (50) are $-\Theta(1)$. As a result, we can find a constant $C_3 > 0$, such that, for $t \leq R$ and $N \geq T_{good}$,

$$\mathcal{I}_a(N+t) - \mathcal{I}_{b_t}(N+t) \leq -C_3 \varDelta \widetilde{N}_b(N,t) + O(N^{1-3\alpha/8}). \qquad (51)$$

Applying the mean value theorem and using the fact that $t$ can be atmost $O(N)$, we can prove that, (51) implies,

$$\mathcal{I}_a(N+t-1) - \mathcal{I}_{b_t}(N+t-1) \leq -C_3 \varDelta \widetilde{N}_b(N,t) + O(N^{1-3\alpha/8}).$$

As a result, we have a constant $C_4 > 0$ such that,

$$\mathcal{I}_a(N+t-1) - \mathcal{I}_{b_t}(N+t-1) \leq -C_3 \varDelta \widetilde{N}_b(N,t) + C_4 N^{1-3\alpha/8}. \qquad (52)$$

Using (47), we can choose $T_{good}$ suitably, and find constants $D_1, D_2 > 0$, such that, whenever $N \geq T_{good}$,

$$R \geq D_1 N^{1-3\alpha/8} \implies \varDelta \widetilde{N}_{b_R}(N,R) \geq D_2 R \quad (\text{see Lemma G.15 of Appendix G.6}).$$

We consider the case where $R \geq \max\left\{D_1, \frac{2C_4}{C_3 D_2}\right\} \times N^{1-3\alpha/8}$.

Since $R \geq D_1 N^{1-3\alpha/8}$, we have

$$\varDelta \widetilde{N}_{b_R}(N,R) \geq D_2 R \geq D_2 \times \frac{2C_4}{C_3 D_2} N^{1-3\alpha/8} = \frac{2C_4}{C_3} N^{1-3\alpha/8}.$$

For notational simplicity, we use $b$ to denote $b_R$. We consider the iteration $N+S$ where arm $b$ was selected for the last time before iteration $N+R$. Then by definition of $b_R$ and $S$, and using the above inequality, we have

$$\varDelta \widetilde{N}_b(N,S) = \varDelta \widetilde{N}_b(N,R) \geq \frac{2C_4}{C_3} N^{1-3\alpha/8}. \qquad (53)$$

Also, since $\varDelta \widetilde{N}_b(N,S) = \varDelta \widetilde{N}_b(N,R)$ and for every $j \neq 1$ $\varDelta \widetilde{N}_j(N,R) \geq \varDelta \widetilde{N}_j(N,S)$, we conclude $b = b_S$. Therefore,

$$\mathcal{I}_a(N+S-1) - \mathcal{I}_b(N+S-1) = \mathcal{I}_a(N+S-1) - \mathcal{I}_{b_S}(N+S-1)$$
$$(\text{using (52)}) \quad \leq -C_3 \varDelta \widetilde{N}_{b_S}(N,S) + C_4 N^{1-3\alpha/8}$$
$$(\text{since } b = b_S) \quad = -C_3 \varDelta \widetilde{N}_b(N,S) + C_4 N^{1-3\alpha/8}$$
$$(\text{using (53)}) \quad \leq -C_3 \times \frac{2C_4}{C_3} N^{1-3\alpha/8} + C_4 N^{1-3\alpha/8}$$
$$= -C_4 N^{1-3\alpha/8}.$$

The above inequality implies, the AT2 algorithm pulls arm $b$ at iteration $N+S$, even though

$$\mathcal{I}_a(N+S-1) \leq \mathcal{I}_b(N+S-1) - C_4 N^{1-3\alpha/8},$$

which is contradicting the algorithm's description. Hence we must have

$$R \leq \max\left\{D_1, \frac{2C_4}{C_3 D_2}\right\} \times N^{1-3\alpha/8} = O(N^{1-3\alpha/8}).$$

$\square$

# G Algorithmic allocations: non fluid behaviour

In the following sections, unless otherwise stated, the proof of the mentioned results for AT2 (1) and IAT2 (2) algorithms follow a similar argument. Also, the constants introduced while stating the results in the following sections might be different for the two algorithms.

While using the $O(\cdot), \Theta(\cdot)$ and $\Omega(\cdot)$ notations, we imply that the hidden constants can depend on the choice of algorithm among AT2 and IAT2, instance $\boldsymbol{\mu}$, exploration factor $\alpha \in (0, 1)$ and no. of arms $K$, and are independent of the sample path.

## G.1 Convergence of algorithmic allocations to the optimality conditions

In this section, our agenda is to prove the convergence of the allocations of AT2 and IAT2 algorithms to the optimality conditions mentioned in Proposition 2.2. For ease of presentation we state the conditions uniquely characterizing the optimal proportion $\boldsymbol{\omega}^\star$ below according to Proposition 2.2:

$$\sum_{a \neq 1} \frac{d(\mu_1, x_{1,a}^\star)}{d(\mu_a, x_{1,a}^\star)} = 1 \quad \text{and} \quad \forall a \neq 1, \quad \omega_1^\star d(\mu_1, x_{1,a}^\star) + \omega_a d(\mu_a, x_{1,a}^\star) = I^\star = T^\star(\boldsymbol{\mu})^{-1}$$

(54)

$$\text{where} \quad x_{1,a}^\star = \frac{\omega_1^\star \mu_1 + \omega_a^\star \mu_a}{\omega_1^\star + \omega_a^\star}, \quad \text{and} \quad \sum_{a \in [K]} \omega_a^\star = 1.$$

Recall that for every $a \in [K]$ and iteration $N$, $\widetilde{\omega}_a(N) = \frac{\widetilde{N}_a(N)}{N}$ denotes the proportion of samples allocated by the algorithm to arm $a$. Let $\widetilde{\boldsymbol{\omega}}(N) = (\widetilde{\omega}_a(N) : a \in [K])$.

Recall the anchor function $g(\boldsymbol{\mu}, \widetilde{\boldsymbol{N}}(\cdot))$ and index $I_a(\cdot)$ for every alternative arm $a \in [K]/\{1\}$. In Section 5, we defined the normalized index $H_a(\cdot)$ of every arm $a \in [K]/\{1\}$ at iteration $N$ as $H_a(N) = \frac{1}{N} I_a(N)$. In the next two sections, we prove,

$$|g(\boldsymbol{\mu}, \widetilde{\boldsymbol{\omega}}(N))| = \left| \sum_{a \in [K]} \frac{d(\mu_1, x_{1,a}(N))}{d(\mu_a, x_{1,a}(N))} - 1 \right| \longrightarrow 0,$$

(55)

and

$$\max_{a,b \in [K]/\{1\}} |H_a(N) - H_b(N)| \longrightarrow 0$$

(56)

a.s. in $\mathbb{P}_{\boldsymbol{\mu}}$ as $N \to \infty$. Moreover, we show that, after a random time of finite expectation, both the convergences in (55) and (56) happen at a uniform rate over all sample paths. We prove these convergence results in Proposition G.1 and G.2 stated below.

**Proposition G.1** (**Convergence of $g$ to zero**). *There exists constants $M_4 \geq 1$ and $C > 0$ independent of the sample paths, such that, if $T_6$ is defined as the iteration at which $g(\widetilde{\boldsymbol{\mu}}(\cdot), \widetilde{\boldsymbol{N}}(\cdot))$ crosses the value zero after iteration $\max\{M_4, T_5\}$ ($T_5$ is a random time satisfying $\mathbb{E}_{\boldsymbol{\mu}}[T_5] < \infty$ and defined in Definition G.1 of Appendix G.1.1), then for $N \geq T_6$ we have,*

$$\left| g(\boldsymbol{\mu}, \widetilde{\boldsymbol{N}}(N)) \right| \leq CN^{-3\alpha/8}.$$

*Moreover, the random time $T_6$ satisfies $\mathbb{E}_{\boldsymbol{\mu}}[T_6] < \infty$.*

**Proposition G.2** (**Closeness of the indexes**). *There exists a random time $T_8$ (defined in Definition G.4 of Appendix G.1.2) satisfying $\mathbb{E}_{\boldsymbol{\mu}}[T_8] < \infty$, such that, for $N \geq T_8$, every pair of alternative arms $a, b \in [K]/\{1\}$ has,*
$$|I_a(N) - I_b(N)| = O(N^{1-3\alpha/8}).$$

**Proof of Proposition 5.1:** By the definition of $T_{stable}$ in Definition G.5 of Appendix G.2, we have $T_{stable} \geq T_6, T_8$, where $T_6$ and $T_8$ are the random times mentioned, respectively, in Proposition G.1

and G.2. As a result, Proposition 5.1 follows trivially from Proposition G.1 and G.2. $\qquad\square$

Proof of Proposition G.1 is in Appendix G.1.1. We prove a detailed version of Proposition G.2 as Proposition G.3 in Appendix G.1.2. Both these results are crucial later for proving the convergence of the algorithmic proportions $\widetilde{\boldsymbol{\omega}}(N) = (\widetilde{\omega}_a(N) : a \in [K])$ to the optimal proportions $\boldsymbol{\omega}^\star = (\omega_a^\star : a \in [K])$ in Proposition 3.1 from Section 3. We prove a detailed version of Proposition 3.1 as Proposition G.4 in Appendix G.2.

To prove Proposition G.1, G.2, and later Proposition G.4, we need to prove several technical properties related to exploration and the allocations made by the algorithms. The detailed technical results related to exploration are in Appendix G.4 and those related to the algorithmic allocations are in Appendix G.5. The arguments in Appendix G.1.1, G.1.2, and G.2 are self-contained, and we refer the reader to the related technical results whenever necessary. For ease of exposition, we provide below a brief summary of the statements proven in Appendix G.4 and G.5.

**Summary of technical results in Appendix G.4 and G.5**

We summarize below the results proven in Appendix G.4 and G.5 as events happening between the non-decreasing sequence of random times $T_0$, $T_1$, $T_2$, $T_3$, and $T_4$, which are defined in Appendix G.5.

1. $T_0 \overset{\text{def.}}{=} \min\{N' \geq 1 \mid \forall N \geq N', \ \max_{a \in [K]} |\widetilde{\mu}_a(N) - \mu_a| \leq \epsilon(\boldsymbol{\mu}) N^{-3\alpha/8}\}$, where $\epsilon(\boldsymbol{\mu}) > 0$ (defined in Appendix B), is a constant depending on the instance $\boldsymbol{\mu}$. By definition, we have $\epsilon(\boldsymbol{\mu}) \leq \frac{1}{4} \min_{a \neq 1}(\mu_1 - \mu_a)$. As a result, the first arm becomes the empirically best arm and stays that way forever after iteration $T_0$. In Lemma G.7 of Appendix G.3, we prove that $\mathbb{E}_{\boldsymbol{\mu}}[T_0] < \infty$, which implies $T_0 < \infty$ a.s. in $\mathbb{P}_{\boldsymbol{\mu}}$.

2. $T_1 \overset{\text{def.}}{=} \max\{T_{\text{explo}}, T_0\}$, where $T_{\text{explo}} < \infty$ is a constant defined in Definition G.7 of Appendix G.4. After iteration $T_{\text{explo}}$, the algorithm consecutively does exploration over a strech of atmost $K$ iterations. Moreover, over a single such "*epoch*" of consecutive explorations, the algorithm explores every arm atmost once (follows from statement 1 and 3 of Proposition G.5). Note that $\mathbb{E}_{\boldsymbol{\mu}}[T_1] \leq T_{\text{explo}} + \mathbb{E}_{\boldsymbol{\mu}}[T_0] < \infty$.

3. $T_2$ is defined in Lemma G.11 as the iteration at which the anchor function $g(\widetilde{\boldsymbol{\mu}}(\cdot), \widetilde{\boldsymbol{N}}(\cdot))$ crosses the value zero after the iteration $\max\{M_1, T_1\}$ ($M_1 \geq 1$ is a constant independent of the sample paths and defined in the proof of Lemma G.11). By Lemma G.9, there exists a constant $C_1 \geq 1$ independent of the sample paths, such that $T_2 \leq C_1 \max\{M_1, T_1\}$. As a result, $\mathbb{E}_{\boldsymbol{\mu}}[T_2] \leq C_1(M_1 + \mathbb{E}_{\boldsymbol{\mu}}[T_1]) < \infty$. After iteration $T_2$, the empirical anchor function $g(\widetilde{\boldsymbol{\mu}}(\cdot), \widetilde{\boldsymbol{N}}(\cdot))$ remains bounded inside an interval of the form $[-(1 - d_{\min}), d_{\max} - 1]$, where $d_{\min} \in (0, 1)$ and $d_{\max} \in (1, \infty)$ are constants independent of the sample paths (see Lemma G.11). Exploiting this, we argue that both $\widetilde{N}_1(N)$ and $\max_{a \in [K]/\{1\}} \widetilde{N}_a(N)$ become $\Omega(N)$ after iteration $T_2$ (see Corollary G.1).

4. $T_3 \overset{\text{def.}}{=} \max\{M_2, T_2\} + 2$, where $M_2 \geq 1$ is a constant chosen in the proof of Lemma G.12 and is independent of the sample paths. After iteration $T_3$, whenever the algorithm picks an alternative arm $a \in [K]/\{1\}$, then for every other alternative arm $b \in [K]/\{1, a\}$, we have $\widetilde{N}_b(N) \geq \gamma \widetilde{N}_a(N)$, for some constant $\gamma \in (0, 1)$ independent of the sample paths (see Lemma G.12). Note that $\mathbb{E}_{\boldsymbol{\mu}}[T_3] \leq M_2 + 2 + \mathbb{E}_{\boldsymbol{\mu}}[T_2] < \infty$.

5. $T_4 = C_2(T_3 + 1)$ for some constant $C_2 \geq 1$ independent of the sample paths, defined in Lemma G.13. After iteration $T_4$, all the arms $a \in [K]$ have $\widetilde{N}_a(N) = \Theta(N)$ (see Lemma G.13). Note that $\mathbb{E}_{\boldsymbol{\mu}}[T_4] \leq C_2(\mathbb{E}_{\boldsymbol{\mu}}[T_3] + 1) < \infty$.

**G.1.1 Convergence of the anchor function to zero**

The following lemma bounds the fluctuation of $g(\widetilde{\boldsymbol{\mu}}, \widetilde{\boldsymbol{N}})$ around $g(\boldsymbol{\mu}, \widetilde{\boldsymbol{N}})$ due to the noise in the estimate $\widetilde{\boldsymbol{\mu}}$ of $\boldsymbol{\mu}$. We need this lemma later for proving convergence of the anchor function $g$ to zero in Proposition G.1.

**Lemma G.1** (**Bounding the noise in** $g$)**.** *For every $N \geq T_2$ (where $T_2$ is the random time defined in Lemma G.11 and satisfies $\mathbb{E}_{\boldsymbol{\mu}}[T_2] < \infty$), we have,*

$$|g(\widetilde{\boldsymbol{\mu}}(N), \widetilde{\boldsymbol{N}}(N)) - g(\boldsymbol{\mu}, \widetilde{\boldsymbol{N}}(N))| = O(N^{-3\alpha/8}).$$

*Proof.* Using mean value theorem for function of several variables, we have,

$$|g(\widetilde{\boldsymbol{\mu}}(N), \widetilde{\boldsymbol{N}}(N)) - g(\boldsymbol{\mu}, \widetilde{\boldsymbol{N}}(N))| \leq \sum_{a=1}^{K} \left| \frac{\partial g}{\partial \mu_a}(\hat{\boldsymbol{\mu}}, \widetilde{\boldsymbol{N}}(N)) \right| \cdot |\widetilde{\mu}_a(N) - \mu_a|,$$

where $\hat{\mu}_a$ lies between $\mu_a$ and $\widetilde{\mu}_a(N)$ for every $a \in [K]$.

We define

$$\hat{x}_{1,a} = \frac{\widetilde{N}_1(N)\hat{\mu}_1 + \widetilde{N}_a(N)\hat{\mu}_a}{\widetilde{N}_1(N) + \widetilde{N}_a(N)}, \quad \text{for every } a \in [K]/\{1\}.$$

Note that,

$$\frac{\partial g}{\partial \mu_1}(\hat{\boldsymbol{\mu}}, \widetilde{\boldsymbol{N}}(N)) = \sum_{a \neq 1} \left( \frac{d_1(\hat{\mu}_1, \hat{x}_{1,a})}{d(\hat{\mu}_a, \hat{x}_{1,a})} - f(\hat{\boldsymbol{\mu}}, a, \hat{\boldsymbol{N}}) \cdot \frac{\widetilde{N}_1}{\widetilde{N}_1 + \widetilde{N}_a} \right), \quad \text{and,}$$

$$\forall a \neq 1, \quad \frac{\partial g}{\partial \mu_a}(\hat{\boldsymbol{\mu}}, \widetilde{\boldsymbol{N}}(N)) = -\frac{d(\hat{\mu}_1, \hat{x}_{1,a})d_1(\hat{\mu}_a, \hat{x}_{1,a})}{(d(\hat{\mu}_a, \hat{x}_{1,a}))^2} - f(\hat{\boldsymbol{\mu}}, a, \hat{\boldsymbol{N}}) \cdot \frac{\widetilde{N}_a}{\widetilde{N}_1 + \widetilde{N}_a}, \quad (57)$$

$$\text{where} \quad f(\hat{\boldsymbol{\mu}}, a, \hat{\boldsymbol{N}}) = -\frac{d_2(\hat{\mu}_1, \hat{x}_{1,a})}{d(\hat{\mu}_a, \hat{x}_{1,a})} + \frac{d(\hat{\mu}_1, \hat{x}_{1,a})d_2(\hat{\mu}_a, \hat{x}_{1,a})}{(d(\hat{\mu}_a, \hat{x}_{1,a}))^2},$$

and recall that $d_1(\cdot, \cdot)$ and $d_2(\cdot, \cdot)$, respectively, denote the partial derivatives of $d(\cdot, \cdot)$ with respect to its first and second argument.

By (6), for $N > T_2$, we have,

$$d(\hat{\mu}_a, \hat{x}_{1,a}) = \Theta\left((\hat{x}_{1,a} - \hat{\mu}_a)^2\right) = \Theta\left(\frac{\widetilde{N}_1(N)^2}{(\widetilde{N}_1(N) + \widetilde{N}_a(N))^2}\right).$$

By Corollary G.1 from Appendix G.5, we have $\widetilde{N}_1(N) = \Omega(N)$ for $N > T_2$. As a result, $d(\hat{\mu}_a, \hat{x}_{1,a}) = \Theta(1)$ for $N > T_2$.

Moreover, for $N > T_2$, we have: $|d_1(\hat{\mu}_1, \hat{x}_{1,a})| = O(1)$, $|d_1(\hat{\mu}_a, \hat{x}_{1,a})| = O(1)$ (using (7)) ; $|d_2(\hat{\mu}_1, \hat{x}_{1,a})| = O(1)$, $|d_2(\hat{\mu}_a, \hat{x}_{1,a})| = O(1)$ (using (8)) ; and $d(\hat{\mu}_1, \hat{x}_{1,a}) = O(1)$ (using (6)). As a result, for $N > T_2$, all the partial derivatives in (57) are $O(1)$. Therefore, for $N > T_2$,

$$|g(\widetilde{\boldsymbol{\mu}}(N), \widetilde{\boldsymbol{N}}(N)) - g(\boldsymbol{\mu}, \widetilde{\boldsymbol{N}}(N))| = O\left(\sum_{a \in [K]} |\widetilde{\mu}_a(N) - \widetilde{\mu}_a|\right) = O(N^{-3\alpha/8}), \quad (58)$$

and hence completing the proof. $\square$

**Halting of exploration:** By Lemma G.13, for $N \geq T_4$, every arm $a \in [K]$ has $\widetilde{N}_a(N) = \Theta(N)$. As a result, we can find a constant $\lambda \in (0, 1)$ such that $\widetilde{N}_a(N) \geq \lambda N$ for every $a \in [K]$ and $N \geq T_4$. We choose $M_3$ large enough such that, for every $N \geq M_3$, $\lambda(N - 1) > N^\alpha$. Then we have $\min_{a \in [K]/\{1\}} \widetilde{N}_a(N - 1) > N^\alpha$ for every $N \geq \max\{M_3, T_4 + 1\}$. As a result, the algorithm doesn't do any exploration after iteration $\max\{M_3, T_4 + 1\}$. With this, we define the following random time,

**Definition G.1.** *We define $T_5 = \max\{M_3, T_4 + 1\}$.*

Note that $\mathbb{E}_{\boldsymbol{\mu}}[T_5] < \infty$, since, $\mathbb{E}_{\boldsymbol{\mu}}[T_4] < \infty$.

We restate Proposition G.1 below,

**Statement of Proposition G.1.** *There exists constants $M_4 \geq 1$ and $C > 0$ independent of the sample paths, such that, if $T_6$ denotes the iteration at which $g(\widetilde{\boldsymbol{\mu}}(\cdot), \widetilde{\boldsymbol{N}}(\cdot))$ crosses the value zero after iteration $\max\{M_4, T_5\}$, then for $N \geq T_6$ we have,*

$$\left| g(\boldsymbol{\mu}, \widetilde{\boldsymbol{N}}(N)) \right| \leq CN^{-3\alpha/8}. \tag{59}$$

*Moreover, the random time $T_6$ satisfies $\mathbb{E}_{\boldsymbol{\mu}}[T_6] < \infty$.*

*Proof.* We prove the proposition via an inductive argument consisting of two main steps,

1. **Initialization:** We start with a choice of the constants $C > 0$ and $M_4 \geq 1$ and show that $g(\boldsymbol{\mu}, \widetilde{\boldsymbol{N}}(\cdot))$ satisfies (59) at iteration $T_6$.

2. **Induction:** We show that, for every $N \geq T_6$, $\left| g(\boldsymbol{\mu}, \widetilde{\boldsymbol{N}}(N)) \right| \leq CN^{-3\alpha/8}$ implies $\left| g(\boldsymbol{\mu}, \widetilde{\boldsymbol{N}}(N+1)) \right| \leq C(N+1)^{-3\alpha/8}$.

By Lemma G.1, we have a constant $C_1 > 0$ independent of the sample path, such that,

$$\left| g(\widetilde{\boldsymbol{\mu}}(N), \widetilde{\boldsymbol{N}}(N)) - g(\widetilde{\boldsymbol{\mu}}, \widetilde{\boldsymbol{N}}(N)) \right| \leq C_1 N^{-3\alpha/8}, \quad \text{for } N \geq T_5. \tag{60}$$

By Lemma G.13, we have $\widetilde{N}_a(N) = \Theta(N)$ for every $a \in [K]$ and $N \geq T_5$. As a result, by (15), we have constants $C_2, C_2' > 0$ independent of the sample paths, such that: for all $N \geq T_5$, and $\hat{N}_a \in \left[ \widetilde{N}_a(N-1), \widetilde{N}_a(N) \right]$,

$$-C_2' N^{-1} \leq \frac{\partial g}{\partial N_1}(\boldsymbol{\mu}, \hat{\boldsymbol{N}}) \leq -C_2 N^{-1}, \quad \text{and}$$

$$\text{for } a \in [K]/\{1\}, \quad C_2' N^{-1} \geq \frac{\partial g}{\partial N_a}(\boldsymbol{\mu}, \hat{\boldsymbol{N}}) \geq C_2 N^{-1}, \tag{61}$$

where $\hat{\boldsymbol{N}} = (\hat{N}_a : a \in [K])$.

We use the constants $C_1, C_2$, and $C_2'$ as defined above in the rest of our proof.

**Initialization:** We choose $C = 4C_1 + C_2'$ and $M_4 = \max\{M_{41}, M_{42}, M_{43}, M_{44}\}$, where $M_{41}, M_{42}, M_{43}, M_{44}$ are defined as,

1. $M_{41} \geq 1$ is the smallest number such that, for every $N \geq M_{41}$ we have $2C_1 N^{-3\alpha/8} > C_1(N-1)^{-3\alpha/8}$,

2. $M_{42} \geq 1$ is the smallest number such that, for every $N \geq M_{42}$ we have $C(N+1)^{-3\alpha/8} \geq (C_1 + C_2')N^{-3\alpha/8}$,

3. $M_{43} \geq 1$ is the smallest number such that, for every $N \geq M_{43}$ we have $C(N+1)^{-3\alpha/8} \geq C_2' N^{-1}$, and

4. $M_{44} \geq 1$ is the smallest number such that, for every $N \geq M_{44}$ we have $\frac{3C\alpha}{8}(N+1)^{-(1+\frac{3\alpha}{8})} < C_2 N^{-1}$.

By definition of $T_6$, $g(\widetilde{\boldsymbol{\mu}}(\cdot), \widetilde{\boldsymbol{N}}(\cdot))$ has opposite signs at iterations $T_6 - 1$ and $T_6$. Therefore,

$$\begin{aligned}
\left| g(\widetilde{\boldsymbol{\mu}}(T_6), \widetilde{\boldsymbol{N}}(T_6)) \right| &\leq \left| g(\widetilde{\boldsymbol{\mu}}(T_6), \widetilde{\boldsymbol{N}}(T_6)) - g(\widetilde{\boldsymbol{\mu}}(T_6 - 1), \widetilde{\boldsymbol{N}}(T_6 - 1)) \right| \\
&\leq \left| g(\boldsymbol{\mu}, \widetilde{\boldsymbol{N}}(T_6)) - g(\boldsymbol{\mu}, \widetilde{\boldsymbol{N}}(T_6 - 1)) \right| + C_1 T_6^{-3\alpha/8} + C_1(T_6 - 1)^{-3\alpha/8} \quad \text{(using (60))} \\
&\leq \left| g(\boldsymbol{\mu}, \widetilde{\boldsymbol{N}}(T_6)) - g(\boldsymbol{\mu}, \widetilde{\boldsymbol{N}}(T_6 - 1)) \right| + 3C_1 T_6^{-3\alpha/8} \quad \text{(using the definition of $M_{41}$)}.
\end{aligned} \tag{62}$$

Let $a \in [K]$ be the arm pulled at iteration $T_6$. Applying the mean value theorem we can find $\hat{N}_a$ between $\widetilde{N}_a(T_6 - 1)$ and $\widetilde{N}_a(T_6)$, can take $\hat{N}_b = \widetilde{N}_b(T_6)$ for all $b \neq a$, and define the tuple $\hat{\boldsymbol{N}} = (\hat{N}_b)_{b \in [K]}$, such that, (62) is bounded by,

$$\left| \frac{\partial g}{\partial N_a}(\boldsymbol{\mu}, \hat{\boldsymbol{N}}) \right| + 3C_1 T_6^{-3\alpha/8}.$$

Using (60) and the above upper bound, we have,

$$\left| g(\boldsymbol{\mu}, \widetilde{\boldsymbol{N}}(T_6)) \right| \leq \left| g(\boldsymbol{\mu}, \widetilde{\boldsymbol{N}}(T_6)) - g(\widetilde{\boldsymbol{\mu}}(T_6), \widetilde{\boldsymbol{N}}(T_6)) \right| + \left| g(\widetilde{\boldsymbol{\mu}}(T_6), \widetilde{\boldsymbol{N}}(T_6)) \right|$$

$$\leq C_1 T_6^{-3\alpha/8} + \left| \frac{\partial g}{\partial N_a}(\boldsymbol{\mu}, \hat{\boldsymbol{N}}) \right| + 3C_1 T_6^{-3\alpha/8}$$

$$\leq 4C_1 T_6^{-3\alpha/8} + C_2' T_6^{-1} \quad \text{(using (61))}$$

$$\leq (4C_1 + C_2') T_6^{-3\alpha/8} = C T_6^{-3\alpha/8}.$$

**Induction:** Note that at a given iteration $N$ the algorithm can only see $g(\widetilde{\boldsymbol{\mu}}(N), \widetilde{\boldsymbol{N}}(N))$. By (60), for $N \geq T_6$, $g(\widetilde{\boldsymbol{\mu}}, \widetilde{\boldsymbol{N}})$ and $g(\boldsymbol{\mu}, \widetilde{\boldsymbol{N}})$ may have different signs only when $\left| g(\boldsymbol{\mu}, \widetilde{\boldsymbol{N}}(N)) \right| \leq C_1 N^{-3\alpha/8}$. Based on this, we consider two cases.

**Case I:** $|g(\boldsymbol{\mu}, \widetilde{\boldsymbol{N}}(N))| \leq C_1 N^{-3\alpha/8}$: We assume $a \in [K]$ to be the arm pulled in iteration $N + 1$. Using the mean value theorem, we can find $\hat{N}_a \in \left[ \widetilde{N}_a(N), \widetilde{N}_a(N+1) \right]$, can take $\hat{N}_b = \widetilde{N}_b(N)$ for all $b \neq a$, and define the tuple $\hat{\boldsymbol{N}} = (\hat{N}_b)_{b \in [K]}$, such that,

$$\left| g(\boldsymbol{\mu}, \widetilde{\boldsymbol{N}}(N+1)) \right| \leq \left| g(\boldsymbol{\mu}, \widetilde{\boldsymbol{N}}(N)) \right| + \left| \frac{\partial g}{\partial N_a}(\boldsymbol{\mu}, \hat{\boldsymbol{N}}) \right|$$

$$\overset{(1)}{\leq} C_1 N^{-3\alpha/8} + C_2' N^{-1} \leq (C_1 + C_2') N^{-3\alpha/8},$$

where (1) follows from (61).

Note that $N \geq T_6 \geq M_4 \geq M_{42}$. By the definition of $M_{42}$, we have

$$\left| g(\boldsymbol{\mu}, \widetilde{\boldsymbol{N}}(N+1)) \right| \leq (C_1 + C_2') N^{-3\alpha/8} \leq (4C_1 + C_2')(N+1)^{-3\alpha/8} = C(N+1)^{-3\alpha/8},$$

for every $N \geq T_6$.

**Case II:** $|g(\boldsymbol{\mu}, \widetilde{\boldsymbol{N}}(N))| > C_1 N^{-3\alpha/8}$: In this case $g(\widetilde{\boldsymbol{\mu}}(N), \widetilde{\boldsymbol{N}}(N))$ and $g(\boldsymbol{\mu}, \widetilde{\boldsymbol{N}}(N))$ have the same sign. Let arm $a$ has been sampled from in iteration $N + 1$. Using the mean value theorem, we have $\hat{N}_a \in \left[ \widetilde{N}_a(N), \widetilde{N}_a(N+1) \right]$, can take $\hat{N}_b = \widetilde{N}_b(N)$ for all $b \neq a$, and define the tuple $\hat{\boldsymbol{N}} = (\hat{N}_b)_{b \in [K]}$, such that,

$$g(\boldsymbol{\mu}, \widetilde{\boldsymbol{N}}(N+1)) = g(\boldsymbol{\mu}, \widetilde{\boldsymbol{N}}(N)) + \frac{\partial g}{\partial N_a}(\boldsymbol{\mu}, \hat{\boldsymbol{N}}). \tag{63}$$

We first consider the case when $g(\boldsymbol{\mu}, \widetilde{\boldsymbol{N}}(N)) > 0$. After the algorithm sees $g(\widetilde{\boldsymbol{\mu}}(N), \widetilde{\boldsymbol{N}}(N)) > 0$, it pulls the first arm. As a result, by (61) and (63), $g(\boldsymbol{\mu}, \widetilde{\boldsymbol{N}}(\cdot))$ decreases in iteration $N + 1$ atmost by $C_2' N^{-1}$ and atleast by $C_2 N^{-1}$. Now there can be two possibilities:

1. If $g(\boldsymbol{\mu}, \widetilde{\boldsymbol{N}}(N + 1)) < 0$, we must have $g(\boldsymbol{\mu}, \widetilde{\boldsymbol{N}}(N + 1)) \geq -C_2' N^{-1}$. Since $N \geq T_6 \geq M_4 \geq M_{43}$, we have $C(N+1)^{-3\alpha/8} \geq C_2' N^{-1}$ by the definition of $M_{43}$. As a result,

$$g(\boldsymbol{\mu}, \widetilde{\boldsymbol{N}}(N + 1)) \geq -C_2' N^{-1} \geq -C(N+1)^{-3\alpha/8}.$$

2. If $g(\boldsymbol{\mu}, \widetilde{\boldsymbol{N}}(N+1)) \geq 0$, then $g(\boldsymbol{\mu}, \widetilde{\boldsymbol{N}}(\cdot))$ has moved towards zero by atleast $C_2 N^{-1}$. Whereas, by iteration $N + 1$, the interval $\left[ -CN^{-3\alpha/8}, CN^{3\alpha/8} \right]$ has reduced from both ends by

$$CN^{-3\alpha/8} - C(N+1)^{-3\alpha/8} \leq \frac{3C\alpha}{8} N^{-(1+\frac{3\alpha}{8})}.$$

Since $N \geq T_6 \geq M_4 \geq M_{44}$, by the definition of $M_{44}$, we have $\frac{3C\alpha}{8}(N+1)^{-(1+\frac{3\alpha}{8})} < C_2 N^{-1}$ for every $N \geq T_6$. As a result, we can ensure $g(\boldsymbol{\mu}, \widetilde{\boldsymbol{N}}(N+1)) \leq C(N+1)^{-3\alpha/8}$ at iteration $N + 1$.

In the other case, when $g(\boldsymbol{\mu}, \widetilde{\boldsymbol{N}}(N)) < 0$, the algorithm sees $g(\widetilde{\boldsymbol{\mu}}(N), \widetilde{\boldsymbol{N}}(N)) < 0$, and hence pulls some arm $a \in [K]/\{1\}$. As a result, by (61) and (63), $g(\boldsymbol{\mu}, \widetilde{\boldsymbol{N}}(\cdot))$ increases in iteration $N + 1$ atmost by $C_2' N^{-1}$ and atleast by $C_2 N^{-1}$. Then we apply the same argument as for the case $g(\boldsymbol{\mu}, \widetilde{\boldsymbol{N}}(N)) > 0$, but by reversing the signs. Therefore, the inductive statement holds true for this case as well. Hence (59) stands proved.

$T_6$ **has finite expectation:** By Lemma G.9, we can have a constant $C_3 > 0$, such that $T_6 \leq C_3 \max\{M_4, T_5\}$. As a result, since $\mathbb{E}_{\boldsymbol{\mu}}[T_5] < \infty$, we have $\mathbb{E}_{\boldsymbol{\mu}}[T_6] \leq C_1(M_4 + \mathbb{E}_{\boldsymbol{\mu}}[T_5]) < \infty$. $\square$

### G.1.2  Closeness of the indexes

Lemma G.2 is a detailed version of Lemma 5.1 mentioned in Section 5, and is essential for proving closeness of the indexes under the allocations made by AT2 and IAT2 algorithms. Recall that $T_6$ is the random time defined in Proposition G.1 and satisfies $\mathbb{E}_{\boldsymbol{\mu}}[T_6] < \infty$.

**Lemma G.2.** *For both AT2 and IAT2 algorithms, there exists constants $M_5 \geq 1$ and $C_1 > 0$ independent of the sample paths, such that, for every $N \geq \max\{M_5, T_6\}$, if the algorithm picks an arm $a \in [K]/\{1\}$ at iteration $N$, then it again picks arm $a$ within the next $\lceil C_1 N^{1-3\alpha/8}\rceil$ iterations.*

Proof of Lemma G.2 is in Appendix G.6, and requires proving several technical lemmas. Some of those supporting lemmas involve arguments similar to the ones used for proving closeness of the indexes while the algorithm operates under an idealized fluid model (discussed in Section 4). In the rest of this section, we use Lemma G.2 to prove closeness of indexes for alternative arms in Proposition G.2.

**Definition G.2.** *We define the random time $T_7 = \max\{M_5, T_6\}$.*

Note that $\mathbb{E}_{\boldsymbol{\mu}}[T_7] < \infty$, since $\mathbb{E}_{\boldsymbol{\mu}}[T_6] < \infty$.

**Definition G.3.** *For every $M \geq 1$, define $T_{7,M} = \max\{M, T_7\}$, and $T_{8,M}$ as the smallest iteration after $T_{7,M}$ by which all the alternative arms in $[K]/\{1\}$ have been picked atleast once by the algorithm.*

Below we state a detailed version of Proposition G.2.

**Proposition G.3.** *For every $M \geq 1$, we have $\mathbb{E}_{\boldsymbol{\mu}}[T_{7,M}] < \infty$ and $\mathbb{E}_{\boldsymbol{\mu}}[T_{8,M}] < \infty$. Moreover, for every $M \geq 1$ and $N \geq T_{8,M}$, every pair of arms $a, b \in [K]/\{1\}$ satisfy,*

$$|I_a(N) - I_b(N)| = O(N^{1-3\alpha/8}),$$

*where the constant hidden in $O(\cdot)$ is independent of $M$ and the sample path after $T_7$.*

**Definition G.4.** *We define $T_8 = T_{8,1}$, where $T_{8,1}$ is defined according to Proposition G.3.*

By the defintion of $T_8$ above, Proposition G.2 follows trivially from Proposition G.3.

The following lemma helps us to bound the deviation of the empirical index $\mathcal{I}_a(N)$ from the index $I_a(N)$ due to the noise in the estimates $\widetilde{\boldsymbol{\mu}}$, for every alternative arm $a \in [K]/\{1\}$.

**Lemma G.3.** *For $a \in [K]/\{1\}$ and $N \geq T_0$, we have,*

$$|\mathcal{I}_a(N) - I_a(N)| = O(N^{1-3\alpha/8}).$$

*Proof.* Proof of this lemma uses mean value theorem. For any arm $a \in [K]/\{1\}$, upon expanding the indexes,

$$|\mathcal{I}_a(N) - I_a(N)| \leq \widetilde{N}_1 \cdot |d(\widetilde{\mu}_1, \widetilde{x}_{1,a}) - d(\mu_1, x_{1,a})| + \widetilde{N}_a \cdot |d(\widetilde{\mu}_a, \widetilde{x}_{1,a}) - d(\mu_a, x_{1,a})|, \quad (64)$$

where $\widetilde{N}_1, \widetilde{N}_a, \widetilde{\mu}_1, \widetilde{\mu}_a$, and $\widetilde{x}_{1,a}$ are evaluated at $N$. Since $\widetilde{N}_1, \widetilde{N}_a \leq N$, the difference (64) is bounded above by,

$$|\mathcal{I}_a(N) - I_a(N)| \leq N \cdot (|d(\widetilde{\mu}_1, \widetilde{x}_{1,a}) - d(\mu_1, x_{1,a})| + |d(\widetilde{\mu}_a, \widetilde{x}_{1,a}) - d(\mu_a, x_{1,a})|).$$

Now considering the first term in the RHS, and applying mean value theorem, we get

$$|d(\widetilde{\mu}_1, \widetilde{x}_{1,a}) - d(\mu_1, x_{1,a})| = \left| d_1(\hat{\mu}_1, \hat{x}_{1,a}) + d_2(\hat{\mu}_1, \hat{x}_{1,a}) \cdot \frac{\widetilde{N}_1}{\widetilde{N}_1 + \widetilde{N}_a} \right| \cdot |\widetilde{\mu}_1 - \mu_1|$$

$$+ \left| d_2(\hat{\mu}_1, \hat{x}_{1,a}) \cdot \frac{\widetilde{N}_a}{\widetilde{N}_1 + \widetilde{N}_a} \right| \cdot |\widetilde{\mu}_a - \mu_a|,$$

where $\hat{\mu}_1, \hat{\mu}_a$, respectively, lie between $\widetilde{\mu}_1, \mu_1$, and $\widetilde{\mu}_a, \mu_a$, and $\hat{x}_{1,a} = \frac{\widetilde{N}_1 \hat{\mu}_1 + \widetilde{N}_a \hat{\mu}_a}{\widetilde{N}_1 + \widetilde{N}_a}$. Using (7) and (8), all the partial derivatives in the above upper bound are $O(1)$ for $N \geq T_0$. Therefore,

$$|d(\widetilde{\mu}_1, \widetilde{x}_{1,a}) - d(\mu_1, x_{1,a})| = O(|\widetilde{\mu}_1 - \mu_1| + |\widetilde{\mu}_a - \mu_a|)$$
$$= O(N^{-3\alpha/8}). \tag{65}$$

Following a similar procedure, we can argue using (7) and (8), that the partial derivatives of $d(\widetilde{\mu}_j, \widetilde{x}_{1,j})$ with respect to $\widetilde{\mu}_1$ and $\widetilde{\mu}_j$ are $O(1)$ in magnitude. As a result, using the mean value theorem,

$$|d(\widetilde{\mu}_a, \widetilde{x}_{1,a}) - d(\mu_a, x_{1,a})| = O(N^{-3\alpha/8}). \tag{66}$$

Therefore, we have,

$$|\mathcal{I}_a(N) - I_a(N)| = O(N^{1-3\alpha/8}),$$

for $N \geq T_0$ and completing the proof. $\qquad\square$

**Proof of Proposition G.2:** We have $\mathbb{E}_{\boldsymbol{\mu}}[T_{7,M}] \leq M + \mathbb{E}_{\boldsymbol{\mu}}[T_7] < \infty$. By Lemma G.13, $\widetilde{N}_a(N) = \Theta(N)$ for $N \geq T_{7,M}$. Hence, by the definition of $T_{8,M}$, there exists a constant $C' > 0$ independent of $M$, such that, for every $M \geq 1$, $T_{8,M} \leq C'T_{7,M}$. As a result, $\mathbb{E}_{\boldsymbol{\mu}}[T_{8,M}] \leq C'\mathbb{E}_{\boldsymbol{\mu}}[T_{7,M}] < \infty$.

Note that $T_{7,1} = T_7$. Also, for every $M \geq 1$, $T_{8,M} \geq T_{8,1} = T_8$ ($T_8$ is defined in Definition G.4). It is sufficient to prove the proposition for every $N \geq T_8$.

We now argue for the algorithms AT2 and IAT2 separately.

**AT2:** We consider any two alternative arms $a, b \in [K]/\{1\}$, and define the time $\tau_{a,b}(N)$ as,

$$\tau_{a,b}(N) = \min\left\{ t \geq 1 \;\middle|\; \mathcal{I}_b(N+t) - \mathcal{I}_a(N+t) \text{ and } \mathcal{I}_b(N) - \mathcal{I}_a(N) \text{ have opposite signs} \right\}.$$

Note that $N + \tau_{a,b}(N)$ must be before the iteration after $N$ by which the algorithm has picked both $a$ and $b$ atleast once. By the definition of $T_7$ and $T_8$, for every $N \geq T_8$, all alternative arms in $[K]/\{1\}$ has been sampled from atleast once between iterations $T_7$ and $N$. Therefore, by Lemma G.2, we have $\tau_{a,b}(N) = O(N^{1-3\alpha/8})$.

Since $\mathcal{I}_a(N) - \mathcal{I}_b(N)$ and $\mathcal{I}_a(N + \tau_{a,b}(N)) - \mathcal{I}_b(N + \tau_{a,b}(N))$ have opposite signs, we have,

$$|\mathcal{I}_a(N) - \mathcal{I}_b(N)| \leq |(\mathcal{I}_a(N) - \mathcal{I}_b(N)) - (\mathcal{I}_a(N + \tau_{a,b}(N)) - \mathcal{I}_b(N + \tau_{a,b}(N)))|$$
$$\leq |\mathcal{I}_a(N + \tau_{a,b}(N)) - \mathcal{I}_a(N)| + |\mathcal{I}_b(N + \tau_{a,b}(N)) - \mathcal{I}_b(N)|$$
$$\leq |I_a(N + \tau_{a,b}(N)) - I_a(N)| + |I_b(N + \tau_{a,b}(N)) - I_b(N)|$$
$$+ O((N + \tau_{a,b}(N))^{1-3\alpha/8}),$$

where the last step follows from Lemma G.3. Now,

$$O\left((N + \tau_{a,b}(N))^{1-3\alpha/8}\right) = O\left(\left(N + O(N^{1-3\alpha/8})\right)^{1-3\alpha/8}\right) = O(N^{1-3\alpha/8}).$$

Therefore,

$$|\mathcal{I}_a(N) - \mathcal{I}_b(N)| \leq |I_a(N + \tau_{a,b}(N)) - I_a(N)| + |I_b(N + \tau_{a,b}(N)) - I_b(N)| + O(N^{1-3\alpha/8}).$$

By Lemma G.3, we know $|I_a(N) - I_b(N)| \leq |\mathcal{I}_a(N) - \mathcal{I}_b(N)| + O(N^{1-3\alpha/8})$. Therefore, the above inequality implies,

$$\begin{aligned} |I_a(N) - I_b(N)| &\leq |\mathcal{I}_a(N) - \mathcal{I}_b(N)| + O(N^{1-3\alpha/8}) \\ &\leq |I_a(N + \tau_{a,b}(N)) - I_a(N)| + |I_b(N + \tau_{a,b}(N)) - I_b(N)| \\ &\quad + O(N^{1-3\alpha/8}). \end{aligned} \tag{67}$$

Using mean value theorem, for $j \in \{a, b\}$, we have,

$$|I_j(N + \tau_{a,b}(N)) - I_j(N)| \leq \left( \sum_{i \in \{1,j\}} \frac{\partial I_j}{\partial N_i}(\hat{N}_1, \hat{N}_j) \right) \cdot \tau_{a,b}(N), \tag{68}$$

where $\hat{N}_i \in \left[ \widetilde{N}_i(N), \ \widetilde{N}_i(N + \tau_{a,b}(N)) \right]$ for $i = 1, a, b$.

We know,

$$\frac{\partial I_j}{\partial N_1}(\hat{N}_1, \hat{N}_j) = d(\mu_1, \hat{x}_{1,j}) \quad \text{and} \quad \frac{\partial I_j}{\partial N_j}(\hat{N}_1, \hat{N}_j) = d(\mu_j, \hat{x}_{1,j}),$$

where $\hat{x}_{1,j} = \frac{\hat{N}_1\mu_1 + \hat{N}_j\mu_j}{\hat{N}_1 + \hat{N}_j}$. Note that both the partial derivatives above are bounded from above by $\max\{d(\mu_1, \mu_a), d(\mu_a, \mu_1)\}$, and therefore $O(1)$. As a result, since $\tau_{a,b}(N) = O(N^{1-3\alpha/8})$, we have,

$$|I_j(N + \tau_{a,b}(N)) - I_j(N)| \leq O(N^{1-3\alpha/8}) \quad \text{for } j = a, b. \tag{69}$$

Using (69) in (67), we get

$$|I_a(N) - I_b(N)| = O(N^{1-3\alpha/8}), \quad \text{for } N \geq T_8.$$

**IAT2:** First we define the modified empirical index of every alternative arm $a \in [K]/\{1\}$ using the notation $\mathcal{I}_a^{(m)}(N)$ as,

$$\mathcal{I}_a^{(m)}(N) = \mathcal{I}_a(N) + \log(\widetilde{N}_a(N)).$$

We define the time $\tau_{a,b}^{(m)}(N)$ as,

$$\begin{aligned} \tau_{a,b}^{(m)}(N) = \min \Big\{ t \geq 1 \ \Big| \ &\mathcal{I}_b^{(m)}(N + t) - \mathcal{I}_a^{(m)}(N + t) \quad \text{and} \\ &\mathcal{I}_b^{(m)}(N) - \mathcal{I}_a^{(m)}(N) \quad \text{have opposite signs} \Big\}. \end{aligned}$$

Note that, for every $a \in [K]/\{1\}, \mathcal{I}_a^{(m)}(N)$ differs from $\mathcal{I}_a(N)$ by atmost $\log(N)$ and $\mathcal{I}_a(N)$ differs from $I_a(N)$ by atmost $O(N^{1-3\alpha/8})$ for $N \geq T_0$. Therefore,

$$|\mathcal{I}_a^{(m)}(N) - I_a(N)| = O(N^{1-3\alpha/8}) \quad \text{for } N \geq T_0 \text{ and every } a \in [K]/\{1\}.$$

Now $N + \tau_{a,b}^{(m)}(N)$ must be earlier than the iteration after $N$ by which the algorithm has picked both $a$ and $b$ atleast once. Using the same argument as AT2, by Lemma G.2, we have $\tau_{a,b}^{(m)}(N) = O(N^{1-3\alpha/8})$. Also, following the same steps as AT2, by replacing the empirical index $\mathcal{I}$ with the modified empirical index $\mathcal{I}^{(m)}$ for every alternative arm, we obtain,

$$|I_a(N) - I_b(N)| \leq |I_a(N + \tau_{a,b}^{(m)}(N)) - I_a(N)| + |I_b(N + \tau_{a,b}^{(m)}(N)) - I_b(N)| + O(N^{1-3\alpha/8}).$$

Using the mean value theorem, since the parital derivatives of $I_a$ and $I_b$ with respect to $\widetilde{N}_1, \widetilde{N}_a$ and $\widetilde{N}_b$ are $O(1)$, we have

$$|I_j(N + \tau_{a,b}^{(m)}(N)) - I_j(N)| \leq O\left(\tau_{a,b}^{(m)}(N)\right) = O(N^{1-3\alpha/8}) \quad \text{for } j = a, b.$$

From the last two observations, we conclude

$$|I_a(N) - I_b(N)| \leq O(N^{1-3\alpha/8}) \quad \text{for } N \geq T_8.$$

$\square$

## G.2   Convergence of algorithm to optimal proportions

In this appendix we prove a slightly detailed version of Proposition 3.1 from Section 3. In Proposition 3.1, we argue that the proportion of samples allocated by the algorithm converges to the optimal proportions for the instance, a.s. in $\mathbb{P}_{\boldsymbol{\mu}}$, as the no. of samples grows to $\infty$.

For every $M \geq 1$, we use $T_{7,M}$ and $T_{8,M}$ as defined in Definition G.3 in Appendix G.1.2. Recall that $T_7 = T_{7,1}$ and $T_8 = T_{8,1}$. By Proposition G.3, we have $\mathbb{E}_{\boldsymbol{\mu}}[T_{8,M}] < \infty$, and $T_{8,M} \geq T_8$ for every $M \geq 1$.

Recall that $\boldsymbol{\omega}^\star$ is the unique optimal allocation according to Proposition 2.2, and $\widetilde{\boldsymbol{\omega}}(N) = (\widetilde{\omega}_a(N) : a \in [K])$ with $\widetilde{\omega}_a(N) = \frac{\widetilde{N}_a(N)}{N}$ is the algorithms allocation at iteration $N$. We now state a slightly detailed version of Proposition 3.1 from Section 3,

**Proposition G.4.** *There exists constants $C_1 > 0$ and $M_6 \geq 1$ depending on $\boldsymbol{\mu}, \alpha$, and $K$ such that, for every $N \geq T_{8,M_6}$ and $a \in [K]$, we have*

$$|\widetilde{\omega}_a(N) - \omega_a^\star| \leq C_1 N^{-3\alpha/8} \quad \text{and} \quad |\widetilde{\mu}_a(N) - \mu_a| \leq \epsilon(\boldsymbol{\mu}) N^{-3\alpha/8},$$

*where $\epsilon(\boldsymbol{\mu})$ is a constant depending only on $\boldsymbol{\mu}$ and defined in Appendix B.*

Detailed proof of Proposition G.4 is in Appendix G.2.2 and relies on using IFT.

Below we define the random times $T_{good}$ and $T_{stable}$, which are mentioned in the statements of Proposition 3.1, 5.1, and Lemma 5.1 from the main body of the paper.

**Definition G.5** ($T_{stable}$ **and** $T_{good}$)**.** *We define $T_{good} = T_{7,M_6}$ and $T_{stable} = T_{8,M_6}$, where $M_6 \geq 1$ is introduced in Proposition G.4.*

**Remark G.1.** *Note that, by definition, $T_{good} \geq T_4, T_6$. As a result, by Proposition G.1 and Lemma G.13, $\left|g(\boldsymbol{\mu}, \widetilde{\boldsymbol{N}}(N))\right| = O(N^{-3\alpha/8})$ and $\widetilde{N}_j(N) = \Theta(N)$ for every $j \in [K]$ and $N \geq T_{good}$.*

**Proof of Lemma 5.1:**   By the definition of $T_{7,M}$, $T_{8,M}$ in Appendix G.1.2, and since $T_{good} = T_{7,M_6}$, $T_{stable} = T_{8,M_6}$, every alternative arm in $[K]/\{1\}$ gets picked atleast once between the iterations $T_{good}$ and $T_{stable}$. The other part of the statement of Lemma 5.1 follows from Lemma G.2 because $T_{good} \geq T_7$. $\square$

Before proving Proposition G.4 in Appendix G.2.2, we find a tighter upper bound on the time to reach optimal proportion $T_{stable}$ in the following Appendix G.2.1. While doing this, we identify a similarity between the time to reach stabilty in fluid dynamics and that for the algorithm.

### G.2.1   Bounding time to reach stability

Lemma G.4 gives an upper bound on the time to reach stability for the algorithmic allocations. We define $\omega_{\min}^\star = \min_{a \in [K]/\{1\}} \omega_a^\star$.

According to the discussion in Section 4, if the fluid dynamics has state $\widetilde{\boldsymbol{N}}(T_{good})$ at time $T_{good}$, then it hits all the indexes and reaches stability by a time atmost $\frac{T_{good}}{\omega_{\min}^\star}$. In Lemma G.4, we argue that, the algorithm also approximately reaches the optimal proportion $\boldsymbol{\omega}^\star$ by atmost $\approx \frac{T_{good}}{\omega_{\min}^\star}$ iterations.

**Lemma G.4.** *For every $M \geq M_6$ ($M_6$ is a constant defined in the statement of Proposition G.4),*

$$T_{8,M} \leq \frac{T_{7,M} + 1}{\omega_{\min}^\star - C_1 M^{-3\alpha/8}},$$

*which implies*

$$T_{stable} \leq \frac{T_{good} + 1}{\omega_{\min}^\star - C_1 M_6^{-3\alpha/8}}.$$

*Moreover, we have*

$$\limsup_{M \to \infty} \frac{T_{8,M}}{T_{7,M}} \leq \frac{1}{\omega_{\min}^\star} \ a.s. \ in \ \mathbb{P}_{\boldsymbol{\mu}}.$$

*Proof.* From Proposition G.4, it follows that, for every $M \geq M_6$ and $N \geq T_{8,M} \geq T_{8,M_6}$, we have,

$$\max_{a \in [K]} |\widetilde{\omega}_a(N) - \omega_a^\star| \leq C_1 N^{-3\alpha/8}. \tag{70}$$

Since $T_{8,M}$ is the first iteration after $T_{7,M}$ by which every alternative arm has been picked atleast once, we have some arm $a \in [K]/\{1\}$ such that,

$$\widetilde{N}_a(T_{8,M}) = \widetilde{N}_a(T_{7,M}) + 1 \leq T_{7,M} + 1.$$

Now by (70), we have

$$\widetilde{N}_a(T_{8,M}) \geq (\omega_a^\star - C_1 T_{8,M}^{-3\alpha/8}) T_{8,M} \geq (\omega_{\min}^\star - C_1 M^{-3\alpha/8}) T_{8,M}.$$

Combining the last two observation, we have,

$$T_{8,M} \leq \frac{T_{7,M} + 1}{\omega_{\min}^\star - C_1 M^{-3\alpha/8}} \quad \text{a.s. in } \mathbb{P}_{\boldsymbol{\mu}}$$

for every $M \geq M_6$.

Since $T_{8,M}, T_{7,M} \to \infty$ as $M \to \infty$ a.s. in $\mathbb{P}_{\boldsymbol{\mu}}$, we have,

$$\limsup_{M \to \infty} \frac{T_{8,M}}{T_{7,M}} \leq \frac{1}{\omega_{\min}^\star} \quad \text{a.s. in } \mathbb{P}_{\boldsymbol{\mu}}.$$

$\square$

### G.2.2 Proving Proposition G.4

By Proposition G.1 and G.3, there exists a constant $C > 0$ independent of the sample paths, such that $\widetilde{\boldsymbol{\omega}}(N) = (\widetilde{\omega}_a(N))_{a \in [K]}$ satisfies,

$$|g(\boldsymbol{\mu}, \widetilde{\boldsymbol{\omega}}(N))| = \left| \sum_{a \in [K]/\{1\}} \frac{d(\mu_1, x_{1,a}(N))}{d(\mu_a, x_{1,a}(N))} - 1 \right| \leq CN^{-3\alpha/8}, \quad \text{and}$$

$$\max_{a,b \in [K]/\{1\}} |I_a(N) - I_b(N)| \leq CN^{-3\alpha/8}, \tag{71}$$

for all $N \geq T_8$ a.s. in $\mathbb{P}_{\boldsymbol{\mu}}$.

Proof of Proposition G.4 relies on using the implicit function theorem. Before proving the proposition, we describe below the framework over which we apply the implicit function theorem. We define the following functions,

$$\Psi_1(\boldsymbol{\omega}, \boldsymbol{\eta}) = g(\boldsymbol{\mu}, \boldsymbol{\omega}) - \eta_1 = \sum_{a \neq 1} \frac{d(\mu_1, x_{1,a}(\omega_1, \omega_a))}{d(\mu_a, x_{1,a}(\omega_1, \omega_a))} - 1 - \eta_1,$$

for $a \in [K]/\{1\}$, $\quad \Psi_a(\boldsymbol{\omega}, I, \boldsymbol{\eta}) = W_a(\omega_1, \omega_a) - I - \eta_a$, and

$$\Psi_{K+1}(\boldsymbol{\omega}) = \sum_{a \in [K]} \omega_a - 1,$$

where $\boldsymbol{\omega} = (\omega_a)_{a\in[K]} \in \mathbb{R}_{\geq 0}^K$, $\quad \boldsymbol{\eta} = (\eta_a)_{a\in[K]} \in \mathbb{R}^K$, $\quad I \in \mathbb{R}$, $\quad$ and for every $a \in [K]/\{1\}$, $\quad x_{1,a}(\omega_1,\omega_a) = \frac{\omega_1\mu_1 + \omega_a\mu_a}{\omega_1+\omega_a}$ and $W_a(\omega_1,\omega_a) = \omega_1 d(\mu_1, x_{1,a}(\omega_1,\omega_a)) + \omega_a d(\mu_a, x_{1,a}(\omega_1,\omega_a))$.

Using the functions defined above, we define the vector valued function $\boldsymbol{\Psi}(\boldsymbol{\omega}, I, \boldsymbol{\eta})$ as follows,

$$\boldsymbol{\Psi}(\boldsymbol{\omega}, I, \boldsymbol{\eta}) = (\ \Psi_1(\boldsymbol{\omega}, \boldsymbol{\eta}),\ \Psi_2(\boldsymbol{\omega}, I, \boldsymbol{\eta}),\ \Psi_3(\boldsymbol{\omega}, I, \boldsymbol{\eta}),\ \dots,\ \Psi_K(\boldsymbol{\omega}, I, \boldsymbol{\eta}),\ \Psi_{K+1}(\boldsymbol{\omega})\ ).$$

$\boldsymbol{\Psi}$ maps tuples of the form $\quad (\boldsymbol{\omega}, I, \boldsymbol{\eta}) \in \mathbb{R}_{\geq 0}^K \times \mathbb{R} \times \mathbb{R}^K \quad$ to $\quad \mathbb{R}^{K+1}$.

Its easy to observe that for every $\boldsymbol{\omega} = (\omega_a : a \in [K]) \in \mathbb{R}_{\geq 0}^K$ satisfying $\sum_{a\in[K]} \omega_a = 1$, and $I \in \mathbb{R}$, there is a unique $\boldsymbol{\eta} \in \mathbb{R}^K$ for which $\boldsymbol{\Psi}(\boldsymbol{\omega}, I, \boldsymbol{\eta}) = \mathbf{0}_{K+1}$. We refer to the quantity $\max_{a\in[K]} |\eta_a|$ as the *violation* caused by the pair $(\boldsymbol{\omega}, I)$ to the optimality conditions in (54).

By (54), all the alternative arms in $[K]/\{1\}$ have equal normalized index under the optimal allocation $\boldsymbol{\omega}^\star$. Let $I^\star = W_a(\omega_1^\star, \omega_a^\star)$ for every $a \in [K]/\{1\}$. Then Proposition 2.2 implies $(\boldsymbol{\omega}^\star, I^\star)$ is the unique tuple satisfying

$$\boldsymbol{\Psi}(\boldsymbol{\omega}^\star, I^\star, \mathbf{0}_K) = \mathbf{0}_{K+1}.$$

To prove Proposition G.4 we need the two technical lemmas: Lemma G.5 and G.6. Let us define $\|\boldsymbol{x}\|_\infty = \max_{a\in[K]} |x_a|$ for every $\boldsymbol{x} \in \mathbb{R}^K$.

Lemma G.5 shows that, the set of allocations satisfying the optimality conditions in (54) upto a maximum violation of $r > 0$ shrinks to $\boldsymbol{\omega}^\star$ as $r$ decreases to zero. In Lemma G.6, we use Lemma G.5 and IFT to argue that if the perturbation vector $\boldsymbol{\eta}$ satisfies $\|\boldsymbol{\eta}\|_\infty \leq \eta_{\max}$, where $\eta_{\max} > 0$ is a constant depending only on $\boldsymbol{\mu}$, then there is a unique pair $(\boldsymbol{\omega}, I)$ satisfying $\boldsymbol{\Psi}(\boldsymbol{\omega}, I, \boldsymbol{\eta}) = \mathbf{0}_{K+1}$. Moreover, the function mapping a perturbation vector $\boldsymbol{\eta} \in [-\eta_{\max}, \eta_{\max}]^K$ to the unique pair $(\boldsymbol{\omega}, I)$ solving $\boldsymbol{\Psi}(\boldsymbol{\omega}, I, \boldsymbol{\eta}) = \mathbf{0}_{K+1}$ is Lipschitz continuous.

It is now easy to see Proposition G.4 follows from Lemma G.5 and G.6. By (71), the violation caused by the algorithmic allocation $\widetilde{\boldsymbol{\omega}}(N)$ to the optimality conditions in (54) converges to zero uniformly at a rate $O(N^{-3\alpha/8})$. We wait for sufficiently many iterations such that, the violation becomes smaller than $\eta_{\max}$. Then using Lipschitzness of the allocation as a function of perturbation (proven in Lemma G.6), we have $\|\widetilde{\boldsymbol{\omega}}(N) - \boldsymbol{\omega}^\star\|_\infty = O(N^{-3\alpha/8})$.

**Lemma G.5.** *For every $r \geq 0$, we define the quantity,*

$$dist(\boldsymbol{\omega}^\star, r) = \max\Big\{\ \max\{\|\boldsymbol{\omega} - \boldsymbol{\omega}^\star\|_\infty,\ |I - I^\star|\} \ \Big|\ \boldsymbol{\omega} \in \mathbb{R}_{\geq 0}^K,\ I \in \mathbb{R},\ and$$

$$\exists\ \boldsymbol{\eta} \in [-r, r]^K \ such\ that\ \boldsymbol{\Psi}(\boldsymbol{\omega}, I, \boldsymbol{\eta}) = \mathbf{0}_{K+1}\ \Big\}.$$

*The following statements are true about the mapping $r \mapsto dist(\boldsymbol{\omega}^\star, r)$,*

1. *$dist(\boldsymbol{\omega}^\star, 0) = 0$,*

2. *$dist(\boldsymbol{\omega}^\star, r)$ is non-decreasing in $r$, and*

3. *$\lim_{r\to 0} dist(\boldsymbol{\omega}^\star, r) = 0$.*

*Proof.* **Statement 1:** Statement 1 follows directly from the fact that $(\boldsymbol{\omega}^\star, I^\star)$ is the unique tuple satisfying $\boldsymbol{\Psi}(\boldsymbol{\omega}^\star, I^\star, \mathbf{0}_K) = \mathbf{0}_{K+1}$, as proven in Proposition 2.2.

**Statement 2:** Follows directly from the definition of $dist(\boldsymbol{\omega}^\star, r)$.

**Statement 3:** By statement 2, $\lim_{r\to 0} dist(\boldsymbol{\omega}^\star, r)$ exists and is non-negative. We consider a contradiction to statement 3 and assume that $\lim_{r\to 0} dist(\boldsymbol{\omega}^\star, r) = d > 0$.

Since $r \to dist(\boldsymbol{\omega}^\star, r)$ is non-decreasing, we can construct a decreasing sequence $\{r_n\}_{n\geq 1}$ such that, for every $n \geq 1$, $r_n > 0$, $dist(\boldsymbol{\omega}^\star, r_n) \geq d$, and $\lim_{n\to\infty} r_n = 0$. As a result, using the definition of $dist(\boldsymbol{\omega}^\star, r)$, we have a sequence of tuples $\{(\boldsymbol{\omega}_n, I_n, \boldsymbol{\eta}_n)\}_{n\geq 1}$, such that,

$$\text{for every } n \geq 1,\quad \|\boldsymbol{\eta}_n\|_\infty \leq r_n,\quad \boldsymbol{\Psi}(\boldsymbol{\omega}_n, I_n, \boldsymbol{\eta}_n) = \mathbf{0}_{K+1},\quad \text{and}$$

$$\liminf_{n\to\infty} \max\left\{\|\boldsymbol{\omega}_n - \boldsymbol{\omega}^\star\|_\infty, |I_n - I^\star|\right\} \geq d.$$

Since $\Psi_{K+1}(\boldsymbol{\omega}_n) = 0$ for every $n \geq 1$, the whole sequence $(\boldsymbol{\omega}_n)_{n\geq 1}$ lies in the set

$$\left\{ (\omega_1, \omega_2, \ldots, \omega_K) \in \mathbb{R}_{\geq 0}^K \;\middle|\; \sum_{i\in[K]} \omega_i = 1 \right\},$$

which is compact with respect to the norm $\|\cdot\|_\infty$.

Let for every $n \geq 1$ and $a \in [K]$, $\omega_{a,n}$ and $\eta_{a,n}$ be, respectively, the $a$-th component of the vectors $\boldsymbol{\omega}_n$ and $\boldsymbol{\eta}_n$. For every $n \geq 1$ and $a \in [K]/\{1\}$, we have $I_n = W_a(\omega_{1,n}, \omega_{a,n}) - \eta_{a,n}$. Since $W_a(\cdot, \cdot)$ always lies in the interval $[0, d(\mu_1, \mu_a) + d(\mu_a, \mu_1)]$ and $|\eta_{a,n}| \leq r_n$, we have

$$-r_n \;\leq\; I_n \;\leq\; d(\mu_1, \mu_a) + d(\mu_a, \mu_1) + r_n \quad \text{for every } n \geq 1.$$

By our assumption, we already have $r_n \to 0$, which also implies, the sequence $r_n$ is bounded from above. As a result, $I_n$ is also bounded. Therefore, we can have a convergent subsequence $\{(\boldsymbol{\omega}_{n_k}, I_{n_k})\}_{k\geq 1}$, with limits $\boldsymbol{\omega}_{n_k} \to \boldsymbol{\omega}'$ and $I_{n_k} \to I'$ in the $\|\cdot\|_\infty$-norm.

For every $k \geq 1$, we have $\boldsymbol{\Psi}(\boldsymbol{\omega}_{n_k}, I_{n_k}, \boldsymbol{\eta}_{n_k}) = \mathbf{0}_{K+1}$. As a result, using the continuity of $\boldsymbol{\Psi}(\cdot)$ with respect to its arguments, we have $\boldsymbol{\Psi}(\boldsymbol{\omega}', I', \mathbf{0}_K) = \mathbf{0}_{K+1}$, implying $\boldsymbol{\omega}'$ is an optimal allocation for the instance $\boldsymbol{\mu}$.

Hence, our assumption $\liminf_{n\to\infty} \max\{\|\boldsymbol{\omega}_n - \boldsymbol{\omega}^\star\|_\infty, |I_n - I^\star|\} \geq d$ implies $\max\{\|\boldsymbol{\omega}' - \boldsymbol{\omega}^\star\|_\infty, |I' - I^\star|\} \geq d > 0$, which further implies $\boldsymbol{\omega}' \neq \boldsymbol{\omega}^\star$. As a result, the instance $\boldsymbol{\mu}$ has two distinct optimal allocations $\boldsymbol{\omega}'$ and $\boldsymbol{\omega}^\star$, which contradicts Proposition 2.2. $\qquad\square$

**Lemma G.6.** *There exists $\eta_{\max} > 0$ depending only on the instance $\boldsymbol{\mu}$, such that the following statements are true,*

1. *For every $\boldsymbol{\eta} \in [-\eta_{\max}, \eta_{\max}]^K$, there exists a unique tuple $(\boldsymbol{\omega}, I) \in \mathbb{R}_{\geq 0}^K \times \mathbb{R}$ which satisfies, $\boldsymbol{\Psi}(\boldsymbol{\omega}, I, \boldsymbol{\eta}) = \mathbf{0}_{K+1}$.*

2. *For every $\boldsymbol{\eta} \in [-\eta_{\max}, \eta_{\max}]^K$, we call the unique tuple mentioned in statement 1 as $(\overline{\boldsymbol{\omega}}(\boldsymbol{\eta}), \overline{I}(\boldsymbol{\eta}))$. Then the function*

$$(\overline{\boldsymbol{\omega}}, \overline{I}) : [-\eta_{\max}, \eta_{\max}]^K \mapsto \mathbb{R}_{\geq 0}^K \times \mathbb{R}$$

*is $L$-Lipschitz, for some $L > 0$ depending on the instance $\boldsymbol{\mu}$.*

*Proof.* By Proposition 2.2, we know that the optimal allocation $\boldsymbol{\omega}^\star$ is the unique allocation satisfying,

$$\boldsymbol{\Psi}(\boldsymbol{\omega}^\star, I^\star, \mathbf{0}_K) \;=\; \mathbf{0}_{K+1}$$

for some $I^\star > 0$. Note that $\boldsymbol{\Psi}(\boldsymbol{\omega}, I, \boldsymbol{\eta}) = \boldsymbol{\Phi}(\boldsymbol{\omega}, I, \boldsymbol{\eta}, 1)$ for every tuple $(\boldsymbol{\omega}, I, \boldsymbol{\eta})$, where $\boldsymbol{\Phi}$ is the function defined in Appendix C. By statement 3 of Lemma C.1, the Jacobian $\frac{\partial \boldsymbol{\Psi}}{\partial(\boldsymbol{\omega}, I)}$ of the function $\boldsymbol{\Psi}(\boldsymbol{\omega}, I, \boldsymbol{\eta})$ is invertible at the tuple $(\boldsymbol{\omega}^\star, I^\star, \mathbf{0}_K)$.

Therefore, applying the Implicit function theorem, we can find $\delta_0, \delta_1 > 0$, and continuously differentiable functions

$$(\overline{\boldsymbol{\omega}}(\cdot), \overline{I}(\cdot)) \;:\; (-\delta_0, \delta_0)^K \mapsto \mathbb{R}_{\geq 0}^K \times \mathbb{R},$$

such that,

1. $\overline{\boldsymbol{\omega}}(\mathbf{0}_K) = \boldsymbol{\omega}^\star, \overline{I}(\mathbf{0}_K) = I^\star$, and

2. for every $\boldsymbol{\eta} \in (-\delta_0, \delta_0)^K$, $(\overline{\boldsymbol{\omega}}(\boldsymbol{\eta}), \overline{I}(\boldsymbol{\eta}))$ is the unique tuple in $\mathbb{R}_{\geq 0}^K \times \mathbb{R}$ to satisfy,

$$\max\left\{\|\overline{\boldsymbol{\omega}}(\boldsymbol{\eta}) - \boldsymbol{\omega}^\star\|_\infty, |\overline{I}(\boldsymbol{\eta}) - I^\star|\right\} \leq \delta_1 \quad \text{and} \quad \boldsymbol{\Psi}(\overline{\boldsymbol{\omega}}(\boldsymbol{\eta}), \overline{I}(\boldsymbol{\eta}), \boldsymbol{\eta}) = \mathbf{0}_{K+1}.$$

By statement 3 of Lemma G.5, we can find a $\delta_2 > 0$ such that, $dist(\boldsymbol{\omega}^\star, r) < \delta_1$ for $r \in [0, \delta_2]$. We define $\eta_{\max} = \min\left\{\frac{\delta_0}{2}, \delta_2\right\}$.

By the definition of $dist(\boldsymbol{\omega}^\star, \cdot)$, for every $\boldsymbol{\eta} \in [-\eta_{\max}, \eta_{\max}]^K$, if a tuple $(\boldsymbol{\omega}, I)$ satisfies $\boldsymbol{\Psi}(\boldsymbol{\omega}, I, \boldsymbol{\eta}) = \mathbf{0}_{K+1}$, then it also satisfies $\max\{\|\boldsymbol{\omega} - \boldsymbol{\omega}^\star\|_\infty, |I - I^\star|\} < \delta_1$.

On the other hand, by IFT, since $\eta_{\max} < \delta_0$, $(\overline{\boldsymbol{\omega}}(\boldsymbol{\eta}), \overline{I}(\boldsymbol{\eta}))$ is the only such tuple possible. Therefore, for every $\boldsymbol{\eta} \in [-\eta_{\max}, \eta_{\max}]^K$, $(\overline{\boldsymbol{\omega}}(\boldsymbol{\eta}), \overline{I}(\boldsymbol{\eta}))$ is the unique element in $\mathbb{R}^K_{\geq 0} \times \mathbb{R}$ such that $\boldsymbol{\Psi}(\overline{\boldsymbol{\omega}}(\boldsymbol{\eta}), \overline{I}(\boldsymbol{\eta}), \boldsymbol{\eta}) = \mathbf{0}_{K+1}$. This proves the first statement of Lemma G.6.

Since $\overline{\boldsymbol{\omega}}(\cdot), \overline{I}(\cdot)$ is continuously differentiable in $(-\delta_0, \delta_0)^K$, every component of this mapping must be $L$-Lipschitz for some $L > 0$ in $\left[-\frac{\delta_0}{2}, \frac{\delta_0}{2}\right]^K$ equipped with $\|\cdot\|_\infty$-norm. We can take $L$ to be the maximum of the $\|\cdot\|_1$-norm of the gradients of different components of $(\overline{\boldsymbol{\omega}}(\cdot), \overline{I}(\cdot))$ over the set $\left[-\frac{\delta_0}{2}, \frac{\delta_0}{2}\right]^K$. Since the gradients are all continuous, their $\|\cdot\|_1$-norm must be bounded in a compact set like $\left[-\frac{\delta_0}{2}, \frac{\delta_0}{2}\right]^K$, and hence $L < \infty$. Therefore, the second part of Lemma G.6 follows from our assumption $\eta_{\max} \leq \frac{\delta_0}{2}$. $\qquad\square$

We now proceed on proving Proposition G.4.

**Proof of Proposition G.4:** Recall that in Section 5, for every $a \in [K]/\{1\}$, we defined the normalized index as $H_a(N) = \frac{I_a(N)}{N}$.

Taking $H(N) = H_2(N) = \frac{I_2(N)}{N}$, let $\widetilde{\boldsymbol{\eta}}(N)$ be the unique $\boldsymbol{\eta} \in \mathbb{R}^K$ to satisfy, $\boldsymbol{\Psi}(\widetilde{\boldsymbol{\omega}}(N), H(N), \boldsymbol{\eta}) = \mathbf{0}_{K+1}$.

Note that for every $a \in [K]/\{1\}$, we have $W_a(\widetilde{\omega}_1(N), \widetilde{\omega}_a(N)) = H_a(N)$. As a result, by (71), we have $\|\widetilde{\boldsymbol{\eta}}(N)\|_\infty \leq C N^{-3\alpha/8}$ for all $N \geq T_8$.

Now we pick $M_6 \geq 1$ large enough, such that,
$$CM_6^{-3\alpha/8} < \eta_{\max},$$
where $\eta_{\max}$ is introduced in Lemma G.6. We define $T_{stable} = T_{8,M_6}$. Note that $T_{stable} \geq T_6 \geq T_0$. As a result, by the definition of $T_0$ in Appendix G.3, we have $\max_{a \in [K]} |\widetilde{\mu}_a(N) - \mu_a| \leq \epsilon(\boldsymbol{\mu}) N^{-3\alpha/8}$ for every $N \geq T_{stable}$.

Now, by (71), for $N \geq T_{stable}$, the allocations $\widetilde{\boldsymbol{\omega}}(N)$ satisfies, $\boldsymbol{\Psi}(\widetilde{\boldsymbol{\omega}}(N), H(N), \widetilde{\boldsymbol{\eta}}(N)) = \mathbf{0}_{K+1}$ with
$$\|\widetilde{\boldsymbol{\eta}}(N)\|_\infty \ \leq \ C N^{-3\alpha/8} \ \leq \ C M_6^{-3\alpha/8} \ < \ \eta_{\max}.$$

As a result, by Lemma G.6, we have
$$\widetilde{\omega}_a(N) \ = \ \overline{\omega}_a(\widetilde{\boldsymbol{\eta}}(N)) \quad \text{for every } a \in [K], \text{ and } N \geq T_{stable},$$
where $\overline{\omega}_a(\cdot)$ is the $a$-th component of the vector valued function $\overline{\boldsymbol{\omega}}(\cdot)$ introduced in Lemma G.6.

By Lemma G.6, for every $a \in [K]$, $\overline{\omega}_a(\cdot)$ is $L$-Lipschitz in $[-\eta_{\max}, \eta_{\max}]^K$ equipped with $\|\cdot\|_\infty$-norm. As a result, for $N \geq T_{stable}$, we have,
$$\max_{a \in [K]} |\widetilde{\omega}_a(N) - \omega_a^\star| \ = \ \max_{a \in [K]} |\overline{\omega}_a(\widetilde{\boldsymbol{\eta}}(N)) - \overline{\omega}_a(\mathbf{0}_K)| \ \leq \ L \|\widetilde{\boldsymbol{\eta}}(N) - \mathbf{0}_K\|_\infty$$
$$= \ L \|\widetilde{\boldsymbol{\eta}}(N)\|_\infty \ \leq \ L C N^{-3\alpha/8}.$$

Taking $C_1 = LC$, we have the desired result. $\qquad\square$

## G.3 $T_0$ has finite expectation

In Section 5, we introduced the random time $T_0$ as,
$$T_0 \ = \ \min\left\{ \ N' \geq 1 \ \bigg| \ \forall N \geq N', \ \max_{a \in [K]} |\widetilde{\mu}_a(N) - \mu_a| \leq \epsilon(\boldsymbol{\mu}) N^{-3\alpha/8} \right\},$$

where $\epsilon(\boldsymbol{\mu})$ is a positive constant defined in Appendix B and depends only on the instance $\boldsymbol{\mu}$.

**Lemma G.7.** *The random time $T_0$ satisfies $\mathbb{E}_{\boldsymbol{\mu}}[T_0] < \infty$ and hence $T_0 < \infty$ a.s. in $\mathbb{P}_{\boldsymbol{\mu}}$.*

*Proof.* To avoid notational clutter, let $\mathbb{P} = \mathbb{P}_{\boldsymbol{\mu}}$ and $\epsilon = \epsilon(\boldsymbol{\mu})$. Then for any $N$,

$$
\begin{aligned}
\mathbb{P}(T_0 = N+1) &\leq \mathbb{P}\left(\exists a \in [K], \ |\widetilde{\mu}_a(N) - \mu_a| > \epsilon N^{-3\alpha/8}\right) \\
&\leq \sum_{a \in [K]} \mathbb{P}\left(|\widetilde{\mu}_a(N) - \mu_a| > \epsilon N^{-3\alpha/8}\right) \\
&\leq \sum_{a \in [K]} \sum_{t=(N^\alpha - C_1)_+}^{N} \mathbb{P}\left(|\hat{\mu}_{a,t} - \mu_a| > \epsilon N^{-3\alpha/8}\right),
\end{aligned}
$$

where $\hat{\mu}_{a,t}$ denotes the empirical mean of $t$ i.i.d. samples drawn from the $a$-th arm, and the last step follows from statement 2 of Proposition G.5, which says $\widetilde{N}_a(N) \geq N^\alpha - C_1$ for some constant $C_1 > 0$.

Using Chernoff's bound (like in the proof of Lemma 19 of [11]), we have,

$$
\begin{aligned}
\mathbb{P}\left(|\hat{\mu}_{a,t} - \mu_a| > \epsilon N^{-3\alpha/8}\right) &\leq \mathbb{P}\left(\hat{\mu}_{a,t} > \mu_a + \epsilon N^{-3\alpha/8}\right) + \mathbb{P}\left(\hat{\mu}_{a,t} < \mu_a - \epsilon N^{-3\alpha/8}\right) \\
&\leq \exp\left(-t \cdot d(\mu_a + \epsilon N^{-3\alpha/8}, \mu_a)\right) + \exp\left(-t \cdot d(\mu_a - \epsilon N^{-3\alpha/8}, \mu_a)\right) \\
&\leq 2\exp\left(-t \cdot \min\left\{d(\mu_a + \epsilon N^{-3\alpha/8}, \mu_a), \ d(\mu_a - \epsilon N^{-3\alpha/8}, \mu_a)\right\}\right)
\end{aligned}
$$

Using (6), we have a constant $C_2 > 0$ depending on the instance $\boldsymbol{\mu}$ and such that,

$$
\min\left\{d(\mu_a + \epsilon N^{-3\alpha/8}, \mu_a), \ d(\mu_a - \epsilon N^{-3\alpha/8}, \mu_a)\right\} \geq \epsilon^2 C_2 N^{-\frac{3\alpha}{4}}.
$$

Therefore, we have,

$$
\mathbb{P}\left(|\hat{\mu}_{a,t} - \mu_a| > \epsilon N^{-3\alpha/8}\right) \leq 2\exp\left(-t\epsilon^2 C_2 N^{-\frac{3\alpha}{4}}\right).
$$

Therefore,

$$
\begin{aligned}
\mathbb{P}(T_0 = N+1) &\leq \sum_{i \in [K]} \sum_{t=(N^\alpha - C_1)_+}^{N} 2\exp\left(-t\epsilon^2 C_2 N^{-\frac{3\alpha}{4}}\right) \\
&\leq \sum_{i \in [K]} \sum_{t=N^\alpha - C_1}^{N} 2\exp\left(-t\epsilon^2 C_2 N^{-\frac{3\alpha}{4}}\right) \\
&\leq 2\sum_{i \in [K]} \exp\left(-\epsilon^2 C_2 (N^\alpha - C_1) N^{-\frac{3\alpha}{4}}\right) \cdot \left(\sum_{t=N^\alpha - C_1}^{N} \exp\left(-\epsilon^2 C_2 N^{-\frac{3\alpha}{4}}(t - N^\alpha + C_1)\right)\right) \\
&\leq 2KN \exp\left(-\epsilon^2 C_2 (N^\alpha - C_1) N^{-\frac{3\alpha}{4}}\right) = 2KN \exp(-\Omega(N^{\frac{\alpha}{4}})),
\end{aligned}
$$

where the constant hidden in $\Omega(\cdot)$ depends only on $\boldsymbol{\mu}$. Using the obtained upper bound,

$$
\begin{aligned}
\mathbb{E}[T_0] &= \mathbb{P}(T_0 = 1) + \sum_{N \geq 1}(N+1)\mathbb{P}(T_0 = N+1) \\
&\leq 1 + \sum_{N \geq 1} 2KN(N+1)\exp(-\Omega(N^{\frac{\alpha}{4}})).
\end{aligned}
$$

Note that the series on the RHS is convergent for any $\alpha \in (0, 1)$. Therefore $\mathbb{E}_{\boldsymbol{\mu}}[T_0] < \infty$. $\qquad\square$

## G.4 Properties of exploration

In the following discussion, the set of iterations in which the algorithm does exploration is defined as all iterations $N$ where $\min_{a\in[K]} \widetilde{N}_a(N-1) < N^\alpha$ (which is equivalent to having $\mathcal{V}_N \neq \emptyset$, where $\mathcal{V}_N$ denotes the set of starved arms at iteration $N$, and is defined in Section 3). We define the epoch of exploration at some arbitrary iteration as follows,

**Definition G.6.** *If the algorithm does exploration at iteration $N$, the epoch of exploration at $N$ is the maximum no. of consecutive iterations including $N$ in which the algorithm has done exploration. More precisely, if $N_1$ and $N_2$ are, respectively, defined as,*

$$N_1 = \max\{t \leq N \mid t-1 \text{ is not an exploration}\} \quad \text{and}$$
$$N_2 = \min\{t \geq N \mid t+1 \text{ is not an exploration}\}$$

*then the epoch of exploration at iteration $N$ is $N_2 - N_1 + 1$.*

**Proposition G.5.** *The following statements are true:*

1. *For every iteration which is an exploration, the epoch of exploration at that iteration is upper bounded by a constant depending on $K$ and $\alpha$. We denote this constant using $T_{epoch}$.*

2. *There exists a constant $C$ depending on $K$ and $\alpha$ such that , over every sample path, we have*

$$\min_{a\in[K]} \widetilde{N}_a(N) \geq N^\alpha - C.$$

   *As a result, $\widetilde{N}_a(N) = \Omega(N^\alpha)$ for every arm $a \in [K]$.*

3. *There exists a $M$ depending on $K$ and $\alpha$ such that, every epoch of exploration starting after iteration $M$ has length atmost $K$, and every arm can get pulled atmost once in that epoch. We call the constant $M$ as $M_{explo}$.*

4. *If an epoch of exploration starts from some $N \geq M_{explo}$, then the next epoch of exploration doesn't start before another $\Theta(N^{1-\alpha})$ iterations.*

5. *Let $\hat{N} \geq M_{explo}$ be such that, $\hat{N}$ is an exploration. Define the following sequence, $N_0 = \hat{N}$, and for $k \geq 1$,*

$$N_k = \min\left\{ N > N_{k-1} \ \middle| \ N_k \text{ is the begining of an epoch of exploration} \right\}.$$

   *Then $N_k = \hat{N} + \Omega(k^{1/\alpha})$. In other words, for any $N \geq M_{explo}$, the $k$-th epoch of exploration after iteration $N$ starts after $N + \Omega(k^{1/\alpha})$ iterations.*

6. *For any $N \geq M_{explo} + T_{epoch} + 1$ and $T \geq 1$, the no. of epochs of exploration intersecting with the set of iterations $\{N, N+1, N+2, \dots, N+T\}$ is $O(T^\alpha)$.*

*Proof.* **Statement 1:** Let the algorithm does exploration at iteration $N$. We can always choose $N$ in such a way that, iteration $N-1$ was not an exploration, by choosing $N$ to be the iteration at which an epoch begins. If the epoch of exploration starting at iteration $N$ continues till iteration $N+t$, *i.e.*, the iterations $N, N+1, \dots, N+t$ are exploration, then,

$$(N+t)^\alpha \geq \min_{a\in[K]} \widetilde{N}_a(N+t-1) \overset{(1)}{\geq} \min_{a\in[K]} \widetilde{N}_a(N-1) + \frac{t}{K}$$
$$\geq \min_{a\in[K]} \widetilde{N}_a(N-2) + \frac{t}{K} \overset{(2)}{\geq} (N-1)^\alpha + \frac{t}{K},$$

where (1) follows from the fact that, $\min_{a\in[N]} \widetilde{N}_a(\cdot)$ increments by atleast 1 over every $K$ consecutive iterations in an epoch of exploration, and (2) from the fact that iteration $N-1$ is not an exploration. From the above inequality, we have,

$$\frac{t}{K} \leq (N+t)^\alpha - (N-1)^\alpha.$$

Note that, $N \to (N+t)^\alpha - (N-1)^\alpha$ is decreasing in $N$. Hence $RHS \leq (1+t)^\alpha \leq 1 + t^\alpha \leq 2t^\alpha$ (since $t \geq 1$). Therefore, we have,

$$\frac{t}{K} \leq 2t^\alpha \quad \text{implying} \quad t \leq (2K)^{1/(1-\alpha)}.$$

Therefore every epoch of exploration is atmost $(2K)^{1/(1-\alpha)}$ iterations long.

**Statement 2:** In the following discussion we use $[i : j]$ for a pair of integers $i < j$ to denote the set $\{i, i+1, i+2, \ldots, j\}$. We consider only those iterations where $\min_{a \in [K]} \widetilde{N}_a(N-1) < N^\alpha$. By statement 1, if we consider $N_1 \leq N \leq N_2$ such that, $N_1$, $N_1 + 1$, $\ldots$, $N_2 - 1$, $N_2$ are all explorations and $N_1 - 1$, $N_2 + 1$ are not explorations, then we must have, $N_2 - N_1 \leq T_{\text{epoch}}$. As a result, we have,

$$N^\alpha - \min_{a \in [K]} \widetilde{N}_a(N) \leq \max_{N \in [N_1:N_2]} (N^\alpha - \min_{a \in [K]} \widetilde{N}_a(N)).$$

Now, the slowest rate at which $\min_{a \in [K]} \widetilde{N}_a(N)$ can grow while iterations $N \in \{N_1, N_1 + 1, \ldots, N_2 - 1, N_2\}$ is if the algorithm pulls the $K$ arms consecutively in those iterations. Therefore, we have,

$$\min_{a \in [K]} \widetilde{N}_a(N) \geq \min_{a \in [K]} \widetilde{N}_a(N_1 - 1) + \frac{N - N_1 + 1}{K} \quad \text{for every } N \in [N_1 : N_2].$$

Using this, we have,

$$N^\alpha - \min_{a \in [K]} \widetilde{N}_a(N) \leq \max_{N \in [N_1:N_2]} \left( N^\alpha - \min_{a \in [K]} \widetilde{N}_a(N_1 - 1) - \frac{N - N_1 + 1}{K} \right).$$

Since iteration $N_1 - 1$ is not an exploration, we have

$$\min_{a \in [K]} \widetilde{N}_a(N_1 - 1) \geq \min_{a \in [K]} \widetilde{N}_a(N_1 - 2) \geq (N_1 - 1)^\alpha.$$

Using this, the upper bound becomes,

$$\max_{N \in [N_1:N_2]} \left( N^\alpha - (N_1 - 1)^\alpha - \frac{N - N_1 + 1}{K} \right) \leq \max_{N \in [N_1:N_2]} \left( (N - N_1 + 1)^\alpha - \frac{N - N_1 + 1}{K} \right)$$

$$\leq \max_{z \in [0, T_{\text{epoch}}]} \left( (z+1)^\alpha - \frac{1+z}{K} \right) = C,$$

where $C$ depends only on $K$ and $\alpha$.

Hence we get,

$$\min_{a \in [K]} \widetilde{N}_a(N) \geq N^\alpha - C = \Omega(N^\alpha).$$

**Statement 3:** If the epoch of exploration starts from $T$ and continues for more than $K$ iterations, note that $\min_{a \in [K]} \widetilde{N}_a(\cdot)$ gets incremented by atleast 1 during the iterations $T$, $T + 1$, $\ldots$, $T + K$. As a result, we have,

$$\min_{a \in [K]} \widetilde{N}_a(T + K - 1) - \min_{a \in [K]} \widetilde{N}_a(T - 1) \geq 1.$$

Since iteration $T - 1$ is not an exploration, we have $\widetilde{N}_a(T - 1) \geq \widetilde{N}_a(T - 2) \geq (T - 1)^\alpha$. Similarly, since iteration $T + K$ is an exploration, $\min_{a \in [K]} \widetilde{N}_a(T + K - 1) < (T + K)^\alpha$. Using these two observations, we have,

$$1 \leq (T + K)^\alpha - (T - 1)^\alpha \leq (T - 1)^\alpha \times \left( \left( 1 + \frac{K + 1}{T - 1} \right)^\alpha - 1 \right) \leq \alpha(K + 1)(T - 1)^{-(1-\alpha)}.$$

Therefore,
$$T \leq (\alpha(K+1))^{1/(1-\alpha)} + 1.$$

Let $M_{\text{explo}} = (\alpha(K+1))^{1/(1-\alpha)} + 2$. Then from the above argument, if $T \geq M_{\text{explo}}$, the epoch of exploration starting at $T$ will last for at most $K$ iterations.

Let us assume that, the epoch begining from some $T_i \geq M_{\text{explo}}$ lasts till iteration $T_f$. Therefore, we have,
$$\min_{a \in [K]} \widetilde{N}_a(T_f - 1) < T_f^\alpha \quad \text{and} \quad \min_{a \in [K]} \widetilde{N}_a(T_i - 2) \geq (T_i - 1)^\alpha.$$

Using this,
$$\min_{a \in [K]} \widetilde{N}_a(T_f - 1) - \min_{a \in [K]} \widetilde{N}_a(T_i - 2) \leq T_f^\alpha - (T_i - 1)^\alpha \leq \alpha(T_i - 1)^{-(1-\alpha)}(T_f - T_i + 1).$$

We argued earlier that $T_f - T_i \leq K$. We also have, $T_i - 1 \geq M_{\text{explo}} - 1 \geq (\alpha(K+1))^{1/(1-\alpha)} + 1$. Using this observations, we get
$$\min_{a \in [K]} \widetilde{N}_a(T_f - 1) - \min_{a \in [K]} \widetilde{N}_a(T_i - 2) \leq \frac{\alpha(K+1)}{((\alpha(K+1))^{1/(1-\alpha)} + 1)^{1-\alpha}} < 1.$$

Since $\min_{a \in [K]} \widetilde{N}_a(T_f) \leq \min_{a \in [K]} \widetilde{N}_a(T_f - 1) + 1$ and $\min_{a \in [K]} \widetilde{N}_a(T_i - 2) \leq \min_{a \in [K]} \widetilde{N}_a(T_i - 1)$, we obtain,
$$\min_{a \in [K]} \widetilde{N}_a(T_f) - \min_{a \in [K]} \widetilde{N}_a(T_i - 1) < 2,$$

implying $\min_{a \in [K]} \widetilde{N}_a(T_f) - \min_{a \in [K]} \widetilde{N}_a(T_i - 1) \leq 1$, which is possible only if every arm is pulled at most once in iterations $T_i, T_i + 1, \ldots, T_f - 1, T_f$.

**Statement 4:** Let an epoch of exploration starts from $N \geq M_{\text{explo}}$ and the next epoch starts from $N + T$ for some $T \geq 1$. The epoch starting from $N$ continues till atmost $\min\{N + K, N + T - 2\}$ by statement 3. Moreover in that epoch, every arm gets pulled atmost once and $\min_{a \in [K]} \widetilde{N}_a(\cdot)$ increments by 1. Therefore,
$$\min_{a \in [K]} \widetilde{N}_a(N + T - 1) - \min_{a \in [K]} \widetilde{N}_a(N - 1) \geq 1.$$

Since iteration $N - 1$ is not an exploration, we have, $\min_{a \in [K]} \widetilde{N}_a(N-1) \geq \min_{a \in [K]} \widetilde{N}_a(N-2) \geq (N-1)^\alpha$. Since iteration $N + T$ is an exploration, we have $\min_{a \in [K]} \widetilde{N}_a(N + T - 1) < (N+T)^\alpha$. Using these in the above inequality, we obtain,
$$1 \leq (N+T)^\alpha - (N-1)^\alpha \leq \alpha(N-1)^{-(1-\alpha)}(T+1),$$

which implies $T \geq \frac{1}{\alpha}(N-1)^{1-\alpha} - 1 = \Theta(N^{1-\alpha})$.

**Statement 5:** Using statement 4, we can have a constant $C_1$ such that,
$$N_k \geq N_{k-1} + C_1 N_{k-1}^{1-\alpha},$$

for $k \geq 1$, where $N_0 = 1$. We now inductively argue that, there exists some constant $C_2$ independent of $k$ and $N$ such that, $N_k \geq N + C_2 k^{1/\alpha}$ for $k \geq 1$.

- For $k = 1$, we choose $C_2 \leq 1$.

- Now for some $k \geq 2$, if $N_{k-1} \geq N + C_2(k-1)^{1/\alpha}$, we have,
$$\begin{aligned} N_k &\geq N_{k-1} + C_1 N_{k-1}^{1-\alpha} \geq N + C_2(k-1)^{1/\alpha} + C_1(N + C_2(k-1)^{1/\alpha})^{1-\alpha} \\ &\geq N + C_2 k^{1/\alpha} + \left( C_1(N + C_2(k-1)^{1/\alpha})^{1-\alpha} - C_2(k^{1/\alpha} - (k-1)^{1/\alpha}) \right) \\ &\geq N + C_2 k^{1/\alpha} + \left( C_1(N + C_2(k-1)^{1/\alpha})^{1-\alpha} - \frac{C_2}{\alpha} k^{\frac{1}{\alpha} - 1} \right). \end{aligned}$$

Upon choosing $C_2 \leq \left( \frac{\alpha C_1}{2^{\frac{1}{\alpha}-1}} \right)^{1/\alpha}$, since we already have $k \geq 2$, we obtain

$$C_1 (N + C_2(k-1)^{1/\alpha})^{1-\alpha} \geq C_1 \left( C_2 \left( \frac{k}{2} \right)^{1/\alpha} \right)^{1-\alpha} = \left( C_2^{-\alpha} \frac{\alpha C_1}{2^{\frac{1}{\alpha}-1}} \right) \frac{C_2}{\alpha} k^{1/\alpha}$$

$$\geq \frac{C_2}{\alpha} k^{1/\alpha}.$$

Therefore, $N_k \geq N + C_2 k^{\frac{1}{\alpha}}$.

Therefore, choosing $C_2 = \min \left\{ 1, \left( \frac{\alpha C_1}{2^{\frac{1}{\alpha}-1}} \right)^{\frac{1}{\alpha}} \right\}$, we have $N_k \geq N + C_2 k^{\frac{1}{\alpha}}$ for all $k \geq 1$.

**Statement 6:** If $N \geq M_{\text{explo}} + T_{\text{epoch}} + 1$ and $T \geq 1$, every epoch of explorations intersecting with the iterations $\{N, N+1, \ldots, N+T-1, N+T\}$ has length atmost $K$ (by statement 3). Hence, every such epoch must have started on or after iteration $N - K$. Let $N_0$ be the time when the first such epoch has started and the sequence $(N_k)_{k \geq 1}$ be defined similar to statement 5. Then $N_0 \geq N - K$ and $N_k \geq N_0 + C_2 k^{\frac{1}{\alpha}} \geq N - K + C_2 k^{\frac{1}{\alpha}}$. Now, if the $k$-th epoch starting after $N_0$ intersects with the iterations $\{N, N+1, \ldots, N+T-1, N+T\}$, then,

$$N + T \geq N - K + C_2 k^{\frac{1}{\alpha}}, \quad \text{which implies}, \quad k \leq \frac{(T+K)^\alpha}{C_2^\alpha} = O(T^\alpha).$$

$\square$

**Definition G.7.** *We define the constant* $T_{\text{explo}} = M_{\text{explo}} + T_{\text{epoch}} + 1$, *where the constants* $M_{\text{explo}}$, *and* $T_{\text{epoch}}$ *are defined in Proposition G.5.*

**Lemma G.8.** *For* $N \geq T_{\text{explo}}$ *and* $T \geq 1$, *the no of times an arm* $a \in [K]$ *is pulled for exploration by the algorithm during iterations* $N, N+1, \ldots, N+T-1, N+T$ *is* $O(T^\alpha)$.

*Proof.* By statement 3 of Proposition G.5, for $N \geq T_{\text{explo}}$ and $T \geq 1$, every epoch of exploration intersecting with the set of iterations $N, N+1, N+2, \ldots, N+T$ is of length atmost $K$. In every such epoch, every arm is pulled atmost once. As a result, no. of times an arm is pulled during the iterations $N, N+1, \ldots, N+T$ is upper bounded by the no. of epoch of iterations intersecting with the set $\{N, N+1, \ldots, N+T-1, N+T\}$. The later quantity is $O(T^\alpha)$ by statement 6 of Proposition G.5. Hence the lemma stands proved. $\square$

### G.5 Technical lemmas related to algorithmic allocations

In this appendix we prove several properties about the anchor function ($g$) and the algorithmic allocations $\widetilde{\boldsymbol{N}}(N) = (\widetilde{N}_a(N) : a \in [K])$. We exploit the results proven in Appendix B and G.4 to prove that the following properties hold after a random time of finite expectation,

- if the algorithm has $g \neq 0$ at some iteration $N$, then $g$ crosses the value zero withing an $O(N)$ iterations, where the constant hidden in $O(\cdot)$ is independent of the sample paths (in Lemma G.9), and

- every arm $a \in [K]$ has $\widetilde{N}_a(N) = \Theta(N)$ samples (in Lemma G.13).

Each of the properties stated above are used extensively in the proofs of Proposition G.1 and G.2.

**Definition G.8.** We define the random time $T_1 = \max\{T_0, T_{\text{explo}}\}$ (where $T_{\text{explo}}$ and $T_0$ are defined, respectively, in Appendix G.4 and G.3).

Note that $\mathbb{E}_{\boldsymbol{\mu}}[T_1] \leq \mathbb{E}_{\boldsymbol{\mu}}[T_0] + T_{\text{explo}} < \infty$.

We can have $g$ far from the value zero at iteration $T_1$. $g$ will still be finite at $T_1$ because of exploration. Lemma G.9 bounds the no. of iterations the algorithm takes to reach the value zero.

**Lemma G.9** (**Upper bound to the time to reach** $g = 0$). *If* $g(\widetilde{\boldsymbol{\mu}}(N), \widetilde{\boldsymbol{N}}(N)) \neq 0$ *at iteration* $N \geq T_1$, *and*

$$U = \min \left\{ t > 0 \ \middle| \ g\left(\widetilde{\boldsymbol{\mu}}(N+t), \widetilde{\boldsymbol{N}}(N+t)\right) \quad \text{and} \quad g\left(\widetilde{\boldsymbol{\mu}}(N), \widetilde{\boldsymbol{N}}(N)\right) \quad \text{have opposite signs} \right\},$$

*then there exists a constant* $C_1 > 0$ *independent of the sample paths such that* $U \leq C_1 N$.

*Proof.* Using (10), for $N \geq T_1$, we have,

$$g\left(\widetilde{\boldsymbol{\mu}}(N), \widetilde{\boldsymbol{N}}(N)\right) \ = \ \Theta \left( \sum_{a \neq 1} \frac{\widetilde{N}_a(N)^2}{\widetilde{N}_1(N)^2} \right) - 1 \ = \ \Theta \left( \left( \sum_{a \neq 1} \frac{\widetilde{N}_a(N)}{\widetilde{N}_1(N)} \right)^2 \right) - 1$$

$$= \ \Theta \left( \left( \frac{N}{\widetilde{N}_1(N)} - 1 \right)^2 \right) - 1. \quad (72)$$

We now consider two situations separately,

**Case I:** $g\left(\widetilde{\boldsymbol{\mu}}(N), \widetilde{\boldsymbol{N}}(N)\right) > 0$: We know that, there is a constant $D_1 > 0$, such that,

$$g\left(\widetilde{\boldsymbol{\mu}}(N+t), \widetilde{\boldsymbol{N}}(N+t)\right) \ \leq \ D_1 \left( \frac{N+t}{\widetilde{N}_1(N+t)} - 1 \right)^2 - 1,$$

for every $t \geq 1$. Now for $t < U$, the algorithm selects an alternative arm from $[K]/\{1\}$ only while exploring. Therefore, using statement 6 of Proposition G.5, we have a constant $c_1 > 0$ such that, $\widetilde{N}_1(N+t) \geq t - c_1 t^\alpha$. Hence,

$$g\left(\widetilde{\boldsymbol{\mu}}(N+t), \widetilde{\boldsymbol{N}}(N+t)\right) \ \leq \ D_1 \left( \frac{N+t}{t - c_1 t^\alpha} - 1 \right)^2 - 1 \ = \ D_1 \left( \frac{N + c_1 t^\alpha}{t - c_1 t^\alpha} \right)^2 - 1 \quad \text{for } t < U.$$

Since $g\left(\widetilde{\boldsymbol{\mu}}(N+U-1), \widetilde{\boldsymbol{N}}(N+U-1)\right) \geq 0$, RHS of the above inequality is non-negative at $t = U - 1$. After some algebraic manipulation, this implies,

$$U - 1 - c_1(1 + D_1^{1/2})(U-1)^\alpha \ \leq \ D_1^{1/2} N,$$

Since LHS of the above inequality is linear in $U$, we can find a constant $C_{11}$ such that $U \leq C_{11} N$.

**Case II:** $g\left(\widetilde{\boldsymbol{\mu}}(N), \widetilde{\boldsymbol{N}}(N)\right) < 0$: Using (72), we have a constant $D_2$ such that,

$$g\left(\widetilde{\boldsymbol{\mu}}(N+t), \widetilde{\boldsymbol{N}}(N+t)\right) \ \geq \ D_2 \left( \frac{N+t}{\widetilde{N}_1(N+t)} - 1 \right)^2 - 1$$

for all $t \geq 1$. Now for $t < U$, the algorithm pulls arm 1 only for exploration. As a result, using statement 6 of Proposition G.5, we have, a constant $c_1 > 0$ such that $\widetilde{N}_1(N+t) \leq N + c_1 t^\alpha$. Using this,

$$g\left(\widetilde{\boldsymbol{\mu}}(N+t), \widetilde{\boldsymbol{N}}(N+t)\right) \ \geq \ D_2 \left( \frac{N+t}{N + c_1 t^\alpha} - 1 \right)^2 - 1 \ \geq \ D_2 \left( \frac{t - c_1 t^\alpha}{N + c_1 t^\alpha} \right)^2 - 1,$$

for $t < U$. Since $g(\widetilde{\boldsymbol{\mu}}(N+U-1), \widetilde{\boldsymbol{N}}(N+U-1)) \leq 0$, the RHS of the above inequality is not positive at $t = U - 1$. After some algebraic manipulation, this implies,

$$U - 1 - c_1(1 + D_2^{-1/2})(U-1)^\alpha \leq D_2^{-1/2} N.$$

Again the LHS of the above inequality is linear in $U$. As a result, we can find a constant $C_{12}$ such that, $U \leq C_{12} N$.

We take $C_1 = \max\{C_{11}, C_{12}\}$ and have $U \leq C_1 N$. $\qquad\square$

Lemma G.10, G.11, and G.12 are necessary for proving Lemma G.13, which says $\widetilde{N}_a(N) = \Theta(N)$ for every arm $a \in [K]$, after a random time of finite expectation. This property is essential for proving Proposition G.1 and G.2 stated earlier.

**Lemma G.10.** *There exists constants $\gamma_1, \gamma_2 \in (0, 1)$ independent of the sample path, such that, for $N \geq T_1$, whenever $g\left(\widetilde{\boldsymbol{\mu}}(N), \widetilde{\boldsymbol{N}}(N)\right)$ and $g\left(\widetilde{\boldsymbol{\mu}}(N+1), \widetilde{\boldsymbol{N}}(N+1)\right)$ have opposite signs, we have,*

*1. $\widetilde{N}_1(N) \geq \gamma_1 N$, and*

*2. $\max_{a \in [K]/\{1\}} \widetilde{N}_a(N) \geq \gamma_2 N$.*

*Proof.* For $N \geq T_1$, we have constants $D_1, D_2 > 0$ such that,

$$D_1 \left( \frac{N}{\widetilde{N}_1(N)} - 1 \right)^2 - 1 \leq g\left(\widetilde{\boldsymbol{\mu}}(N), \widetilde{\boldsymbol{N}}(N)\right) \leq D_2 \left( \frac{N}{\widetilde{N}_1(N)} - 1 \right)^2 - 1$$

We consider only the situation where $g\left(\widetilde{\boldsymbol{\mu}}(N), \widetilde{\boldsymbol{N}}(N)\right) \geq 0$ and $g\left(\widetilde{\boldsymbol{\mu}}(N+1), \widetilde{\boldsymbol{N}}(N+1)\right) \leq 0$. Extending this to the other case follows similar argument.

From $g(\widetilde{\boldsymbol{\mu}}(N+1), \widetilde{\boldsymbol{N}}(N+1)) \leq 0$, we have,

$$D_1 \left( \frac{N+1}{\widetilde{N}_1(N+1)} - 1 \right)^2 - 1 \leq 0, \text{ which implies}$$

$$\widetilde{N}_1(N+1) \geq (1 + D_1^{-1/2})^{-1} N. \tag{73}$$

Since $\widetilde{N}_1(N) \geq 1$ (for exploration), we have $\widetilde{N}_1(N+1) \leq \widetilde{N}_1(N) + 1 \leq 2\widetilde{N}_1(N)$. Therefore, using (73) we get $\widetilde{N}_1(N) \geq \frac{1}{2}(1 + D_1^{-1/2})^{-1} N$ and we can take $\gamma_1 = \frac{1}{2}(1 + D_1^{-1/2})^{-1}$.

Similarly from, $g(\widetilde{\boldsymbol{\mu}}(N), \widetilde{\boldsymbol{N}}(N)) \geq 0$, we have,

$$D_2 \left( \frac{N}{\widetilde{N}_1(N)} - 1 \right)^2 - 1 \geq 0, \text{ which implies}$$

$$\widetilde{N}_1(N) \leq (1 + D_2^{-1/2})^{-1} N. \tag{74}$$

Using (74), we have, $\sum_{a \in [K]/\{1\}} \widetilde{N}_a(N) \geq (1 + D_2^{1/2})^{-1} N$. This further implies

$$\max_{a \in [K]/\{1\}} \widetilde{N}_a(N) \geq \frac{1}{K-1}(1 + D_2^{1/2})^{-1} N.$$

Hence we can take $\gamma_2 = \frac{1}{K-1}(1 + D_2^{1/2})^{-1}$. $\qquad \square$

**Lemma G.11.** *There exists constants $M_1 \geq 1$, $d_{max} \in [1, \infty)$ and $d_{min} \in (0, 1]$ independent of the sample path, such that, if $T_2$ is the time at which $g$ crosses zero after the iteration $\max\{M_1, T_1\}$, then, for $N > T_2$, we have:*

$$d_{min} \leq \sum_{a \neq 1} \frac{d(\widetilde{\mu}_1(N), \widetilde{x}_{1,a}(N))}{d(\widetilde{\mu}_a(N), \widetilde{x}_{1,a}(N))} \leq d_{max}.$$

*Moreover, we have $\mathbb{E}_{\boldsymbol{\mu}}[T_2] < \infty$.*

*Proof.* Let,

$$D(\widetilde{\boldsymbol{\mu}}(N), \widetilde{\boldsymbol{N}}(N)) = \sum_{a \neq 1} \frac{d(\widetilde{\mu}_1(N), \widetilde{x}_{1,a}(N))}{d(\widetilde{\mu}_a(N), \widetilde{x}_{1,a}(N))}.$$

Also for every $M \geq 1$, we define $T_{1,M} = \max\{M, T_1\}$ and $T_{2,M}$ as the iteration at which $g$ crosses zero for the first time after the iteration $T_{1,M}$, *i.e.*,

$$T_{2,M} \overset{\text{def.}}{=} \min \left\{ N \geq T_{1,M} \,\middle|\, g(\widetilde{\boldsymbol{\mu}}(N+1), \widetilde{\boldsymbol{N}}(N+1)) \text{ and} \right.$$
$$\left. g(\widetilde{\boldsymbol{\mu}}(T_{1,M}), \widetilde{\boldsymbol{N}}(T_{1,M})) \text{ are of opposite signs} \right\}.$$

**Existence of $d_{\max}$:** If $g(\widetilde{\boldsymbol{\mu}}(N), \widetilde{\boldsymbol{N}}(N)) \leq 0$, then $D(\widetilde{\boldsymbol{\mu}}(N), \widetilde{\boldsymbol{N}}(N))$ is bounded above trivially by $1$.

Otherwise if $g(\widetilde{\boldsymbol{\mu}}(N), \widetilde{\boldsymbol{N}}(N)) > 0$ for some $N > T_{2,M}$, let $T = \max\{ t \leq N \mid g(\widetilde{\boldsymbol{\mu}}(t), \widetilde{\boldsymbol{N}}(t)) \leq 0 \}$ *i.e.* the time before iteration $N$ when $g$ has crossed the level zero. As a result, $T \geq T_{2,M}$ (by definition of $T_{2,M}$), and by Lemma G.10, $\widetilde{N}_1(T) \geq \gamma_1 T$ for some $\gamma_1 \in (0,1)$. Let $S = N - T$. During iterations $T+1, \ldots, T+S$, since $g$ is positive, the algorithm pulls alternative arms in $[K]/\{1\}$ only for exploration. As a result, using statement 6 of Proposition G.5, we have

$$\widetilde{N}_1(T+S) \geq \widetilde{N}_1(T) + S - c_1 S^\alpha \geq \gamma_1 T + S - c_1 S^\alpha$$

for some $c_1 > 0$. Using (72), we have a constant $D_2 > 0$ such that,

$$D(\widetilde{\boldsymbol{\mu}}(T+S), \widetilde{\boldsymbol{N}}(T+S)) \leq D_2 \left( \frac{T+S}{\gamma_1 T + S - c_1 S^\alpha} - 1 \right)^2 = D_2 \left( \frac{(1-\gamma_1)T + c_1 S^\alpha}{\gamma_1 T + S - c_1 S^\alpha} \right)^2.$$

We take $M_{11} = -\frac{2}{\gamma_1} \min_{S \geq 0}(S - c_1 S^\alpha)$. It is easy to observe that for $T \geq M_{11}$ and $S \geq 0$, the function of $T, S$ in the RHS is bounded above. Therefore, we take,

$$d_{\max} = \max \left\{ 1, \max_{T \geq M_{11}, S \geq 0} D_2 \left( \frac{(1-\gamma_1)T + c_1 S^\alpha}{\gamma_1 T + S - c_1 S^\alpha} \right)^2 \right\} < \infty.$$

Hence, if $M \geq M_{11}$, we have $g(\widetilde{\boldsymbol{\mu}}(N), \widetilde{\boldsymbol{N}}(N)) \leq d_{\max}$ for every $N > T_{2,M}$.

**Existence of $d_{\min}$:** If $g(\widetilde{\boldsymbol{\mu}}(N), \widetilde{\boldsymbol{N}}(N)) \geq 0$, we have $1$ has the trivial lower bound. Otherwise, if $g(\widetilde{\boldsymbol{\mu}}(N), \widetilde{\boldsymbol{N}}(N)) < 0$, we define $T = \max\{ t \leq N \mid g(\widetilde{\boldsymbol{\mu}}(t), \widetilde{\boldsymbol{N}}(t)) \geq 0 \}$ and $S = N - T$. As a result, $T \geq T_{2,M}$ (by definition of $T_{2,M}$). By Lemma G.10, we have $\widetilde{N}_1(T) \leq (1 - \gamma_2)N$ for some $\gamma_2 \in (0,1)$. Also, the algorithm pulls the first arm only for exploration during the iterations $T+1, T+2, \ldots, T+S$, since $g$ is negative. Therefore, using statement 6 of Proposition G.5, we have, $\widetilde{N}_1(T+S) \leq \widetilde{N}_1(T) + c_1 S^\alpha \leq (1-\gamma_2)T + c_1 S^\alpha$. By (72), we have $D_1 > 0$ such that,

$$D(\widetilde{\boldsymbol{\mu}}(T+S), \widetilde{\boldsymbol{N}}(T+S)) \geq D_1 \left( \frac{T+S}{(1-\gamma_2)T + c_1 S^\alpha} - 1 \right)^2 = D_1 \left( \frac{\gamma_2 T + S - c_1 S^\alpha}{(1-\gamma_2)T + c_1 S^\alpha} \right)^2.$$

Let $M_{12} = -\frac{2}{\gamma_2} \min_{S \geq 0}(S - c_1 S^\alpha)$. The function of $T, S$ in the RHS is bounded below by a positive constant for $T \geq M_{12}$ and $S \geq 0$. Let,

$$d_{\min} = \min \left\{ 1, \min_{T \geq M_{12}, S \geq 0} D_1 \left( \frac{\gamma_2 T + S - c_1 S^\alpha}{(1-\gamma_2)T + c_1 S^\alpha} \right)^2 \right\} > 0.$$

Then for $M \geq M_{12}$, we have $g(\widetilde{\boldsymbol{\mu}}(N), \widetilde{\boldsymbol{N}}(N)) \geq d_{\min} > 0$ for every $N > T_{2,M}$.

Now, upon taking $M_1 = \max\{M_{11}, M_{12}\}$, and $T_2 = T_{2,M_1}$, the lemma follows. By Lemma G.9, we have a constant $C_1 > 0$ such that, $T_2 \leq C_1 \max\{M_1, T_1\}$. As a result, $\mathbb{E}_{\boldsymbol{\mu}}[T_2] \leq C_1(M_1 + \mathbb{E}_{\boldsymbol{\mu}}[T_1]) < \infty$. $\qquad\square$

Using a technique similar to the one used in the proof of Lemma G.10 using (72), Lemma G.11 implies the following corollary. We define the random time $T_2$ as the one introduced in Lemma G.11.

**Corollary G.1.** *There exists constants $\beta_1, \beta_2 \in (0,1)$ independent of the sample paths, such that, for every $N > T_2$,*

1. $\widetilde{N}_1(N) \geq \beta_1 N$, and

2. $\max_{a \in [K]/\{1\}} \widetilde{N}_a(N) \geq \beta_2 N$.

**Lemma G.12.** *There exists constants $\gamma > 0$ and $M_2 \geq 1$ independent of the sample path, such that, for $N > \max\{M_2, T_2\} + 1$, if the algorithm pulls some arm $a \in [K]/\{1\}$, then for every alternate arm $b \in [K]/\{1, a\}$, $\widetilde{N}_b(N) \geq \gamma \widetilde{N}_a(N)$.*

*Proof.* We consider two separate cases.

**Case I: Iteration $N$ is an exploration:** Then we have a constant $C_1$ (introduced in statement 2 of Proposition G.5) such that,

$$\widetilde{N}_b(N) \geq N^\alpha - C_1 \geq \widetilde{N}_a(N-1) - C_1 = \widetilde{N}_a(N) - 1 - C_1. \tag{75}$$

We also have $\widetilde{N}_a(N) \geq N^\alpha - C_1$. Now let $M_2 \geq 2$ be chosen such that, $M_2^\alpha > 3C_1 + 2$. As a result, for $N \geq \max\{M_2, T_2\}$ we have, $\widetilde{N}_a(N) \geq N^\alpha - C_1 \geq 2(1 + C_1)$. This together with (75) implies, $\widetilde{N}_b(N) \geq \frac{1}{2}\widetilde{N}_a(N)$.

**Case II: Iteration $N$ is not an exploration:** We consider the AT2 (1) and IAT2 (2) algorithms separately.

**Case II.I AT2 Algorithm:** If iteration $N$ is not an exploration, we have, $\mathcal{I}_a(N-1) \leq \mathcal{I}_b(N-1)$. Using (11), for $N \geq \max\{M_2, T_2\}$, we have constants $C_2$ and $C_3$ such that,

$$C_2 \frac{\widetilde{N}_1(N-1) \cdot \widetilde{N}_a(N-1)}{\widetilde{N}_1(N-1) + \widetilde{N}_a(N-1)} \leq \mathcal{I}_a(N-1)$$

$$\leq \mathcal{I}_b(N-1) \leq C_3 \frac{\widetilde{N}_1(N-1) \cdot \widetilde{N}_b(N-1)}{\widetilde{N}_1(N-1) + \widetilde{N}_b(N-1)}.$$

As a result, we have,

$$\widetilde{N}_a(N-1) \leq \frac{C_3}{C_2} \frac{\widetilde{N}_1(N-1) + \widetilde{N}_a(N-1)}{\widetilde{N}_1(N-1) + \widetilde{N}_b(N-1)} \widetilde{N}_b(N-1) \overset{(1)}{\leq} \frac{2C_3}{\beta_1 C_2} \widetilde{N}_b(N-1) \leq \frac{2C_3}{\beta_1 C_2} \widetilde{N}_b(N),$$

where, for (1), we first note that $N > T_2 + 1$. As a result, using Corollary (G.1), $\widetilde{N}_1(N-1) \geq \beta_1(N-1)$ ($\beta_1 \in (0,1)$ is the constant introduced in Corollary G.1), and on the other hand $\widetilde{N}_1(N-1) + \widetilde{N}_b(N-1) \leq 2(N-1)$. Hence, we get

$$\widetilde{N}_b(N) \geq \frac{\beta_1 C_2}{2C_3} \widetilde{N}_a(N-1) = \frac{\beta_1 C_2}{2C_3}(\widetilde{N}_a(N) - 1).$$

Again using statement 2 of Proposition G.5, $\widetilde{N}_a(N) \geq N^\alpha - C_1 \geq M_2^\alpha - C_1 \geq 2$ (since $M_2^\alpha \geq 3C_1 + 2$). As a result,

$$\widetilde{N}_b(N) \geq \frac{\beta_1 C_2}{4C_3} \widetilde{N}_a(N).$$

Taking $\gamma = \min\left\{\frac{1}{2}, \frac{\beta_1 C_2}{4C_3}\right\}$, we have the desired result for AT2 algorithm.

**Case II.II IAT2 Algorithm:** In this case, we have,

$$\log(\widetilde{N}_a(N-1)) + \mathcal{I}_a(N-1) \leq \log(\widetilde{N}_b(N-1)) + \mathcal{I}_b(N-1).$$

Using (11), we have constants $C_2$ and $C_3$ such that,

$$C_2 \frac{\widetilde{N}_1(N-1) \cdot \widetilde{N}_a(N-1)}{\widetilde{N}_1(N-1) + \widetilde{N}_a(N-1)} \leq \log(\widetilde{N}_a(N-1)) + \mathcal{I}_a(N-1)$$

$$\leq \mathcal{I}_b(N-1) + \log(\widetilde{N}_b(N-1)) \leq C_3 \frac{\widetilde{N}_1(N-1) \cdot \widetilde{N}_b(N-1)}{\widetilde{N}_1(N-1) + \widetilde{N}_b(N-1)} + \frac{1}{e}\widetilde{N}_b(N-1).$$

As a result, using the same arguments used for the AT2 algorithm in Case II.I, we can conclude,

$$\widetilde{N}_b(N) \geq \frac{\beta_1}{4} \left( \frac{C_3}{C_2} + \frac{1}{e} \right)^{-1} \widetilde{N}_a(N),$$

for $N > \max\{M_2, T_2\} + 1$.

Now taking $\gamma = \min \left\{ \frac{1}{2}, \frac{\beta_1}{4} \left( \frac{C_2}{C_3} + \frac{1}{e} \right)^{-1} \right\}$, we have the desired result for IAT2 algorithm. $\qquad\square$

**Definition G.9.** We define the random time $T_3 = \max\{M_2, T_2\} + 2$, where $M_2$ is the constant introduced in Lemma G.12, and $T_2$ is the random time defined in Lemma G.11.

**Definition G.10.** We define the random time $T_4$ as $T_4 = \frac{1}{\beta_2}(T_3 + 1)$, where $\beta_2 \in (0, 1)$ is the constant introduced in Corollary G.1.

Note that $\mathbb{E}_{\boldsymbol{\mu}}[T_3] < \infty$, since $\mathbb{E}_{\boldsymbol{\mu}}[T_2] < \infty$. For the same reason, by Definition G.10, $T_4$ satisfies $\mathbb{E}_{\boldsymbol{\mu}}[T_4] < \infty$ since $\mathbb{E}_{\boldsymbol{\mu}}[T_3] < \infty$.

**Lemma G.13 (Sufficient sampling).** *For every $N \geq T_4$, we have $\widetilde{N}_a(N) = \Theta(N)$ for every $a \in [K]$.*

*Proof.* For $a = 1$ and every $N \geq T_4$, we have $\widetilde{N}_1(N) = \Theta(N)$ by Corollary G.1.

Otherwise for $a \neq 1$ and every $N \geq T_4$, we define $A'_N = \arg\max_{b \in [K]/\{1\}} \widetilde{N}_b(N)$. By Corollary G.1,

$$\widetilde{N}_{A'_N}(N) \geq \beta_2 N \geq \beta_2 T_4 = T_3 + 1.$$

Therefore, arm $A'_N$ must have been pulled by the algorithm somewhere between iterations $T_3$ and $N$. Let us define the time $N'$ to be the last time before $N$ when $A'_N$ was pulled. Then $N' \geq T_3 > \max\{M_2, T_2\} + 1$. As a result, using Lemma G.12, we have,

$$\widetilde{N}_a(N') \geq \gamma \widetilde{N}_{A'_N}(N') \overset{(i)}{=} \gamma \widetilde{N}_{A'_N}(N),$$

where (i) follows by definition of $N'$. Since $N \geq N'$, we have $\widetilde{N}_a(N) \geq \widetilde{N}_a(N')$. As a result, we obtain, for every $a \in [K]/\{1\}$ and $N \geq T_4$,

$$\widetilde{N}_a(N) \geq \gamma \widetilde{N}_{A'_N}(N) = \gamma \max_{b \in [K]/\{1\}} \widetilde{N}_b(N) \geq \gamma \beta_2 N,$$

where $\beta_2 \in (0, 1)$ is the constant mentioned in Corollary G.1. Hence we have $\widetilde{N}_a(N) = \Omega(N)$ for $N \geq T_4$ and the lemma follows. $\qquad\square$

### G.6 Proving Lemma G.2

In this appendix, we prove Lemma G.2 from Appendix G.1.2. For improved clarity, we reiterate Lemma G.2 from Appendix G.1.2.

**Statement of Lemma G.2.** *For both AT2 and IAT2 algorithms, there exists constants $M_5 \geq 1$ and $C_1 > 0$ independent of the sample paths, such that for every $N \geq \max\{M_5, T_6\}$, if the algorithm picks an arm $a \in [K]/\{1\}$ at iteration $N$, then it again picks arm $a$ within the next $\lceil C_1 N^{1-3\alpha/8} \rceil$ iterations.*

For every arm $a \in [K]$, iteration $N \geq 1$ and $R \geq 1$, we define the following quantities,

$$\Delta \widetilde{N}_a(N, R) = \widetilde{N}_a(N + R) - \widetilde{N}_a(N),$$

$$b(N, R) = \arg\max_{j \in [K]/\{1\}} \frac{\Delta \widetilde{N}_j(N, R)}{\widetilde{N}_j(N)}, \quad \text{and}$$

$$\tau(N, R) = \max\{ t \leq N + R \mid A_t = b(N, R) \}.$$

For proving Lemma G.2, we need the three technical lemmas: Lemma G.14, G.15 and G.16.

**Lemma G.14.** *For $N \geq T_6$ ($T_6$ is the random time defined in the statement of Proposition G.1 and it satisfies $\mathbb{E}_{\boldsymbol{\mu}}[T_6] < \infty$), we have,*

$$\frac{\Delta \widetilde{N}_1(N,R)}{\Delta \widetilde{N}_{b(N,R)}(N,R)} \leq \frac{\widetilde{N}_1(N)}{\widetilde{N}_{b(N,R)}(N)} + O\left(RN^{-1} + (\Delta \widetilde{N}_{b(N,R)}(N,R))^{-1} N^{1-3\alpha/8}(1+RN^{-1})^3\right).$$
(76)

*Proof.* In the proof, for readability, we use $b$ to denote $b(N,R)$. Also, for every arm $a \in [K]$, we use $\Delta \widetilde{N}_a$ to denote $\Delta \widetilde{N}_a(N,R)$.

By Proposition G.1, we have a constant $C > 0$ independent of the sample paths, such that for $N \geq T_6$,

$$\left| g(\boldsymbol{\mu}, \widetilde{\boldsymbol{N}}(N+R)) - g(\boldsymbol{\mu}, \widetilde{\boldsymbol{N}}(N)) \right| \leq 2CN^{-3\alpha/8}.$$

Therefore, applying the mean value theorem, we have,

$$\left| \frac{\partial g}{\partial N_1}(\boldsymbol{\mu}, \hat{\boldsymbol{N}}) \Delta \widetilde{N}_1 + \sum_{a \in [K]/\{1\}} \frac{\partial g}{\partial N_a}(\boldsymbol{\mu}, \hat{\boldsymbol{N}}) \Delta \widetilde{N}_a \right| \leq 2CN^{-3\alpha/8},$$

where $\hat{\boldsymbol{N}} = (\hat{N}_a)_{a \in [K]}$, with $\hat{N}_a \in \left[ \widetilde{N}_a(N), \widetilde{N}_a(N+R) \right]$ for every $a \in [K]$.

Now expanding the partial derivatives of $g$ with respect to $N_a$'s for $a \in [K]$, we get

$$\left| \Delta \widetilde{N}_1 \sum_{a \in [K]/\{1\}} f(\boldsymbol{\mu}, a, \hat{\boldsymbol{N}}) \frac{\hat{N}_a \Delta_a}{(\hat{N}_1 + \hat{N}_a)^2} - \sum_{a \in [K]/\{1\}} f(\boldsymbol{\mu}, a, \hat{\boldsymbol{N}}) \Delta \widetilde{N}_a \frac{\hat{N}_1 \Delta_a}{(\hat{N}_1 + \hat{N}_a)^2} \right| \leq 2CN^{-3\alpha/8},$$
(77)

where, $f(\boldsymbol{\mu}, a, \hat{\boldsymbol{N}})$ were defined in (13) of Appendix B.

Letting

$$\hat{p}_j = \frac{f(\boldsymbol{\mu}, j, \hat{\boldsymbol{N}}) \frac{\hat{N}_j \Delta_j}{(\hat{N}_1 + \hat{N}_j)^2}}{\sum_{a \neq 1} f(\boldsymbol{\mu}, a, \hat{\boldsymbol{N}}) \frac{\hat{N}_a \Delta_a}{(\hat{N}_1 + \hat{N}_a)^2}}$$

for every arm $j \in [K]/\{1\}$, we have,

$$\left| \frac{\Delta \widetilde{N}_1}{\hat{N}_1} - \sum_{a \neq 1} \hat{p}_a \frac{\Delta \widetilde{N}_a}{\hat{N}_a} \right| \leq 2CN^{-3\alpha/8} \times \hat{N}_1^{-1} \left( \sum_{a \in [K]/\{1\}} f(\boldsymbol{\mu}, a, \hat{\boldsymbol{N}}) \frac{\hat{N}_a \Delta_a}{(\hat{N}_1 + \hat{N}_a)^2} \right)^{-1}. \quad (78)$$

We know from (14) of Appendix B that $f(\boldsymbol{\mu}, a, \boldsymbol{N}) = \Theta\left( \frac{N_a}{N_1}\left(1 + \frac{N_a}{N_1}\right)^2 \right)$. As a result,

$$\hat{N}_1^{-1} \left( \sum_{a \neq 1} f(\boldsymbol{\mu}, a, \hat{\boldsymbol{N}}) \frac{\hat{N}_a \Delta_a}{(\hat{N}_1 + \hat{N}_a)^2} \right)^{-1} = \Theta\left( \frac{\hat{N}_1^2}{\sum_{a \in [K]/\{1\}} \hat{N}_a^2} \right)$$

By Lemma G.13, we have $\hat{N}_a \geq \widetilde{N}_a = \Theta(N)$ and $\hat{N}_1 \leq N + R$ for $N \geq T_6$. As a result, we get,

$$\widetilde{N}_1^{-1} \left( \sum_{a \neq 1} f(\boldsymbol{\mu}, a, \hat{\boldsymbol{N}}) \frac{\hat{N}_a \Delta_a}{(\hat{N}_1 + \hat{N}_a)^2} \right)^{-1} = O\left((1 + RN^{-1})^2\right).$$

Putting this in (78), we obtain

$$\left| \frac{\Delta \widetilde{N}_1}{\hat{N}_1} - \sum_{a \neq 1} \hat{p}_a \frac{\Delta \widetilde{N}_a}{\hat{N}_a} \right| = O\left( N^{-3\alpha/8}(1 + RN^{-1})^2 \right),$$

which further implies,

$$\frac{\Delta\widetilde{N}_1}{\hat{N}_1} \leq \sum_{a \neq 1} \hat{p}_a \frac{\Delta\widetilde{N}_a}{\hat{N}_a} + O\left(N^{-3\alpha/8}(1 + RN^{-1})^2\right)$$

$$\leq \sum_{a \neq 1} \hat{p}_a \frac{\Delta\widetilde{N}_a}{\widetilde{N}_a} + O\left(N^{-3\alpha/8}(1 + RN^{-1})^2\right) \quad \text{(since } \widetilde{N}_a \geq \hat{N}_a\text{)}.$$

Let $b = \arg\max_{a \neq 1} \frac{\Delta\widetilde{N}_a}{\widetilde{N}_a}$. The above inequality implies,

$$\frac{\Delta\widetilde{N}_1}{\hat{N}_1} \leq \frac{\Delta\widetilde{N}_b}{\widetilde{N}_b} + O\left(N^{-3\alpha/8}(1 + RN^{-1})^2\right).$$

Now multiplying both sides by $\hat{N}_1/\Delta\widetilde{N}_b$, we get,

$$\frac{\Delta\widetilde{N}_1}{\Delta\widetilde{N}_b} \leq \frac{\hat{N}_1}{\widetilde{N}_b} + O\left(\Delta\widetilde{N}_b^{-1}\hat{N}_1 N^{-3\alpha/8}(1 + RN^{-1})^2\right)$$

$$\leq \frac{\hat{N}_1}{\widetilde{N}_b} + O\left(\Delta\widetilde{N}_b^{-1} N^{1-3\alpha/8}(1 + RN^{-1})^3\right) \quad \text{(since } \hat{N}_1 \leq N + R\text{)}.$$

(76) follows from the above inequality upon observing that, $\hat{N}_1 \leq \widetilde{N}_1 + R$ and $\widetilde{N}_b = \Omega(N)$ (by Lemma G.13). $\qquad\square$

**Lemma G.15.** *There exists constants $M_{51} \geq 1$, $D_1, D_2 > 0$ independent of the sample paths such that, for $N \geq \max\{M_{51}, T_6\}$, we have $\lceil D_1 N^{1-3\alpha/8} \rceil < N$, and for all $R \in \{\lceil D_1 N^{1-3\alpha/8} \rceil, \lceil D_1 N^{1-3\alpha/8} \rceil + 1, \ldots, N\}$, we have $\Delta\widetilde{N}_{b(N,R)}(N, R) \geq D_2 R$.*

*Proof.* To simplify notation, we adopt $b$ to denote $b(N, R)$ and, for every arm $j \in [K]$, we use $\Delta\widetilde{N}_j$ to denote $\Delta\widetilde{N}_j(N, R)$.

By (76) of Lemma G.14, there exists a constant $D_3 > 0$ independent of the sample paths, such that, for all $N \geq T_6$ and $R \in \{1, 2, \ldots, N\}$,

$$\Delta\widetilde{N}_1 \leq \Delta\widetilde{N}_b \times \left(\frac{\widetilde{N}_1}{\widetilde{N}_b} + D_3\left(1 + \Delta\widetilde{N}_b^{-1} N^{1-3\alpha/8}\right)\right) \quad \text{(we have } RN^{-1} \leq 1\text{)}$$

$$\leq \left(\frac{\widetilde{N}_1}{\widetilde{N}_b} + D_3\right) \Delta\widetilde{N}_b + D_3 N^{1-3\alpha/8}$$

Since $\widetilde{N}_1(N)$ and $\widetilde{N}_b(N)$ are both $\Theta(N)$ (by Lemma G.13), we have a constant $D_4 > 0$ such that, $\frac{\widetilde{N}_1(N)}{\widetilde{N}_b(N)} + D_3 \leq D_4$ for every $N \geq T_6$. Using this in the above inequality, we get,

$$\Delta\widetilde{N}_1 \leq D_4 \Delta\widetilde{N}_b + D_3 N^{1-3\alpha/8}. \tag{79}$$

We take $D_1 = 2D_3$ and $M_{51} \geq 1$ to be the smallest number such that, every $N \geq M_{51}$ satisfies $\lceil D_1 N^{1-3\alpha/8} \rceil < N$.

Now if $N \geq \max\{M_{51}, T_6\}$ and $R \in \{\lceil D_1 N^{1-3\alpha/8} \rceil, \lceil D_1 N^{1-3\alpha/8} \rceil + 1, \ldots, N\}$, from (79) we get,

$$\Delta\widetilde{N}_1 \leq D_4 \Delta\widetilde{N}_b + D_3 N^{1-3\alpha/8}$$

$$\leq D_4 \Delta\widetilde{N}_b + \frac{R}{2}. \tag{80}$$

By definition of $b$, we have,

$$\frac{\widetilde{N}_a}{\widetilde{N}_b} \Delta\widetilde{N}_b \geq \Delta\widetilde{N}_a \quad \text{for every } a \in [K]/\{1\}.$$

Again, since $\widetilde{N}_a(N) = \Theta(N)$ for every $a \in [K]$ (by Lemma G.13), there exists a constant $D_5 > 0$ independent of the sample paths such that,

$$D_5 \Delta \widetilde{N}_b \geq \Delta \widetilde{N}_a \quad \text{for every } a \in [K]/\{1, b\}.$$

As a result,

$$
\begin{aligned}
R &= \sum_{a \in [K]/\{1,b\}} \Delta \widetilde{N}_a + \Delta \widetilde{N}_1 + \Delta \widetilde{N}_b \\
&\leq (K-2)D_5 \Delta \widetilde{N}_b + D_4 \Delta \widetilde{N}_b + \frac{R}{2} + \Delta \widetilde{N}_b \quad \text{(using (80))} \\
&\leq (1 + (K-2)D_5 + D_4) \Delta \widetilde{N}_b + \frac{R}{2}, \\
\implies \Delta \widetilde{N}_b &\geq \frac{1}{2(1 + (K-2)D_5 + D_4)} R.
\end{aligned}
$$

Now taking $D_2 = \frac{1}{2(1+(K-2)D_5+D_4)} > 0$, we have the desired conclusion. $\qquad \square$

**Lemma G.16.** *For AT2 (1) and IAT2 (2) algorithms, there exists constants $C_3, C_4 > 0$ independent of the sample paths, such that, for $N \geq T_6$ and $R \in \{1, 2, \ldots, N\}$, if the algorithm pulls some arm $a \in [K]/\{1\}$ at iteration $N$ and doesn't pull $a$ for the next $R$ iterations, then,*

$$
\begin{aligned}
\textbf{AT2:} \quad \mathcal{I}_a(N+R) - \mathcal{I}_{b(N,R)}(N+R) &\leq -C_3 \Delta \widetilde{N}_{b(N,R)}(N,R) + C_4 N^{1-3\alpha/8} \\
&\quad + O\left(\left(N^{-3\alpha/8} + RN^{-1}\right)R\right), \quad (81)
\end{aligned}
$$

$$
\begin{aligned}
\textbf{IAT2:} \quad \mathcal{I}_a^{(m)}(N+R) - \mathcal{I}_{b(N,R)}^{(m)}(N+R) &\leq -C_3 \Delta \widetilde{N}_{b(N,R)}(N,R) + C_4 N^{1-3\alpha/8} \\
&\quad + O\left(\left(N^{-3\alpha/8} + RN^{-1}\right)R\right), \quad (82)
\end{aligned}
$$

*where for every alternative arm $j \in [K]/\{1\}$,*

$$\mathcal{I}_j^{(m)}(N) = \mathcal{I}_j(N) + \log(\widetilde{N}_j(N))$$

*is the modified empirical index of that arm at iteration $N$*

*Proof.* For cleaner presentation, we use $b$ to denote $b(N, R)$. We also use $\widetilde{\mu}_j$, $\widetilde{x}_{1,j}$, $\Delta \widetilde{N}_j$ and $\widetilde{N}_j$, respectively, to denote $\widetilde{\mu}_j(N)$, $\widetilde{x}_{1,j}(N)$, $\Delta \widetilde{N}_j(N, R)$ and $\widetilde{N}_j(N)$ for every arm $j \in [K]$, whenever it doesn't cause confusion.

**AT2 Algorithm:** Let

$$S_{a,b}(\widetilde{\boldsymbol{N}}(N), \widetilde{\boldsymbol{\mu}}(N)) = \mathcal{I}_a(N) - \mathcal{I}_b(N)$$

denote the the difference between the empirical indexes of the two arms $a$ and $b$ at iteration $N$. Note that $S_{a,b}(\widetilde{\boldsymbol{N}}, \widetilde{\boldsymbol{\mu}})$ depends only on the tuple of variables $(\widetilde{N}_1, \widetilde{N}_a, \widetilde{N}_b, \widetilde{\mu}_1, \widetilde{\mu}_a, \widetilde{\mu}_b)$. In the following argument, we apply mean value theorem over $S_{a,b}$ and the empirical indexes $\mathcal{I}_a$ and $\mathcal{I}_b$, treating them as functions of $\widetilde{N}_1, \widetilde{N}_a, \widetilde{N}_b, \widetilde{\mu}_1, \widetilde{\mu}_a$, and $\widetilde{\mu}_b$.

Using the multivariate mean value theorem, we have,

$$
\begin{aligned}
S_{a,b}(\widetilde{\boldsymbol{N}}(N+R), \widetilde{\boldsymbol{\mu}}(N+R)) - S_{a,b}(\widetilde{\boldsymbol{N}}(N), \widetilde{\boldsymbol{\mu}}(N)) &= \sum_{j=1,a,b} \frac{\partial S_{a,b}}{\partial \mu_j}(\hat{\boldsymbol{N}}, \hat{\boldsymbol{\mu}}) \Delta \widetilde{\mu}_j \\
&\quad + \frac{\partial S_{a,b}}{\partial N_1}(\hat{\boldsymbol{N}}, \hat{\boldsymbol{\mu}}) \cdot \Delta \widetilde{N}_1 + \frac{\partial S_{a,b}}{\partial N_b}(\hat{\boldsymbol{N}}, \hat{\boldsymbol{\mu}}) \cdot \Delta \widetilde{N}_b, \quad (83)
\end{aligned}
$$

where $\Delta \widetilde{\mu}_j = \widetilde{\mu}_j(N+R) - \widetilde{\mu}_j(N)$ for $j = 1, a, b$, and $(\hat{\boldsymbol{N}}, \hat{\boldsymbol{\mu}}) = (\hat{N}_1, \hat{N}_a, \hat{N}_b, \hat{\mu}_1, \hat{\mu}_a, \hat{\mu}_b)$ such that, for $j = 1, a, b$, $\hat{\mu}_j$ lies between $\widetilde{\mu}_j(N)$ and $\widetilde{\mu}_j(N+R)$, and $\hat{N}_j$ lies in $\left[\widetilde{N}_j(N), \widetilde{N}_j(N+R)\right]$.

Note that there is no contribution on the RHS in (83) due to $\widetilde{N}_a$, because $\widetilde{N}_a(\cdot)$ doesn't change during

iterations $N+1, N+2, \ldots, N+R$, owing to our assumption that the algorithm doesn't pull arm $a$ during the mentioned iterations.

First we consider the partial derivatives of $S_{j,b}$ with respect to $\mu_1, \mu_j$ and $\mu_b$ in (83). We have

$$\frac{\partial S_{a,b}}{\partial \mu_1}(\hat{\boldsymbol{N}}, \hat{\boldsymbol{\mu}}) = \hat{N}_1 \left( d_1(\hat{\mu}_1, \hat{x}_{1,a}) - d_1(\hat{\mu}_1, \hat{x}_{1,b}) \right),$$

where $\hat{x}_{1,j} = \frac{\hat{N}_1 \hat{\mu}_1 + \hat{N}_j \hat{\mu}_j}{\hat{N}_1 + \hat{N}_j}$ for $j = a, b$.

Since $R \leq N$, we have for $j = 1, a, b$, $\hat{N}_j \leq \widetilde{N}_j + N = \Theta(N)$ (by Lemma G.13). As a result, using (7) of Appendix B, both $d_1(\hat{\mu}_1, \hat{x}_{1,a})$ and $d_1(\hat{\mu}_1, \hat{x}_{1,b})$ are $\Theta(1)$. Using this, $\left| \frac{\partial S_{a,b}}{\partial \mu_1}(\hat{\boldsymbol{N}}, \hat{\boldsymbol{\mu}}) \right|$ is $O(N)$.

Similarly,

$$\frac{\partial S_{a,b}}{\partial \mu_a}(\hat{\boldsymbol{N}}, \hat{\boldsymbol{\mu}}) = \hat{N}_a d_1(\hat{\mu}_a, \hat{x}_{1,a}) \text{ and } \frac{\partial S_{a,b}}{\partial \mu_b}(\hat{\boldsymbol{N}}, \hat{\boldsymbol{\mu}}) = -\hat{N}_b d_1(\hat{\mu}_b, \hat{x}_{1,b}).$$

Using the same argument as for the partial derivative of $S_{a,b}$ with respect to $\mu_1$, we have $\left| \frac{\partial S_{a,b}}{\partial \mu_j}(\hat{\boldsymbol{N}}, \hat{\boldsymbol{\mu}}) \right| = O(N)$ for both $j = a, b$.

Therefore, the contribution in the RHS of (83) due to the noisiness in the empirical means $\widetilde{\mu}_1, \widetilde{\mu}_a, \widetilde{\mu}_b$ is bounded above by,

$$\sum_{j=1,a,b} \frac{\partial S_{a,b}}{\partial \mu_j}(\hat{\boldsymbol{N}}, \hat{\boldsymbol{\mu}}) \Delta \widetilde{\mu}_j \leq \sum_{j=1,a,b} \left| \frac{\partial S_{a,b}}{\partial \mu_j}(\hat{\boldsymbol{N}}, \hat{\boldsymbol{\mu}}) \right| \cdot |\Delta \widetilde{\mu}_j|$$

$$\overset{(1)}{=} O(N) \times O(N^{-3\alpha/8})$$

$$= O(N^{1-3\alpha/8}), \tag{84}$$

where (1) follows since for $N \geq T_0$ and $j = 1, a, b$, $|\Delta \widetilde{\mu}_j| = O(N^{-3\alpha/8})$.

Now considering the partial derivative of $S_{a,b}(\cdot)$ with respect to $N_1$, we get

$$\frac{\partial S_{a,b}}{\partial N_1}(\hat{\boldsymbol{N}}, \hat{\boldsymbol{\mu}}) = d(\hat{\mu}_1, \hat{x}_{1,a}) - d(\hat{\mu}_1, \hat{x}_{1,b}).$$

Using Lemma G.13, $\widetilde{N}_j(N)$ and $\widetilde{N}_j(N+R)$ are both $\Theta(N)$ for all $j = 1, a, b$ and $N \geq T_6$, since $R \leq N$. As a result, using (7), (8), we have,

$$\left| \frac{\partial^2 S_{a,b}}{\partial \mu_j \partial N_k}(\boldsymbol{N}', \boldsymbol{\mu}') \right| = O(1), \quad \text{and} \quad \left| \frac{\partial^2 S_{a,b}}{\partial N_j \partial N_k}(\boldsymbol{N}', \boldsymbol{\mu}') \right| = O(N^{-1})$$

for every $j, k \in \{1, a, b\}$, and tuple $(\boldsymbol{N}', \boldsymbol{\mu}') = (N_1', N_a', N_b', \mu_1', \mu_a', \mu_b')$ having $N_i' \in \left[ \widetilde{N}_i(N), \widetilde{N}_i(N+R) \right]$ and $\mu_i'$ lying between $\hat{\mu}_i$ and $\widetilde{\mu}_i$, for every $i = 1, a, b$.

Therefore, applying the mean value theorem, we get,

$$\frac{\partial S_{a,b}}{\partial N_1}(\hat{\boldsymbol{N}}, \hat{\boldsymbol{\mu}}) = d(\widetilde{\mu}_1, \widetilde{x}_{1,a}) - d(\widetilde{\mu}_1, \widetilde{x}_{1,b}) + O\left( \sum_{j=1,a,b} |\hat{\mu}_j - \widetilde{\mu}_j| \right) + O\left( N^{-1} \sum_{j=1,b} \Delta \widetilde{N}_j \right)$$

$$= d(\widetilde{\mu}_1, \widetilde{x}_{1,a}) - d(\widetilde{\mu}_1, \widetilde{x}_{1,b}) + O\left( N^{-3\alpha/8} + RN^{-1} \right).$$

Similarly, considering the partial derivative with respect to $N_b$,

$$\frac{\partial S_{a,b}}{\partial N_b}(\hat{\boldsymbol{N}}, \hat{\boldsymbol{\mu}}) = -d(\widetilde{\mu}_b, \widetilde{x}_{1,b}) + O\left( N^{-3\alpha/8} + RN^{-1} \right).$$

Now, using all these upper bounds in the RHS of (83), we obtain

$$S_{a,b}(\widetilde{\boldsymbol{N}}(N+R), \widetilde{\boldsymbol{\mu}}(N+R)) - S_{a,b}(\widetilde{\boldsymbol{N}}(N), \widetilde{\boldsymbol{\mu}}(N))$$
$$\leq \ \Delta\widetilde{N}_1 \cdot (d(\widetilde{\mu}_1, \widetilde{x}_{1,a}) - d(\widetilde{\mu}_1, \widetilde{x}_{1,b})) - \Delta\widetilde{N}_b \cdot d(\widetilde{\mu}_b, \widetilde{x}_{1,b})$$
$$+ O\Big(N^{1-3\alpha/8} + R(N^{-3\alpha/8} + RN^{-1})\Big). \tag{85}$$

Since arm $a$ was pulled in iteration $N$, we have $\mathcal{I}_a(N-1) \leq \mathcal{I}_b(N-1)$. Also since arm $b$ was not pulled in iteration $N$, its empirical index remains unchanged, *i.e.* $\mathcal{I}_b(N-1) = \mathcal{I}_b(N)$. Combining these two observations, we have $\mathcal{I}_a(N-1) \leq \mathcal{I}_b(N)$.

As a result,

$$\begin{aligned}
S_{a,b}(\widetilde{\boldsymbol{N}}(N), \widetilde{\boldsymbol{\mu}}(N)) \ &= \ \mathcal{I}_a(N) - \mathcal{I}_b(N) \\
&= \ (\mathcal{I}_a(N) - \mathcal{I}_a(N-1)) + (\mathcal{I}_a(N-1) - \mathcal{I}_b(N)) \\
&\leq \ \mathcal{I}_a(N) - \mathcal{I}_a(N-1) \\
&= \ I_a(N) - I_a(N-1) + O(N^{1-3\alpha/8}) \quad \text{(using Lemma G.3)} \\
&\leq \ d(\mu_a, \mu_1) + O(N^{1-3\alpha/8}) = O(N^{1-3\alpha/8}), \tag{86}
\end{aligned}$$

where the last step follows from the fact that, the partial derivatives of $I_a(\cdot)$ with respect to $N_a$ is $d(\mu_a, x_{1,a}(N)) \leq d(\mu_a, \mu_1)$. As a result, since arm $a$ was pulled in iteration $N$, the increment $I_a(N) - I_a(N-1)$ in $I_a(\cdot)$ is upper bounded by $d(\mu_a, \mu_1)$.

Therefore (85) implies,

$$\begin{aligned}
S_{a,b}(\widetilde{\boldsymbol{N}}(N+R), \widetilde{\boldsymbol{\mu}}(N+R)) \ &\leq \ \Delta\widetilde{N}_1 \cdot (d(\widetilde{\mu}_1, \widetilde{x}_{1,a}) - d(\widetilde{\mu}_1, \widetilde{x}_{1,b})) - \Delta\widetilde{N}_b \cdot d(\widetilde{\mu}_b, \widetilde{x}_{1,b}) \\
&\quad + S_{a,b}(\widetilde{\boldsymbol{N}}(N), \widetilde{\boldsymbol{\mu}}(N)) + O\left(\left(N^{-3\alpha/8} + RN^{-1}\right)R\right) \\
&\leq \ \Delta\widetilde{N}_1 \cdot (d(\widetilde{\mu}_1, \widetilde{x}_{1,a}) - d(\widetilde{\mu}_1, \widetilde{x}_{1,b})) - \Delta\widetilde{N}_b \cdot d(\widetilde{\mu}_b, \widetilde{x}_{1,b}) \\
&\quad + O\left(N^{1-3\alpha/8} + \left(N^{-3\alpha/8} + RN^{-1}\right)R\right). \tag{87}
\end{aligned}$$

Now we consider two possibilities:

**Case I** $d(\widetilde{\mu}_1, \widetilde{x}_{1,a}) - d(\widetilde{\mu}_1, \widetilde{x}_{1,b}) < 0$ **:** In this case, by (85), $S_{a,b}(\widetilde{\boldsymbol{N}}(N+R), \widetilde{\boldsymbol{\mu}}(N+R))$ is upper bounded by,

$$-\Delta\widetilde{N}_b \cdot d(\widetilde{\mu}_b, \widetilde{x}_{1,b}) + O\left(N^{1-3\alpha/8} + \left(N^{-3\alpha/8} + RN^{-1}\right)R\right).$$

By Lemma G.13 and (6), since both $\widetilde{N}_1(N)$ and $\widetilde{N}_b(N)$ are $\Theta(N)$, we have $d(\widetilde{\mu}_b, \widetilde{x}_{1,b}) = \Theta(1)$. As a result, (81) follows.

**Case II** $d(\widetilde{\mu}_1, \widetilde{x}_{1,a}) - d(\widetilde{\mu}_1, \widetilde{x}_{1,b}) \geq 0$ **:** In this case, the RHS of (85) can be rewritten as,

$$\Delta\widetilde{N}_b\left(\frac{\Delta\widetilde{N}_1}{\Delta\widetilde{N}_b}(d(\widetilde{\mu}_1, \widetilde{x}_{1,a}) - d(\widetilde{\mu}_1, \widetilde{x}_{1,b})) - d(\widetilde{\mu}_b, \widetilde{x}_{1,b})\right) + O\left(N^{1-3\alpha/8} + \left(N^{-3\alpha/8} + RN^{-1}\right)R\right). \tag{88}$$

Using (76) in Lemma G.14, the upper bound becomes

$$\begin{aligned}
\Delta\widetilde{N}_b&\left(\frac{\widetilde{N}_1}{\widetilde{N}_b}(d(\widetilde{\mu}_1, \widetilde{x}_{1,a}) - d(\widetilde{\mu}_1, \widetilde{x}_{1,b})) - d(\widetilde{\mu}_b, \widetilde{x}_{1,b})\right) \\
&+ O\left(N^{1-3\alpha/8}(1 + RN^{-1})^3 + \left(N^{-3\alpha/8} + RN^{-1}\right)R\right) \\
= \Delta\widetilde{N}_b&\left(\frac{\widetilde{N}_1}{\widetilde{N}_b}(d(\widetilde{\mu}_1, \widetilde{x}_{1,a}) - d(\widetilde{\mu}_1, \widetilde{x}_{1,b})) - d(\widetilde{\mu}_b, \widetilde{x}_{1,b})\right) \\
&+ O\left(N^{1-3\alpha/8} + \left(N^{-3\alpha/8} + RN^{-1}\right)R\right) \quad \text{(since } R \leq N\text{)}. \tag{89}
\end{aligned}$$

By (86), we have,

$$\mathcal{I}_a(N) \ \le \ \mathcal{I}_b(N) + O(N^{1-3\alpha/8}). \tag{90}$$

Expanding the empirical index terms, we get,

$$\widetilde{N}_1 d(\widetilde{\mu}_1, \widetilde{x}_{1,a}) + \widetilde{N}_a d(\widetilde{\mu}_a, \widetilde{x}_{1,a}) \ \le \ \widetilde{N}_1 d(\widetilde{\mu}_1, \widetilde{x}_{1,b}) + \widetilde{N}_b d(\widetilde{\mu}_b, \widetilde{x}_{1,b}) + O(N^{1-3\alpha/8}).$$

By Lemma G.13 we have $\widetilde{N}_b = \Theta(N)$. As a result, upon dividing both sides of the above inequality by $\widetilde{N}_b$ and after some re-arrangement of terms, we obtain,

$$\frac{\widetilde{N}_1}{\widetilde{N}_b}(d(\widetilde{\mu}_1, \widetilde{x}_{1,j}) - d(\widetilde{\mu}_1, \widetilde{x}_{1,b})) - d(\widetilde{\mu}_b, \widetilde{x}_{1,b}) \ \le \ -\frac{\widetilde{N}_j}{\widetilde{N}_b}d(\widetilde{\mu}_j, \widetilde{x}_{1,j}) + O(N^{-3\alpha/8}).$$

Using the above inequality in (89), we get the upper bound,

$$-\Delta\widetilde{N}_b\left(\frac{\widetilde{N}_j}{\widetilde{N}_b}d(\widetilde{\mu}_a, \widetilde{x}_{1,a})\right) + O\left(N^{1-3\alpha/8} + \left(N^{-3\alpha/8} + RN^{-1}\right)R\right) \tag{91}$$

By Lemma G.13 and (6), we have, $\frac{\widetilde{N}_j}{\widetilde{N}_b}d(\widetilde{\mu}_j, \widetilde{x}_{1,j}) = \Theta(1)$. Therefore (81) follows.

**IAT2 Algorithm:** The proof of (82) for IAT2 is very similar to the proof of (81) for AT2. We first consider the difference of the modified empirical indexes between the two arms $a$ and $b$,

$$S_{a,b}^{(m)}(\widetilde{\boldsymbol{N}}(N), \widetilde{\boldsymbol{\mu}}(N)) \ = \ \mathcal{I}_a^{(m)}(N) - \mathcal{I}_b^{(m)}(N).$$

Following a similar procedure as the AT2 algorithm, we apply the mean value theorem to obtain,

$$S_{a,b}^{(m)}(\widetilde{\boldsymbol{N}}(N+R), \widetilde{\boldsymbol{\mu}}(N+R)) - S_{a,b}^{(m)}(\widetilde{\boldsymbol{N}}(N), \widetilde{\boldsymbol{\mu}}(N)) \ = \ \sum_{j=1,a,b} \frac{\partial S_{a,b}^{(m)}}{\partial \mu_j}(\hat{\boldsymbol{N}}, \hat{\boldsymbol{\mu}}) \cdot \Delta\widetilde{\mu}_j$$

$$+ \frac{\partial S_{a,b}^{(m)}}{\partial N_1}(\hat{\boldsymbol{N}}, \hat{\boldsymbol{\mu}}) \cdot \Delta\widetilde{N}_1 + \frac{\partial S_{a,b}^{(m)}}{\partial N_b}(\hat{\boldsymbol{N}}, \hat{\boldsymbol{\mu}}) \cdot \Delta\widetilde{N}_b, \tag{92}$$

where $\Delta\widetilde{\mu}_j = \widetilde{\mu}_j(N+R) - \widetilde{\mu}_j(N)$, and $(\hat{\boldsymbol{N}}, \hat{\boldsymbol{\mu}}) = (\hat{N}_1, \hat{N}_a, \hat{N}_b, \hat{\mu}_1, \hat{\mu}_a, \hat{\mu}_b)$, such that, $\hat{\mu}_j$ lies between $\widetilde{\mu}_j(N)$ and $\widetilde{\mu}_j(N+R)$, and $\hat{N}_j \in \left[\widetilde{N}_j(N), \widetilde{N}_j(N+R)\right]$, for every $j = 1, a, b$.

The contribution to (92) due to noise in $\widetilde{\boldsymbol{\mu}}$ is $O(N^{1-3\alpha/8})$ by the same argument we proved (84) for AT2.

Now we consider the partial derivatives of $S_{a,b}^{(m)}$ with respect to $N_1, N_a, N_b$. Following the same steps as used for AT2 algorithm, we have,

$$\frac{\partial S_{a,b}^{(m)}}{\partial N_1}(\hat{\boldsymbol{N}}, \hat{\boldsymbol{\mu}}) \ = \ \frac{\partial S_{a,b}}{\partial N_1}(\hat{\boldsymbol{N}}, \hat{\boldsymbol{\mu}}) \ = \ d(\widetilde{\mu}_1, \widetilde{x}_{1,a}) - d(\widetilde{\mu}_1, \widetilde{x}_{1,b}) + O(N^{-3\alpha/8} + RN^{-1}), \text{ and}$$

$$\frac{\partial S_{a,b}^{(m)}}{\partial N_b}(\hat{\boldsymbol{N}}, \hat{\boldsymbol{\mu}}) \ = \ \frac{\partial S_{a,b}}{\partial N_b}(\hat{\boldsymbol{N}}, \hat{\boldsymbol{\mu}}) - \frac{1}{\hat{N}_b} \ \le \ -d(\widetilde{\mu}_b, \widetilde{x}_{1,b}) + O(N^{-3\alpha/8} + RN^{-1}).$$

As a result, the contribution to (92) due to $\widetilde{N}_1$ and $\widetilde{N}_b$ is,

$$\frac{\partial S_{a,b}^{(m)}}{\partial N_1}(\hat{\boldsymbol{N}}, \hat{\boldsymbol{\mu}}) \cdot \Delta\widetilde{N}_1 + \frac{\partial S_{a,b}^{(m)}}{\partial N_b}(\hat{\boldsymbol{N}}, \hat{\boldsymbol{\mu}}) \cdot \Delta\widetilde{N}_b$$

$$\le \ \Delta\widetilde{N}_1 \cdot (d(\widetilde{\mu}_1, \widetilde{x}_{1,a}) - d(\widetilde{\mu}_1, \widetilde{x}_{1,b})) - \Delta\widetilde{N}_b \cdot d(\widetilde{\mu}_b, \widetilde{x}_{1,b}) + O\left(\left(N^{-3\alpha/8} + RN^{-1}\right)R\right).$$

Therefore, adding the contributions of the noise in $(\widetilde{\mu}_j)_{j=1,a,b}$, (92) can be further upper bounded by,

$$S_{a,b}^{(m)}(\widetilde{\boldsymbol{N}}(N+R), \widetilde{\boldsymbol{\mu}}(N+R)) - S_{a,b}^{(m)}(\widetilde{\boldsymbol{N}}(N), \widetilde{\boldsymbol{\mu}}(N))$$
$$\leq \ \Delta\widetilde{N}_1 \cdot (d(\widetilde{\mu}_1, \widetilde{x}_{1,a}) - d(\widetilde{\mu}_1, \widetilde{x}_{1,b})) - \Delta\widetilde{N}_b \cdot d(\widetilde{\mu}_b, \widetilde{x}_{1,b})$$
$$+ O\left(N^{1-3\alpha/8} + \left(N^{-3\alpha/8} + RN^{-1}\right)R\right). \tag{93}$$

Now, we find an upper bound to $S_{a,b}^{(m)}(\widetilde{\boldsymbol{N}}(N), \widetilde{\boldsymbol{\mu}}(N))$. Since the algorithm pulls arm $a$ and doesn't pull arm $b$ at iteration $N$, we must have

$$\mathcal{I}_a^{(m)}(N-1) \ \leq \ \mathcal{I}_b^{(m)}(N-1) \ = \ \mathcal{I}_b^{(m)}(N).$$

Therefore,

$$\begin{aligned} S_{a,b}^{(m)}(\widetilde{\boldsymbol{N}}(N), \widetilde{\boldsymbol{\mu}}(N)) &= \mathcal{I}_a^{(m)}(N) - \mathcal{I}_b^{(m)}(N) \\ &= (\mathcal{I}_a^{(m)}(N) - \mathcal{I}_a^{(m)}(N-1)) + (\mathcal{I}_a^{(m)}(N-1) - \mathcal{I}_b^{(m)}(N)) \\ &\leq \mathcal{I}_a^{(m)}(N) - \mathcal{I}_a^{(m)}(N-1) \\ &= \mathcal{I}_a(N) - \mathcal{I}_a(N-1) + \log\left(\frac{\widetilde{N}_a(N)}{\widetilde{N}_a(N-1)}\right) \\ &\stackrel{(1)}{=} \mathcal{I}_a(N) - \mathcal{I}_a(N-1) + O(1) \\ &\stackrel{(2)}{=} O(N^{1-3\alpha/8}), \end{aligned} \tag{94}$$

where (1) follows from the fact that $\widetilde{N}_a(N) = \Theta(N)$ for $N \geq T_6$, and (2) follows using the arguments used while bounding $\mathcal{I}_a(N) - \mathcal{I}_a(N-1)$ in (86).

Putting this in (93), we get,

$$S_{a,b}^{(m)}(\widetilde{\boldsymbol{N}}(N+R), \widetilde{\boldsymbol{\mu}}(N+R)) \leq \Delta\widetilde{N}_1 \cdot (d(\widetilde{\mu}_1, \widetilde{x}_{1,a}) - d(\widetilde{\mu}_1, \widetilde{x}_{1,b})) - \Delta\widetilde{N}_b \cdot d(\widetilde{\mu}_b, \widetilde{x}_{1,b})$$
$$+ O\left(N^{1-3\alpha/8} + R(N^{-3\alpha/8} + RN^{-1})\right). \tag{95}$$

Now there can be two cases.

**Case I** $d(\widetilde{\mu}_1, \widetilde{x}_{1,a}) - d(\widetilde{\mu}_1, \widetilde{x}_{1,b}) < 0$ **:** Then, using the same argument as was used for Case I of AT2 algorithm, we get (82).

**Case II** $d(\widetilde{\mu}_1, \widetilde{x}_{1,a}) - d(\widetilde{\mu}_1, \widetilde{x}_{1,b}) \geq 0$ **:** Using (76) of Lemma G.12, the RHS in (95) is upper bounded by,

$$\Delta\widetilde{N}_b\left(\frac{\widetilde{N}_1}{\widetilde{N}_b}(d(\widetilde{\mu}_1, \widetilde{x}_{1,a}) - d(\widetilde{\mu}_1, \widetilde{x}_{1,b})) - d(\widetilde{\mu}_b, \widetilde{x}_{1,b})\right) + O\left(N^{1-3\alpha/8} + \left(N^{-3\alpha/8} + RN^{-1}\right)R\right).$$

Using (94), we have

$$\mathcal{I}_a^{(m)}(N) \ \leq \ \mathcal{I}_b^{(m)}(N) + O\left(N^{1-3\alpha/8}\right),$$

which implies,

$$\mathcal{I}_a(N) \ \leq \ \mathcal{I}_b(N) + \log\left(\frac{\widetilde{N}_b}{\widetilde{N}_a}\right) + O(N^{1-3\alpha/8}) \ \stackrel{(1)}{=} \ \mathcal{I}_b(N) + O(N^{1-3\alpha/8}),$$

where (1) follows from the fact that $\widetilde{N}_a(N)$ and $\widetilde{N}_b(N)$ are both $\Theta(N)$, causing $\log\left(\frac{\widetilde{N}_a(N)}{\widetilde{N}_b(N)}\right) = O(1)$ for $N \geq T_6$. Note that the above inequality is same as (90) obtained in Case II of AT2. Now (82) follows using exactly the same argument as in the Case II of AT2 after (90). $\qquad\square$

For every $a \in [K]/\{1\}$, $N \geq T_6$ and $R \in \{1, 2, \ldots, N\}$, we have,

$$
\begin{aligned}
|\mathcal{I}_a(N+R) - \mathcal{I}_a(N+R-1)| &\overset{(1)}{=} |I_a(N+R) - I_a(N+R-1)| + O((N+R)^{1-3\alpha/8}) \\
&\overset{(2)}{=} |I_a(N+R) - I_a(N+R-1)| + O(N^{1-3\alpha/8}) \\
&\overset{(3)}{\leq} \max\{d(\mu_1, \mu_a), d(\mu_a, \mu_1)\} + O(N^{1-3\alpha/8}) = O(N^{1-3\alpha/8}),
\end{aligned}
\tag{96}
$$

where: (1) follows by Lemma G.3; (2) follows since $R \leq N$; and (3) follows from the fact that $I_a(\cdot)$ can increment by atmost $\max\{d(\mu_1, \mu_a), d(\mu_a, \mu_1)\}$ in one iteration.

Since, at every iteration $N \geq 1$ and for every arm, the modified empirical index differ from the empirical index by atmost $\log(N)$, we have,

$$
\begin{aligned}
\left|\mathcal{I}_a^{(m)}(N+R) - \mathcal{I}_a^{(m)}(N+R-1)\right| &\leq |\mathcal{I}_a(N+R) - \mathcal{I}_a(N+R-1)| + 2\log(N+R) \\
&\overset{(i)}{\leq} |\mathcal{I}_a(N+R) - \mathcal{I}_a(N+R-1)| + O(\log(N)) \\
&= O(N^{1-3\alpha/8}) + O(\log(N)) = O(N^{1-3\alpha/8}),
\end{aligned}
\tag{97}
$$

where (i) follows because $R \leq N$.

Using (96) and (97), the following corollary follows from Lemma G.16,

**Corollary G.2.** *For AT2 (1) and IAT2 (2) algorithms, for $N \geq T_6$ and $R \in \{1, 2, \ldots, N\}$, if the algorithm pulls some arm $a \in [K]/\{1\}$ at iteration $N$ and doesn't pull $a$ for the next $R$ iterations, then,*

$$
\begin{aligned}
\textbf{AT2:} \quad \mathcal{I}_a(N+R-1) - \mathcal{I}_{b(N,R)}(N+R-1) &\leq -C_3 \Delta \widetilde{N}_{b(N,R)}(N,R) + C_5 N^{1-3\alpha/8} \\
&\quad + O\left(\left(N^{-3\alpha/8} + RN^{-1}\right)R\right),
\end{aligned}
\tag{98}
$$

$$
\begin{aligned}
\textbf{IAT2:} \quad \mathcal{I}_a^{(m)}(N+R-1) - \mathcal{I}_{b(N,R)}^{(m)}(N+R-1) &\leq -C_3 \Delta \widetilde{N}_{b(N,R)}(N,R) + C_5 N^{1-3\alpha/8} \\
&\quad + O\left(\left(N^{-3\alpha/8} + RN^{-1}\right)R\right),
\end{aligned}
\tag{99}
$$

*where $C_3, C_5 > 0$ are constants independent of the sample paths.*

**Proof of Lemma G.2:** We prove the proposition for the AT2 algorithm. The proof for IAT2 algorithm follows the exact same argument by replacing $\mathcal{I}$ with $\mathcal{I}^{(m)}$.

In the proof, we argue via contradiction. We show that, there exists constants $M_5 \geq 1$ and $C_1 > 0$ independent of the sample paths, such that for $N \geq \max\{M_5, T_6\}$, if the algorithm pulls some arm $a \in [K]/\{1\}$ at iteration $N$ and doesn't pull it for the next $R(N) \overset{\text{def.}}{=} \lceil C_1 N^{1-3\alpha/8} \rceil$ iterations, then at iteration $\tau(N, R(N))$, the algorithm pulls arm $b(N, R(N))$, even though arm $a$ has empirical index strictly less than that of arm $b(N, R(N))$.

Using Corollary G.2, there exists constants $C_3, C_5, C_6 > 0$ independent of the sample paths, such that, for $N \geq T_6$ and $R \in \{1, 2, \ldots, N\}$,

$$
\begin{aligned}
\mathcal{I}_a(N+R-1) - \mathcal{I}_{b(N,R)}(N+R-1) &\leq -C_3 \Delta \widetilde{N}_{b(N,R)}(N,R) + C_4 N^{1-3\alpha/8} \\
&\quad + C_5 R(N^{-3\alpha/8} + RN^{-1}).
\end{aligned}
\tag{100}
$$

We define

$$
C_1 = \max\left\{D_1, \frac{2C_4}{D_2 C_3}\right\} + 1,
$$

where $D_1$ and $D_2$ are the constants introduced in Lemma G.15. Let $M_{52} \geq 1$ to be the smallest number such that, every $N \geq M_{52}$ satisfies $\lceil C_1 N^{1-3\alpha/8} \rceil < N$. Since $C_1 > D_1$, by the definition of $M_{51}$ in the proof of Lemma G.15, we have $M_{52} > M_{51}$.

Now let $R(N) = \lceil C_1 N^{1-3\alpha/8} \rceil$. Then by Lemma G.15, for $N \geq \max\{M_{52}, T_6\}$, since $R(N) > \lceil D_1 N^{1-3\alpha/8} \rceil$, we have,

$$\Delta \widetilde{N}_{b(N,R(N))}(N, R(N)) \geq D_2 R(N) \geq \frac{2C_4}{C_3} N^{1-3\alpha/8}. \tag{101}$$

Since arm $b(N, R)$ is not pulled between the iterations $\tau(N, R)$ and $N + R$, we have

$$\Delta \widetilde{N}_{b(N,R)}(N, t) = \Delta \widetilde{N}_{b(N,R)}(N, R)$$

for all $t \in \{ \tau(N, R) - N, \ \tau(N, R) - N + 1, \ \ldots, \ R \}$.

As a result, by definition of $b(N, R)$ we have,

$$b(N, R(N)) = b(N, \tau(N, R(N)) - N) \quad \text{and}$$
$$\Delta \widetilde{N}_{b(N,R(N))}(N, R(N)) = \Delta \widetilde{N}_{b(N,R)}(N, \tau(N, R(N)) - N).$$

In the rest of the proof, we denote $b(N, R(N))$ and $\tau(N, R(N)) - N$, respectively, using $b(N)$ and $\tau_b(N)$.

Therefore, using Corollary G.2, we have, for $N \geq \max\{M_{52}, T_6\}$,

$$\begin{aligned}
&\mathcal{I}_a(N + \tau_b(N) - 1) - \mathcal{I}_{b(N)}(N + \tau_b(N) - 1) \\
&\leq \ - C_3 \Delta \widetilde{N}_{b(N)}(N, \tau_b(N)) + C_4 N^{1-3\alpha/8} + C_5 \tau_b(N) \times (N^{-3\alpha/8} + \tau_b(N) N^{-1}) \\
&\overset{(1)}{\leq} \ - C_3 \times \frac{2C_4}{C_3} N^{1-3\alpha/8} + C_4 N^{1-3\alpha/8} + C_5 R(N) \times (N^{-3\alpha/8} + R(N) N^{-1}) \\
&\overset{(2)}{\leq} \ - C_4 N^{1-\alpha/8} + C_5 (C_1 + 1)(C_1 + 2) N^{1-3\alpha/8} \times N^{-3\alpha/8} \\
&\overset{(3)}{=} \ - (C_4 - C_6 N^{-3\alpha/8}) N^{1-3\alpha/8}, \tag{102}
\end{aligned}$$

where: (1) follows using (101) and $\tau_b(N) \leq R(N)$, (2) follows since $R(N) \leq (C_1 + 1) N^{-3\alpha/8}$, and (3) follows by letting $C_6 = C_5 (C_1 + 1)(C_1 + 2)$.

We now take $M_{53} \geq 1$ to be large enough, such that, $C_6 M_{53}^{-3\alpha/8} < C_4$. Let $M_5 = \max\{M_{52}, M_{53}\}$. Then (102) implies, for $N \geq \max\{M_5, T_6\}$ and $R(N) = \lceil C_1 N^{1-3\alpha/8} \rceil$, if the algorithm picks some arm $a \in [K]/\{1\}$ at iteration $N$ and doesn't pick $a$ for the next $R(N)$ iterations, then, at iteration $N + \tau_b(N)$, the algorithm picks arm $b(N)$, even though, $\mathcal{I}_a(N + \tau_b(N) - 1) - \mathcal{I}_{b(N)}(N + \tau_b(N) - 1) < 0$. Thus we arrive at a contradiction. $\qquad \square$

## H   Proof of Theorem 3.1

By Proposition 3.1, we know that, for every $a \in [K]$ and $N \geq T_{stable}$,

$$|\widetilde{\omega_a}(N) - \omega_a^\star| \leq C_1 N^{-3\alpha/8}.$$

Recall the functions $W_a(\cdot, \cdot)$ defined in Appendix G.2.2. Note that for every allocation $\boldsymbol{\omega} = (\omega_a)_{a \in [K]}$, and $a \in [K]/\{1\}$, we have,

$$\frac{\partial W_a}{\partial \omega_1}(\omega_1, \omega_a) = d(\mu_1, x_{1,a}(\omega_1, \omega_a)) \quad \text{and} \quad \frac{\partial W_a}{\partial \omega_a}(\omega_1, \omega_a) = d(\mu_a, x_{1,a}(\omega_1, \omega_a)),$$

where $x_{1,a}(\omega_1, \omega_a) = \frac{\omega_1 \mu_1 + \omega_a \mu_a}{\omega_1 + \omega_a}$.

As a result, for every $a \in [K]/\{1\}$, the partial derivatives of $W_a(\omega_1, \omega_a)$ with respect to $\omega_1$ and $\omega_a$ are both $O(1)$. Therefore, using the mean value theorem, for every $a \in [K]/\{1\}$ and $N \geq T_{stable}$, we have,

$$|W_a(\widetilde{\omega}_1(N), \widetilde{\omega}_a(N)) - W_a(\omega_1^\star, \omega_a^\star)| = O(N^{-3\alpha/8}). \tag{103}$$

Define the normalized index of every arm as

$$H_a(N) \;=\; \frac{I_a(N)}{N} \;=\; W_a(\widetilde{\omega}_1(N), \widetilde{\omega}_a(N)).$$

Also by (54), $W_a(\omega_1^\star, \omega_a^\star) = I^\star = T^\star(\boldsymbol{\mu})^{-1}$ for every alternative arm $a \in [K]/\{1\}$.

Therefore (103) gives us,

$$|H_a(N) - T^\star(\boldsymbol{\mu})^{-1}| \;=\; O(N^{-3\alpha/8}), \quad \text{for } a \in [K]/\{1\} \text{ and } N \geq T_{stable}. \tag{104}$$

By Lemma G.3, we also know,

$$|\mathcal{I}_a(N) - NH_a(N)| \;=\; |\mathcal{I}_a(N) - I_a(N)| \;=\; O(N^{1-3\alpha/8}). \tag{105}$$

Combining (104) and (105), we get

$$
\begin{aligned}
|\mathcal{I}_a(N) - NT^\star(\boldsymbol{\mu})^{-1}| \;&\leq\; |\mathcal{I}_a(N) - NH_a(N)| + N|H_a(N) - T^\star(\boldsymbol{\mu})^{-1}| \\
&=\; O(N^{1-3\alpha/8}),
\end{aligned}
$$

for every $a \in [K]/\{1\}$. Hence, we can find a constant $C_2 > 0$, independent of the sample paths, such that,

$$\mathcal{I}_a(N) \;\geq\; \frac{N}{T^\star(\boldsymbol{\mu})} - C_2 N^{1-3\alpha/8},$$

for every $N \geq T_{stable}$ and $a \in [K]/\{1\}$. As a result, for $N \geq T_{stable}$, we have,

$$\min_{a \in [K]/\{1\}} \mathcal{I}_a(N) \;\geq\; \frac{N}{T^\star(\boldsymbol{\mu})} - C_2 N^{1-3\alpha/8}. \tag{106}$$

The threshold function $\beta(N, \delta)$ deciding our stopping condition satisfies,

$$\log(1/\delta) \;\leq\; \beta(N, \delta) \;\leq\; \log(1/\delta) + C_3 \log\log(1/\delta) + C_4 \log\log(N) + C_5$$

for constants $C_3, C_4, C_5 > 0$. For every $\delta > 0$, we define the deterministic quantity,

$$t_{\max,\delta} \;=\; \min\left\{ N \geq T_{stable} \;\bigg|\; \frac{N}{T^\star(\boldsymbol{\mu})} - C_2 N^{1-3\alpha/8} > \beta(N, \delta) \right\}. \tag{107}$$

Now we make the following observations about $t_{\max,\delta}$,

1. Note that $\frac{N}{T^\star(\boldsymbol{\mu})} - C_2 N^{1-3\alpha/8}$ increases linearly in $N$ and $\beta(N, \delta)$ is $O(\log\log(N) + \log(1/\delta))$, for a fixed $\delta > 0$. Hence, $t_{\max,\delta}$ is finite for every $\delta > 0$.

2. We have $\beta(N, \delta) \geq \log(1/\delta)$ and $\frac{N}{T^\star(\boldsymbol{\mu})} - C_2 N^{1-3\alpha/8} < \frac{N}{T^\star(\boldsymbol{\mu})}$. As a result, $t_{\max,\delta}$ is atleast the iteration at which $\frac{N}{T^\star(\boldsymbol{\mu})}$ exceeds $\log(1/\delta)$, which is atleast $T^\star(\boldsymbol{\mu})\log(1/\delta)$. This implies $t_{\max,\delta} \geq T^\star(\boldsymbol{\mu})\log(1/\delta)$. As a result, $t_{\max,\delta} \to \infty$ as $\delta \to 0$.

3. If $\tau_\delta > T_{stable}$, then $\min_{a \in [K]/\{1\}} \mathcal{I}_a(N)$ exceed $\beta(N, \delta)$ before the lower bound of $\min_{a \in [K]/\{1\}} \mathcal{I}_a(N)$ in (106) exceeds $\beta(N, \delta)$. As a result, $\tau_\delta \leq t_{\max,\delta}$, whenever $\tau_\delta \geq T_{stable}$. This gives us the upper bound,

$$\tau_\delta \;\leq\; \max\{T_{stable}, t_{\max,\delta}\} \quad \text{a.s. in } \mathbb{P}_{\boldsymbol{\mu}}. \tag{108}$$

We have $\mathbb{E}_{\boldsymbol{\mu}}[T_{stable}] < \infty$, which also implies $T_{stable} < \infty$ a.s. in $\mathbb{P}_{\boldsymbol{\mu}}$. Now using (108), we get,

$$\limsup_{\delta \to 0} \frac{\mathbb{E}_{\boldsymbol{\mu}}[\tau_\delta]}{\log(1/\delta)} \;\leq\; \limsup_{\delta \to 0} \frac{\mathbb{E}_{\boldsymbol{\mu}}[T_{stable}] + t_{\max,\delta}}{\log(1/\delta)} \;=\; \limsup_{\delta \to 0} \frac{t_{\max,\delta}}{\log(1/\delta)}.$$

Similarly, we have,

$$\limsup_{\delta \to 0} \frac{\tau_\delta}{\log(1/\delta)} \leq \limsup_{\delta \to 0} \frac{T_{stable} + t_{\max,\delta}}{\log(1/\delta)} = \limsup_{\delta \to 0} \frac{t_{\max,\delta}}{\log(1/\delta)} \quad \text{a.s. in } \mathbb{P}_{\boldsymbol{\mu}}.$$

Therefore to prove asymptotic optimality, it is sufficient to prove $\limsup_{\delta \to 0} \frac{t_{\max,\delta}}{\log(1/\delta)} \leq T^\star(\boldsymbol{\mu})$.

Let $s_{\max,\delta} = t_{\max,\delta} - 1$. Note that $\limsup_{\delta \to 0} \frac{t_{\max,\delta}}{\log(1/\delta)} = \limsup_{\delta \to 0} \frac{s_{\max,\delta}}{\log(1/\delta)}$. By definition of $t_{\max,\delta}$, we have

$$\frac{s_{\max,\delta}}{T^\star(\boldsymbol{\mu})} - C_2 s_{\max,\delta}^{1-3\alpha/8} \leq \beta(s_{\max,\delta}, \delta) = \log(1/\delta) + C_3 \log\log(1/\delta) + C_4 \log\log(s_{\max,\delta}) + C_5.$$

After some rearrangement of terms, upon dividing both sides by $\log(1/\delta)$, we get,

$$\frac{1}{T^\star(\boldsymbol{\mu})} \frac{s_{\max,\delta}}{\log(1/\delta)} \Big( 1 - C_2 s_{\max,\delta}^{-3\alpha/8} - C_3 s_{\max,\delta}^{-1} \log\log(s_{\max,\delta})$$

$$- C_5 s_{\max,\delta}^{-1} \Big) \leq 1 + C_3 \frac{\log\log(1/\delta)}{\log(1/\delta)}.$$

By Observation 2, $s_{\max,\delta} = t_{\max,\delta} - 1 \to \infty$ as $\delta \to 0$. As a result, the above inequality implies,

$$\frac{1}{T^\star(\boldsymbol{\mu})} \limsup_{\delta \to 0} \frac{s_{\max,\delta}}{\log(1/\delta)} \leq 1.$$

We already argued that $\limsup_{\delta \to 0} \frac{t_{\max,\delta}}{\log(1/\delta)} \leq T^\star(\boldsymbol{\mu})$. As a result, we have a constant $C_6 > 0$, such that $t_{\max,\delta} \leq C_6 \log(1/\delta)$ for all $\delta$. By (108) we have:

$$\frac{t_{\max,\delta} - 1}{T^\star(\boldsymbol{\mu})} - C_2 (t_{\max,\delta} - 1)^{1-3\alpha/8} \leq \beta(t_{\max,\delta} - 1, \delta)$$

which implies,

$$t_{\max,\delta} \leq T^\star(\boldsymbol{\mu}) \log(1/\delta) + O\left( \log\log(1/\delta) + t_{\max,\delta}^{1-3\alpha/8} \right).$$

Putting $t_{\max,\delta} \leq C_6 \log(1/\delta)$ in the above inequality,

$$t_{\max,\delta} \leq T^\star(\boldsymbol{\mu}) \log(1/\delta) + O\left( (\log(1/\delta))^{1-3\alpha/8} \right). \tag{109}$$

Since $\tau_\delta \leq \max\{T_{stable}, 1 + t_{\max,\delta}\}$, we can find a constant $C > 0$ such that $\tau_\delta \leq \max\left\{ T_{stable}, T^\star(\boldsymbol{\mu}) \log(1/\delta) + C(\log(1/\delta))^{1-3\alpha/8} \right\}$. Hence Theorem 3.1 stands proved.

# I  Extending the proposed algorithm to distributions with bounded support

We describe a natural extension of AT2 and IAT2 algorithms to bandit instances from a non-parametric family. We conduct experiments to compare the proposed algorithm with the existing ones. We consider the class of distributions having their supports contained in $[0, 1]$, which we denote by $\mathcal{F}_{[0,1]}$. This is similar to the assumptions made in [16]. Some definitions are in order. We use $\mu(G)$ to denote the mean of distribution $G \in \mathcal{F}_{[0,1]}$.

For every $F \in \mathcal{F}_{[0,1]}$ and $x \in [0, 1]$, we define $\text{KL}_{\text{inf}}^+$ and $\text{KL}_{\text{inf}}^-$ as:

$$\text{KL}_{\text{inf}}^+(F, x) = \inf\{ \text{KL}(F, G) \mid \mu(G) > x \} \quad \text{and}$$
$$\text{KL}_{\text{inf}}^-(F, x) = \inf\{ \text{KL}(F, G) \mid \mu(G) < x \}.$$

At iteration $N$ of the algorithm, let $\widetilde{F}_a(N)$ be the empirical distribution of the samples collected from some arm $a \in [K]$, $\widetilde{\boldsymbol{F}}(N) = (\widetilde{F}_a(N) : a \in [K])$, $\widetilde{N}_a(N)$ be the total no. of samples collected from $a$ till $N$, and $\widetilde{\boldsymbol{N}}(N) = (\widetilde{N}_a(N) : a \in [K])$. Let $\widetilde{\mu}_a(N) = \mu(\widetilde{F}_a(N))$ and $\hat{i}_N = \arg\max_{a \in [K]} \widetilde{\mu}_a(N)$. We now define:

$$x_{\hat{i}_N, a}(N) = \arg\min_{x \in [0,1]} \left\{ \widetilde{N}_{\hat{i}_N}(N) \cdot \text{KL}_{\text{inf}}^-(\widetilde{F}_{\hat{i}_N}(N), x) + \widetilde{N}_a(N) \cdot \text{KL}_{\text{inf}}^+(\widetilde{F}_a(N), x) \right\}.$$

At every iteration, we compute the *empirical index* of every arm $a \neq \hat{i}_N$ as:

$$\mathcal{I}_a(N) = \min_{x \in [0,1]} \left\{ \widetilde{N}_{\hat{i}_N}(N) \cdot \mathrm{KL}_{\mathrm{inf}}^-(\widetilde{F}_{\hat{i}_N}(N), x) + \widetilde{N}_a(N) \cdot \mathrm{KL}_{\mathrm{inf}}^+(\widetilde{F}_a(N), x) \right\},$$

and the *anchor function* as,

$$g(\widetilde{\boldsymbol{F}}(N), \widetilde{\boldsymbol{N}}(N)) = \sum_{a \neq \hat{i}_N} \frac{\mathrm{KL}_{\mathrm{inf}}^-(\widetilde{F}_{\hat{i}_N}(N), x_{\hat{i}_N, a}(N))}{\mathrm{KL}_{\mathrm{inf}}^+(\widetilde{F}_a(N), x_{\hat{i}_N, a}(N))} - 1.$$

The AT2 and IAT2 algorithm for the class $\mathcal{F}_{[0,1]}$, respectively, follows the same steps as in (1) and (2) with the anchor and index functions defined as above.

We experimentally demonstrate the performance of the proposed algorithms in Appendix J.5.

## J  Experiments

### J.1  Dynamics of the algorithms

**Experiment 1 (Gaussian and Bernoulli bandits with well-separated arms):** In the main text (Figure 1), we presented the evolution of normalized indexes for the sub-optimal arms for AT2, when run without the stopping rule. Numerically, we observe similar plots for normalized indexes even for the other algorithms: 0.5-EB-TCB (proposed in [16] with $\beta = 0.5$), and TCB (proposed in [22]). Hence, we do not report them. However, we do observe differences in the evolution of the anchor function value across these algorithms. We present this in two different settings in the current section.

Interestingly, as per our implementations, we observe that only AT2 satisfies the asymptotic optimality conditions, maintaining the anchor function close to 0, in addition to maintaining the equality of the normalized indexes.

In this section, we consider the following two examples:

1. *Gaussian bandit.* In the first setup (Figure 4), we consider a 4 armed Gaussian bandit with unit variance and mean vector $\mu = [10, 8, 7, 6.5]$. This is the same setting as in Section 6 from the main text.

2. *Bernoulli bandit.* In the second setup (Figure 5), we consider a Bernoulli bandit with means $\mu = [0.91, 0.73, 0.64, 0.59]$.

In Figures 4 and 5, we plot the evolution of anchor function value for the three algorithms in the two settings, without implementing the stopping rule. The solid lines in the figure represent the average of anchor function over 4,000 independent runs. The shaded bands around the sold lines (almost invisible in these figures), represent 2 standard deviation bands around the mean.

We observe that only AT2 maintains the anchor function value close to 0. Our experiments suggest that that TCB algorithm, as per our implementation, doesn't satisfy the asymptotic optimality conditions.

### J.2  Sample complexity comparison

In Section 6 in the main text, we compared the sample complexities (SC) of the three algorithms on a well-separated Gaussian bandit (Figure 2). In this section, we compare the SC of all the algorithms, as a function of different parameters. We consider harder Gaussian as well as Bernoulli bandit instances (with means close to each other), presented below.

1. *Gaussian bandit.* A 4 armed Gaussian bandit with unit variance and mean vector $\mu = [7.25, 7.05, 7, 7.1]$ so that the means are closer together.

2. *Bernoulli bandit.* As a second example, we consider a 4-armed Bernoulli bandit with close-by means: $\mu = [0.99, 0.96, 0.95, 0.97]$.

**Experiment 2 (SC as function of $\beta$):** In this experiment, we compare SC of (I)AT2, (I)TCB, $\beta$-EB-(I)TCB algorithms, for $\beta \in [0.2, 0.3, 0.4, 0.5, 0.6, 0.7, 0.8]$, on the Gaussian instance (Figure 6) and the Bernoulli instance (Figure 7), described above. The error probability $\delta$ in both these experiments is set to 0.001. All the algorithms use the same forced exploration rule and stopping rules.

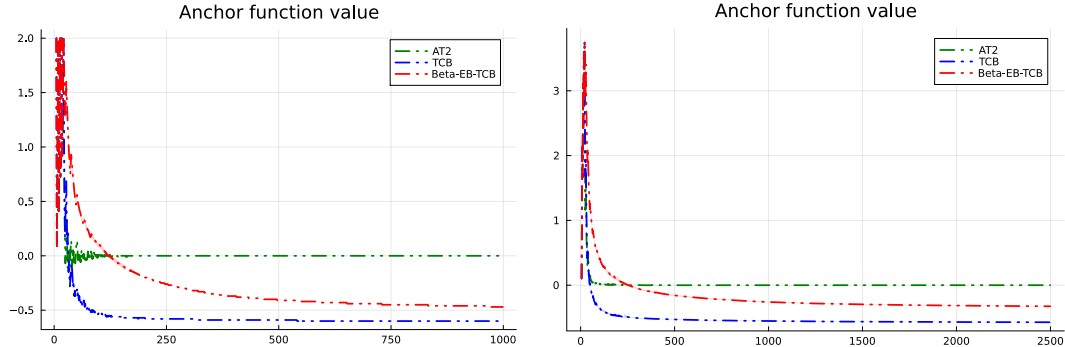

Figure 4: Anchor function value for easy Gaussian bandit (Exp.1), averaged over 4,000 sample paths.

Figure 5: Anchor function value for easy Bernoulli bandit (Exp.1), averaged over 4,000 sample paths.

The lines with the markers in the figures represent the average number of samples generated before stopping, averaged over $4,000$ independent simulations, while the shaded regions denote 2 standard deviations around the mean. We also report the average sample complexity and the standard deviation of the average sample complexity for AT2, IAT2, TCB, and ITCB, across $4,000$ independent simulations.

In both these simulations, we observe that AT2 and IAT2, respectively, have about 5% lower sample complexity compared to TCB and ITCB.

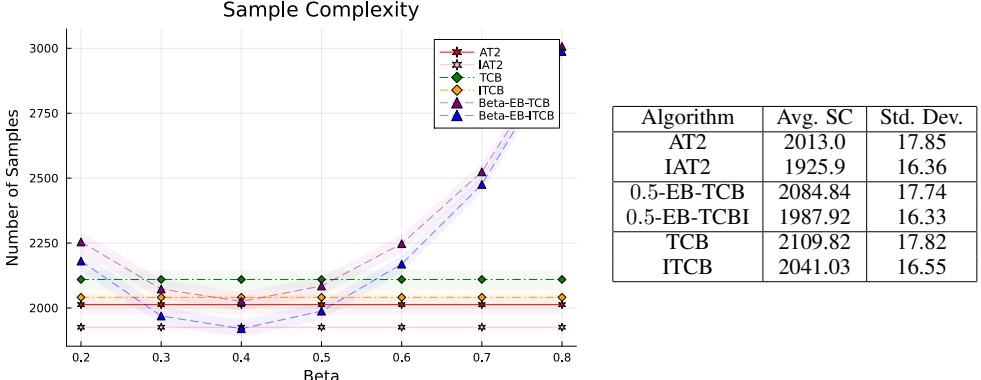

| Algorithm | Avg. SC | Std. Dev. |
|---|---|---|
| AT2 | 2013.0 | 17.85 |
| IAT2 | 1925.9 | 16.36 |
| 0.5-EB-TCB | 2084.84 | 17.74 |
| 0.5-EB-TCBI | 1987.92 | 16.33 |
| TCB | 2109.82 | 17.82 |
| ITCB | 2041.03 | 16.55 |

Figure 6: Sample complexity (SC) on Gaussian bandit from Exp. 2, averaged over $4,000$ independent sample paths. $\delta = 0.001$

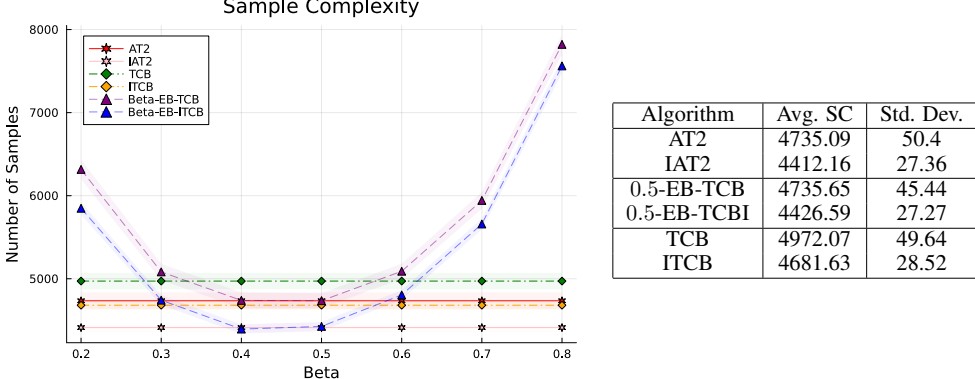

| Algorithm | Avg. SC | Std. Dev. |
|---|---|---|
| AT2 | 4735.09 | 50.4 |
| IAT2 | 4412.16 | 27.36 |
| 0.5-EB-TCB | 4735.65 | 45.44 |
| 0.5-EB-TCBI | 4426.59 | 27.27 |
| TCB | 4972.07 | 49.64 |
| ITCB | 4681.63 | 28.52 |

Figure 7: Sample complexity (SC) on Bernoiulli bandit from Exp. 2, averaged over $4,000$ independent sample paths. $\delta = 0.001$

**Experiment 3 (SC as function of $\delta$):** In Figure 8 and Figure 9, we plot the sample complexities of the three algorithms — AT2, 0.5-EB-TCB, and TCB — as a function of $\delta$, for the Gaussian and Bernoulli bandits considered in Experiment 2 above. All the algorithms use the same forced exploration and stopping rules. We observe that AT2 consistently outperforms both the previously known algorithms, and the gap in performance increases as we reduce $\delta$.

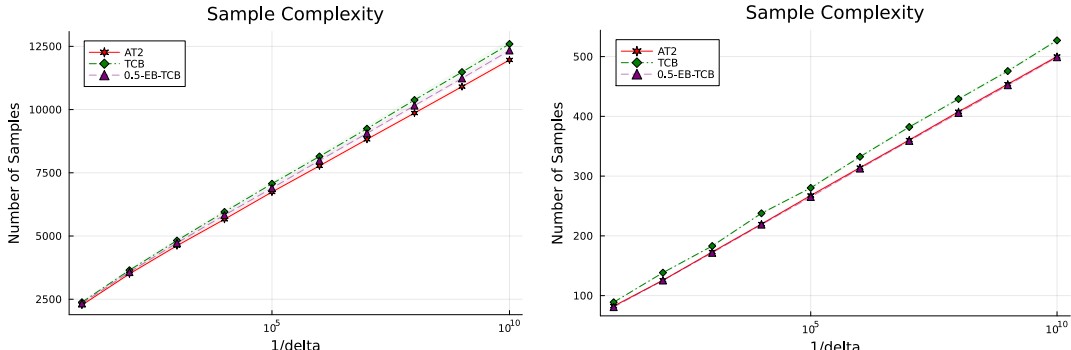

Figure 8: Sample complexity of Gaussian bandit (Exp.3), averaged over 4,000 sample paths.

Figure 9: Sample complexity for Bernoulli bandit (Exp.3), averaged over 4,000 sample paths.

**Experiment 4 (SC as a function of number of arms).** We plot the number of samples needed by the three algorithms, as a function of number of arms in the bandit instance. $\delta$ is set to $0.001$ in this experiment.

For scalability, in this experiment, we consider a simple Gaussian bandit (well-separated means) with all arms having a unit variance. Arm 1 is optimal with mean 10. To study the effect of number of arms on sample complexity, we choose all the other arms to be same with mean 8. Thus, the bandit instances have Gaussian arms with unit variance, and means

$$\mu = [10, 8, \dots].$$

As in the earlier experiments, for fair comparison, all the algorithms are implemented with the same forced exploration and stopping rules. Results are presented in Figure 10. We observe that the sample complexity increases linearly with number of arms for the three algorithms. In this experiment, the performance of TCB and AT2 looks comparable, with TCB requiring slightly more number of samples. However, the gap in their performance is expected to increase for smaller values of $\delta$.

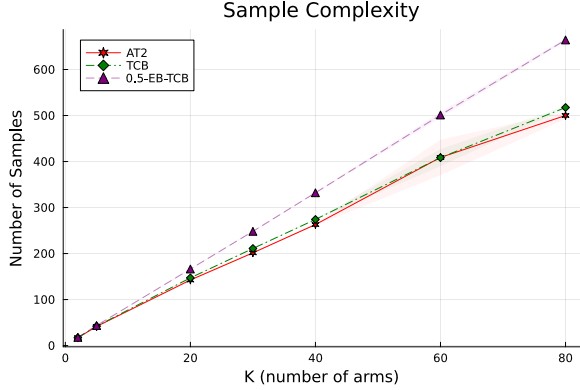

Figure 10: Sample complexity on Gaussian bandit (Exp.4), averaged over $4,000$ independent sample paths.

### J.3   Runtime comparison

**Experiment 5:** In this experiment, we compare the run-time of (I)AT2 and (I)TCB algorithms on a 4 armed Gaussian bandit with means $\mu = [10, 9.4, 7, 6.5]$ and unit variance, averaged over longer

100,000 simulations. $\delta$ is set to 0.001, and the four algorithms use the same forced exploration and stopping rules.

Table 1 represents the average run-time of the two algorithms. We observe that TCB and ITCB take roughly two times more computational time compared to AT2 and IAT2, respectively.

| Algorithm | Avg. Sample Complexity | Std. Dev. | Avg. Run Time (microsec.) | Run Time Std. Dev. |
|---|---|---|---|---|
| AT2 | 90.53 | 0.2 | 129.76 | 32.34 |
| IAT2 | 90.63 | 0.2 | 310.76 | 55.88 |
| TCB | 96.55 | 0.21 | 501.22 | 82.60 |
| ITCB | 96.69 | 0.21 | 845.19 | 145.97 |

Table 1: Runtime of (I)AT2 and (I)TCB on Gaussian bandit with $\mu = [10, 9.4, 7, 6.5]$ and unit variances (Exp.6). Results reported are for $100,000$ independent runs of each algorithm.

## J.4 Effect of forced-exploration parameter $\alpha$ on sample complexity

In this section, we provide a numerical evaluation of the impact of forced exploration on the performance of AT2 and IAT2. Our experiments suggest that unlike IAT2, AT2 needs forced exploration. On instances where the second and third best arms have equal means, AT2 might see some bad samples from the best arm in the beginning. As a result, without forced exploration, it will sample the second and the third best arms forever. However, we see that AT2 performs sufficient exploration for instances having all arms with different means. Note that similar observations were made in [16] for $\beta$-EB-TCB.

**Experiment 6:** To see the above mentioned behavior of AT2, we study the performance of AT2 and IAT2 on the following two bandit instances.

1. A $4$ armed Gaussian bandit with unit variance and mean vector $\mu = [7.25, 7.05, 7, 7.1]$, so that the means are close together, yet all different.

2. A $4$ armed Gaussian bandit with unit variance and mean vector $\mu = [7.25, 7, 7, 7]$, so that the three suboptimal arms have equal means.

Intuitively, it might appear that forced exploration could significantly increase the sample complexity of AT2 and IAT2. However, in Figure 11, we see that IAT2's performance remains unaffected and AT2 performs at least as well as IAT2 with moderate amount of forced exploration on the first instance, where the gap between the means is small.

Figure 12 shows that without forced exploration, AT2's sample complexity blows up on the instance with equal sub-optimal arm means. We see that on this instance, IAT2 performs significantly better than AT2. Furthermore, IAT2's sample complexity remains almost unaffected with respect to the amount of forced exploration done.

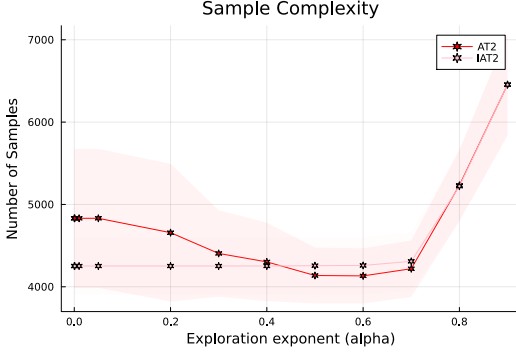

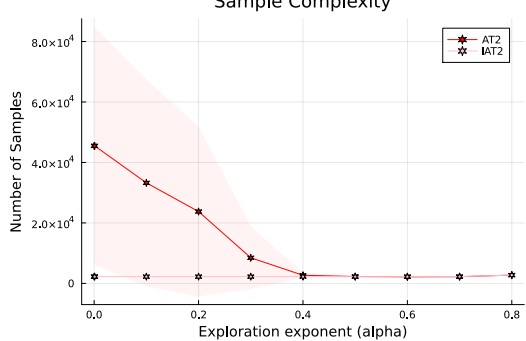

Figure 11: Sample complexity of Gaussian bandit with means $\mu = [7.25, 7.05, 7, 7.1]$, as a function of $\alpha$ (Exp.6)

Figure 12: Sample complexity of Gaussian bandit with means $\mu = [7.25, 7, 7, 7]$, as a function of $\alpha$ (Exp.6)

## J.5 Experiments for bandits with bounded-support distributions

In this section, we experimentally demonstrate the performance of a natural extension of AT2 to a non-parametric setting of bandits with arms having distributions supported in $[0, 1]$ (see Appendix I for the modified AT2). This is the setting considered in, for example, [16].

**Experiment 7:** Consider a $4$-armed bandit with the following arm distributions:

$$\mathrm{Beta}(1.5, 1), \; \mathrm{Beta}(2, 6), \; \mathrm{Beta}(1, 1.5), \; \text{and} \; \mathrm{Beta}(1, 7).$$

Here, the arms have means

$$\mu = [0.6, 0.25, 0.4, 0.125].$$

While we do not provide the analysis of the algorithm for this setting, numerically we observe that even in this non-parametric setting, extension of AT2 to this setting outperforms $\beta$-EB-TCB, and a corresponding natural extension of TCB to this setting.

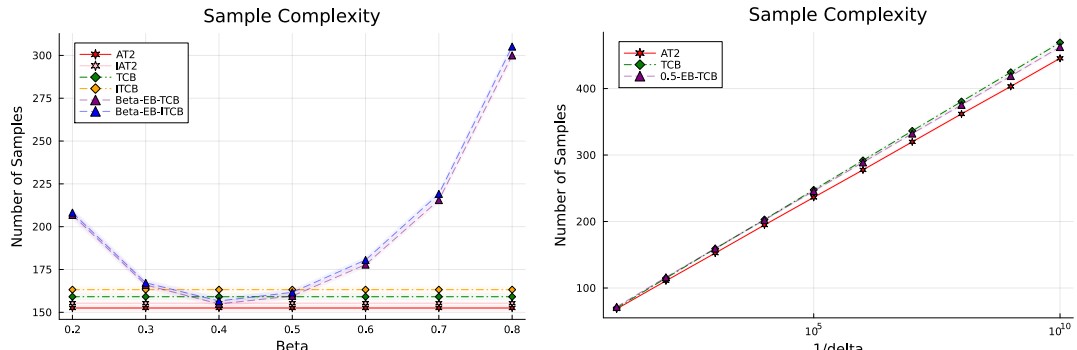

Figure 13: Sample complexity (Exp.6), averaged over 4,000 sample paths. We set $\delta = 0.001$.

Figure 14: Sample complexity (Exp.6), averaged over 4,000 sample paths.

**Reproducibility:** Our code is implemented in `Julia 1.7.1`, and the plots are generated with the `Plots.jl` package. Other dependencies are listed in the `Readme.md` file, which also includes instructions to reproduce the figures and tables presented here. We build upon the publicly available code for [16]. Our experiments are conducted on an institutional cluster computing facility having an Intel Xeon Gold 6130 2.1GHz CPU with 32 cores.

