# OpenReview forum: "Optimal Top-Two Method for Best Arm Identification and Fluid Analysis"
_NeurIPS.cc/2024/Conference — NeurIPS 2024 poster_

### Official Review · Reviewer_mata · 2024-06-27

**Soundness:** 3
**Presentation:** 3
**Contribution:** 3
**Rating:** 5
**Confidence:** 4

**Summary:**

This paper considers the top-2 algorithm for identifying the best arm in multi-armed bandits. The authors introduce a new approach for determining the optimal $\beta$ for sampling the empirical best and best challenger arms. This novel method relies on a function of allocations anchored at a threshold, which is simple in concept.

**Strengths:**

1. The paper introduces a novel method for determining the optimal $\beta$ in Top-2 algorithms, which has been a popular problem in the field. Empirical results demonstrate that the proposed method offers improved sample complexity and computational efficiency compared to existing approaches.
2. The analysis leverages fluid dynamics and the implicit function theorem to establish the algorithm's optimality, marking a unique and intriguing contribution to the field of bandit problems.

**Weaknesses:**

1. One weakness of the paper is the use of the forced exploration stage in the AT2 algorithm (Line 1-2), which ensures that all the arm means are estimated with sufficient accuracy. It is not clear whether this is crucial for the effectiveness of the algorithm or simply for the ease of theoretical proof. Indeed, forced exploration might harm the empirical performance of BAI but significantly simplifies the asymptotic theoretical analysis.
2. The second weakness of the paper is that all the analyses are asymptotic, although I understand that achieving non-asymptotic results might be challenging for the top-2 algorithm.

**Questions:**

1. Line 634-637: What about a Bernoulli distribution with mean zero?
2. Line 144: Another widely used definition for a $\delta$-correct algorithm is $\mathbb{P}\left(\tau_\delta<\infty\right) = 1$ and $\mathbb{P}\left( k_{\tau_\delta} \neq 1\right) \leq \delta$. Are there any concerns with this definition?
3. It is surprising that TCB does not satisfy the asymptotic optimality conditions. Have you verified the correctness of the implementation? Additionally, did you use the same forced exploration rule and stopping rule for all algorithms in the numerical results? Finally, I suggest including a theoretical analysis of the computational complexity for the proposed algorithm.

**Limitations:**

The main contributions of this work are theoretical in nature. I don't foresee any direct negative social impact.

---

> ### Author Rebuttal · Authors · 2024-08-07
>
> We thank the reviewer for the suggestions and provide our responses below.
>
> We first address the weaknesses.
>
> **[Use of forced exploration in analysis and experiments]** We address this weakness in **(1)** of the global rebuttal and in our rebuttal to Reviewer 4f2z at "Use of foced exploration in analysis".  We used the same forced exploration rule and stopping rule for all algorithms. We have added this detail to the experiments section of the revision.
>
>
> **[Non-asymptotic results]** We address this weakness at **(3)** in the global rebuttal.
>
> **Response to the questions:**
>
> **[Bernoulli distribution with mean zero]** If we consider the SPEF generated by some non-degenerate Bernoulli distribution, then all members in that SPEF will also be non-degenerate, i.e., they will have their mean in the open interval $(0,1)$. Note that Bernoulli with mean zero is a degenerate distribution and therefore cannot be a part of that SPEF. For a Bernoulli SPEF, $\mathcal{H}(\mu)$ in lines 634-637 is only defined for instances having their mean in $(0,1)$. Also by construction, $\mathcal{H}(\mu)$ is compact and a subset of $(0,1)$. As a result all distributions in $\mathcal{H}(\mu)$ are non-degenrate and, as argued in lines 635-637, the minimum variance of such distributions is positive.
>
> **[Alternate definition of** $\delta$-**correctness]** The suggested definition $\mathbb{P}(\tau_\delta<\infty)=1$ and $\mathbb{P}(k_{\tau_\delta}\neq 1)\leq \delta$ of $\delta$-correctness is subsumed in the definition $\mathbb{P}(\tau_\delta<\infty,k_{\tau_\delta}\neq 1)\leq \delta$ in lines 143-144. The non-asymptotic bound in Theorem 3.1 implies that the AT2 and IAT2 algorithms satisfy $\mathbb{P}(\tau_\delta<\infty)=1$ and $\mathbb{P}(k_{\tau_\delta}\neq 1)\leq \delta$.
>
>
> **[About sub-optimality of (I)TCB in [21]]** As we mentioned in our response "Sub-optimality of (I)TCB [21]" in our rebuttal to Reviewer Uo1U, (I)TCB may not be asymptotically optimal since it does not reinforce $g=0$ condition. Our experiments, as per our implementation, also demonstrate this (Figures 3, 4), where we see only AT2 satisfies the optimality conditions ($g=0$).
>
>
> **[Computational complexity of AT2 and IAT2]**
>
> - **Per iteration computation:** If we consider that every $KL$-divergence computation takes only a constant amount of time independent of the instance $\mu$ and the no. of arms $K$, then every iteration of the AT2 and IAT2 algorithms require only $O(K)$ time to execute. We have mentioned this in the revision.
>
> - **Overall expected computational complexity:** Since getting a non-asymptotic bound of $\tau_\delta$ is challenging in this setting, we do not have a overall bound of the expected computational complexity (see **(3)** of global rebuttal on non-asymptotic bounds over sample complexity). However, experiment 4 in Appendix J.2 shows that, the expected sample complexity of the proposed algorithms increases linearly as a function of the no. of arms.

---

> > ### Comment · Reviewer_mata · 2024-08-13
> >
> > Thank you for your response. After reading the rebuttal, I would like to maintain my original rating.

---

### Official Review · Reviewer_Uo1U · 2024-07-04

**Soundness:** 4
**Presentation:** 2
**Contribution:** 4
**Rating:** 6
**Confidence:** 4

**Summary:**

The paper investigates the problem of fixed-confidence BAI in stochastic multi-armed bandits. Simply stated, the problem entails finding the best arm (the arm with the largest mean reward) as quickly as possible, subject to an upper bound on the error probability. There exist a plethora fo works that investigate this problem, derive problem instance-dependent lower bounds on the asymptotic growth rate of the expected stopping time, and devise algorithms that achieve the asymptotic lower bound; here, the asymptotics is one of vanishing error probabilities. One class of efficiently implementable algorithms for this problem that has gained significant attention in the recent years is that of top-2 sampling algorithms. The basic premise on which such algorithms function is to pull the empirical best arm at any given time with a certain constant probability, say $\beta$, and to pull the empirically second-best arm (or the challenger arm) with the remaining probability. Despite its elegance and ease of implementability, the aforementioned $\beta$-top-2 sampling algorithm an algorithm falls short of achieving the instance-dependent lower bound. The key to achieving the lower bound, as it turns out, is to set $\beta=\beta^*$, where $\beta^*$ depends on the underlying problem instance and is therefore unknown to the algorithm. While several attempts to learn the value of $\beta^*$ as part of the algorithm have taken place in the recent past, the current work accomplishes the same objective elegantly, using the theory of ODEs. Notably, the paper proposes an algorithm for BAI, and analyses the fluid dynamics (ODE) corresponding to a continuous version of the same. Implicit function theorem is used to demonstrate the existence and uniqueness of the solutions to the ODE, and to further demonstrate that the arm allocations under the algorithm closely follows the trajectory coming from the ODE. It is then argued that the limiting solution of the ODE (and hence the asymptotic arm allocations under the algorithm) matches with the optimum allocations corresponding to the underlying instance, thereby proving the asymptotic optimality of the proposed algorithm.

**Strengths:**

The use of the theory of ODEs and the implicit function theorem to demonstrate the asymptotic optimality of the proposed algorithm is the novel aspect of this paper. To the best of my knowledge, such an approach to the demonstration of asymptotic optimality does not exist in the literature on fixed-confidence BAI. Furthermore, while the concept of *anchor function* appearing in the paper has its roots in Garivier and Kaufmann's paper on BAI, a careful analysis of its behaviour using the theory of ODE to demonstrate asymptotic optimality of the proposed algorithm is impressive.

(I must acknowledge, though, that I was unable to go through the extensive and detailed appendix due to paucity of time.)

**Weaknesses:**

The writing of the paper can be slightly improved. While the authors have done a commendable job in explaining the intuition behind their detailed proofs, I have some minor suggestions that might possibly improve the readability of the paper.

1. In the paragraphs leading up to Proposition 2.1, the authors introduce the notation $x\_{1,a}=\dfrac{\omega\_1 \ x\_1 + \omega\_a \ x\_a}{\omega\_1 + \omega\_a}$. Shortly thereafter, the authors use the phrase "define $N\_{1,1}$ as the value of $N\_1$ at which $\sum\_{a \neq 1} \dfrac{d(\mu\_1, x\_{1,a})}{d(\mu\_1, x\_{1,a})}=1$ for a given allocation $\boldsymbol{N}\_{\bar{B}^c}$", without stating the (new) definition of $x\_{1,a}$. The actual definition of $x\_{1,a}$ only appears later, in the first part of Proposition 2.1. I suggest that the authors move the definition of $x\_{1,a}$ to an earlier paragraph if possible.

Some minor comments:

2. "alocation" in line 160 --> "allocation".

3. $f(\boldsymbol{\mu}, a, N)$ in line 291 --> $f(\boldsymbol{\mu}, a, \boldsymbol{N})$.

**Questions:**

Can the authors explain what role the anchor function $g$ plays at a higher level. It seems almost unintuitive that this function, together with the empirical means and allocations, should dictate the sampling of the best empirical arm or challenger arm at any given time (items 3,4 in lines 208-211), in order to match with the optimal allocations in the long run. For a contextual comparison, in the TCB and ITCB algorithms of [21], the place of the anchor function is taken by the empirical problem complexity. Given that the empirical terms are expected to be close to their true values in the long run, the empirical problem complexity is likely to be close to the true complexity. It is then intuitive to expect that any sampling rule that is based on the empirical problem complexity, is likely to lead to sampling the arms according to the optimal allocation in the long run.

Along these/similar lines, can the authors provide a high-level intuition for why looking at the anchor function and its long-term behaviour is pivotally connected to learning the optimal $\beta$ and thereby matching the arm selection frequencies with the optimal allocation?

**Limitations:**

NA.

---

> ### Author Rebuttal · Authors · 2024-08-07
>
> We thank the reviewer for the suggestions and provide our responses below.
>
> The typos have been corrected in the revision. Thanks.
>
> **Response to the questions:**
>
> **[Intuitive reason of using the anchor function]** The proposed algorithms are motivated by the first order conditions which uniquely identify the optimal proportion (see Proposition 2.1). In **(2)** of the global rebuttal we explained the intuition behind the algorithm, choice of the particular anchor function and setting it at threshold zero. Also see our response "Intuition behind the steps of the algorithm" to Reviewer Aqmy.
>
> **[Sub optimality of (I)TCB [21]]** [21] implicitly assumes that having all indexes equal and the proportion weights summing to 1 uniquely identifies the optimal allocation ([21, Lemma 2]). This uniqueness is true with the constraint $w_1=\beta$ as in [Russo, Simple Bayesian Algorithms for Best Arm Identification]. However, simply having all indexes equal and sum of the weights equal to one ([21, Lemma 2]) imposes K-1 conditions on K variables allows one degree of freedom, which is optimized at $g=0$.  See the discussion at **(2)** in the global rebuttal.

---

> > ### Comment · Reviewer_Uo1U · 2024-08-13
> > **Response to Authors' Rebuttal**
> >
> > I thank the authors for their detailed and insightful response. All my questions have been fully addressed. I will maintain my current score.

---

### Official Review · Reviewer_jbje · 2024-07-07

**Soundness:** 2
**Presentation:** 1
**Contribution:** 1
**Rating:** 5
**Confidence:** 2

**Summary:**

This paper considers the problem of identifying an optimal "top-2" algorithm for the best arm identification problem in the bandits framework. The question of how to allocate pulls between incumbents and challengers is considered. An algorithm for achieving the best balance is described.

**Strengths:**

"Top-2" approaches for best arm identification are powerful and increasingly popular, so this paper's focus is an important area of study. Posing the problem in terms of fluid dynamics could potentially be a powerful way of modeling the problem.

**Weaknesses:**

I found the paper to be generally very hard to follow. The language is awkward in many places. I do not find that the introduction makes it clear how the paper's contributions are achieved. Explanations are very math-heavy and hard to follow and I find the theoretical results (e.g., Proposition 2.1) are not understandable. The paper also references many concepts without explaining them well (e.g., IFT, GLLR, ITCB).

The presentation of this paper needs to be significantly improved before it is ready for publication.

**Questions:**

If we were to pull both of the top-2 arms instead of randomizing between them, how different would the sample complexity be?

---

> ### Author Rebuttal · Authors · 2024-08-07
>
> We thank the reviewer for the suggestions and provide our responses below.
>
> We first address the weaknesses pointed out.
>
>  In the revision we have attempted to improve the exposition, including the mathematical presentation.  We would appreciate further specific suggestions that you may have to improve the presentation.
>
> **[About Proposition 2.1 statement]**  It will be really helpful
> if the reviewer specifies the parts of Proposition 2.1 statement that are not understandable. We explained the notations and quantities involved in the statement of Proposition 2.1 in the discussion before the proposition. As Reviewer Uo1U pointed out in the first weakness, the only mild confusion in the proposition statement was the meaning of $x_{1,a}$ in the new space of allocations $\mathbf{N}$. We now clarify it in the revision.
>
>
> **[Unexplained concepts]**
>
> - **Implicit function theorem (IFT):** This is a classical result in real analysis and is widely used in optimization theory. We now add a reference [Luenberger and Ye, Linear and non-linear programming] to the revision.
>
> - **Generalized log likelihood ratio (GLLR):** The GLLR test is a standard result in the best arm identification literature.  While defining the stopping rule in Section 3, we referred to [13, Section 3.2] for a detailed derivation of the GLLR test. We couldn't provide this derivation in the main body of our paper due to space constraints.
>
>
> **Response to the question:**
>
> **[Pulling both leader and challenger]** This algorithm essentially a stratified version of $\beta$-EB-TC policy in [16] with $\beta=1/2$ (see, e.g., [Paul Glasserman, Monte Carlo methods in financial engineering] for use of stratified sampling in simulation). By modifying the anchor function to $g=1/2-\frac{N_1}{\sum_{a\in[K]}N_a}$, our analysis in Appendix E.2 implies that the proposed strategy will converge to the $\beta$-optimal proportion with $\beta=1/2$. The proposed strategy won't be asymptotically optimal if the optimal $\beta$ (which is same as the optimal value of $\omega_1$ in the min-max problem in Eq. (1)) is different from $1/2$.

---

> > ### Comment · Reviewer_jbje · 2024-08-11
> >
> > Thanks for your response. In general, as a non-expert in the area, I found this paper quite inaccessible. I believe that improving the exposition and mathematical presentation will help a lot, as will adding definitions or at least references for IFT, GLLR, etc.
> >
> > In terms of Proposition 2.1 specifically, it is stated that it characterizes the unique optimal allocation, but it is very hard or me to connect this to the wall of math in the proposition. I don't think saying "The following statements are true" and then listing a sequence of technicality true statements is the clearest way to express a theorem. I would find more verbal explanation very helpful.
> >
> > Given that the authors have expressed a commitment to strengthening the exposition, and that other more expert reviewers seem to feel positively about the work, I am happy to increase my score from 3 to 5.

---

> > > ### Author Response · Authors · 2024-08-14
> > >
> > > Thank you for increasing the score. We very much appreciate your comment that exposition and mathematical presentation needs to be improved. We are tweaking the exposition at a few places of the paper to improve the presentation. We illustrate the changes below through an updated version of Proposition 2.1:
> > >
> > > Proposition 2.1 is crucial for constructing the fluid dynamics. Proposition 2.2 provides a set of conditions which uniquely identify the optimal allocation $\omega^\star$. We define the quantity $N_{\min}=N_{1,1}+\sum_{a\in\bar{B}^c}N_a$.
> > >
> > >
> > > **Proposition 2.1:** For every positive $N$ satisfying $N\geq N_{\min}$, there is a unique set of variables $N_{\bar{B}}(N)=\left(N_a(N)\geq 0:a\in\bar{B}\right)$ and $I_B(N)$ satisfying the following conditions:
> > >
> > > $\sum_{a\neq 1}\frac{d(\mu_1,x_{1,a})}{d(\mu_a,x_{1,a})}=1$ where $x_{1,a}=\frac{N_1(N)\cdot\mu_1+N_a(N)\cdot \mu_a}{N_1(N)+N_a(N)}$ and
> > >
> > > $N_1(N)\cdot d(\mu_1,x_{1,a})+N_a(N)\cdot d(\mu_a,x_{1,a})=I_B(N)$ for every $a\in B$.
> > >
> > > Furthermore, $N_{\bar{B}}(\cdot)$  and $I_B(\cdot)$ are continuously differentiable w.r.t. $N$ for every $N>N_{\min}$.
> > >
> > >
> > > **Proposition 2.2:** Upon taking $B=[K]/{1}$ and $N=1$, $N_{\bar{B}}(1)$, as defined in Proposition 2.1, is same as the unique optimal allocation $\omega^\star$ solving the max-min problem in Eq.(1), and we also have $I_B(1)=T^\star(\mu)^{-1}$. Moreover, for every $N>0$, the unique solution $N_{\bar{B}}(N)=(N_a(N):a\in[K])$ satisfies $N_a(N)=N\omega_a^\star$.
> > >
> > >
> > > **[Providing references]** We have added references to the IFT in the revision. About the GLLR, we referred to the discussion in [13, Section 3.2] in Section 3 of the submitted version of our paper. Furthermore, in the revision, we have made a section in the appendix explaining the intuition behind GLLR.

---

### Official Review · Reviewer_Aqmy · 2024-07-08

**Soundness:** 4
**Presentation:** 3
**Contribution:** 3
**Rating:** 6
**Confidence:** 3

**Summary:**

This paper focuses on best arm identification (BAI) under the fixed confidence setting. It assumes the underlying distributions are from the single parameter exponential function. It discusses the problem of how to find the optimal $\beta$ for Top-Two type of algorithm for BAI, where $\beta$ denotes the probability of pulling the empirical best arm at each step. It proposes two algorithms, called Anchored Top-2 (AT2) algorithm and an improved version called Improved AT2 (IAT2) algorithm. It shows that both of the algorithms reaches the optimal sample complexity when $\delta \to 0$. A key idea the authors propose is to identift the underlying fluid dynamics using the ODEs tracked by the proposed algorithms. This idea helps to prove the convergence of the algorithms. It also does simulations to show the index values for each arm at different time steps for AT2 and compares the sample complexity of AT2 with other popular Top-Two algorithm.

**Strengths:**

1. The paper discusses an important question which is to find the optimal $\beta$ in the Top-Two algorithm and shows a sound solution (AT2 algorithm) to solve this problem.
2. The contributions are solid and they novelly use the fluid dynamics to describe the behaviors of the proposed algorithms.
3. The tools for proving the upper bound of the sample complexity of the proposed algorithms are also novel enough.

**Weaknesses:**

1. The paper is rather technical and lacks some intuitive explanations of different parts. It would be nice to include a picture to desribe the actual behavior of the paths captured by the ODEs if possible. Or just provide more intuitions for the steps of the algorithms.
2. The experiments lack some more explanations to discuss the meanings behind.

**Questions:**

1. possible typos: "There" -> "Their" in line 88?
2. What is $a$ in the equation at the end of page 3?
3. Why the threshold for function $g(\cdot)$ in the algorithm is 0? Is it possible or trivial to change to other thresholds?
4. Why the stopping condition is ignored as mention in line 249 for the fluid dynamics?

**Limitations:**

See weakness.

---

> ### Author Rebuttal · Authors · 2024-08-07
>
> We thank the reviewer for the suggestions and provide our responses below.
>
> We first address the weaknesses pointed out.
>
> **[Dynamics of the ODEs]** We illustrate the evolution of anchor function and indexes in the fluid dynamics in Figure 2 of the pdf attached with the global response. In the figure, we can see that, the fluid dynamics start at an allocation with positive value of the anchor function. After that, as $N_1$ increases, the anchor function decreases and becomes zero nearly at time $50$. After that, the fluid dynamics keeps on maintaining the anchor at zero and gives samples to the first arm and minimum index arms. We can observe in the plot that all the indexes are hit eventually and the black index, corresponding to the worst arm, is the last index to be hit. Moreover, after the last index is hit, the fluid dynamics follows the optimal proportions.
>
> **[Intuition behind steps of the algorithm]** The intuition behind the algorithms is derived from the first order conditions in the lower bound optimization problem and is discussed in **(2)** of the global response. We have added this to the revision. We had provided some intuitions in the earlier draft in the introduction in the lines 54-70.
>
> **[Explanation of the experiments]** Owing to space constraints, we restricted our discussion in Section 6 of the main body to: (1) Illustrating that the algorithm closely mimics the fluid path, and (2) Comparing the proposed algorithm with the existing $\beta$-EB-TC(I) [16] and (I)TCB [21] policies. We provide the detailed experiments in Appendix J.
>
> We were a little concise in our attempt to explain the underlying intuitions governing the algorithm. With the addition of Fig 2 in the pdf attached to global rebuttal and the associated discussion, we now explain the intuitions better.
>
> **Response to the questions:**
>
> **[Keeping the anchor at threshold zero]** By Proposition 2.1, the optimal proportion is uniquely identified by the first order conditions: the anchor $g=0$ and indexes of sub-optimal arms are equal. If we modify the threshold of the anchor function to some value other than zero, then the algorithm will converge to some allocation which won't be optimal. As a result, the proposed algorithms will suffer a higher sample complexity if we don't use the threshold zero, and will not be assymptotically optimal. In  **(2)** of the global rebuttal, we explain the intuitive reason behind choosing the specific threshold zero for the anchor function based on the first order conditions.
>
> **[Ignoring stopping condition in the fluid dynamics]** The stopping time condition was ignored to keep the discussion simple. It is easy to incorporate it. We simply keep track of the idealized GLLR (where the means are known) and stop when it exceeds the threshold $\beta(N,\delta)$ defined in line 200. The fluid dynamics may stop before the system reaches stability. We have added this comment to the revision.

---

> > ### Comment · Reviewer_Aqmy · 2024-08-11
> >
> > Thanks to the authors for addressing my questions and weaknesses. I think adding the specifications and explanations to the experiments and question 2 would make the paper more intuitive to understand. I intend to keep my score and good luck for the submission.

---

> > > ### Author Response · Authors · 2024-08-13
> > >
> > > Thanks again for your suggestions. These will improve the paper's presentation.

---

### Official Review · Reviewer_4f2z · 2024-07-17

**Soundness:** 3
**Presentation:** 3
**Contribution:** 3
**Rating:** 7
**Confidence:** 4

**Summary:**

The paper is about fixed confidence bandit best arm identification. The authors study the top-two family of algorithms. Those algorithms identify at each time step a leader and a challenger arm, and sample one of those two arms: typical top-two algorithms in the literature sample the leader with a fixed probability $\beta$ chosen in advance. A top-two algorithm that uses a fixed $\beta$ cannot have optimal expected sample complexity for all problems, since the optimal proportion of samples for the best arm might not be $\beta$. In order to derive an asymptotically optimal top-two algorithm, the authors introduce an adaptative choice of $\beta$, and study its dynamic through the study of a continuous time fluid dynamic.

**Strengths:**

- The method proposed is the first fully adaptive choice of the proportion $\beta$ that ensures asymptotic optimality for all single parameter asymptotic families. Another optimal method was already proposed in [28], but only for Gaussian distributions.

- The fluid dynamic model is particularly interesting: it gives a good intuition of why the dynamic for $\beta$ converges to the right proportions. It is also a very original approach to the study of bandit identification algorithms.

- The authors propose a full algorithm (leader and challenger choice as well as the new adaptive proportions $\beta$), but the new adaptive proportions could presumably be used with other choices for the leader and challenger, and could become a generic tool for asymptotically optimal top-two algorithms.

- The paper is clear and explains the idea behing the new method very well.

**Weaknesses:**

- The analysis is very asymptotic. It uses that after a while, the forced exploration of steps 1 and 2 in the algorithm ensures that all means are well estimated. That implicitly requires that $\delta$ should be small enough that the stopping condition is not attained before that time. Most of the literature on top-two algorithms is also asymptotic so that is not very surprising, but there are also non-asymptotic results (for fixed $\beta$): see [Jourdan & Degenne, Non-asymptotic analysis of a ucb-based top two algorithm].

- the algorithm AT2 uses forced exploration, which could hurt in practice: if the parameter $\alpha$ is set too high, the algorithm will explore some bad arms too much. On the other hand if it is set too low, the algorithm can get stuck sampling sub-optimal arms. Indeed, if $\alpha$ is zero the algorithm is close EB-TC from [16] which can have horrible practical performance, and the new algorithms will likely get stuck in the same way: if there is one good arm and two bad arms with equal means but the good arm gets low empirical mean at the beginning, the algorithm will sample the two bad arms forever (see [16]). IAT2 should not suffer from the same issue, thanks to its added exploration mechanism.
I did not find the value of $\alpha$ used in the experiments and the effect of that parameter was not studied experimentally.

- The analysis is very involved and might be hard to reuse or extend in other contexts. In particular, it uses a lot the particular shape of the optimality conditions for the sampling proportions. It is not clear if the right properties to use the same kind of technique will hold in other settings like structured bandits. This is more an open problem than a weakness of the paper.

Remarks and typos:

- A remark about this review: the paper including the appendix is 75 pages long, hence I could not check every proof.

- An algorithm $\beta$-EB-TCB is mentionned repeatedly, supposedly coming from [16]. There is no algorihtm called $\beta$-EB-TCB in [16]. Is it $\beta$-EB-TC? Likewise, $\beta$-EB-ITCB does not exist in that paper: is it $\beta$-EB-TCI?

**Questions:**

- The algorithm uses forced exploration to ensure that every arm is sampled some minimal amount (steps 1 and 2, lines 206 and 207). Suppose that instead, we use some variant of top-two without forced exploration, with the adaptive $\beta$ that you propose, and prove that it naturally explores (we prove a "sufficient exploration" lemma as in [16]]). That is not possible with AT2 without forced exploration unless all arms have different means since it uses greedy leader and challenger, but it could be possible with IAT2. Would the analysis still apply? Or is the forced exploration also used for something else in the analysis?

- The upper bound on the time needed to reach proportions close to the optimal proportions depends on the inverse of the minimal optimal proportion over the arms (line 262, line 321). That upper bound can be very large if the minimum is close to 0, which can happen if there is a very bad arm. Can this be avoided?

**Limitations:**

The limitations are adequately addressed.

---

> ### Author Rebuttal · Authors · 2024-08-07
>
> We thank the reviewer for the suggestions and provide our responses below.
>
> We first address the weaknesses.
>
> **[Asymptotic results of the paper]** We thank the reviewer for pointing us to the recent reference [Jourdan and Degenne, '23]. We respond to it in **(3)** of the global rebuttal.
>
> **[Experiments involving exploration parameter** $\alpha$ **]** This is an extremely important point. We address it empirically in **(1)** of the global rebuttal and have added the experimental details in the revision.
>
> **[Extending the analysis to other problems]** Our analysis can be used for proving asymptotic $\beta$-optimality of the $\beta$-EB-TC (I) algorithms in [16] by using the anchor function $g=\beta-\frac{N_1}{\sum_{a\in[K]}N_a}$ (see Appendix E.2). We also expect it to work for best $k$-arm selection as in [28]. In general, many algorithms in the bandit setting rely on keeping indexes close to each other. Our fluid approach may be useful while analyzing these settings.
>
> We now update to $\beta$-EB-TC (I) from [16] in the revision.
>
> **Response to questions:**
>
> **[Use of forced exploration in analysis]** We need forced exploration **only to prove**: The estimated means converge to the actual means at rate $O(N^{-3\alpha/8})$ ($\alpha$ being the exploration parameter) after a random time $T_0$ of finite expectation independent of $\delta$. Our analysis will work if we can prove a sufficient exploration lemma (like in [16] showing $\Omega(\sqrt{N})$ exploration of every arm for the $\beta$-EB-TC (I) policies) for the proposed algorithms, and asymptotic optimality will follow.
>
> **[Bounding the time to reach optimal proportion]** The argument in Section 4 to bound the time to reach optimal proportion implies that: The fluid dynamics reach optimal proportion at time $N_a^0/\omega_a^\star$ where $a$ is the sub-optimal arm whose index is met last by the fluid dynamics, which will give us an upper bound $N^0/\omega_a^\star$. If the worst arm (which also has minimum optimal proportion) has the largest no. of samples to begin with (i.e. largest value of $N_a^0$), then the worst arm has largest index throughout the fluid dynamics. As a result, the arm with minimum optimal proportion is hit last, giving us the upper bound $N_K^0/\omega_{\min}^\star$. Since the worst arm has maximum no. of initial samples, $N_K^0$ can be arbitrarily close to $N^0$, giving us the tight upper bound $N^0/\omega_{\min}^\star$. The above argument gives us a tight upper bound for the time to reach optimal proportions for the fluid dynamics. Since the algorithm closely tracks the fluid dynamics after a random time $T_0$ of finite expectation, similar worst case bounds hold for the algorithm as well.
>
> Note that for $\delta$ large (say 5%), the algorithm may reach its stopping time before the largest index is hit. In that case, even in the worst case as above the upper bound $N^0/\omega_{\min}^\star$ may be loose.

---

### Author Rebuttal · Authors · 2024-08-07

Here we address questions/weaknesses raised by more than 1 reviewer. The manuscript will be updated as per our reponses.

**1) Conjectures on sufficient exploration and numerical experiments supporting them:** As reviewer 4f2z rightly pointed out, AT2 needs forced exploration. Otherwise, for instances where the second and third best arms have equal means, AT2 might see some bad samples from the best arm in the beginning. After that, AT2 will sample forever from the second and third best arms. Following Reviewer 4f2z's comment, we **conjecture** that: AT2 will do sufficient exploration for instances having all arms with different means. Intuitively, it might appear that force exploration significantly increases the sample complexity of AT2 and IAT2. However, in the experiments, we see IAT2's performance remains unaffected and AT2 performs atleast as well as IAT2 with moderate amount of forced exploration even for instances where gap between the means is small (see Figure 1.a of the attached pdf). We further conjecture that IAT2 performs sufficient exploration for all instances, including those where multiple sub-optimal arms have equal means. In Figure 1.b of the attached pdf, we consider a four armed instance where the three sub-optimal arms have equal mean. We observe that without forced exploration, AT2's sample complexity blows up, and IAT2 performs significantly better than AT2. Furthermore, IAT2's sample complexity remains almost unaffected w.r.t. the amount of forced exploration done. Proving sufficient exploration for the proposed algorithms appears to be technically challenging and the proofs are likely to be lengthier.


**2) Intuition behind the choice of anchor function and algorithm:** Statement 2 of Proposition 2.1 implies: The optimal proportion solving the min-max problem in Eq.1 is uniquely identified by the first order conditions: (1) Anchor function $g=\sum_{a\neq 1}\frac{d(\mu_1,x_{1,a})}{d(\mu_a,x_{1,a})}-1$ evaluated at the proportion is zero, and (2) Index of all the sub-optimal arms are equal. The proposed algorithms (AT2 and IAT2) chase these first order conditions. At every iteration, the sampling strategy of AT2 and IAT2 pushes the anchor function $g$ towards zero. Once the anchor function becomes approximately zero, the proposed algorithms attempts to maintain equality of the indexes by choosing the minimum index arm as challenger (see Section 5 for detailed arguments). Thus the proposed algorithms converge to the first order conditions (see Proposition 5.1), which also implies convergence to the optimal proportion (see Proposition 3.1), and hence, asymptotic optimality follows (see Theorem 3.1). We made an attempt to explain these intuitions in the "contributions" section of the introduction.

Below we provide a simplier argument to support the appearance of the anchor function $g$ in the first order conditions. The lower bound problem of sample complexity for a chosen confidence level $\delta$ can be alternatively expressed as solution to the problem $\mathbf{O}$ defined in Eq. 20 of Appendix D. Note that $\mathbf{O}$ is an optimization problem over $K$ variables. We reduce this to a single variable optimization problem with $N_1$ being the only free variable. In Theorem D.1 we prove that the index constraints are tight at the optimal solution of $\mathbf{O}$ i.e. $W_a(N_1,N_a)=\log(1/(2.4\delta))$.  So, we can restrict to allocations where index of the sub-optimal arms are $\log(1/(2.4\delta))$. Fixing some $N_1\geq 0$, let $\bar{N}_a(N_1)$ be the *unique* value of $N_a$ at which $W_a(N_1,N_a)=\log(1/(2.4\delta))$. We take $\bar{N}_a(N_1)=\infty$ if no such $N_a$ exists.

Thus $\mathbf{O}$ is equivalent to the single variable optimization problem of minimizing $f(N_1)=N_1+\sum_{a\neq 1}\bar{N}_a(N_1)$ over $N_1\geq 0$. Each $\bar{N}_a(N_1)$ is strictly decreasing and differentiable with derivative $-\frac{d(\mu_1,x_a(N_1))}{d(\mu_a,x_a(N_1))}$ where $x_a=\frac{N_1 \mu_1+\bar{N}_a(N_1) \mu_a}{N_1+\bar{N}_a(N_1)}$.

Hence $f^\prime(N_1)=1-\sum_{a\neq 1}\frac{d(\mu_1,x_a(N_1))}{d(\mu_a,x_a(N_1))}$, which is exactly the negative of the anchor $g(\cdot)$ at the allocation $N_1$ and, $N_a=\bar{N}_a(N_1)$ for $a\neq 1$. Through differentiation it can be seen that $f(N_1)$ is strictly convex w.r.t. $N_1$ and attains its unique minimum when $g=0$. The (I)TCB algorithm's sampling strategy can be sub-optimal. We discuss this further in response to reviewer Uo1U.

**3) Getting non-asymptotic upper bounds:** Proving non-asymptotic  upper bounds to sample complexity which asymptotically matches the lower bound is a challenging research direction in the best arm identification literature.  In Theorem 2.1, we prove an almost sure non-asymptotic upper bound of the stopping time $\tau_\delta$, where we hide the order of dependence of the instance dependent constants. Understanding exact dependence of these constants requires more technically involved arguments.  [Jourdan and Degenne, Non-Asymptotic Analysis of UCB-based Top Two Algorithm] provides a non-asymptotic analysis of a $\beta$-Top two policy, which uses UCB leader and is asymptotically $\beta$-optimal over the family of Gaussian instances with fixed variance. Their upper bound also holds for sub-Gaussian instances. Moreover, their upper bound asymptotically grows like $2T^{\star}_{1/2}(\mu)\log(1/\delta)$ for $\beta=1/2$ and doesn't match the lower bound. We expect that, their approach of choosing the UCB-based leader and  constructing concentration events over the empirical mean and the indexes can be extended for proving non-asymptotic upper bounds for the AT2 and IAT2 algorithms over a Gaussian family, but again we may have a higher constant compared to the lower bound.

---

### Decision · Program_Chairs · 2024-09-25

**Decision:**

Accept (poster)

**Comment:**

This is a very nice paper that I read in detail and decided to accept together with the strong recommendation from the reviewers. It is the first paper that provides a fully adaptive choice of the proportion $\beta$ in top two algorithms. This is an important advancement. Furthermore, the new techniques that involve the notion of anchor function and the use of the theory of ODEs and the implicit function theorem to demonstrate the asymptotic optimality of the proposed algorithm may pave the way for further theoretical advancements in the area of fixed-confidence BAI.